# A Survey of Optimization Methods for Training DL Models: Theoretical Perspective on Convergence and Generalization

## Abstract

As data sets grow in size and complexity, it is becoming more difficult to pull useful features from them using hand-crafted feature extractors. For this reason, deep learning (DL) frameworks are now widely popular. DL frameworks process input data using multi-layer networks. Importantly, DL approaches, as opposed to traditional machine learning (ML) methods, automatically find high-quality representation of complex data useful for a particular learning task. The Holy Grail of DL and one of the most mysterious challenges in all of modern ML is to develop a fundamental understanding of DL optimization and generalization. While numerous optimization techniques have been introduced in the literature to navigate the exploration of the highly non-convex DL optimization landscape, many survey papers reviewing them primarily focus on summarizing these methodologies, often overlooking the critical theoretical analyses of these methods. In this paper, we provide an extensive summary of the theoretical foundations of optimization methods in DL, including presenting various methodologies, their convergence analyses, and generalization abilities. This paper not only includes theoretical analysis of popular generic gradient-based first-order and second-order methods, but it also covers the analysis of the optimization techniques adapting to the properties of the DL loss landscape and explicitly encouraging the discovery of well-generalizing optimal points. Additionally, we extend our discussion to distributed optimization methods that facilitate parallel computations, including both centralized and decentralized approaches. We provide both convex and non-convex analysis for the optimization algorithms considered in this survey paper. Finally, this paper aims to serve as a comprehensive theoretical handbook on optimization methods for DL, offering insights and understanding to both novice and seasoned researchers in the field.

## 1 Introduction

The rapid evolution of DL models (LeCun et al., 2015; Goodfellow et al., 2016; Vaswani et al., 2017; OpenAI, 2023; Achiam et al., 2023; Touvron et al., 2023), combined with the exponential growth of available data (Chen & Lin, 2014), has pushed the field of machine learning into new frontiers. As these models grow in complexity and scale (Kasneci et al., 2023; Chang et al., 2024), optimizing their performance becomes paramount to harnessing their full potential. Consequently, the goal of improving the existing optimization methods for training DL models has attracted significant attention among researchers who aim at developing efficient (fast-converging), accurate (well-generalizing), scalable (applicable to large data sets and models and heavily parallelizable), and consistent (minimizing the effect of non-deterministic calculations in heavily parallelized systems) DL optimization strategies (Sutskever et al., 2013; Kingma & Ba, 2014; You et al., 2019; Shoeybi et al., 2019; Yang et al., 2023), and furthermore designing DL model architectures that properly condition the optimization problem such that a high-quality solution is easier to find (He et al., 2016; Ioffe & Szegedy, 2015; Klambauer et al., 2017; Vaswani et al., 2017; OpenAI, 2023).

While the existing literature contains plethora of optimization techniques designed for DL applications, there remains a conspicuous gap in the theory of DL optimization. Many survey papers and studies mainly focus on summarizing the methodologies that are employed (Sun, 2020; Ruder, 2016; Le et al., 2011), and often overlook their theoretical foundations. This oversight limits the comprehensive understanding of DL optimization

techniques, which hinders the progress in the field of DL. This papers fills the existing theoretical gap and provides an extensive summary of the theoretical foundations of optimization methods in DL, including describing methodologies, providing convergence analyses, and showing generalization abilities.

Next we summarize the optimizers that have been used in a DL setting to train DL models.

Gradient-based optimization methods are widely used for their computational efficiency in enhancing the performance of DL models. These methods can generally be categorized into two categories: i) First-order methods (Robbins & Monro, 1951; Polyak, 1964; Nesterov, 1983; Liu et al., 2020), which are extensively employed in stochastic versions to reduce the computational burden associated with large datasets and complex model architectures in DL (Bottou, 2010; Sekhari et al., 2021; Tian et al., 2023); ii) Second-order methods (Broyden, 1967; Fletcher, 1970; Goldfarb, 1970; Shanno, 1970; Liu & Nocedal, 1989; Yuan, 1991; Yuan et al., 2022; 2024), which utilize second-order information, such as the Hessian matrix, to inform the search direction during optimization and are applied in DL (Bollapragada et al., 2018; Wang & Choromanska, 2020; Yousefi & Martínez Calomardo, 2022; Niu et al., 2023). In recent years, some advancements have been made in first-order optimization algorithms. (Jin et al., 2021; Huang & Becker, 2020; Yang et al., 2022) focuses on introducing stochastic perturbations to the gradients, which helps the first-order method escape saddle points—flat regions where optimization may stagnate. This is particularly beneficial in non-convex landscapes commonly found in deep learning. By effectively navigating the geometry of the loss surface, pretubed SGD allows for more efficient exploration and exploitation of the parameter space, leading to faster and more reliable convergence to optimal solutions. (Raginsky et al., 2017; Dalalyan & Karagulyan, 2019; Huang & Becker, 2021) focuses on the integration of Langevin dynamics into first-order optimization methods. These works investigate the theoretical underpinnings and practical implementations of Langevin dynamics, highlighting its potential to improve convergence rates and robustness in the presence of noise. Compared with the first-order methods, second-order methods exhibit faster convergence rates in terms of iterations. Popular Newton's method achieves a quadratic convergence rate under nonconvex assumptions, which is much faster that the linear convergence rate of first-order gradient descent (GD) method under the same assumptions. However, the requirement of computing the inverse of the Hessain matrix reslts in the time complexity that is cubic in the number of parameters, which makes it quite challenging to use in practical DL settings. In order to decrease the time complexity of second-order methods and at the same time preserve its fast convergence rate, quasi-Newton's methods (Dennis & Moré, 1977) have been proposed. Broyden–Fletcher–Goldfarb–Shannon (BFGS) (Broyden, 1967; Fletcher, 1970; Goldfarb, 1970; Shanno, 1970) algorithm is the most famous quasi-Newton's method that decreases the time complexity from cubic to quadratic while keeps superlinear convergence rate. However, BFGS still suffers from infavorable memory requirements due to the necessity of storing large dimenssional pseudo-Hessian matrix. In order to reduce the memory storage and the computation load more, (Liu & Nocedal, 1989) proposed the limited-memory BFGS (LBFGS) which decreases the time complexity to linear by compressing the pseudo-Hessian matrix. These second-order methods are also used as optimizers in training DL networks (Bollapragada et al., 2018; Wang & Choromanska, 2020; Yousefi & Martínez Calomardo, 2022; Niu et al., 2023). In this paper, we discuss the convergence rate of all the aforementioned second-order methods. Except for the (quasi-)Newton's methods we mentioned before, there is another track of second-order algorithm, the Hessian-free (Martens et al., 2010; Martens & Sutskever, 2011) algorithm. While quasi-Newton methods rely on constructing an approximate Hessian matrix to guide parameter updates, Hessian-free optimization avoids this by directly estimating curvature through iterative processes, such as conjugate gradient methods. This results in reduced computational and memory requirements, making it particularly suitable for the high-dimensional landscapes often encountered in deep neural networks. We do not focus on Hessain-free methods in our review, because works on this track come with no theoretical guarantees in the literature, they are purely empirical.

Furthermore, our exploration extends beyond conventional gradient-based first-order and second-order optimization methods. We delve into the analysis of innovative techniques grounded in the understanding of the properties of the DL loss landscape (Li et al., 2018; Cooper, 2018; Chaudhari & Soatto, 2015; Keskar et al., 2017b; Bisla et al., 2022; Orvieto et al., 2022). They aim at identifying optimal points located in the flat valleys in the DL optimization landscape, which, as they argue, correspond to lower generalization errors. The landscape-aware DL optimization methods provides valuable insights into the underlying mechanisms

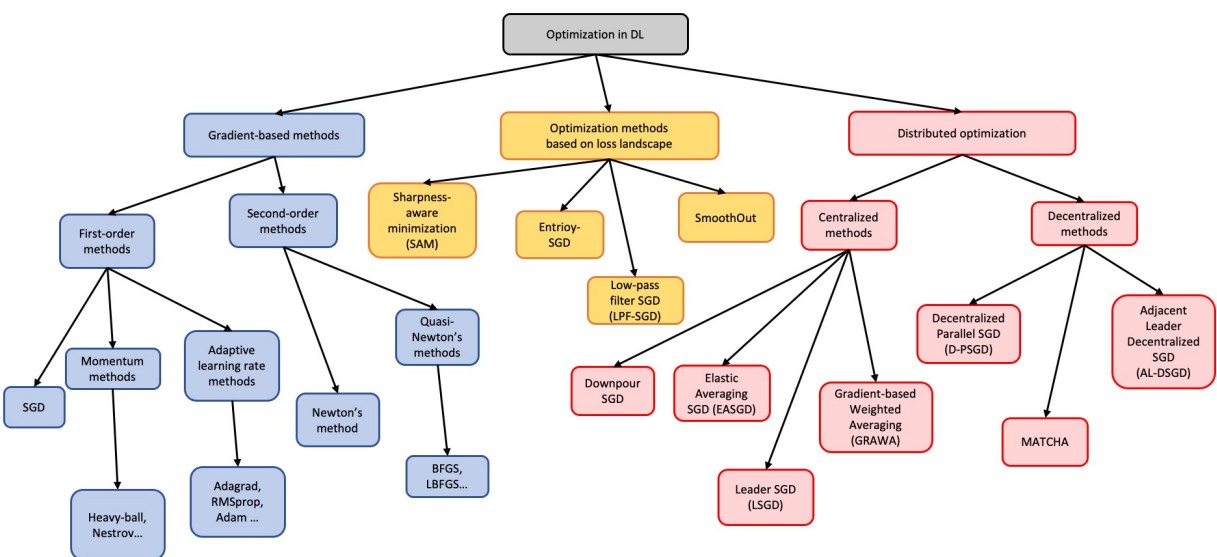

Figure 1: The DL optimization methods discussed in this survey paper.

that govern the exploration of non-convex loss surfaces, contributing to the the development of more effective and robust DL optimization strategies.

The approaches adapting the DL optimization strategy to the properties of the loss landscape and encouraging the recovery of flat optima could be generally classified as follows: i) regularization-based methods that regularize gradient descent strategy with sharpness measures such as the Minimum Description Length (Hochreiter & Schmidhuber, 1997), local entropy (Chaudhari et al., 2017), or variants of $\epsilon$-sharpness (Foret et al., 2021; Keskar et al., 2017b), low-pass-filter-based norm (Bisla et al., 2022) ii) surrogate methods that evolve the objective function according to the diffusion equation (Mobahi, 2016), iii) averaging strategies that average model weights across training epochs (Izmailov et al., 2018; Cha et al., 2021), and iv) smoothing strategies that smooth the loss landscape by introducing noise in the model weights and average model parameters across multiple workers run in parallel (Wen et al., 2018; Haruki et al., 2019; Lin et al., 2020) (such methods focus on distributed training of DL models with an extremely large batch size). In this paper, we choose four most representative methods from among the above mentioned ones: Sharpness-Aware Minimization (SAM) (Foret et al., 2021; Li et al., 2023), Entropy-SGD (Chaudhari et al., 2017), Low-pass filter SGD (LPF-SGD) (Bisla et al., 2022) and SmoothOut (Wen et al., 2018). We discuss principles of these methodologies and theoretically analyze their generalization abilities.

Our investigation of DL optimization further continues to distributed optimization schemes, which facilitate parallel computations. Our survey includes a thorough summary of both centralized and decentralized approaches, highlighting their respective advantages and limitations in optimizing DL models across distributed computing environments.

For centralized methods, we start with the Downpour SGD (Dean et al., 2012; Ben-Nun & Hoefler, 2018; Gholami et al., 2018), an efficient data-paralleled distributed optimization technique with a central worker gathering and leveraging the information from local distributed workers to coordinate the optimization process. The method relies on frequent information exchange between the center worker and local workers and overlooks the potential variations in problem settings arising from differing data shards across each local worker. In order to solve this, (Zhang et al., 2015) propose the Elastic Averaging SGD (EASGD) algorithm which introduces an elastic force between the center worker and local workers to encourage more exploration within each local worker by allowing their parameters to fluctuate further from the center parameter. The Leader Stochastic Gradient Decent (LSGD) (Teng et al., 2019) algorithm further improves over EASGD. LSGD is well-aligned with current hardware architecture, where local workers forming a group lie within

| | Method | Assumptions | | | | Step size | Convergence rate | Comment |
|---|---|---|---|---|---|---|---|---|
| | | Convex | Lip. smooth | Hessian bound | Variance bound | | | |
| First-order Methods | SGD | strongly convex | ✗ | ✗ | ✓ | $\propto \frac{1}{k+1}$ | sub-linear: $\mathcal{O}(\frac{1}{K+1})$ | The step size of SGD/SGD-M is inversely proportional to the iteration counter while that of Adaptive methods is constant or bounded, which makes it hard theoretically to compare these methods in terms of converegence speed. However, several studies (Kingma & Ba, 2014; Reddi et al., 2019; Schmidt et al., 2021) have empirically shown that Adaptive methods converge faster time- and iteration-wise and achieve lower training loss in fewer iterations than SGD in various applications. |
| | | ✗ | ✓ | ✗ | ✓ | $\propto \frac{1}{\sqrt{k}}$ | sub-linear: $\mathcal{O}(\frac{1}{\sqrt{K}})$ | |
| | SGD-M | convex | ✗ | ✗ | ✓ | $\propto \frac{1}{\sqrt{k+1}}$ | sub-linear: $\mathcal{O}(\frac{1}{\sqrt{K+1}})$ | |
| | | ✗ | ✓ | ✗ | ✓ | $\propto \frac{1}{\sqrt{k}}$ | sub-linear: $\mathcal{O}(\frac{1}{\sqrt{K}})$ | |
| | Adagrad | ✗ | ✓ | ✗ | ✓ | Const. | sub-linear: $\mathcal{O}(\frac{\ln K}{\sqrt{K}})$ | |
| | Adam | ✗ | ✓ | ✗ | ✓ | $\propto\sqrt{\frac{1-\beta_2^k}{1-\beta_2}}$ | sub-linear: $\mathcal{O}(\frac{\ln K}{\sqrt{K}})$ | |
| Second-order Method | Newton's Method | ✗ | ✗ | ✓ | ✗ | ✗ | quadratic | Although second-order methods converge faster than first-order methods iteration-wise, the computation of the Hessian is expensive, which makes these methods slower and less preferred than first-order techniques, especially in massive data and model settings typical for deep learning. |
| | BFGS | ✗ | ✓ | ✓ | ✗ | ✗ | super-linear | |
| | LBFGS | convex | ✗ | ✓ | ✗ | ✗ | R-linear | |

Table 1: Comparison of convergence results for different optimization methods (Part 1). For convex functions, the stopping criterion is based on the condition that the difference between the current function value $F(x)$ and the optimal value $F(x^*)$ becomes sufficiently small. In the case of non-convex functions, the stopping criterion is determined by the gradient norm $\|\nabla F(x)\|$, which should be sufficiently small.

| | | Assumptions | | | | Step size | Convergence rate | Comment |
|---|---|---|---|---|---|---|---|---|
| | Method | Convex | Lip. smooth | Hessian bound | Variance bound | | | |
| Distributed Methods (Centralized) | Downpour-SGD | Strongly convex ✓ | ✗ | ✓ | ✗ | $\propto \frac{1}{\sqrt{k}}$ | sub-linear: $\mathcal{O}(\frac{1}{K})$ | The literature theoretically shows that centralized distributed optimization methods have constant difference convergence rate. However, (Teng et al., 2019; Dimlioglu & Choromanska, 2024) empirically show that LSGD and GRAWA achieve faster convergence time-wise and iteration-wise and recover better quality and flatter local optima. |
| | | ✗ | ✓ | ✗ | ✓ | $\propto \frac{1}{\sqrt{k}}$ | sub-linear: $\mathcal{O}(\frac{1}{\sqrt{K}})$ | |
| | EASGD | Strongly convex ✓ | ✗ | ✓ | ✗ | $\propto \frac{1}{\sqrt{k}}$ | sub-linear: $\mathcal{O}(\frac{1}{K})$ | |
| | | ✗ | ✓ | ✗ | ✓ | $\propto \frac{1}{\sqrt{k}}$ | sub-linear: $\mathcal{O}(\frac{1}{\sqrt{K}})$ | |
| | LSGD | Strongly convex ✓ | ✗ | ✓ | ✗ | $\propto \frac{1}{\sqrt{k}}$ | sub-linear: $\mathcal{O}(\frac{1}{K})$ | |
| | | ✗ | ✓ | ✗ | ✓ | $\propto \frac{1}{\sqrt{k}}$ | sub-linear: $\mathcal{O}(\frac{1}{\sqrt{K}})$ | |
| | GRAWA | Strongly convex ✓ | ✗ | ✓ | ✗ | $\propto \frac{1}{\sqrt{k}}$ | sub-linear: $\mathcal{O}(\frac{1}{K})$ | |
| | | ✗ | ✓ | ✗ | ✓ | $\propto \frac{1}{\sqrt{k}}$ | sub-linear: $\mathcal{O}(\frac{1}{\sqrt{K}})$ | |
| Distributed Methods (Decentralized) | D-PSGD | ✗ | ✓ | ✗ | ✓ | $\propto \sqrt{\frac{m}{k}}$ | sub-linear: $\mathcal{O}(\frac{1}{\sqrt{mK}}) + \mathcal{O}(\frac{m}{K})$ | m is the number of local workers in the system. Although AL-DSGD theoretically converges slower than D-PSGD and MATCHA (though they are all sub-linear), it empirically shows comparable generalization results. |
| | MATCHA | ✗ | ✓ | ✗ | ✓ | $\propto \sqrt{\frac{m}{k}}$ | sub-linear: $\mathcal{O}(\frac{1}{\sqrt{mK}}) + \mathcal{O}(\frac{m}{K})$ | |
| | AL-DSGD | ✗ | ✓ | ✗ | ✓ | $\propto \sqrt{\frac{m}{k}}$ | sub-linear: $\mathcal{O}(\frac{1}{\sqrt{mK}}) + \mathcal{O}(\sqrt{\frac{m}{K}}) + \mathcal{O}(\sqrt{\frac{m}{K^3}})$ | |

Table 2: Comparison of convergence results for different optimization methods (Part 2). For convex functions, the stopping criterion is based on the condition that the difference between the current function value $F(x)$ and the optimal value $F(x^*)$ becomes sufficiently small. In the case of non-convex functions, the stopping criterion is determined by the gradient norm $\|\nabla F(x)\|$, which should be sufficiently small.

a single computational node and different groups correspond to different nodes. The algorithm introduces multiple local leaders for each group of local workers, and occasionally pulls all the local workers towards the current local leader as well as global leader to ensure fast convergence. Finally, inspired by EASGD and LSGD, Gradient-based Weighted Averaging (GRAWA) Dimlioglu & Choromanska (2024) applies a pulling

| | Method | Assumptions | | | | Generalization Error Bound |
|---|---|---|---|---|---|---|
| | | Convex | Lip. Cont. | Lip. smooth | Others | |
| First-order Methods | SGD | convex | M-Lip. | L-smooth | Step size: $\alpha_k \leq \frac{2}{L}$ | $\epsilon_{\text{SGD}} \lesssim \frac{M^2}{n} \sum_{k=1}^{K} \alpha_k$ |
| | | ✗ | M-Lip. | L-smooth | step size: $\alpha_k \leq \frac{1}{k}$ | $\epsilon_{\text{SGD}} \lesssim \frac{1}{n} M^{\frac{1}{1+L}} K^{\frac{L}{L+1}}$ |
| | SGD-M | $\mu$-strongly convex | M-Lip. | L-smooth | Step size: $\alpha_k = \alpha$, momentum $\beta$ | $\epsilon_{\text{SGD-M}} \lesssim \frac{\alpha M^2 (L+\mu)}{n(\alpha L \mu - 3\beta(L+\mu))}$ |
| | | ✗ | M-Lip. | L-smooth | $\alpha_k = \frac{1}{k}$, momentum $\beta$ | $\epsilon_{\text{SGD-M}} \lesssim \frac{\exp(\beta) K^{(1-1/n)L}}{n}$ |
| Landscape-aware Methods | SAM | No theoretical, only empirical generalization guarantees in literature. | | | | |
| | Entropy-SGD | ✗ | M-Lip. | L-smooth | step size: $\alpha_k \leq \frac{1}{k}$, Hessain bounded | $\frac{\epsilon_{\text{Entropy-SGD}}}{\epsilon_{\text{SGD}}} \lesssim \left(\frac{M}{K}\right)^L$ |
| | LPF-SGD | ✗ | M-Lip. | L-smooth | step size: $\alpha_k \leq \frac{1}{k}$, kernel: $\mu \sim \mathcal{N}(0, \sigma^2 I), \sigma > \frac{M}{L}$ | $\frac{\epsilon_{\text{LPF-SGD}}}{\epsilon_{\text{SGD}}} \lesssim \frac{1}{K^{\frac{1}{M/\sigma+1} - \frac{1}{L+1}}}$ |
| | Smooth Out | ✗ | M-Lip. | L-smooth | step size: $\alpha_k \leq \frac{1}{k}$, kernel: $\mu \sim U[-a, a], a > \frac{M\sqrt{d}}{L}$ | $\frac{\epsilon_{\text{SmoothOut}}}{\epsilon_{\text{SGD}}} \lesssim \frac{1}{K^{\frac{1}{M\sqrt{d}/\sigma+1} - \frac{1}{L+1}}}$ |
| Comment | The generalization ability of optimization methods is closely related to the Lipschitz continuity and Lipschitz smoothness properties of the loss function. Landscape-aware optimization methods yield tighter generalization error bounds when the number of iterations $K$ is sufficiently large. | | | | | |

Table 3: The comparison of the generalization results for different optimization methods.

force on all workers towards the center worker, but using the weighted average computed across local workers based on their gradients. In our survey we summarize the centralized methods and provide their theoretical convergence guarantees.

Different from centralized methods, decentralized methods eliminate the need for a central worker, distributing tasks among multiple workers that communicate according to a certain topology. Decentralized Parallel SGD (D-PSGD) (Lian et al., 2017; 2018) is the most fundamental decentralized method which performs local gradient updates and averages parameters of each worker with its neighbors given by the topology. Since the topology is fixed, D-PSGD encounters the error-runtime trade-off issue. In particular, note that the extremely dense topology will cause a large communication load while too sparse topology will face the problem of poor performance of the model on each worker. In order to solve this issue, MATCHA (Wang et al., 2019) provides a win-win strategy by utilizing the matching decomposition technique on a dense topology, which carries the potential to reduce the per-node communication complexity for each iteration while keeping the good performance at the same time. Adjacent Learder Decentralized SGD (AL-DSGD) (He et al., 2024) furthermore improves this approach by proposing dynamic communication graphs instead of the fixed topology and applying a corrective force on the workers dictated by both the currently best-performing neighbor and the neighbor with the maximal degree. In this paper, we discuss all the mentioned decentralized methods and provide theoretical convergence proofs for these schemes.

By synthesizing these diverse perspectives on DL optimization, this paper aims to serve as a comprehensive theoretical handbook of optimization methods for DL. We provide convex and/or non-convex analysis for the optimization algorithms considered in this survey paper. In the context of non-convex optimization, our analysis goes beyond DL setting, and in many cases can be applied more broadly in other, simpler, non-convex scenarios. We however dedicate this manuscript specifically to general DL optimization, since

our non-convex analysis makes no simplifying assumptions on the network architecture, as opposed to many past works (Dauphin et al., 2014; Baldi & Hornik, 1989; Saxe et al., 2014; Baldassi et al., 2015; 2016a;b). Finally, with this work we endeavor to provide a general proof framework that not only summarizes existing knowledge but also fosters new insights and avenues for future research.

Figure 1 provides a chart containing methods analyzed in this paper. Table 1 & 2 summarize the convergence results for the first and second-order optimization methods and distributed (centralized and decentralized) optimization methods. Table 3 summarizes the generalization results for the first-order methods and landscape-aware optimization methods. This survey paper is primarily a theoretical review of optimization methods for deep learning. The theory-experiment gap however often arises in real-world scenarios because theoretical guarantees are based on idealized conditions that may not hold in practice. Below let us briefly mention several representative assumptions required for the theoretical guarantees we discuss in this survey paper and comment whether they are commonly satisfied in applications.

1.  **Convexity:** Deep learning often deals with non-convex loss landscapes. All methods we mention except for BFGS have the theoretical guarantees under nonconvex assumption.

2.  **Smoothness:** All methods we listed require the smoothness of the loss function. In practice, loss functions may not be smooth, especially in the presence of noise or irregular data distributions. To solve this problem, deep learning researchers use techniques that improve the smoothness of the loss, among them regularization, dropout, batch normalization and so on.

3.  **Variance Bound:** All the stochastic methods assume that the variance of the gradient estimates is bounded. In practice, training loss functions may not always satisfy this assumption, particularly in scenarios involving noisy or heterogeneous data. For instance, datasets with outliers or imbalanced classes can lead to higher gradient variance, complicating the training process. In order to solve this problem, we could implement mini-batch training instead of point-wise stochastic training to average out noise and stabilize gradient estimates. This is indeed done in practice.

4.  **Step Size:** The step size used in practice often differs from that in the convergence proof. For instance, in the proof of SGD and SGD-momentum for nonconvex settings, we assume a learning rate that decreases inversely proportionally ($\alpha_t = 1/t$). However, in empirical applications, decreasing the learning rate too quickly can terminate the learning process too early, resulting in suboptimal solutions. Instead of an inversely proportional learning rate, practitioners typically employ step-wise, linear, or cosine annealing rates, which drop much slower.

We can approach the assumptions for convergence guarantees from a different perspective than viewing soft assumptions as the ones implying more useful bounds and strict assumptions as the ones implying worse bounds. Instead, these assumptions should be viewed as indicators of the specific conditions under which an algorithm is likely to perform optimally. For example, the smoothness assumption is crucial for convergence guarantees of all methods, it implies we could achieve better performance by smoothing the loss landscape of deep learning models. This is also the motivation of landscape-aware optimization methods we introduced in this paper (e.g. SAM, Entropy-SGD, etc) which could reach better generalization ability.

This survey paper is organized as follows: Section 2 discusses different first-order and second-order gradient-based methods and provides their convergence analysis and generalization abilitie. Section 3 discusses optimization methods adapting to the properties of the DL loss landscape and provides their generalization guarantees. Section 4 discusses both centralized and decentralized distributed optimization along with their convergence analysis. Section 5 concludes the paper.

## 2   Gradient-based Optimization methods

Neural networks play a pivotal role in machine learning due to their ability to effectively approximate a broad range of functions, denoted as $f(x)$. The method of backpropagation, derived from the application of the chain rule, enables easy computation of gradients of the loss function across network layers and facilitates gradient-based DL optimization algorithms. This section specifically explores the theoretical guarantees

(convergence and/or generalizaqtion guarantees) associated with state-of-the-art gradient-based optimization algorithms. Notably, it investigates the applicability and reliability of the first-order methods like SGD, Momentum methods, AdaGrad, and Adam, as well as second-order methods such as BFGS and L-BFGS.

## 2.1 Preliminaries

We begin by introducing important definitions.

Preliminaries for assumptions of functions In this section, we define the key assumptions that are commonly made about objective functions in optimization problems. These assumptions are essential for analyzing the convergence properties of optimization algorithms. The most commonly used assumptions include strong convexity, $L$-Lipschitz continuity, and $L$-smoothness. Below are the formal definitions for these properties:

**Definition 1** (strongly convex). *A differentiable function $f : \mathbb{R}^n \to \mathbb{R}$ is $\mu$-strongly convex with constant $\mu > 0$ if there exists a constant $\mu > 0$ such that for all $x, y \in \mathbb{R}^n$, the following inequality holds:*

$$f(y) \geq f(x) + \nabla f(x)^\top (y - x) + \frac{\mu}{2} \|y - x\|^2$$

*This condition implies that $f(x)$ has a unique minimizer, and the function is "curved upwards" around its minimizer, making optimization easier and ensuring that gradient-based methods converge faster. Strong convexity provides a lower bound on the function's curvature.*

**Definition 2** (L-Lipschitz continuity). *A differentiable function $f : \mathbb{R}^n \to \mathbb{R}$ is L-Lipschitz continuous if there exists a constant $L > 0$ such that for all $x, y \in \mathbb{R}^n$, the following inequality holds:*

$$\|\nabla f(x) - \nabla f(y)\| \leq L \|x - y\|$$

*This condition ensures that the gradient does not change too rapidly, which is important for ensuring stability in gradient-based optimization algorithms. The constant $L$ is known as the Lipschitz constant.*

**Definition 3** (L-smoothness). *A differentiable function $f : \mathbb{R}^n \to \mathbb{R}$ is L-smooth if its gradient $\nabla f(x)$ is L-Lipschitz continuous, meaning that there exists a constant $L > 0$ such that for all $x, y \in \mathbb{R}^n$, the following holds:*

$$\|\nabla f(x) - \nabla f(y)\| \leq L \|x - y\|$$

*In other words, $f(x)$ is L-smooth if its gradient changes at most linearly with respect to the distance between $x$ and $y$. This property is important for proving the convergence rates of gradient-based methods, as it bounds the rate of change of the gradient, ensuring that optimization algorithms converge at a reasonable pace.*

### 2.1.1 Preliminaries for the Convergence Analysis

One of the most fundamental way of describing any optimization method is through convergence guarantee. This guarantee is of vital importance since it assures that the algorithm will reach a global/local optimum under certain conditions. A typical way to compare the performance of different algorithms is to compare their rates of convergence. Following (Schatzman, 2002), below we define four rates of convergence that we will focus on — sub-linear, linear, superlinear, and quadratic. They are ordered from the slowest to the fastest one.

**Definition 4** (Sublinear convergence rate). *Suppose we have a sequence $\{x_k\} \subset \mathbb{R}^d$ such that $x_k \to x^*$ when $k \to \infty$. We say that the convergence is sub-linear if*

$$\limsup_{k \to \infty} \frac{\|x_{k+1} - x^*\|}{\|x_k - x^*\|} = 1$$

**Remark.** If $\frac{\|x_{k+1} - x^*\|}{\|x_k - x^*\|} \leq \left(\frac{k}{k+1}\right)^{\frac{1}{p}}$ where $p$ is a constant, by Definition 4 $\{x_k\}$ is sub-linearly convergent. Then $\|x_k - x^*\| \leq \frac{1}{k^{\frac{1}{p}}} \|x_0 - x^*\| = \mathcal{O}(\frac{1}{k^{\frac{1}{p}}})$ also implies sub-linear rate.

**Definition 5** (Linear convergence rate). *Suppose we have a sequence $\{x_n\} \subset \mathbb{R}^d$ such that $x_n \to x^*$ when $n \to \infty$. We say that the convergence is linear if there exists $r \in (0,1)$ such that*

$$\frac{\|x_{k+1} - x^*\|}{\|x_k - x^*\|} \leq r$$

*for all $k$ sufficiently large.*

**Definition 6** (Superlinear convergence rate). *Suppose we have a sequence $\{x_k\} \subset \mathbb{R}^d$ such that $x_k \to x^*$ when $n \to \infty$. We say that the convergence is superlinear if*

$$\lim_{k \to \infty} \frac{\|x_{k+1} - x^*\|}{\|x_k - x^*\|} = 0$$

**Definition 7** (Quadratic convergence rate). *Suppose we have a sequence $\{x_k\} \subset \mathbb{R}^d$ such that $x_k \to x^*$ when $n \to \infty$. We say that the convergence is quadratic if there exists $0 < M < \infty$ such that*

$$\frac{\|x_{k+1} - x^*\|}{\|x_k - x^*\|^2} \leq M$$

*for all $k$ sufficiently large.*

Second-order methods exhibit faster convergence rates in terms of iterations compared to the first-order techniques. Specifically, Newton's method, a prominent second-order optimization algorithm, achieves a quadratic convergence rate. Quasi-Newton techniques attempt to avoid time consuming computations of the Hessian matrix of the Newton's method by approximating this matrix with a second-order term that can however be derived from the first-order information about the function (its value and gradient). Consequently, time-wise they are faster than the Newton's method, but still less efficient that the first-order methods that do not rely on the computations of the second-order term at all. Note however that using low-rank representation of the second-order term and utilizing Sherman–Morrison formula to compute its inverse, one can significantly accelerate the implementation of the Quasi-Newton methods. Iteration-wise, Quasi-Newton methods, including BFGS and L-BFGS, demonstrate a superlinear convergence rate, which is worst than the Newton method, but still better than in the case of the first-order optimization tools.

In contrast, first-order methods such as the gradient descent (GD) method are characterized by a linear convergence rate. Stochastic methods like Stochastic Gradient Descent (SGD), SGD-momentum, and Adam, on the other hand, exhibit a sublinear convergence rate. Instead of computing the exact gradient at each iteration by averaging over the whole data set, stochastic first-order methods compute its noisy approximation with respect to a single data sample or a mini-batch of samples. This implies that they converge more slowly than their full-batch counterparts when considering the number of iterations alone. In practice however they are much faster time-wise. The trade-off between convergence speed and time complexity highlights the nuanced considerations in selecting the most suitable optimization approach for specific applications.

A comprehensive discussion of the convergence rates of different methods will be presented in Sections 2.2 and 2.3.

### 2.1.2 Preliminaries for the Generalization Error

Let $\mathcal{S} = \{\xi_1, \cdots, \xi_n\}$ denotes the set of samples of size $n$ drawn i.i.d from some space $\mathcal{Z}$ with unknown distribution $D$. We assume a learning model parametrized with the parameters $x$. Let $f(x; \xi)$ denote the loss of the model for a specific data example $\xi$.

Our ultimate goal is to minimize the true risk:

$$F(x) := \mathbb{E}_{\xi \sim D} f(x; \xi). \tag{2.1}$$

Since the distribution $D$ is unknown, it is common to replace the objective by the empirical risk:

$$F_{\mathcal{S}}(x) := \frac{1}{n} \sum_{i=1}^{n} f(x; \xi_i). \tag{2.2}$$

We assume $x = A(\mathcal{S})$ for a potentially randomized algorithm $A(\cdot)$. In order to find an upper-bound on the true risk, we consider the generalization error, which is the expected difference of the empirical and the true risk:

$$\epsilon_g := \mathbb{E}_{\mathcal{S},A}[F(A(\mathcal{S})) - F_{\mathcal{S}}(A(\mathcal{S}))] \tag{2.3}$$

In order to find an upper-bound on the generalization error of algorithm A, we consider the uniform stability property.

**Definition 8** ($\epsilon_s$-uniform stability)**.** *Let $\mathcal{S}$ and $\mathcal{S}'$ denote two datasets from the space $\mathcal{Z}^n$ such that $\mathcal{S}$ and $\mathcal{S}'$ differ in at most one example. Algorithm A is $\epsilon_s$-uniformly stable if and only if for all data sets $\mathcal{S}$ and $\mathcal{S}'$ we have*

$$\sup_{\xi} \mathbb{E}[f(A(\mathcal{S});\xi) - f(A(\mathcal{S}');\xi)] \le \epsilon_s. \tag{2.4}$$

**Theorem 9** ((Ramezani-Kebrya et al., 2018))**.** *If A is an $\epsilon_s$-uniformly stable algorith, then the generalization error (2.3) of A is upper-bounded by $\epsilon_s$.*

Above we have established a fundamentals of the theoretical framework that will be used to analyze the algorithms considered in this survey paper. Convergence and generalization properties will be considered through the lenses of the above established definitions.

## 2.2 First-order methods

First-order methods are optimization algorithms that rely on gradient information to find the minimum or maximum of a function. They are widely used in machine learning and optimization due to their simplicity and efficiency. These methods are particularly effective in high-dimensional and large-scale problems, making them fundamental tools for various applications in data analysis and model training. In this section, we discuss Stochstic Gradient Descent (SGD), Stochastic Gradient Descent with Momentum (SGD-Momentum) and adaptive learning rate methods. Hyperparameter settings vary widely for first-order methods, but common choices include using a learning rate of 0.001 for Adam and a learning rate of 0.01 or 0.1 for SGD. Other parameters, like momentum (typically set around 0.9) for SGD, can also significantly affect performance. Recent studies (Robbins & Monro, 1951; Nesterov, 1983; Liu et al., 2020) emphasize the need for careful tuning in practical applications.

### 2.2.1 Stochastic Gradient Decent

Recall the update formula of SGD:

$$x_{k+1} = x_k - \alpha_k g(x_k;\xi_k), \tag{2.5}$$

where $g(x_k;\xi_k)$ is a stochastic gradient (resp. subgradient) of $F(x)$ at $x_k$ depends on a random variable $\xi_k$ such that $\mathbb{E}[g(x_k;\xi_k)] = \nabla F(x_k)$ (resp. $\mathbb{E}[g(x_k;\xi_k)] \in \nabla F(x_k)$). In this section, we are going to prove the sublinear convergence rate of SGD and show the generalization error of SGD.

**Covergence Analysis**   We are going to discuss the convergence rate of SGD for the convex setting first and then our analysis will extend to the non-convex setting.

**Theorem 10** (Convergence of SGD; Convex Setting)**.** *Suppose $F(x)$ is $\mu$-strongly convex, $\mathbb{E}[\|g(x;\xi) - \mathbb{E}[g(x;\xi)]\|] \le \delta^2$ and $\|\nabla F(x)\| \le G$ for any $x$. Let the update*

$$x_{k+1} = x_k - \alpha_k g(x_k;\xi_k) \tag{2.6}$$

*run for K iterations. Set $\alpha_k = \alpha_0/(k+1)$ and $\alpha_0 > \frac{1}{2\mu}$. Then for all $k > 0$*

$$\mathbb{E}[\|x_k - x^*\|_2^2] \le \frac{Q(\alpha_0)}{k+1}, \tag{2.7}$$

*where $Q(\alpha_0) = \max\{\|x_0 - x^*\|_2^2, \frac{(G^2+\delta^2)\alpha_0^2}{2\mu\alpha_0-1}\}$.*

*Proof.*

$$\|x_{k+1} - x^*\|_2^2 = \|x_k - \alpha_k g(x_k; \xi_k) - x^*\|^2 \tag{2.8}$$
$$= \|x_k - x^*\|_2^2 - 2\alpha_k(x_k - x^*)^T g(x_k; \xi_k) + \alpha_k^2 \|g(x_k; \xi_k)\|_2^2.$$

Furthermore,

$$\mathbb{E}[(x_k - x^*)^T g(x_k; \xi_k)] = \mathbb{E}[\mathbb{E}[(x_k - x^*)^T g(x_k; \xi_k)|\xi_1, \cdots, \xi_{k-1}]] \tag{2.9}$$
$$= \mathbb{E}[(x_k - x^*)^T \mathbb{E}[g(x_k; \xi_k)|\xi_1, \cdots, \xi_{k-1}]]$$
$$= \mathbb{E}[(x_k - x^*)^T \nabla F(x_k)].$$

$f$ is $\mu$-strongly convex and $x^*$ is the optimal point, thus

$$(x_k - x^*)^T \nabla F(x_k) = [\nabla F(x_k) - \nabla F(x^*)]^T (x_k - x^*) \geq \mu \|x_k - x^*\|_2^2. \tag{2.10}$$

From formula (2.9) and (2.10), we conclude

$$\mathbb{E}[(x_k - x^*)^T g(x_k; \xi_k)] \geq \mu \|x_k - x^*\|_2^2. \tag{2.11}$$

Compute expectation on the left-hand side and right-hand side of Equation (2.97) to obtain

$$\mathbb{E}[\|x_{k+1} - x^*\|_2^2] \tag{2.12}$$
$$= \mathbb{E}[\|x_k - x^*\|_2^2] - 2\alpha_k \mathbb{E}[(x_k - x^*)^T g(x_k; \xi_k)] + \alpha_k^2 \mathbb{E}[\|g(x_k; \xi_k)\|_2^2]$$
$$\leq \mathbb{E}[\|x_k - x^*\|_2^2] - 2\alpha_k \mathbb{E}[(x_k - x^*)^T g(x_k; \xi_k)] + \alpha_k^2 \left( \mathbb{E}[\|g(x_k; \xi_k) - \mathbb{E}[g(x_k; \xi_k)]\|_2^2] + \mathbb{E}[\|g(x_k; \xi_k)\|_2^2] \right)$$
$$\leq (1 - 2\mu\alpha_k)\mathbb{E}[\|x_k - x^*\|_2^2] + \alpha_k^2(G^2 + \delta^2)$$

We are going to prove the bound with mathematical induction. Define $\phi_t = \mathbb{E}[\|x_k - x^*\|_2^2]$ and $Q(\alpha_0) = \max\{\|x_0 - x^*\|_2^2, \frac{(G^2+\delta^2)\alpha_0^2}{2\mu\alpha_0-1}\}$. When $t = 0$, note that

$$\phi_0 = \mathbb{E}[\|x_0 - x^*\|_2^2] = \|x_0 - x^*\|_2^2 \leq Q(\alpha_0)/1.$$

Assume that $\phi_k \leq \frac{Q(\alpha_0)}{k+1}$, we are going to show $\phi_{k+1} \leq \frac{Q(\alpha_0)}{k+2}$. By Formula (2.12)

$$\phi_{k+1} \leq (1 - 2\mu\alpha_k)\phi_k + \alpha_k^2(G^2 + \delta^2) \tag{2.13}$$
$$= \left(1 - 2\mu\alpha_0 \frac{1}{k+1}\right)\phi_k + \alpha_0^2(G^2 + \delta^2)\left(\frac{1}{k+1}\right)^2.$$

Therefore we can simplify (2.13) as

$$\phi_{k+1} \leq (1 - \frac{2\mu\alpha_0}{k+1})\frac{Q(\alpha_0)}{k+1} + \frac{\alpha_0^2(G^2 + \delta^2)}{(k+1)^2} \tag{2.14}$$
$$= \frac{k+1-2\mu\alpha_0}{(k+1)^2}Q(\alpha_0) + \frac{\alpha_0^2(G^2 + \delta^2)}{(k+1)^2}$$
$$= \frac{t}{(k+1)^2}Q(\alpha_0, k) - \frac{2\mu\alpha_0 - 1}{(k+1)^2}Q(\alpha_0) + \frac{\alpha_0^2(G^2 + \delta^2)}{(k+1)^2}$$
$$\leq \frac{t}{(k+1)^2}Q(\alpha_0) - \frac{2\mu\alpha_0 - 1}{(k+1)^2} \times \frac{(G^2 + \delta^2)\alpha_0^2}{2\mu\alpha_0 - 1} + \frac{\alpha_0^2(G^2 + \delta^2)}{(k+1)^2}$$
$$= \frac{t}{(k+1)^2}Q(\alpha_0)$$
$$\leq \frac{1}{k+2}Q(\alpha_0)$$

By mathematical induction we have $\phi_t \leq \frac{Q(\alpha_0, k)}{k+1}$ for all $k > 0$, which is equivalent to

$$\mathbb{E}[\|x_k - x^*\|_2^2] \leq \frac{Q(\alpha_0)}{k+1}, \tag{2.15}$$

where $Q(\alpha_0) = \max\{\|x_0 - x^*\|_2^2, \frac{(G^2 + \delta^2)\alpha_0^2}{2\mu\alpha_0 - 1}\}$. $\qquad\qquad\qquad\qquad\qquad\qquad\qquad\qquad\square$

**Remark.** According to Theorem 10, SGD has $\mathcal{O}(\frac{1}{K})$ sublinear convergence rate in the convex setting.

**Theorem 11** (Convergence of SGD; Nononvex Setting)**.** *Suppose the objective function $F(x)$ is $L$-smooth, $\mathbb{E}[\|g(x;\xi) - \mathbb{E}[g(x;\xi)]\|] \leq \delta^2$ and $\|\nabla F(x)\| \leq G$ for any $x$. Let the update*

$$x_{k+1} = x_k - \alpha g(x_k; \xi_k) \tag{2.16}$$

*run for $K$ iterations. If $\alpha < \frac{1}{L}$, Then for all $K > 0$*

$$\frac{1}{K} \sum_{k=0}^{K-1} \mathbb{E}[\|\nabla F(x_k)\|^2] \leq \frac{2\mathbb{E}[F(x_0) - F(x_K)]}{\alpha K} + \alpha L \delta^2. \tag{2.17}$$

*When setting the learning rate as $\alpha = \sqrt{\frac{1}{K}}$, we obtain sublinear convergence rate $\mathcal{O}(\sqrt{\frac{1}{K}})$.*

*Proof.* For notation simplicity, we omit $\xi_k$ in stochastic gradient and denote $g(x_k; \xi_k)$ as $g(x_k)$. Since function $F(x)$ is $L$-smooth, we have:

$$
\begin{aligned}
F(x_{k+1}) =& F(x_k - \alpha g(x_k)) \\
\leq& F(x_k) - \alpha \langle \nabla F(x_k), g(x_k) \rangle + \frac{\alpha^2 L}{2} \|g(x_k)\|^2 \\
\leq& F(x_k) - \alpha \langle \nabla F(x_k), g(x_k) \rangle + \frac{\alpha^2 L}{2} \|\nabla F(x_k)\|^2 + \frac{\alpha^2 L}{2} \|g(x_k) - \nabla F(x_k)\|^2
\end{aligned} \tag{2.18}
$$

Take the expectation on both sides to obtain:

$$\mathbb{E}[F(x_{k+1})] \leq F(x_k) - \alpha(1 - \frac{\alpha L}{2})\mathbb{E}[\|\nabla F(x_k)\|^2] + \frac{\alpha^2 L \delta^2}{2}. \tag{2.19}$$

Since $\alpha < \frac{1}{L}$, we have:

$$\mathbb{E}[F(x_{k+1}) - F(x_k)] \leq -\frac{1}{2}\mathbb{E}[\|\nabla F(x_k)\|^2] + \frac{\alpha^2 L \delta^2}{2}. \tag{2.20}$$

Summing $k = 0, ..., K - 1$ and divide over $K$, we have

$$\frac{1}{K} \sum_{k=1}^{K-1} \mathbb{E}[\|\nabla F(x_k)\|^2] \leq \frac{2\mathbb{E}[F(x_0) - F(x_K)]}{\alpha K} + \alpha L \delta^2. \tag{2.21}$$

$$\square$$

**Remark.** According to Theorem 11, SGD is of characterized by the $\mathcal{O}(\frac{1}{\sqrt{K}})$ sublinear convergence rate in the nonconvex setting.

**Generalization Ability** We are next going to discuss the generalization error of SGD under first the convex and then nonconvex assumptions. We organize the theorems and proofs in (Hardt et al., 2016) in this section.

**Theorem 12** (Generalization Guarantee of SGD; Convex Setting)**.** *(Theorem 3.7 in (Hardt et al., 2016))*
*Assume that $f(\cdot;\xi) \in [0,1]$ is a convex, M-Lipschitz and L-smooth loss function for every example $\xi$. Suppose*
*that we run SGD for K steps with step size $\alpha_k \leq 2/L$. Then SGD has uniform stability with*

$$\epsilon_s \leq \frac{2M^2}{n} \sum_{k=1}^{K} \alpha_k. \tag{2.22}$$

*Proof.* Let $S$ and $S'$ be two samples of size n that differ on at most one example. Consider the sequence of
gradient updates $g_1, ..., g_K$ and $g'_1, ..., g'_K$ induced by running SGD on sample set $S$ and $S'$, repectively. Let
$x_k$ and $x'_k$ denote corresponding outputs of SGD.

We now fix an example $\xi$ and apply the Lipschitz condition on $f(\cdot;\xi)$ to get

$$\mathbb{E}|f(x_k;\xi) - f(x'_k;\xi)| \leq M\mathbb{E}[\delta_K],$$

where $\delta_k = \|x_k - x'_k\|$. Observe that at step $t$; with probability $1 - \frac{1}{n}$ the example selected by SGD is the same
in both $S$ and $S'$, where $g_k = g'_k$. Otherwise, with probability $\frac{1}{n}$ the selected example is different. By the
property that $f$ is L-lipschitz continuous we have

$$\mathbb{E}[\delta_{k+1}] \leq (1 - \frac{1}{n})\mathbb{E}[\delta_k] + \frac{1}{n}\mathbb{E}[\delta_k] + \frac{2\alpha_k M}{n} = \mathbb{E}[\delta_k] + \frac{2\alpha_k M}{n}.$$

Therefore,

$$\mathbb{E}|f(x_K;\xi) - f(x'_K;\xi)| \leq M\mathbb{E}[\delta_K] \leq \frac{2M^2}{n} \sum_{k=1}^{K} \alpha_k.$$

$\square$

**Theorem 13** (Generalization Guarantee of SGD; Nonconvex Setting)**.** *(Theorem 3.8 in (Hardt et al.,*
*2016))Assume that $f(\cdot;\xi) \in [0,1]$ is a M-Lipschitz and L-smooth loss function for every example $\xi$. Suppose*
*that we run SGD for K steps with monotonically non-increasing step size $\alpha_k \leq c/k$. Then SGD has uniform*
*stability with*

$$\epsilon_s \leq \frac{1 + 1/Lc}{n-1}(2cM^2)^{\frac{1}{Lc+1}} K^{\frac{Lc}{Lc+1}}. \tag{2.23}$$

*Proof.* Let $S$ and $S'$ be two samples of size n that differ on at most one example. Consider the gradient
update $g_1, ..., g_K$ and $g'_1, ..., g'_K$ induced by running SGD on sample set $S$ and $S'$, repectively. Let $x_K$ and $x'_K$
denote corresponding outputs of SGD.

$$\mathbb{E}|f(x_K;\xi) - f(x'_K;\xi)| \leq \frac{k_0}{n} + M\mathbb{E}[\delta_K|\delta_{k_0} = 0], \tag{2.24}$$

where $\delta_K = \|x_K - x'_K\|$. To simplify the notation, let $\Delta_k = \mathbb{E}[\delta_k|\delta_{k_0} = 0]$. We will bound $\Delta_k$ as function of $k_0$
and then minimize for $k_0$.

Observe that at step $k$; with probability $1 - \frac{1}{n}$ the example selected by SGD is the same in both $S$ and $S'$,
where $g_t = g'_t$. Otherwise, with probability $\frac{1}{n}$ the selected example is different. By the property that $f$ is
L-lipschitz continuous we have

$$\begin{aligned}
\Delta_{k+1} &\leq (1 - \frac{1}{n})(1 + \alpha_k L)\Delta_k + \frac{1}{n}\Delta_k + \frac{2\alpha_k M}{n} \\
&\leq \left(\frac{1}{n} + (1 - \frac{1}{n})(1 + cL/k)\right)\Delta_k + \frac{2cM}{kn} \\
&= \left(1 + (1 - \frac{1}{n})cL/k\right)\Delta_k + \frac{2cM}{kn} \\
&\leq \exp\left((1 - \frac{1}{n})\frac{cL}{k}\right)\Delta_k + \frac{2cM}{kn},
\end{aligned}$$

Therefore,

$$\Delta_K \le \sum_{k=k_0+1}^{K} \left\{ \prod_{m=k+1}^{T} \exp\left( (1-\frac{1}{n})\frac{cL}{m} \right) \right\} \frac{2cM}{kn}$$

$$= \sum_{k=k_0+1}^{K} \exp\left( (1-\frac{1}{n})cL \sum_{m=k+1}^{T} \frac{1}{m} \right) \frac{2cM}{kn}$$

$$\le \sum_{k=k_0+1}^{K} \exp\left( (1-\frac{1}{n})cL \log(\frac{K}{k}) \right) \frac{2cM}{kn}$$

$$= \frac{2cM}{n} K^{Lc(1-1/n)} \sum_{k=k_0+1}^{K} k^{-Lc(1-1/n)-1}$$

$$\le \frac{1}{(1-1/n)Lc} \frac{2cM}{n} \left( \frac{K}{k_0} \right)^{Lc(1-1/n)}$$

$$\le \frac{2M}{L(n-1)} \left( \frac{K}{k_0} \right)^{Lc}$$

Plugging the pobtained Inequlity into the Equation (2.24), we have

$$\mathbb{E}|f(x_K;\xi) - f(x_K';\xi)| \le \frac{k_0}{n} + \frac{2M^2}{L(n-1)} \left( \frac{K}{k_0} \right)^{Lc}.$$

The right-hand side is approximately minimized by $k_0 = (2cM^2)^{\frac{1}{Lc+1}} T^{\frac{Lc}{Lc+1}}$, and then we have

$$\mathbb{E}|f(x_K;\xi) - f(x_K';\xi)| \le \frac{1+1/Lc}{n-1} (2cM^2)^{\frac{1}{Lc+1}} K^{\frac{Lc}{Lc+1}}.$$

$\square$

### 2.2.2 Stochastic Gradient Descent with Momentum

SGD faces difficulties when moving through valleys in the optimization landscape (Ruder, 2016), which are the areas where the curvature sharply inclines in one direction, often near the local optimal points. In these scenarios, SGD tends to oscillate along the steep slopes of the valley, resulting in slow progress towards the local optimum at the bottom, as depicted in Figure (2a).

The introduction of the momentum term successfully addresses this issue by accelerating SGD in the pertinent direction and mitigating oscillations, as depicted in Figure (2b). Momentum method follows the gradient direction augmented by the term that corresponds to the fraction of the direction from the previous time step.

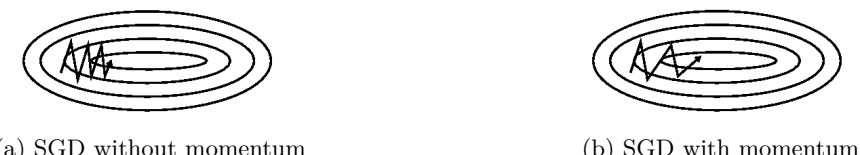

(a) SGD without momentum          (b) SGD with momentum

Figure 2: Figures for SGD with and without momentum. This Figure is taken from Ruder (2016) (Figure 2).

In this section, we consider the two most popular types of momentum methods: Heavy-Ball (HB) method and Nestrov accelerated gradient (NAG) method. The difference between these methods lies in how the momentum term $\beta m_k$ is used to move the parameter $x_k$ when computing the gradient. In this paper, we only consider the stochastic versions of Momentum methods, stochastic Heavey-Ball (SHB) and stochastic Nestrov accelerated gradient (SNAG) As depicted in (Yang et al., 2016), the update rules for SHB and SNAG can be represented as:

$$\text{SHB:} \begin{cases} m_{k+1} = \beta m_k + \alpha \nabla g(x_k;\xi_k) \\ x_{k+1} = x_k - m_{k+1}, \end{cases} \qquad \text{SNAG:} \begin{cases} m_{k+1} = \beta m_k + \alpha \nabla g(x_k - \beta m_k;\xi_k) \\ x_{k+1} = x_k - m_{k+1}, \end{cases}$$

which can be rewritten as

$$\text{SHB:} \quad x_{k+1} = x_k - \alpha g(x_k; \xi_k) + \beta(x_k - x_{k-1}) \tag{2.25}$$

$$\text{SNAG:} \quad \begin{cases} y_{k+1} = x_k - \alpha g(x_k) \\ x_{k+1} = y_{k+1} + \beta(y_{k+1} - y_k). \end{cases} \tag{2.26}$$

Both formulas can be unified in the form of the stochastic unified momentum (SUM), as suggested by (Yang et al., 2016), as follows:

$$\text{SUM}: \quad \begin{cases} y_{k+1} = x_k - \alpha g(x_k) \\ y_{k+1}^s = x_k - s\alpha g(x_k) \\ x_{k+1} = y_{k+1} + \beta(y_{k+1}^s - y_k^s). \end{cases} \tag{2.27}$$

Note that when $s = 0$ SUM method becomes SHB and when $s = 1$ it reduces to SNAG.

**Covergence Analysis**   As was done for SGD, the convergence analysis of stochastic Momentum methods will first consider convex assumptions, and will then be generalized to the nonconvex setting.

**Assumption 14** (Convex Setting). *The following properties holds for objective function $F$:*

- *$F(x)$ is convex function.*

- *(Variance bounded) $\mathbb{E}\left[\left\|g(x;\xi) - \mathbb{E}[g(x;\xi)]^2\right\|\right] \le \delta^2$.*

- *(Subgradient Bounded) $\|\nabla F(x)\| \le G$.*

**Theorem 15** (Convergence of Unified Momentum; Convex Setting). *(Theorem 1 in (Yang et al., 2016)) Suppose function $F$ satisfies Assumption 14. Let the update of the UM method be run for $K$ iterations with stochastic gradients $g(x_k; \xi_k)$. By setting $\alpha = \frac{C}{\sqrt{K+1}}$ one can obtain*

$$\mathbb{E}[F(\hat{x}_K) - F(x^*)] \le \frac{\beta(F(\hat{x}_0) - F(x^*))}{(1-\beta)(K+1)} + \frac{(1-\beta)\|x_0 - x^*\|^2}{2C\sqrt{K+1}} + \frac{C(1+2s\beta)(G^2 + \delta^2)}{2(1-\beta)\sqrt{K+1}}, \tag{2.28}$$

*where $C$ is a positive constant, $\hat{x}_K = \sum_{k=0}^{K} x_k/(K+1)$, and $x^*$ is optimal solution.*

*Proof.* For notation simplicity, we denote $g(x_k; \xi_k) = g(x_k) = g_k$. The update Formula (2.27) implies the following recursions:

$$x_{k+1} + p_{k+1} = x_k + p_k - \frac{\alpha}{1-\beta} g(x_k) \tag{2.29}$$

$$v_{k+1} = \beta v_k + ((1-\beta)s - 1)\alpha g(x_k), \tag{2.30}$$

where $v_k = \frac{1-\beta}{\beta} p_k$ and $p_k$ is given by

$$p_k = \begin{cases} \dfrac{\beta}{1-\beta}(x_k - x_{k-1} + s\alpha g(x_{k-1})), & k \ge 1 \\ 0, & k = 0. \end{cases} \tag{2.31}$$

Define $\delta_k = g_k - \nabla F(x_k)$ and recall that $x^*$ is the optimal point. From the above recursions we have

$$\|x_{k+1} + p_{k+1} - x^*\|^2 \tag{2.32}$$

$$= \|x_k + p_k - x^*\|^2 - \frac{2\alpha}{1-\beta}(x_k + p_k - x^*)^T g_k + \left(\frac{\alpha}{1-\beta}\right)^2 \|g_k\|^2$$

$$= \|x_k + p_k - x^*\|^2 - \frac{2\alpha}{1-\beta}(x_k - x^*)^T g_k - \frac{2\alpha\beta}{(1-\beta)^2}(x_k - x_{k-1})^T g_k$$

$$- \frac{2s\alpha^2\beta}{(1-\beta)^2} g_{k-1}^T g_k + \left(\frac{\alpha}{1-\beta}\right)^2 \|g_k\|^2$$

$$= \|x_k + p_k - x^*\|^2 - \frac{2\alpha}{1-\beta}(x_k - x^*)^T(\delta_k + \nabla F(x_k)) - \frac{2\alpha\beta}{(1-\beta)^2}(x_k - x_{k-1})^{(}\delta_k + \nabla F(x_k))$$

$$- \frac{2s\alpha^2\beta}{(1-\beta)^2}(\delta_{k-1} + \nabla F(x_{k-1}))^T(\delta_k + \nabla F(x_k)) + \left(\frac{\alpha}{1-\beta}\right)^2 \|\delta_k + \nabla F(x_k)\|^2. \tag{2.33}$$

Note that

$$\mathbb{E}[(x_k - x^*)^T(\delta_k + \nabla F(x_k))] = \mathbb{E}[(x_k - x^*)^T \nabla F(x_k)]$$

$$\mathbb{E}[(x_k - x_{k-1})^T(\delta_k + \nabla F(x_k))] = \mathbb{E}[(x_k - x_{k-1})^\nabla F(x_k)]$$

$$\mathbb{E}[(\delta_{k-1} + \nabla F(x_{k-1}))^T(\delta_k + \nabla F(x_k))] = \mathbb{E}[(\delta_{k-1} + \nabla F(x_{k-1}))^T \nabla F(x_k)] = \mathbb{E}[g_{k-1}^T \nabla F(x_k)]$$

$$\mathbb{E}[\|\delta_k + \nabla F(x_k)\|^2] = \mathbb{E}[\|\delta_k\|^2] + \mathbb{E}[\|\nabla F(x_k)\|^2].$$

Taking expectation on both sides gives

$$\mathbb{E}[\|x_{k+1} + p_{k+1} - x^*\|^2]$$

$$= \mathbb{E}[\|x_k + p_k - x^*\|^2] - \frac{2\alpha}{1-\beta}\mathbb{E}[(x_k - x^*)^T \nabla F(x_k)] - \frac{2\alpha\beta}{(1-\beta)^2}\mathbb{E}[(x_k - x_{k-1})^T \nabla F(x_k)]$$

$$- \frac{2s\alpha^2\beta}{(1-\beta)^2}\mathbb{E}[g_{k-1}^T \nabla F(x_k)] + \left(\frac{\alpha}{1-\beta}\right)^2 (\mathbb{E}[\|\delta_k\|^2] + \mathbb{E}[\|\nabla F(x_k)\|^2]). \tag{2.34}$$

Moreover, since F is convex and $\mathbb{E}[\|g(x;\xi) - \mathbb{E}[g(x;\xi)]\|] \le \delta^2$ and $\|\nabla F(x)\| \le G$ for any $x$, we have that

$$F(x_k) - F(x^*) \le (x_k - x^*)^T \nabla F(x_k)$$

$$F(x_k) - F(x_{k-1}) \le (x_k - x_{k-1})^T \nabla F(x_k)$$

$$- \mathbb{E}[g_{k-1}^T \nabla F(x_k)] \le \frac{\mathbb{E}[\|g_{k-1}\|^2 + \|\nabla F(x_k)\|^2]}{2} \le \delta^2/2 + G^2 \le \delta^2 + G^2$$

$$\mathbb{E}[\|\delta_k\|^2] \le \delta^2, \quad \mathbb{E}[\|\nabla F(x_k)\|^2] \le G^2.$$

Therefore, (2.34) can be rewritten as

$$\mathbb{E}[\|x_{k+1} + p_{k+1} - x^*\|^2] \le \mathbb{E}[\|x_k + p_t - x^*\|^2] - \frac{2\alpha}{1-\beta}\mathbb{E}[F(x_k) - F(x^*)] \tag{2.35}$$

$$- \frac{2\alpha\beta}{(1-\beta)^2}\mathbb{E}[F(x_k) - F(x_{k-1})] + \left(\frac{\alpha}{1-\beta}\right)^2 (2s\beta + 1)(G^2 + \delta^2)$$

Let $x_{-1} = x_0$, then Inequality (2.35) also holds for $k = 0$. In all, (2.35) holds for all $k \ge 0$. Summing the Inequality (2.35) for $k = 0, \cdots, K$, we obtain

$$\frac{2\alpha}{1-\beta}\sum_{k=0}^{K}\mathbb{E}[F(x_k) - F(x^*)]$$

$$\le \frac{2\alpha\beta}{(1-\beta)^2}(F(x_0) - F(x_K)) + \|x_0 - x^*\|^2 + \left(\frac{\alpha}{1-\beta}\right)^2 (2s\beta + 1)(G^2 + \delta^2). \tag{2.36}$$

Define $\hat{x}_K = \sum_{k=0}^{K} x_k$, by convexity of $L$ we have $\sum_{k=0}^{K} \mathbb{E}[F(x_k)] \le \mathbb{E}[F(\hat{x}_K)]$. Therefore

$$\mathbb{E}[F(\hat{x}_K) - F(x^*)] \le \frac{\beta(F(x_0) - F(x^*))}{(1-\beta)(K+1)} + \frac{(1-\beta)\|x_0 - x^*\|^2}{2\alpha(K+1)} + \frac{\alpha(2s\beta + 1)(G^2 + \delta^2)}{2(1-\beta)}. \tag{2.37}$$

Since $\alpha = \frac{C}{\sqrt{K+1}}$,

$$\mathbb{E}[F(\hat{x}_K) - F(x^*)] \le \frac{\beta(F(\hat{x}_0) - F(x^*))}{(1-\beta)(K+1)} + \frac{(1-\beta)\|x_0 - x^*\|^2}{2C\sqrt{K+1}} + \frac{C(1+2s\beta)(G^2 + \delta^2)}{2(1-\beta)\sqrt{K+1}}. \tag{2.38}$$

$\square$

**Remark.** The convergence rate for UM methods is $\mathcal{O}(1/\sqrt{K})$. This is a sublinear convergence rate.

**Assumption 16** (Nonconvex Setting). *The following properties holds for objective function $F$:*

- *(Lipschitz gradient) Loss function $F(x)$ is L-smooth.*

$$\|\nabla F(x) - \nabla F(y)\| \le L\|x - y\|.$$

- *(Variance bounded) $\mathbb{E}\left[\left\|g(x;\xi) - \mathbb{E}[g(x;\xi)]^2\right\|\right] \le \delta^2$.*

- *(Gradient Bound) $\|\nabla F(x)\| \le G$.*

**Theorem 17** (Convergence of SUM; Nonconvex Setting). *(Theorem 3 in (Yang et al., 2016)) Suppose $L$ satisfies Assumption 16. Let the UM update run for $K$ iterations with stochastic gradients $g(x_k; \xi_k)$. By setting $\alpha = \min\left\{\frac{1-\beta}{2L}, \frac{C}{\sqrt{K}}\right\}$ we have that*

$$\frac{1}{K}\sum_{k=0}^{K-1} \mathbb{E}[\|\nabla F(x_k)\|^2] \le \frac{2(F(x_0) - L^*)(1-\beta)}{K} \max\left\{\frac{2L}{1-\beta}, \frac{\sqrt{K}}{C}\right\}$$
$$+ \frac{C}{\sqrt{K}} \frac{L\beta^2((1-\beta)s - 1)^2(G^2 + \delta^2) + L\delta^2(1-\beta)^2}{(1-\beta)^3}, \tag{2.39}$$

*where $L^*$ is the minimal value of the loss function.*

*Proof.* The following two useful lemmas will be needed to conduct the proof. Proofs for Lemma 18 and Lemma 19 could be found in the Appendix of (Yang et al., 2016).

**Lemma 18.** *Let $z_k = x_k + p_k$. For SUM, we have that for any $k \ge 0$:*

$$\mathbb{E}[F(z_{k+1}) - F(z_k)] \le \frac{1}{2L}\mathbb{E}[\|\nabla F(z_{k+1} - \nabla F(z_k)\|^2]$$
$$+ \left(\frac{L\alpha^2}{(1-\beta)^2} - \frac{\alpha}{\beta}\right)\mathbb{E}[\|\nabla F(x_k)\|^2] + \frac{L^2\alpha^2}{2(1-\beta)^2}\sigma^2.$$

**Lemma 19.** *For SUM, we have that for any $k \ge 0$,*

$$\mathbb{E}[\|\nabla F(z_k) - \nabla F(x_k)\|^2] \le \frac{L^2\beta^2((1-\beta)s - 1)^2\alpha^2(G^2 + \sigma^2)}{(1-\beta)^4}$$

Then continue the proof for Theorem 17. Let $B$ and $B'$ be defined as

$$B = \frac{\alpha}{1-\beta} - \frac{L\alpha^2}{(1-\beta)^2},$$

$$B' = \frac{L\beta^2((1-\beta)s - 1)^2\alpha^2(G^2 + \sigma^2)}{2(1-\beta)^4} + \frac{L\alpha^2\sigma^2}{2(1-\beta)^2}.$$

Lemma 18 and 19 imply that

$$\mathbb{E}[F(z_{k+1}) - F(z_k)] \le -B\mathbb{E}[\|\nabla F(x_k)\|^2] + B'.$$

By summing the above inequalities for $k = 0, ..., K$ and noting that $\alpha < \frac{1-\beta}{L}$, one can obtain

$$B \sum_{k=0}^{K-1} \mathbb{E}[\|\nabla F(x_k)\|^2] \le \mathbb{E}[F(x_0) - F(x_K)] + KB' \le \mathbb{E}[F(x_0) - F(x^*)] + KB'.$$

Then, rearranging terms yields

$$\frac{1}{K} \sum_{k=0}^{K-1} \mathbb{E}[\|\nabla F(x_k)\|^2] \le \frac{F(x_0) - F(x^*)}{BK} + \frac{B'}{B}.$$

Assuming $\alpha \le \frac{1-\beta}{2L}$ gives

$$B = \frac{\alpha}{1-\beta} - \frac{L\alpha^2}{(1-\beta)^2} = \frac{\alpha}{1-\beta}\left(1 - \frac{\alpha L}{1-\beta}\right) \ge \frac{\alpha}{2(1-\beta)}.$$

Thus:

$$\frac{1}{K} \sum_{k=0}^{K-1} \mathbb{E}[\|\nabla F(x_k)\|^2] \le \frac{2(F(z_0) - L^*)(1-\beta)}{\alpha K} + \frac{2(1-\beta)}{\alpha}B'.$$

Assume $\alpha = \min\{\frac{1-\beta}{2L}, \frac{C}{\sqrt{K}}\}$ and let $z_0 = x_0$. We obtain:

$$\frac{1}{K} \sum_{k=0}^{K-1} \mathbb{E}[\|\nabla F(x_k)\|^2] \le \frac{2(F(x_0) - F(x^*))(1-\beta)}{K} \max\{\frac{2L}{1-\beta}, \frac{\sqrt{K}}{C}\}$$
$$+ \frac{C}{\sqrt{K}} \frac{L\beta^2((1-\beta)s-1)^2(G^2+\delta^2) + L\delta^2(1-\beta)^2}{(1-\beta)^3}. \tag{2.40}$$

$\square$

**Remark.** We could draw the following two conclusions from Theorem 17: (i) Convergence rate $\mathcal{O}(1/\sqrt{K})$ implies stochastic momentum methods maintains the sublinear convergence rate even if objective function $f$ is non-convex.(ii) SHB and SNAG have different variants regarding the term $L\beta^2((1-\beta)s-1)^2(G^2+\delta^2)$ in the convergence bound of SUM because of the different choices of parameter $s$ for the update rule (2.27) of SUM: $L\beta^2$ for SHB ($s = 0$), $L\beta^4$ for SNAG ($s = 1$) and 0 for SGD ($s = 1/(1-\beta)$).

**Generalization guarantees.** For the generalization guarantees of SGD with momentum methods, we focus on the Stochastic Heavy-Ball (SHB) method since we only find generalization guarantee for SHB in the literature. Recall the SHB update rule:

$$x_{k+1} = x_k - \alpha g(x_k; \xi_k) + \beta(x_k - x_{k-1}), \tag{2.41}$$

where $\alpha$ is the learning rate and $\beta$ is the momentum value. We are going to show the generalization ability of HB method first under convex and then under nonconvex setting.

**Assumption 20** (Convex Setting)**.** *The following properties holds for the objective function $f$:*

- *$f(x; \xi)$ is $\mu$-strongly convex.*

- *$L$-smooth: $|\nabla f(x; \xi) - \nabla f(x; \xi)| \le L\|x - y\|$.*

- *$M$-Lipchitz continuous: $\|\nabla f(x; \xi) - \nabla f(y; \xi)\| \le M\|x - y\|$.*

**Theorem 21** (Generalization Guarantee of SHB; Convex Setting). *(Theorem 18 in (Ramezani-Kebrya et al., 2021)) Let the loss function $f(x;\xi)$ satisfies the Assumption 20. Suppose that the HB momentum methods is run for $K$ steps with constant learning rate $\alpha$ and momentum $\beta$. Provided that $\frac{\alpha L\mu}{L+\mu} - \frac{1}{2} \le \beta < \frac{\alpha L\mu}{3(L+\mu)}$ and $\alpha \le \frac{2}{L+\mu}$, the HB momentum method satisfies $\epsilon_s$-uniformly stability with*

$$\epsilon_s \le \frac{2\alpha M^2(L+\mu)}{n(\alpha L\mu - 3\beta(L+\mu))}. \tag{2.42}$$

*Proof.* Let $S$ and $S'$ be two samples of size n with at most one example difference. Let $x_k$ and $x'_k$ denote corresponding outputs of SGD momentum on $S$ and $S'$. We consider the update $x_{k+1} = \mathcal{G}_k(x_k) + \beta(x_k - x_{k-1})$ and $x'_{k+1} = \mathcal{G}'_k(x'_k) + \beta_k(x'_k - x_{k'-1})$, where $\mathcal{G}_k(x_k) := x_k - \alpha g(x_k;\xi_k)$ and $\mathcal{G}'_k(x'_k) := x'_k - \alpha g(x'_k;\xi'_k)$ We denote $\delta_k = \|x_k - x'_k\|$. Suppose $x_0 = x'_0$ and $\delta_0 = 0$.

With probability $1 - \frac{1}{n}$, the seleted examples in $S$ and $S'$ are the same. Because $\mathcal{G}_k$ is $(1 - \frac{\alpha L\mu}{L+\mu})$-expansive for $\le \frac{2}{L+\mu}$, we have

$$
\begin{aligned}
\delta_{k+1} &= \|\beta(x_k - x'_k) - \beta(x_{k-1} - x'_{k-1}) + \mathcal{G}_k(x_k) - \mathcal{G}'_k(x'_k)\| \\
&\le \beta \|x_k - x'_k\| + \beta \|x_{k-1} - x'_{k-1}\| + \|\mathcal{G}_k(x_k) - \mathcal{G}'_k(x'_k)\| \\
&\le (1 + \beta - \frac{\alpha L\mu}{L+\mu})\delta_k + \beta\delta_{k-1}.
\end{aligned}
$$

With probability $\frac{1}{n}$, the selected examples in $S$ and $S'$ are different. Hence, we have

$$
\begin{aligned}
\delta_{k+1} &= \|\beta(x_k - x'_k) - \beta(x_{k-1} - x'_{k-1}) + \mathcal{G}_k(x_k) - \mathcal{G}'_k(x'_k)\| \\
&\le \beta \|x_k - x'_k\| + \beta \|x_{k-1} - x'_{k-1}\| + \|\mathcal{G}_k(x_k) - \mathcal{G}'_k(x'_k)\| + \|\mathcal{G}_k(x'_k) - \mathcal{G}'_t(x'_k)\| \\
&\le (1 + \beta - \frac{\alpha L\mu}{L+\mu})\delta_k + \beta\delta_{k-1} + \|\mathcal{G}'_k(x'_k) - \mathcal{G}'_t(x'_k)\| \\
&\le (1 + \beta - \frac{\alpha L\mu}{L+\mu})\delta_k + \beta\delta_{k-1} + \|x'_k - \mathcal{G}_k(x'_t)\| + \|x'_k - \mathcal{G}'_k(x'_k)\| \\
&\le (1 + \beta - \frac{\alpha L\mu}{L+\mu})\delta_k + \beta\delta_{k-1} + 2\alpha M,
\end{aligned}
$$

Therefore

$$
\begin{aligned}
\mathbb{E}[\delta_{k+1}] &\le (1 - \frac{1}{n})\left((1 + \beta - \frac{\alpha L\mu}{L+\mu})\mathbb{E}[\delta_k] + \beta\mathbb{E}[\delta_{t-1}]\right) \\
&\quad + \frac{1}{n}\left((1 + \beta - \frac{\alpha L\mu}{L+\mu})\mathbb{E}[\delta_k] + \beta\mathbb{E}[\delta_{k-1}] + 2\alpha M\right) \\
&= (1 + \beta - \frac{\alpha L\mu}{L+\mu})\mathbb{E}[\delta_k] + \beta\mathbb{E}[\delta_{k-1}] + \frac{2\alpha M}{n}
\end{aligned}
$$

Let us consider the recursion

$$\mathbb{E}[\widetilde{\delta}_{t+1}] = (1 + \beta - \frac{\alpha L\mu}{L+\mu})\mathbb{E}[\widetilde{\delta}_k] + \beta\mathbb{E}[\widetilde{\delta}_{k-1}] + \frac{2\alpha M}{n} \tag{2.43}$$

Formular (2.43) implies

$$\mathbb{E}[\delta_{k+1}] \ge (1 + \beta - \frac{\alpha L\mu}{L+\mu})\mathbb{E}[\widetilde{\delta}_k],$$

hence we have

$$
\begin{aligned}
\mathbb{E}[\widetilde{\delta}_{t+1}] &\le (1 + \beta + \frac{\beta}{1 + \beta - \frac{\alpha L\mu}{L+\mu}})\mathbb{E}[\widetilde{\delta}_k] + \frac{2\alpha M}{n} \\
&\le (1 + 3\beta - \frac{\alpha L\mu}{L+\mu})\mathbb{E}[\widetilde{\delta}_k] + \frac{2\alpha M}{n}.
\end{aligned}
$$

Since $\mathbb{E}[\widetilde{\delta}_{t+1}] \geq \mathbb{E}[\delta_{k+1}]$ for all $k$, we have

$$\mathbb{E}[\delta_K] \leq \frac{2\alpha M}{n} \sum_{k=1}^{K} \left(1 + 3\beta + \frac{\alpha L\mu}{L+\mu}\right)^k \leq \frac{2\alpha M(L+\mu)}{n(\alpha L\mu - 3\beta(L+\mu))}.$$

$\square$

**Remark.** Proof for Theorem 21 (Generalization guarantee of SHB) resembles the proof of Theorem 12 (Generalization guarantee of SGD). From Theorem 21, we could draw the following two conclusions: i) when $n$ increases, $\epsilon_s$ decreases and algorithm $A$ becomes more stable. It consists with the common sense. When we includes more samples in the training data set, the generalizaton error will be smaller, the algorithm will be more stable. When ii) When $\beta$ decreases, $\epsilon_s$ decreases and algorithm $A$ becomes more stable The smaller momentum parameter $\beta$ leads to smaller generalization error. When $\beta = 0$, the SHB method becomes the vanilla SGD method. Therefore, SGD generalizes better than SHB. It also coincides with the common sense. Although the momentum term have the potential to accelerate SGD, collecting the gradient information from the previous steps introduces the bias to the estimation of the update direction and may cause algorithm's instability.

**Assumption 22** (Nonconvex Setting). *Assume that the following properties hold for the objective function $f$:*

- *L-smooth: $|\nabla f(x;\xi) - \nabla f(x;\xi)| \leq L \|x - y\|$.*

- *M-Lipchitz continuous: $\|\nabla f(x;\xi) - \nabla f(y;\xi)\| \leq M \|x - y\|$.*

**Theorem 23** (Generalization Guarantee of SHB; Nonconvex Setting). *(Theorem 10 in (Ramezani-Kebrya et al., 2021)) Let the loss function $f(x;\xi)$ satisfy Assumption 22. Suppose that the HB momentum method is executed for $K$ steps with the learning rate $\alpha_k = \alpha_0/k$ and some constant momentum $\beta_d \in (0,1]$ in the first $k_d$ steps. Then, for any $1 \leq \widetilde{k} \leq k_d \leq K$, the momentum method satisfies $\epsilon_s$ stability with:*

$$\epsilon_s \leq \frac{2\alpha_0 M}{n} K^{(1-\frac{1}{n})\alpha_0 L} \widetilde{h}(\beta_d, k_d) + \frac{\widetilde{k}B}{n} + \frac{2M}{L(n-1)} \left(\frac{K}{\widetilde{k}}\right)^{(1-\frac{1}{n})\alpha_0 L}, \tag{2.44}$$

*where $\widetilde{h}(\beta_d, k_d) = \exp(2\beta_d(k_d - \widetilde{k}))(\ln(1 + \frac{1}{2\beta_d\widetilde{k}}) - \frac{1}{2}\ln(1 + \frac{2}{\beta_d k_d}))$ and $B = \sup f(x;\xi)$.*

*Proof.* Let $S$ and $S'$ be two samples of size n with at most one example difference. Let $x_k$ and $x'_k$ denote corresponding outputs of SGD momentum on $S$ and $S'$. We consider the update $x_{k+1} = \mathcal{G}_k(x_k) + \beta_k(x_k - x_{k-1})$ and $x'_{k+1} = \mathcal{G}'_k(x'_k) + \beta_k(x'_k - x'_{k'-1})$, where $\beta_k := \beta_d \mathbb{I}(k \leq k_d)$, $\mathcal{G}_k(x_k) := x_k - \alpha_k g(x_k;\xi_k)$ and $\mathcal{G}'_k(x'_k) := x'_k - \alpha_k g(x'_k;\xi'_k)$ We denote $\delta_k = \|x_k - x'_k\|$. Suppose $x_0 = x'_0$ and $\delta_0 = 0$.

First, as a preliminary step, we observe that the expected loss difference under $x_K$ and $x'_K$ for every x and every $\widetilde{k} \in \{1, ..., K\}$ is bounded by:

$$\mathbb{E}[f(x_K;\xi) - f(x'_K;\xi)] \leq \frac{\widetilde{k}B}{n} + L\mathbb{E}[\delta_K|\delta_{\widetilde{k}} = 0]. \tag{2.45}$$

where $B = \sup f(x;)$.This follows from (Hardt et al., 2016). Not let define $\Delta_{k,\widetilde{k}} := \mathbb{E}[\delta_k|\delta_{\widetilde{k}} = 0]$. Our goal is to find a bound on $\Delta_{K,\widetilde{k}}$ and then minimize over $\widetilde{k}$.

At step $k$, with probability $1 - \frac{1}{n}$, the seleted examples in $S$ and $S'$ are the same.

$$\begin{aligned}
\delta_{k+1} &= \|(\beta_k(x_k - x'_k) - \beta_k(x_{k-1} - x'_{k-1}) + \mathcal{G}_k(x_k) - \mathcal{G}'_k(x'_k)\| \\
&\leq \beta_k \|x_k - x'_k\| + \beta_k \|x_{k-1} - x'_{k-1}\| + \|\mathcal{G}_k(x_k) - \mathcal{G}'_k(x'_k)\| \\
&\leq (1 + \beta_k - \alpha L\alpha_k L)\delta_k + \beta_k\delta_{k-1}.
\end{aligned}$$

The last inequality holds by $l$ is L-smooth. With probability $\frac{1}{n}$, the selected examples in $S$ and $S'$ are different. Hence, we have

$$
\begin{aligned}
\delta_{k+1} &= \|\beta_k(x_k - x'_k) - \beta_k(x_{k-1} - x'_{k-1}) + \mathcal{G}_k(x_k) - \mathcal{G}'_k(x'_k)\| \\
&\leq \|(1 + \beta_k)(x_k - x'_k) - \beta_k(x_{k-1} - x'_{k-1}) - \alpha_k(\nabla f(x_k; \xi_k) - \nabla f(x'_k, \xi'_k))\| \\
&\leq (1 + \beta_k)\delta_k + \beta_k\delta_{k-1} + 2\alpha_k M
\end{aligned}
$$

After taking the expectation, for every $k \geq \widetilde{k}$, we have

$$
\Delta_{k+1, \widetilde{k}} \leq (1 + \beta_k + (1 - \frac{1}{n})\alpha_k L)\Delta_{k, \widetilde{k}} + \beta_k \Delta_{k-1, \widetilde{k}} + 2\alpha_k M/n.
$$

Consider the following recursion:

$$
\widetilde{\Delta}_{k+1, \widetilde{k}} = (1 + \beta_k + (1 - \frac{1}{n})\alpha_k L)\widetilde{\Delta}_{k, \widetilde{k}} + \beta_k \Delta_{k-1, \widetilde{k}} + 2\alpha_k M/n.
$$

Note that the definition of recursion guarantees $\widetilde{\Delta}_{k+1, \widetilde{k}} \geq \widetilde{\Delta}_{k, \widetilde{k}}$. Then, we have the following inequality:

$$
\widetilde{\Delta}_{k+1, \widetilde{k}} = (1 + 2\beta_k + (1 - \frac{1}{n})\alpha_k L)\widetilde{\Delta}_{k, \widetilde{k}} + 2\alpha_k M/n.
$$

Note that $\widetilde{\Delta}_{k, \widetilde{k}} \geq \Delta_{k, \widetilde{k}}$, we have $\mathbb{E}[\Delta_{T, \widetilde{k}}] \leq S_1 + S_2$, where:

$$
S_1 = \sum_{k=\widetilde{k}+1}^{k_d} \prod_{p=k+1}^{K} \left(1 + 2\beta_p + (1 - \frac{1}{n})\frac{\alpha_0 L}{p}\right) \frac{2\alpha_0 M}{nk}
$$

and

$$
S_2 = \sum_{k=k_d+1}^{K} \prod_{p=k+1}^{K} \left(1 + 2\beta_p + (1 - \frac{1}{n})\frac{\alpha_0 L}{p}\right) \frac{2\alpha_0 M}{nk}.
$$

Substituting $\beta_p = \beta_d$ for $p = 1, ..., k_d$, we have

$$
\begin{aligned}
S_1 &= \sum_{k=\widetilde{k}+1}^{k_d} \prod_{p=k+1}^{K} \left(1 + 2\beta_p + (1 - \frac{1}{n})\frac{\alpha_0 L}{p}\right) \frac{2\alpha_0 M}{nk} \\
&\leq \sum_{k=\widetilde{k}+1}^{k_d} \prod_{p=k+1}^{K} \exp\left(2\beta_p + (1 - \frac{1}{n})\frac{\alpha_0 L}{p}\right) \frac{2\alpha_0 M}{nk} \\
&\leq \sum_{k=\widetilde{k}+1}^{k_d} \exp\left(2\beta_d(k_d - k) + (1 - \frac{1}{n})\alpha_0 L \log(\frac{K}{k})\right) \frac{2\alpha_0 M}{nk} \\
&\leq \frac{2\alpha_0 M}{n} K^{(1-\frac{1}{n})\alpha_0 L} \exp(2\beta_d k_d) \int_{\widetilde{k}}^{k_d} h_1(k) k^{-(1-\frac{1}{n})\alpha_0 L} dk \\
&\leq \frac{2\alpha_0 M}{n} K^{(1-\frac{1}{n})\alpha_0 L} \exp(2\beta_d k_d) \int_{\widetilde{k}}^{k_d} h_1(k) dk \\
&\leq \frac{2\alpha_0 M}{n} K^{(1-\frac{1}{n})\alpha_0 L} \exp(2\beta_d k_d)(E_1(2\beta_d \widetilde{k}) - E_1(2\beta_d k_d)) \quad (2.46)
\end{aligned}
$$

where $h_1(x) = \frac{\exp(-2\beta_d x)}{x}$ and the integral:

$$
E_1 = \int_x^{\infty} \frac{-\exp(x)}{x} dx.
$$

From (Abramowitz & Stegun, 1948), the following bound holds:

$$
\frac{1}{2} \exp(-x) \ln(1 + \frac{2}{x}) < E_1(x) < \exp(-x) \ln(1 + \frac{1}{x}).
$$

Therefore, we could upper bound $S_1$ as:

$$S_1 \le \frac{2\alpha_0 M}{n} K^{(1-\frac{1}{n})\alpha_0 L} \widetilde{h}(\beta_d, k_d), \tag{2.47}$$

where $\widetilde{h}(\beta_d, k_d) = \exp(2\beta_d(k_d - \widetilde{k}))(\ln(1 + \frac{1}{2\beta_d \widetilde{k}}) - \frac{1}{2}\ln(1 + \frac{2}{\beta_d k_d}))$.

Similarly, we could also upper bound $S_2$ as:

$$\begin{aligned}
S_2 &= \sum_{k=k_d+1}^{K} \prod_{p=k+1}^{K} \left(1 + 2\beta_p + (1 - \frac{1}{n})\frac{\alpha_0 L}{p}\right)\frac{2\alpha_0 M}{nk} \\
&\le \frac{2M}{L(n-1)}\left(\frac{K}{k_d}\right)^{(1-\frac{1}{n})\alpha_0 L} \\
&\le \frac{2M}{L(n-1)}\left(\frac{K}{\widetilde{k}}\right)^{(1-\frac{1}{n})\alpha_0 L} \tag{2.48}
\end{aligned}$$

Regarding to the bound of terms $S_1$ and $S_2$, if we plugging back to (2.45), we have:

$$\epsilon_s \le \frac{2\alpha_0 M}{n} K^{(1-\frac{1}{n})\alpha_0 L} \widetilde{h}(\beta_d, k_d) + \frac{\widetilde{k}B}{n} + \frac{2M}{L(n-1)}\left(\frac{K}{\widetilde{k}}\right)^{(1-\frac{1}{n})\alpha_0 L}. \tag{2.49}$$

$$\square$$

**Remark.** Proof for Theorem 23 (Generalization guarantee of SHB) resembles the proof of Theorem 13 (Generalization guarantee of SGD). Theorem 23 suggests that the stability bound decreases inversely with the size of the training set. It also increases as the momentum parameter $\beta_d$ increases.

### 2.2.3 Adaptive learning rate methods

Adaptive optimization methods, such as AdaGrad (Duchi et al., 2011), RMSprop (Tieleman & Hinton, 2012), AdaDelta (Zeiler, 2012), and Adam (Kingma & Ba, 2014) significantly improved gradient-based optimization algorithms. They revolutionized gradient descent approaches by incorporating two key perspectives: i) adapt the learning rate to the parameters by performing smaller updates (i.e., using low learning rates) for parameters associated with frequently occurring features, and larger updates (i.e., using high learning rates) for parameters associated with infrequent features. Note that this strategy is well-suited for sparse data. ii) introduce a variable, denoted as $v_k$, to capture the exponentially decaying sum/average of past squared gradients. $v_k$ can be viewed as an approximation of the Hessian matrix of the loss function and thus it encapsulates a valuable second-order information about the loss function. Traditionally, second-order methods, like Newton's and quasi-Newton's methods, exhibit faster convergence rates than their first-order counterparts, owing to the additional Hessian information in the update formula. Consequently, adaptive optimization methods converge faster than vanilla first-order gradient-based methods.

(Défossez et al., 2022) provides a unified formulation for adaptive methods: AdaGrad, Adam and AdaDelta, which we denoted as Unified Adaptive (UA)in this section. Suppose we have hyperparameters $\beta_1$ and $\beta_2$ such that $0 \le \beta_1 < \beta_2 \le 1$, and a sequence of learning rates $\{\alpha_k\}_{k\in\mathbb{N}}$. $\beta_1$ is a heavy-ball momentum parameter and $\beta_2$ controls the rate at which the past norms of gradients are forgotten. We define three variables: $x_k, m_k, v_k \in \mathbb{R}^d$, which respectively correspond to the parameters of the model, the exponentially decaying sum of gradients at each iteration, and the exponentially decaying sum of squared gradients at each iteration. Given the initial parameter $x_0$, $m_0 = 0$ and $v_0 = 0$, the update rule of UA method at iteration $t$ can be formulated as

$$\text{UA}: \quad \begin{cases} m_{k,i} = \beta_1 m_{k-1,i} + g(x_k; \xi_k) & i = 1, ..., d \\ v_{k,i} = \beta_2 v_{k-1,i} + (g_i(x_k; \xi_k))^2 & i = 1, ..., d \\ x_{k,i} = x_{k-1,i} - \alpha_k \frac{m_{k,i}}{\sqrt{v_{k,i} + \epsilon}} & i = 1, ..., d, \end{cases} \tag{2.50}$$

where $g_i(x_k; \xi_k) = \mathbb{E}_{\xi_k}[F(x_k)]$ is the unbiased stochastic estimation of the gradient.

Taking $\beta_1 = 0$, $\beta_2 = 1$ and $\alpha_k = \alpha$ ($\alpha$ represents a constant learning rate), UA results in the AdaGrad algorithm.

$$\text{AdaGrad} : \begin{cases} v_{k,i} = v_{k-1,i} + (g_i(x_k; \xi_k))^2 \\ x_{k,i} = x_{k-1,i} - \alpha \dfrac{g_i(x_k; \xi_k)}{\sqrt{v_{k,i} + \epsilon}}, \end{cases} \tag{2.51}$$

Then we are going to verify UA could generate Adam. Taking $\beta_1, \beta_2 \in (0,1)$, $\beta_1 < \beta_2$, and the learning rate $\alpha_k$ as

$$\alpha_k^{\text{Adam}} = \alpha \frac{1 - \beta_1}{\sqrt{1 - \beta_2}} \quad \cdot \quad \underbrace{\frac{1}{1 - \beta_1^k}}_{\text{corrective term for } m_k} \quad \cdot \quad \underbrace{\sqrt{1 - \beta_2^k}}_{\text{corrective term for } v_k} \quad ,$$

UA results in the Adam algorithm:

$$\text{Adam} : \begin{cases} m_{k,i} = \beta_1 m_{k-1,i} + g(x_k; \xi_k) & i = 1, ..., d \\ v_{k,i} = \beta_2 v_{k-1,i} + (g_i(x_k; \xi_k))^2 & i = 1, ..., d \\ x_{k,i} = x_{k-1,i} - \alpha \dfrac{1 - \beta_1}{\sqrt{1 - \beta_2}} \cdot \dfrac{1}{1 - \beta_1^k} \cdot \sqrt{1 - \beta_2^k} \dfrac{m_{k,i}}{\sqrt{v_{k,i} + \epsilon}} & i = 1, ..., d, \end{cases} \tag{2.52}$$

The formulation (2.52) is different from the standard formulation (2.54) of Adam at first glance. We are going to show that both formulations are equivalent with each other. By simple mathematical manipulation, formulation 2.52 could be written as:

$$\begin{cases} (1 - \beta_1) m_{k,i} = \beta_1 \cdot (1 - \beta_1) m_{k-1,i} + (1 - \beta_1) g(x_k; \xi_k) & i = 1, ..., d \\ (1 - \beta_2) v_{k,i} = \beta_2 \cdot (1 - \beta_2) v_{k-1,i} + (1 - \beta_2)(g_i(x_k; \xi_k))^2 & i = 1, ..., d \\ x_{k,i} = x_{k-1,i} - \alpha \dfrac{\frac{1 - \beta_1}{1 - \beta_1^k} m_{k,i}}{\frac{1 - \beta_2}{1 - \beta_2^k} \sqrt{v_{k,i} + \epsilon}} & i = 1, ..., d, \end{cases} \tag{2.53}$$

In UA, $m_{k,i}$ and $v_{k,i}$ are weighted sum of gradients and squared gradients, respectively. However, the standrad formulation for Adam use the weighted average (defined as $\hat{m}_{k,i}$ and $\hat{v}_{k,i}$ in this paper) of gradients and squared gradients, where:

$$\hat{m}_{k,i} = \beta_1 \hat{m}_{k-1,i} + (1 - \beta_1) g_i(x_k)$$
$$\hat{v}_{k,i} = \beta_2 \hat{v}_{k-1,i} + (1 - \beta_2)(g_i(x_k))^2$$

By simple mathematical manipulation, the weighted average and weighted sum could be transferred from each other as:

$$\hat{m}_{k,i} = \beta_1 \hat{m}_{k-1,i} + (1 - \beta_1) g_i(x_k) = \cdots = (1 - \beta_1) \sum_{l=1}^{k} \beta_1^{k-l} g_i(x_k) = (1 - \beta_1) m_{k,i}$$

$$\hat{v}_{k,i} = \beta_2 \hat{v}_{k-1,i} + (1 - \beta_2)(g_i(x_k))^2 = \cdots = (1 - \beta_2) \sum_{l=1}^{t} \beta_1^{k-l} g_i(x_k) = (1 - \beta_2) v_{k,i}$$

The Adam algorithm also introduces two corrective terms to account for the fact that the weighted average $\hat{m}_t$ and $\hat{v}_t$ are biased towards 0 for the first few iterations, i.e.,

$$\hat{m}_{k,i}^{\text{corr}} = \frac{\hat{m}_{k,i}}{1 - \beta_1^k}, \quad \hat{v}_{k,i}^{\text{corr}} = \frac{\hat{v}_{k,i}}{1 - \beta_2^k}.$$

Since the $\epsilon$ term is just a small constant to guarantee that the denominator is larger than zero and the update rule is valid, which does not matter much in the update formulation. Therefore, formula (2.53) could be rewritten as:

$$\text{Adam}: \begin{cases} \hat{m}_{k,i} = \beta_1 \hat{m}_{k-1,i} + (1-\beta_1)g_i(x_k;\xi_k) \\[2mm] \hat{v}_{k,i} = \beta_2 \hat{v}_{k-1,i} + (1-\beta_2)\left(g_i(x_k;\xi_k)\right)^2 \\[2mm] \hat{m}_{k,i}^{\text{corr}} = \dfrac{\hat{m}_{k,i}}{1-\beta_1^k}, \quad \hat{v}_{k,i}^{\text{corr}} = \dfrac{\hat{v}_{k,i}}{1-\beta_2^k} \\[3mm] x_{k,i} = x_{k-1,i} - \alpha\dfrac{\hat{m}_{k,i}^{\text{corr}}}{\sqrt{\hat{v}_{k,i}^{\text{corr}} + \epsilon}}, \end{cases} \tag{2.54}$$

Formula (2.54) is exactly the standard update rule for Adam. We already show that our unified formulation $UA$ could represents the Adam algorithm. Moreover, AdaDelta is a special case of Adam when $\beta_1 = 0$, thus the momentum term is removed. Obviously, UA could also represent the AdaDelta algorithm.

**Convergence Analysis** (Défossez et al., 2022) provides convergence the proof for the UA algorithm. The proof sketch is a little bit different from the previous ones. $f$ function here represents the loss of individual training examples or minibatches, $F$ here is the full training objective function. Therefore, $\mathbb{E}[\nabla f(x)] = F(x)$. The goal is to find the critical point of the function $F$.

In this section, we focus on the convergence proof for UA with $0 \leq \beta_1 < \beta_2 \leq 1$ under the following Assumption 24. We focus on the nonconvex setting because the existing literature primarily provides convergence proofs under nonconvex assumptions for the considered family of algorithms. ALso note that nonconvex assumptions are more general than convex ones.

**Assumption 24** (Nonconvex Setting)**.** *The following properties hold for the objective functions $f$ of individual training examples or minibatches and the full training objective function $F$, where $\mathbb{E}[f(x)] = F(x)$:*

- *(Lipschitz gradient.) Loss function $F(x)$ is L-smooth.*

$$\|\nabla F(x) - \nabla F(y)\| \leq L\|x-y\|.$$

- *(Gradient Bound.) $\|\nabla f(x)\| \leq R - \sqrt{\epsilon}$.*

- *(Bounded value.) $F(x) \geq F^*, \quad \forall x.$*

**Theorem 25** (Convergence of AdaGrad with momentum; Nonconvex Setting)**.** *(Theorem 3 in (Défossez et al., 2022)) Under Assumption 24, consider the UA method defined in (2.50) with hyper-parameters $\beta_2 = 1$, $\alpha_k = \alpha$ with $\alpha > 0$, and $0 \leq \beta_1 < 1$. For a number of iterations $K$ define $\tau$ to be a random index from the index set $\{0, ..., K-1\}$ such that:*

$$\forall j \in \mathbb{N}, \mathbb{P}[\tau = j] \sim 1 - \beta_1^{N-j},$$

*(thus if $\beta_1 = 0$, it is equivalent to sampling $\tau$ uniformly in $\{0, ..., K-1\}$). We have that for any $K \in \mathbb{N}^*$ such that $K > \frac{\beta_1}{1-\beta_1}$,*

$$\mathbb{E}\left[\|\nabla F(x_\tau)\|^2\right] \leq 2R\sqrt{K}\frac{F(x_0) - F^*}{\alpha\tilde{K}} + \frac{\sqrt{K}}{\tilde{K}}E\ln\left(1 + \frac{KR^2}{\epsilon}\right), \tag{2.55}$$

*with $\tilde{K} = K - \frac{\beta_1}{1-\beta_1}$, and,*

$$E = \alpha dRL + \frac{12dR^2}{1-\beta_1} + \frac{2\alpha^2 dL^2\beta_1}{1-\beta_1}.$$

**Theorem 26** (Convergence of Adam with momentum; Nonconvex Setting)**.** *(Theorem 4 in (Défossez et al., 2022)) Under Assumption 24, conisder the UA method defined in (2.50) with hyper-parameters $0 < \beta_2 < 1$,*

$0 \le \beta_1 < \beta_2$, and $\alpha_k = \alpha(1 - \beta_1)\sqrt{\frac{1-\beta_2^k}{1-\beta_2}}$ with $\alpha > 0$. For a number of iterations $K$ define $\tau$ to be a random index from the index set $\{0, ..., K-1\}$ such that:

$$\forall j \in \mathbb{N}, \mathbb{P}[\tau = j] \sim 1 - \beta_1^{N-j} \tag{2.56}$$

(thus if $\beta_1 = 0$, it is equivalent to sampling $\tau$ uniformly in $\{0, ..., K-1\}$). We have that for any $K \in \mathbb{N}^*$ such that $K > \frac{\beta_1}{1-\beta_1}$,

$$\mathbb{E}\left[\|\nabla F(x_\tau)\|^2\right] \le 2R\frac{F(x_0) - F^*}{\alpha\tilde{K}} + E\left(\frac{1}{\tilde{K}s}\ln\left(1 + \frac{R^2}{(1-\beta_2)\epsilon}\right) - \frac{K}{\tilde{K}}\ln(\beta_2)\right) \tag{2.57}$$

with $\tilde{K} = K - \frac{\beta_1}{1-\beta_1}$ and

$$E = \frac{\alpha dRL(1 - \beta_1)}{(1 - \beta_1/\beta_2)(1 - \beta_2)} + \frac{12dR^2\sqrt{1 - \beta_1}}{(1 - \beta_1/\beta_2)^{3/2}\sqrt{1 - \beta_2}} + \frac{2\alpha^2 dL^2\beta_1}{(1 - \beta_1/\beta_2)(1 - \beta_2)^{3/2}}.$$

*Proof for Theorem 25 and 26.* We are going to prove the convergence of Adagrad and Adam jointly. Firstly, we provide the general bound 2.65 that holds for both algorithms, we then split the proof for either algorithm.

**Common part of the proof.** Let us take an iteration $k \in \mathbb{N}^*$. Define $G_{k,i} = \nabla_i F(x_{k-1})$, $g_{k,i} = \nabla_i f_k(x_{k-1})$, $u_{k,i} = \frac{m_{k,i}}{\sqrt{\epsilon + v_{k,i}}}$, and $U_{k,i} = \frac{g_{k,i}}{\sqrt{\epsilon + v_{k,i}}}$. $G_k = (G_{k,1}, G_{k,2}, .., G_{k,d})$, $u_k = (u_{k,1}, u_{k,2}, ..., u_{k,d})$, $U_k = (U_{k,1}, U_{k,2}, ..., U_{k,d})$.

Using the smoothness of $F$ defined in Assumption 24, we have

$$F(x_k) \le F(x_{k-1}) - \alpha_n G_k^T u_k + \frac{\alpha_k^2 L}{2}\|u_k\|_2^2.$$

Taking the full expectation and using Lemma 70 (Shown in Appendix A),

$$\mathbb{E}[F(x_k)] \le \mathbb{E}[F(x_{k-1})] - \frac{\alpha_k}{2}\left(\sum_{i\in[d]}\sum_{l=0}^{k-1}\beta_1^l\mathbb{E}\left[\frac{G_{k-l,i}^2}{2\sqrt{\epsilon + \tilde{v}_{k,l+1,i}}}\right]\right) + \frac{\alpha_k^2 L}{2}\mathbb{E}\left[\|u_k\|_2^2\right]$$
$$+ \frac{\alpha_k^3 L^2}{4R}\sqrt{1 - \beta_1}\left(\sum_{l=1}^{k-1}\|u_{k-l}\|_2^2\sum_{p=l}^{k-1}\beta_1^p\sqrt{p}\right) + \frac{3\alpha_n R}{\sqrt{1 - \beta_1}}\left(\sum_{l=0}^{k-1}\left(\frac{\beta_1}{\beta_2}\right)^l\sqrt{l+1}\|U_{k-l}\|_2^2\right). \tag{2.58}$$

Notice that because of the bound on the $\ell_\infty$ norm of the stochastic gradients at the iterates in Assumption 24, we have for any $l \in \mathbb{N}$, $l < k$, and any coordinate $i \in [d]$, $\sqrt{\epsilon + \tilde{v}_{k,l+1,i}} \le R\sqrt{\sum_{j=0}^{k-1}\beta_2^j}$. Introducing $\Omega_k = \sqrt{\sum_{j=0}^{k-1}\beta_2^j}$, we have

$$\mathbb{E}[F(x_k)] \le \mathbb{E}[F(x_{k-1})] - \frac{\alpha_k}{2R\Omega_n}\sum_{l=0}^{k-1}\beta_1^l\mathbb{E}\left[\|G_{k-l}\|_2^2\right] + \frac{\alpha_k^2 L}{2}\mathbb{E}\left[\|u_k\|_2^2\right]$$
$$+ \frac{\alpha_k^3 L^2}{4R}\sqrt{1 - \beta_1}\left(\sum_{l=1}^{k-1}\|u_{k-l}\|_2^2\sum_{p=l}^{n-1}\beta_1^p\sqrt{p}\right) + \frac{3\alpha_k R}{\sqrt{1 - \beta_1}}\left(\sum_{l=0}^{k-1}\left(\frac{\beta_1}{\beta_2}\right)^l\sqrt{l+1}\|U_{k-l}\|_2^2\right). \tag{2.59}$$

Now summing over all iterations $k \in [K]$ for $K \in \mathbb{N}^*$, and using that for both Adam and Adagrad, $\alpha_k$ is non-decreasing, as well the fact that $L$ is bounded below by $F^*$ from Assumption 24, we get

$$\underbrace{\frac{1}{2R}\sum_{k=1}^K\frac{\alpha_k}{\Omega_k}\sum_{l=0}^{k-1}\beta_1^l\mathbb{E}\left[\|G_{k-l}\|_2^2\right]}_{A} \le F(x_0) - F^* + \underbrace{\frac{\alpha_K^2 L}{2}\sum_{k=1}^K\mathbb{E}\left[\|u_k\|_2^2\right]}_{B}$$
$$+ \underbrace{\frac{\alpha_K^3 L^2}{4R}\sqrt{1 - \beta_1}\sum_{k=1}^K\sum_{l=1}^{k-1}\mathbb{E}\left[\|u_{k-l}\|_2^2\right]\sum_{p=l}^{k-1}\beta_1^p\sqrt{p}}_{C} + \underbrace{\frac{3\alpha_K R}{\sqrt{1 - \beta_1}}\sum_{k=1}^K\sum_{l=0}^{k-1}\left(\frac{\beta_1}{\beta_2}\right)^l\sqrt{l+1}\mathbb{E}\left[\|U_{k-l}\|_2^2\right]}_{D}. \tag{2.60}$$

First looking at $B$, we have using Lemma 72,

$$B \leq \frac{\alpha_K^2 L}{2(1-\beta_1)(1-\beta_1/\beta_2)} \sum_{i\in[d]} \left( \ln\left(1 + \frac{v_{K,i}}{\epsilon}\right) - K\log(\beta_2) \right). \tag{2.61}$$

Then looking at $C$ and introducing the change of index $j = k - l$,

$$
\begin{aligned}
C &= \frac{\alpha_K^3 L^2}{4R} \sqrt{1-\beta_1} \sum_{k=1}^{K} \sum_{j=1}^{k} \mathbb{E}\left[\|u_j\|_2^2\right] \sum_{l=k-j}^{k-1} \beta_1^l \sqrt{l} \\
&= \frac{\alpha_K^3 L^2}{4R} \sqrt{1-\beta_1} \sum_{j=1}^{K} \mathbb{E}\left[\|u_j\|_2^2\right] \sum_{k=j}^{K} \sum_{l=k-j}^{k-1} \beta_1^l \sqrt{l} \\
&= \frac{\alpha_K^3 L^2}{4R} \sqrt{1-\beta_1} \sum_{j=1}^{K} \mathbb{E}\left[\|u_j\|_2^2\right] \sum_{l=0}^{K-1} \beta_1^l \sqrt{l} \sum_{k=j}^{j+l} 1 \\
&= \frac{\alpha_K^3 L^2}{4R} \sqrt{1-\beta_1} \sum_{j=1}^{K} \mathbb{E}\left[\|u_j\|_2^2\right] \sum_{k=0}^{K-1} \beta_1^k \sqrt{k}(k+1) \\
&\leq \frac{\alpha_K^3 L^2}{R} \sum_{j=1}^{K} \mathbb{E}\left[\|u_j\|_2^2\right] \frac{\beta_1}{(1-\beta_1)^2},
\end{aligned}
\tag{2.62}
$$

using Lemma 74. Finally, using Lemma 72, we get

$$C \leq \frac{\alpha_K^3 L^2 \beta_1}{R(1-\beta_1)^3(1-\beta_1/\beta_2)} \sum_{i\in[d]} \left( \ln\left(1 + \frac{v_{K,i}}{\epsilon}\right) - K\log(\beta_2) \right). \tag{2.63}$$

Finally, introducing the same change of index $j = n - k$ for $D$, we get

$$
\begin{aligned}
D &= \frac{3\alpha_K R}{\sqrt{1-\beta_1}} \sum_{k=1}^{K} \sum_{j=1}^{k} \left(\frac{\beta_1}{\beta_2}\right)^{k-j} \sqrt{1+k-j}\, \mathbb{E}\left[\|U_j\|_2^2\right] \\
&= \frac{3\alpha_K R}{\sqrt{1-\beta_1}} \sum_{k=1}^{K} \mathbb{E}\left[\|U_j\|_2^2\right] \sum_{k=j}^{K} \left(\frac{\beta_1}{\beta_2}\right)^{k-j} \sqrt{1+k-j} \\
&\leq \frac{6\alpha_K R}{\sqrt{1-\beta_1}} \sum_{k=1}^{K} \mathbb{E}\left[\|U_j\|_2^2\right] \frac{1}{(1-\beta_1/\beta_2)^{3/2}},
\end{aligned}
\tag{2.64}
$$

using Lemma 73. Finally, using Lemma 72, we get

$$D \leq \frac{6\alpha_K R}{\sqrt{1-\beta_1}(1-\beta_1/\beta_2)^{3/2}} \sum_{i\in[d]} \left( \ln\left(1 + \frac{v_{K,i}}{\epsilon}\right) - K\ln(\beta_2) \right). \tag{2.65}$$

This is as far as we can get without having to use the specific form of $\alpha_K$ given by either equation 2.54 for Adam or equation 2.51 for Adagrad. We will now split the proof for either algorithm.

**Adagrad.** For Adagrad, we have $\alpha_n = (1 - \beta_1)\alpha$, $\beta_2 = 1$ and $\Omega_n \leq \sqrt{N}$ so that,

$$
\begin{aligned}
A &= \frac{1}{2R} \sum_{k=1}^{K} \frac{\alpha_n}{\Omega_n} \sum_{j=1}^{n} \beta_1^{n-j} \mathbb{E}\left[\|G_j\|_2^2\right] \\
&\geq \frac{\alpha(1 - \beta_1)}{2R\sqrt{N}} \sum_{j=1}^{N} \mathbb{E}\left[\|G_j\|_2^2\right] \sum_{n=j}^{N} \beta_1^{n-j} \\
&= \frac{\alpha}{2R\sqrt{N}} \sum_{j=1}^{N} (1 - \beta_1^{N-j+1}) \mathbb{E}\left[\|G_j\|_2^2\right] \\
&= \frac{\alpha}{2R\sqrt{N}} \sum_{j=1}^{N} (1 - \beta_1^{N-j+1}) \mathbb{E}\left[\|\nabla F(x_{j-1})\|_2^2\right] \\
&= \frac{\alpha}{2R\sqrt{N}} \sum_{j=0}^{N-1} (1 - \beta_1^{N-j}) \mathbb{E}\left[\|\nabla F(x_j)\|_2^2\right].
\end{aligned}
\tag{2.66}
$$

Reusing equation 2.71 and equation 2.72 from the Adam proof, and introducing $\tau$ as in equation 2.56, we immediately have

$$
A \geq \frac{\alpha \tilde{K}}{2R\sqrt{N}} \mathbb{E}\left[\|\nabla F(x_\tau)\|_2^2\right].
\tag{2.67}
$$

Further notice that for any coordinate $i \in [d]$, we have $v_N \leq NR^2$, besides $\alpha_N = (1 - \beta_1)\alpha$, so that putting together equation 2.60, equation 2.67, equation 2.61, equation 2.63 and equation 2.65 with $\beta_2 = 1$, we get

$$
\mathbb{E}\left[\|\nabla F(x_\tau)\|_2^2\right] \leq 2R\sqrt{K} \frac{F_0 - F_*}{\alpha \tilde{K}} + \frac{\sqrt{K}}{\tilde{K}} E \ln\left(1 + \frac{KR^2}{\epsilon}\right),
\tag{2.68}
$$

with

$$
E = \alpha dRL + \frac{2\alpha^2 dL^2 \beta_1}{1 - \beta_1} + \frac{12dR^2}{1 - \beta_1}.
\tag{2.69}
$$

This conclude the proof of Theorem 25.

**Adam.** For Adam, using equation 2.54, we have $\alpha_k = (1 - \beta_1)\Omega_k \alpha$. Thus, we can simplify the $A$ term from equation 2.60, also using the usual change of index $j = k - l$, to get

$$
\begin{aligned}
A &= \frac{1}{2R} \sum_{k=1}^{K} \frac{\alpha_k}{\Omega_n} \sum_{j=1}^{k} \beta_1^{k-j} \mathbb{E}\left[\|G_j\|_2^2\right] \\
&= \frac{\alpha(1 - \beta_1)}{2R} \sum_{j=1}^{K} \mathbb{E}\left[\|G_j\|_2^2\right] \sum_{k=j}^{K} \beta_1^{k-j} \\
&= \frac{\alpha}{2R} \sum_{j=1}^{K} (1 - \beta_1^{K-j+1}) \mathbb{E}\left[\|G_j\|_2^2\right] \\
&= \frac{\alpha}{2R} \sum_{j=1}^{K} (1 - \beta_1^{K-j+1}) \mathbb{E}\left[\|\nabla F(x_{j-1})\|_2^2\right] \\
&= \frac{\alpha}{2R} \sum_{j=0}^{K-1} (1 - \beta_1^{K-j}) \mathbb{E}\left[\|\nabla F(x_j)\|_2^2\right].
\end{aligned}
\tag{2.70}
$$

If we now introduce $\tau$ as in the theorem, we can first notice that

$$
\sum_{j=0}^{K-1} (1 - \beta_1^{K-j}) = K - \beta_1 \frac{1 - \beta_1^K}{1 - \beta_1} \geq K - \frac{\beta_1}{1 - \beta_1}.
\tag{2.71}
$$

Introducing

$$\tilde{K} = K - \frac{\beta_1}{1 - \beta_1}, \tag{2.72}$$

we then have

$$A \geq \frac{\alpha\tilde{K}}{2R}\mathbb{E}\left[\|\nabla F(x_\tau)\|_2^2\right]. \tag{2.73}$$

Further notice that for any coordinate $i \in [d]$, we have $v_{K,i} \leq \frac{R^2}{1-\beta_2}$, besides $\alpha_K \leq \alpha\frac{1-\beta_1}{\sqrt{1-\beta_2}}$, so that putting together equation 2.60, equation 2.73, equation 2.61, equation 2.63 and equation 2.65 we get

$$\mathbb{E}\left[\|\nabla F(x_\tau)\|_2^2\right] \leq 2R\frac{F(x_0) - F^*}{\alpha\tilde{K}} + \frac{E}{\tilde{K}}\left(\ln\left(1 + \frac{R^2}{\epsilon(1-\beta_2)}\right) - N\log(\beta_2)\right), \tag{2.74}$$

with

$$E = \frac{\alpha dRL(1-\beta_1)}{(1-\beta_1/\beta_2)(1-\beta_2)} + \frac{2\alpha^2 dL^2\beta_1}{(1-\beta_1/\beta_2)(1-\beta_2)^{3/2}} + \frac{12dR^2\sqrt{1-\beta_1}}{(1-\beta_1/\beta_2)^{3/2}\sqrt{1-\beta_2}}. \tag{2.75}$$

This conclude the proof of Theorem 26. $\qquad\square$

**Remark.** The AdaGrad and Adam is of $\mathcal{O}(\ln K/\sqrt{K})$ sublinear convergence rate. Increasing $\beta_1$ (adding momentum) always deteriorates the bounds. Table 4 concludes the theoretical convergence results of first-order methods. We understand that the convergence order conclusion might not fully capture the intuitive advantages of adaptive algorithms over SGD, especially with respect to their empirical performance. Note however that this theoretical order is consistent with the literature. For example, Robbins and Monro (Robbins & Monro, 1951) established an $\mathcal{O}(1/\sqrt{K})$ convergence rate for stochastic gradient descent (SGD) under certain conditions. Momentum-based SGD methods also achieve a similar $\mathcal{O}(1/\sqrt{K})$ rate, as shown in (Yang et al., 2016). Adaptive learning rate algorithms, such as Adam and Adagrad, have gained popularity due to their empirical acceleration over vanilla SGD. Theoretical analyses, including (Défossez et al., 2022), prove an $\mathcal{O}(\ln K/\sqrt{K})$ convergence rate for Adam and Adagrad, while AMSGrad achieves the same rate as demonstrated in (Reddi et al., 2019). To clarify, while the convergence rate $\mathcal{O}(\ln K/\sqrt{K})$ of Adaptive methods does indeed have a slower asymptotic rate compared to $\mathcal{O}(1/\sqrt{K})$ of SGD-momentum, the presence of the logarithmic term becomes significant primarily for larger values of K. As I stated before, since the assumptions used for SGD and Adaptive methods are different, we could not conclude which algorithm converges faster based on the theoretical results. Importantly, the real-world performance of adaptive algorithms often transcends theoretical convergence rates due to their ability to adjust learning rates based on the data's geometry and the gradient history. In terms of practical performance, adaptive algorithms (such as Adagrad, Adam, etc.) have been shown to perform better than SGD in several scenarios due to the following key reasons:

**Individualized Learning Rates:** Adaptive methods assign different learning rates to each parameter based on its historical gradients. This enables the method to give larger updates to infrequent parameters (which often carry useful information) and smaller updates to frequent parameters, improving overall efficiency.

**Efficient Use of Gradient Information:** Adaptive methods often utilize gradient scaling to control the impact of noisy or sparse gradients, which allows them to avoid the slow progress SGD might experience in such scenarios, especially when dealing with sparse data or noisy gradients.

**Robustness to Hyperparameter Choices:** Adaptive algorithms tend to be more robust to the choice of initial learning rate, as the learning rates are adapted throughout the training process. This contrasts with SGD, where choosing the right learning rate can be crucial for convergence and performance, requiring more manual tuning.

**Empirical Evidence:** Several studies (e.g., Kingma & Ba (2014); Reddi et al. (2019); Schmidt et al. (2021)) have empirically shown that adaptive methods converge faster and require fewer iterations to achieve lower loss compared to SGD, particularly for complex, non-convex optimization problems. These results are often seen in practical deep learning applications, where adaptive methods have become the default choice due to their superior performance in terms of training efficiency.

| | Method | Assumptions | | | | | Convergence rate | Comment |
|---|---|---|---|---|---|---|---|---|
| | | Convex | Lip. smooth | Hessian bound | Variance bound | Step size | | |
| First-order Methods | SGD | strongly convex | ✗ | ✗ | ✓ | $\propto \frac{1}{k+1}$ | sub-linear: $\mathcal{O}(\frac{1}{K+1})$ | The step size of SGD/SGD-M is inversely proportional to the iteration counter while that of Adaptive methods is constant or bounded, which makes it hard theoretically to compare these methods in terms of converegence speed. However, several studies (Kingma & Ba, 2014; Reddi et al., 2019; Schmidt et al., 2021) have empirically shown that Adaptive methods converge faster time- and iteration-wise and achieve lower training loss in fewer iterations than SGD in various applications. |
| | | ✗ | ✓ | ✗ | ✓ | $\propto \frac{1}{\sqrt{k}}$ | sub-linear: $\mathcal{O}(\frac{1}{\sqrt{K}})$ | |
| | SGD-M | convex | ✗ | ✗ | ✓ | $\propto \frac{1}{\sqrt{k+1}}$ | sub-linear: $\mathcal{O}(\frac{1}{\sqrt{K+1}})$ | |
| | | ✗ | ✓ | ✗ | ✓ | $\propto \frac{1}{\sqrt{k}}$ | sub-linear: $\mathcal{O}(\frac{1}{\sqrt{K}})$ | |
| | Adagrad | ✗ | ✓ | ✗ | ✓ | Const. | sub-linear: $\mathcal{O}(\frac{\ln K}{\sqrt{K}})$ | |
| | Adam | ✗ | ✓ | ✗ | ✓ | $\propto \sqrt{\frac{1-\beta_2^k}{1-\beta_2}}$ | sub-linear: $\mathcal{O}(\frac{\ln K}{\sqrt{K}})$ | |

Table 4: Comparison of convergence results for different first-order optimization methods. For convex functions, the stopping criterion is based on the condition that the difference between the current function value $F(x)$ and the optimal value $F(x^*)$ becomes sufficiently small. In the case of non-convex functions, the stopping criterion is determined by the gradient norm $\|\nabla F(x)\|$, which should be sufficiently small.

**Generalization Ability.** It should be noted that theoretical guarantees for generalization are not commonly found in the literature concerning adaptive learning rate methods. Some empirical studies (Wilson et al., 2017; Keskar et al., 2017b; Luo et al., 2019) indicate that adaptive learning rate methods can generalize worse than standard SGD. While (Zhou et al., 2020; Zou et al., 2021) offer some theoretical insights into why Adam may generalize less effectively than SGD, these works do not provide detailed stability bounds for adaptive methods. Given that our survey focuses primarily on theoretical proofs of generalization bounds for optimization methods, we will not explore this issue in depth.

## 2.3 Second-order methods

For the theoretical analysis of the second-order methods, differently than was the case for the previously analysed first-order methods, we focus on the full-batch version instead of the stochastic version. Let $F : \mathbb{R}^n \to \mathbb{R}$ be the continuously differentiable objective function. Consider an unconstrained optimization problem:

$$\min F(x), \quad x \in \mathbb{R}^n.$$

In this section, we discuss the covergence guarantee of Newton's method and BFGS under nonconvex setting and L-BFGS under convex setting.

### 2.3.1 Newton's method

In the Newton's method (2.77), we compute the inverse of Hessian matrix $\nabla^2 F(x_k)$ directly to formulate the update rule:

$$x_{k+1} = x_k - \nabla^2 F(x_k)^{-1} \nabla F(x_k). \tag{2.76}$$

**Theorem 27** (Convergence of Newton's Method; Nonconvex Setting; Section 3.3.2 in (Wright, 2006)). *Let $F \in \mathcal{C}^3$ and Hessian matrix statisfies*

$$\left\| \nabla^2 F(x_k)^{-1} \right\| \le h.$$

$x^*$ *is the optimal point such that $\nabla F(x^*) = 0$. Given a proper initial point $x_0 \in \mathbb{R}^d$, there exists $\alpha$ such that the Newton's method iterates*

$$x_{k+1} = x_k - \nabla^2 F(x_k)^{-1} \nabla F(x_k), \tag{2.77}$$

*converge according to*

$$\|x_{k+1} - x^*\| \le \alpha \|x_k - x^*\|^2.$$

*Proof.* For the $k$-th iteration, perform second-order Taylor expansion of the function $F$ as:

$$F(x) \approx F(x_k) + \nabla F(x_k)^T (x - x_k) + \frac{1}{2}(x - x_k)^T \nabla^2 F(x_k)(x - x_k). \tag{2.78}$$

By Talyor expansion

$$\nabla F(x^*) - \nabla F(x_k) - \nabla^2 F(x_k)^{-1}(x^* - x_t) = \mathcal{O}(\|x^* - x_t\|^2),$$

then there exits $\epsilon > 0$, $c > 0$ such that if $\|x_0 - x^*\| < \epsilon$,

$$\nabla F(x^*) - \nabla F(x_k) - \nabla^2 F(x_k)^{-1}(x^* - x_t) \le c \|x^* - x_k\|^2.$$

$$\begin{aligned}
\|x_{k+1} - x^*\| &= \left\| x_k - x^* - \nabla^2 F(x_k)^{-1} \nabla F(x_k) \right\| \\
&\le \left\| \nabla^2 F(x_k)^{-1} \right\| \| \nabla F(x^*) - \nabla F(x_k) - \\
&\quad \nabla^2 F(x_k)^{-1}(x^* - x_k) \| \\
&\le ch \|x^* - x_k\|^2
\end{aligned}$$

$\square$

Although Newton's method demonstrates a quadratic convergence rate (Theorem 27), surpassing all first-order methods, the computation of the inverse of the Hessian matrix, denoted as $\nabla^2 F(x_t)^{-1}$, incurs a cubic time complexity of $\mathcal{O}(d^3)$. This results in significant computational burdens, particularly when dealing with large model sizes. The heavy computation load associated with the cubic time complexity of computing the inverse Hessian compromises efficiency and scalability of the method.

### 2.3.2 Broyden–Fletcher–Goldfarb–Shanno algorithm (BFGS)

To decrease the time complexity of the Newton's method, but at the same time benefit from the second-order information, Broyden, Fletcher, Goldfarb and Shanno approximate Hessian matrix with pseudo-Hessian matrix $\mathbf{B}_k$, to reduce the time complexity of the method from cubic to quadratic in the dimension $d$ (the time complexity is dictated by the operation of matrix inversion). In this section, we are going to show the motivation and mathematical proof of convergence for BFGS.

Let $F : \mathbb{R}^n \to \mathbb{R}$ be continuously differentiable. Consider an unconstrained optimization problem:

$$\min F(x), \quad x \in \mathbb{R}^n.$$

For the $k$-th iteration, perform second-order Taylor expansion of the function $F$:

$$F(x) = F(x_k) + \nabla F(x_k)^T (x - x_k) + \frac{1}{2}(x - x_k)^T \nabla^2 F(x_k)(x - x_k). \tag{2.79}$$

Define the update of parameter as $\mathbf{p} = x - x_k$ and construct a quadratic function $m_k(\mathbf{p})$ for the $k$-th iteration based on the Taylor expansion (2.79):

$$m_k(\mathbf{p}) = F(x_k) + \nabla F(x_k)^T \mathbf{p} + \frac{1}{2}\mathbf{p}^T \mathbf{B}_k \mathbf{p}. \tag{2.80}$$

We can get the update direction by minimizing (2.80):

$$\mathbf{p}_k = -\mathbf{B}_k^{-1} \nabla F(x_k).$$

Then

$$x_{k+1} = x_k + \alpha_k \mathbf{p}_k,$$

where step size $\alpha_k$ comes from backtracking line search.

Now we have $x_{k+1}$ and we can construct the quadratic function for $(k+1)$-th iteration:

$$m_{k+1}(\mathbf{p}) = F(x_{k+1}) + \nabla F(x_{k+1})^T \mathbf{p} + \frac{1}{2}\mathbf{p}^T \mathbf{B}_{k+1} \mathbf{p}. \tag{2.81}$$

Because $m_{k+1}(\mathbf{p})$ should have the same gradient as function $F$ at point $x_k$,

$$\nabla m_{k+1}(-\alpha_k \mathbf{p}_k) = \nabla F(x_{k+1}) - \alpha_k \mathbf{B}_{k+1} \mathbf{p}_k = \nabla F(x_k).$$

Let $s_k = x_{k+1} - x_k = -\alpha_k \mathbf{p}_k$, $y_k = \nabla F(x_{k+1}) - \nabla F(x_k)$,

$$\mathbf{B}_{k+1} s_k = y_k. \tag{2.82}$$

Lets express the relation between $\mathbf{B}_{k+1}$ and $\mathbf{B}_k$ as follows

$$\mathbf{B}_{k+1} = \mathbf{B}_k + \mathbf{E}_k,$$

where $\mathbf{E}_k$ is a rank 2 matrix. Define

$$\mathbf{E}_k = \alpha u_k u_k^T + \beta v_k v_k^T, \tag{2.83}$$

and combine it with (2.82) to obtain:

$$(\mathbf{B}_k +_k u_k^T + \beta v_k v_k^T) s_k = y_k$$
$$\alpha(u_k^T s_k) u_k + \beta(v_k^T y_k)\mathbf{v_k} = y_k - \mathbf{B}_k s_k. \tag{2.84}$$

Equation (2.84) implies that $u_k$ and $v_k$ are not unique. Assume $u_k$ and $v_k$ are parallel with $\mathbf{B}_k s_k$ and $y_k$, respectively, and denote $u_k = \gamma \mathbf{B}_k s_k$ and $v_k = \theta y_k$. Substitute them into (2.83) and (2.84):

$$\mathbf{E}_k = \alpha \gamma^2 \mathbf{B}_k s_k s_k^T \mathbf{B}_k + \beta \theta^2 y_k y_k^T, \tag{2.85}$$
$$(\alpha \gamma^2 s_k^T \mathbf{B}_k s_k + 1)\mathbf{B}_k s_k + (\beta \theta^2 y_k^T s_k - 1) y_k = \mathbf{0}. \tag{2.86}$$

Therefore,

$$\alpha \gamma^2 s_k^T \mathbf{B}_k s_k + 1 = 0 \implies \alpha \gamma^2 = -\frac{1}{s_k^T \mathbf{B}_k s_k},$$
$$\beta \theta^2 y_k^T s_k - 1 = 0 \implies \beta \theta^2 = \frac{1}{y_k y_k^T}.$$

Further substitute this into (2.85) to obtain

$$\mathbf{B}_{k+1} = \mathbf{B}_k - \frac{\mathbf{B}_k s_k s_k^T \mathbf{B}_k}{s_k^T \mathbf{B}_k s_k} + \frac{y_k y_k^T}{y_k^T s_k}. \tag{2.87}$$

Inverse $\mathbf{B}_{k+1}$ with Sherman Morrison formula $(\mathbf{A} + u\mathbf{C}v)^{-1} = \mathbf{A}^{-1} - (\mathbf{A}^{-1}uv\mathbf{A}^{-1})/(\mathbf{C}^{-1} + v\mathbf{A}^{-1}u)$ (below we remove subscripts to simplify the notation):

$$\left(\mathbf{B} - \frac{\mathbf{B}ss^T\mathbf{B}}{s^T\mathbf{B}s} + \frac{yy^T}{y^Ts}\right)^{-1}$$
$$= \left(\mathbf{B} + \frac{yy^T}{y^Ts}\right)^{-1} + \frac{\left(\mathbf{B} + \frac{yy^T}{y^Ts}\right)^{-1}\mathbf{B}ss^T\mathbf{B}\left(\mathbf{B} + \frac{yy^T}{y^Ts}\right)^{-1}}{s^T\mathbf{B}s - s^T\mathbf{B}(\mathbf{B} + y^T\mathbf{B}^{-1}y)\mathbf{B}s}. \tag{2.88}$$

Since

$$\left(\mathbf{B} + \frac{yy^T}{y^Ts}\right)^{-1}\mathbf{B}ss^T\mathbf{B}\left(\mathbf{B} + \frac{yy^T}{y^Ts}\right)^{-1}$$
$$= \left(\mathbf{B}^{-1} - \frac{\mathbf{B}^{-1}yy^T\mathbf{B}^{-1}}{y^Ts + y^T\mathbf{B}^{-1}y}\right)\mathbf{B}ss^T\mathbf{B}\left(\mathbf{B}^{-1} - \frac{\mathbf{B}^{-1}yy^T\mathbf{B}^{-1}}{y^Ts + y^T\mathbf{B}^{-1}y}\right)$$
$$= ss^T - \frac{(y^Ts)(sy^T\mathbf{B}^{-1} + \mathbf{B}^{-1}ys^T)}{y^Ts + y^T\mathbf{B}^{-1}y} + \frac{(y^Ts)^2\mathbf{B}^{-1}yy^T\mathbf{B}^{-1}}{(y^Ts + y^T\mathbf{B}^{-1}y)^2}, \tag{2.89}$$

and

$$s^T\mathbf{B}s - s^T\mathbf{B}(\mathbf{B} + y^T\mathbf{B}^{-1}y)\mathbf{B}s$$
$$= s^T\mathbf{B}s - s^T\mathbf{B}\left(\mathbf{B}^{-1} - \frac{\mathbf{B}^{-1}yy^T\mathbf{B}^{-1}}{y^Ts + y^T\mathbf{B}^{-1}y}\right)\mathbf{B}s$$
$$= \frac{(y^Ts)^2}{y^Ts + y^T\mathbf{B}^{-1}y}, \tag{2.90}$$

we can now combine (2.88), (2.102) and (2.90) to obtain

$$\left(\mathbf{B} - \frac{\mathbf{B}ss^T\mathbf{B}}{s^T\mathbf{B}s} + \frac{yy^T}{y^Ts}\right)^{-1}$$
$$= \mathbf{B}^{-1} + \frac{ss^T}{y^Ts} + \frac{ss^Ty^T\mathbf{B}^{-1}y}{(y^Ts)^2} - \frac{sy^T\mathbf{B}^{-1}}{y^Ts} - \frac{\mathbf{B}^{-1}ys^T}{y^Ts}$$
$$= \mathbf{B}^{-1}\left(\mathbf{I} - \frac{ys^T}{y^Ts}\right) - \frac{sy^T\mathbf{B}^{-1}}{y^Ts}\left(\mathbf{I} - \frac{ys^T}{y^Ts}\right) + \frac{ss^T}{y^Ts}$$
$$= \left(\mathbf{B}^{-1} - \frac{sy^T\mathbf{B}^{-1}}{y^Ts}\right)\left(\mathbf{I} - \frac{ys^T}{y^Ts}\right) + \frac{ss^T}{y^Ts}$$
$$= \left(\mathbf{I} - \frac{sy^T}{y^Ts}\right)\mathbf{B}^{-1}\left(\mathbf{I} - \frac{ys^T}{y^Ts}\right) + \frac{ss^T}{y^Ts}. \tag{2.91}$$

Therefore,

$$\mathbf{B}_{k+1}^{-1} = \left(\mathbf{I} - \frac{s_k y_k^T}{y_k^T s_k}\right)\mathbf{B}_k^{-1}\left(\mathbf{I} - \frac{y_k s_k^T}{y_k^T s_k}\right) + \frac{s_k s_k^T}{y_k^T s_k}. \tag{2.92}$$

Let $\mathbf{H}_k = \mathbf{B}_k^{-1}$ and $\rho = \frac{1}{y_k^T s_k}$. We finally conclude the update formula for BFGS to be

$$x_{k+1} = x_k - \alpha_k \mathbf{H}_k \nabla F(x_k) \tag{2.93}$$

$$\mathbf{H}_{k+1} = \left(\mathbf{I} - \rho s_k y_k^T\right) \mathbf{H}_k \left(\mathbf{I} - \rho y_k s_k^T\right) + \rho s_k s_k^T. \tag{2.94}$$

**Convergence Analysis**  We are going to show that BFGS is superlinear. The proof in this section is based on the work (Broyden et al., 1973) and is reorganized for easy reading.

The following Dennis-Moré theorem is widely used to prove superlinear convergence rate because it is sufficient and necessary. We show it below but omit the proof.

**Theorem 28** (Dennis-Moré(Dennis & Moré, 1974))**.** *Suppose $F$ is strictly twice differentiable at $x^*$, a zero of $F$, the Jacobian mapping $\nabla^2 F(x^*)$ is nonsigular. Let $\{\mathbf{B}_k\}$ be a sequence of matrices. The sequence $\{\mathbf{x_k}\}$ updating parameters as*

$$x_{k+1} = x_k - \mathbf{B}_k^{-1} \nabla F(x_k)$$

*converges to $x^*$ superlinearly if and only if*

$$\lim_{k \to \infty} \frac{\left\|(\mathbf{B}_k - \nabla^2 F(x^*))(x_k - x^*)\right\|}{\|x_k - x^*\|}.$$

The following Corollary is equivalent to Dennis-Moré Theorem.

**Corollary 29.** *Suppose $F$ is strictly twice differentiable at $x^*$, a zero of $F$, and the Jacobian mapping $\nabla^2 F(x^*)$ is nonsigular. Let $\{\mathbf{H}_k\}$ be a sequence of matrices. The sequence $\{\mathbf{x_k}\}$ updating parameters as*

$$x_{k+1} = x_k - \mathbf{H}_k \nabla F(x_k)$$

*converges to $x^*$ superlinearly if and only if*

$$\frac{\left\|\left[\mathbf{H}_k - \nabla^2 F(x^*)^{-1}\right]\left(\nabla F(x_{k+1}) - \nabla F(x_k)\right)\right\|}{\|\nabla F(x_{k+1}) - \nabla F(x_k)\|} = 0. \tag{2.95}$$

In order to prove BFGS satisfies Corollary 29, we need the following assumption.

**Assumption 30** (Nonconvex Setting)**.** *Assume $F : \mathbb{R}^n \to \mathbb{R}$ is twice differentiable in an open set $\mathcal{D}$ and suppose that for some $x^* \in \mathcal{D}$ and $p > 0$*

$$\left\|\nabla^2 F(x) - \nabla^2 F(x^*)\right\| \le K \|x - x^*\|^p.$$

*Then for each $u, v \in \mathcal{D}$, define $\sigma = \max\left\{\|v - x^*\|^p, \|u - x^*\|\right\}$ and assume*

$$\left\|\nabla F(v) - \nabla F(u) - \nabla^2 F(x^*)(v - u)\right\| \le K\sigma \|v - u\|.$$

*Moreover, if $\nabla^2 F(x^*)$ is invertible, there exists $\epsilon > 0$ and $\rho > 0$ such that $\max\left\{\|v - x^*\|^p, \|u - x^*\|\right\} \le \epsilon$ implies that*

$$(1/\rho) \|v - u\| \le \|\nabla F(v) - \nabla F(u)\| \le \rho \|v - u\|. \tag{2.96}$$

**Theorem 31** (Convergence of BFGS; Nonconvex Setting)**.** *Let the loss function $F$ satisfies Assumption 30. Then for the update rule of BFGS:*

$$x_{k+1} = x_k - \alpha_k \mathbf{H}_k \nabla F(x_k) \tag{2.97}$$

$$\mathbf{H}_{k+1} = \left(\mathbf{I} - \rho s_k y_k^T\right) \mathbf{H}_k \left(\mathbf{I} - \rho y_k s_k^T\right) + \rho s_k s_k^T, \tag{2.98}$$

*the approximate Hessian matrix satisfies*

$$\frac{\left\|\left[\mathbf{H}_k - \nabla^2 F(x^*)^{-1}\right]\left(\nabla F(x_{k+1}) - \nabla F(x_k)\right)\right\|}{\|\nabla F(x_{k+1}) - \nabla F(x_k)\|} = 0. \tag{2.99}$$

*Therefore, the parameter $x_k$ generated by BFGS converges superlinearly to the optimal point $x^*$.*

*Proof.* We first introduce two lemmas that will be used in the final proof. The proof for the following two lemmas could be found in Appendix B.

**Lemma 32.** *If* $\overline{\mathbf{H}}, \mathbf{H} \in L(\mathbb{R}^n)$, $s, y, d \in \mathbb{R}^n$ *satisfy*

$$\overline{\mathbf{H}} = \mathbf{H} + \frac{(s - \mathbf{H}y)d^T + d(s - \mathbf{H}y)^T}{d^T y} - \frac{y^T(s - \mathbf{H}y)dd^T}{(d^T y)^2} \tag{2.100}$$

*and* $\mathbf{M} \in Ł(\mathbb{R}^n)$ *is a non-singular symmetric matrix, then for all* $\mathbf{A} \in L(\mathbb{R}^n)$ *it follows that*

$$\overline{\mathbf{E}} = \mathbf{P}^T \mathbf{E} \mathbf{P} + \frac{\mathbf{M}(s - \mathbf{A}y)(\mathbf{M}d)^T}{d^T y} + \frac{\mathbf{M}d(s - \mathbf{A}y)^T \mathbf{M} \mathbf{P}}{d^T y},$$

$$\mathbf{P} = \mathbf{I} - \frac{(\mathbf{M}^{-1}y)(\mathbf{M}d)^T}{d^T y},$$

*where* $\mathbf{E} = \mathbf{M}(\mathbf{H} - \mathbf{A})\mathbf{M}$, $\overline{\mathbf{E}} = \mathbf{M}(\overline{\mathbf{H}} - \mathbf{A})\mathbf{M}$.

**Lemma 33.** *Let* $\mathbf{M} \in Ł(\mathbb{R}^n)$ *be a non-singular symmetric matrix such that*

$$\left\| \mathbf{M}d - \mathbf{M}^{-1}y \right\| \le \beta \left\| \mathbf{M}^{-1}y \right\| \tag{2.101}$$

*for some* $\beta \in [0, \frac{1}{3}]$ *and* $d, y \in \mathbb{R}^n$ *with* $y \ne 0$. *Then*

*a)* $(1 - \beta) \left\| \mathbf{M}^{-1}y \right\|^2 \le d^T y \le (1 + \beta) \left\| \mathbf{M}^{-1}y \right\|^2$,

*for non-zero* $\mathbf{E} \in L(\mathbb{R}^n)$

*b)* $\left\| \mathbf{E} \left( \mathbf{I} - \frac{(\mathbf{M}^{-1}y)(\mathbf{M}^{-1}y)^T}{d^T y} \right) \right\|_F \le \sqrt{1 - \alpha\theta^2} \left\| \mathbf{E} \right\|_F$,

*c)* $\left\| \mathbf{E}\mathbf{P} \right\|_F \le \left[ \sqrt{1 - \alpha\theta^2} + (1 - \beta)^{-1} \frac{\left\| \mathbf{M}d - \mathbf{M}^{-1}y \right\|}{\left\| \mathbf{M}^{-1}y \right\|} \right] \left\| \mathbf{E} \right\|_F$,

*where*

$$\mathbf{P} = \mathbf{I} - \frac{(\mathbf{M}^{-1}y)(\mathbf{M}d)^T}{d^T y}, \alpha = \frac{1 - 2\beta}{1 - \beta^2}, \quad \theta = \frac{\left\| \mathbf{E}\mathbf{M}^{-1} \right\| y}{\left\| \mathbf{E} \right\|_F \left\| \mathbf{M}^{-1}y \right\|},$$

*moreover, for* $\forall s \in \mathbb{R}^n$ *and* $\forall \mathbf{A} \in L(\mathbb{R}^n)$

*d)* $\left\| \frac{(s - \mathbf{A}y)(\mathbf{M}d)^T}{d^T y} \right\|_F \le 2 \frac{\left\| s - \mathbf{A}y \right\|}{\left\| \mathbf{M}^{-1}y \right\|}$.

After introducing Lemma 32 and 33, we start with the detailed proof. For easy notation, we remove subscripts in approximate Hessian update (2.97) and (2.98) and denote $\mathbf{H}_{k+1}$ and $x_{k+1}$ as $\overline{\mathbf{H}}$ and $\overline{\mathbf{x}}$. Then update formulas can be rewritten as

$$\overline{\mathbf{x}} = x - \alpha_k \mathbf{H} \nabla F(x)$$

$$\overline{\mathbf{H}} = \left( \mathbf{I} - \frac{sy^T}{y^T s} \right) \mathbf{H} \left( \mathbf{I} - \frac{ys^T}{y^T s} \right) + \frac{ss^T}{y^T s},$$

which is equivalent to

$$\overline{\mathbf{H}} = \mathbf{H} + \frac{(s - \mathbf{H}y)s^T + s(s - \mathbf{H}y)^T}{s^T y} - \frac{y^T(s - \mathbf{H}y)ss^T}{(s^T y)^2}. \tag{2.102}$$

Obviously, BFGS is a special case of Lemma 32 with vector $d = s$. From Lemma 33, we know that

$$\left\|\mathbf{P}^T\mathbf{E}\mathbf{P}\right\|_F \le \left\|\mathbf{P}\right\|_F \left\|\mathbf{E}\mathbf{P}\right\|_F$$
$$\le (1 + \frac{\left\|\mathbf{M}d - \mathbf{M}^{-1}y\right\|}{(1 - \beta)\left\|\mathbf{M}^{-1}y\right\|})(\sqrt{1 - \alpha\theta^2}\frac{\left\|\mathbf{M}d - \mathbf{M}^{-1}y\right\|}{(1 - \beta)\left\|\mathbf{M}^{-1}y\right\|})\left\|\mathbf{E}\right\|_F$$
$$\le (\sqrt{1 - \alpha\theta^2} + \frac{5}{2}(1 - \beta)^{-1}\frac{\left\|\mathbf{M}d - \mathbf{M}^{-1}y\right\|}{\left\|\mathbf{M}^{-1}y\right\|})\left\|\mathbf{E}\right\|_F, \tag{2.103}$$

$$\left\|\frac{\mathbf{M}(s - \mathbf{A}y)(\mathbf{M}d)^T}{d^T y}\right\| \le 2\left\|\mathbf{M}\right\|_F\frac{\left\|s - \mathbf{A}y\right\|}{\mathbf{M}^{-1}y}, \tag{2.104}$$

$$\left\|\frac{\mathbf{M}d(s - \mathbf{A}y)^T\mathbf{M}\mathbf{P}}{d^T y}\right\| \le \left\|\mathbf{P}\right\|_F 2\left\|\mathbf{M}\right\|_F\frac{\left\|s - \mathbf{A}y\right\|}{\mathbf{M}^{-1}y}$$
$$\le (1 + \frac{\beta}{1 - \beta})\sqrt{n}2\left\|\mathbf{M}_F\frac{\left\|s - \mathbf{A}y\right\|}{\mathbf{M}^{-1}y}\right\|$$
$$\le 4\sqrt{n}\left\|\mathbf{M}\right\|_F\frac{\left\|s - \mathbf{A}y\right\|}{\mathbf{M}^{-1}y}. \qquad \qquad . \tag{2.105}$$

Substitute (2.103-2.105) into Lemma 32 (note that $\left\|\mathbf{E}\right\|_F = \left\|\mathbf{H} - \mathbf{A}\right\|_M$) and letting $A = \nabla^2 F(x^*)^{-1}$, we get

$$\left\|\overline{\mathbf{H}} - \nabla^2 F(x^*)^{-1}\right\|_M$$
$$\le \left[\sqrt{1 - \alpha\theta^2} + \frac{5}{2}(1 - \beta)^{-1}\frac{\left\|\mathbf{M}d - \mathbf{M}^{-1}y\right\|}{\left\|\mathbf{M}^{-1}y\right\|}\right]\left\|\mathbf{H} - \nabla^2 F(x^*)^{-1}\right\|_M$$
$$+ 2(2\sqrt{n} + 1)\left\|\mathbf{M}\right\|_F\frac{\left\|s - \nabla^2 F(x^*)^{-1}y\right\|}{\left\|\mathbf{M}^{-1}\mathbf{y}\right\|}. \tag{2.106}$$

For BFGS, let $\mathbf{M} = \nabla^2 F(x^*)^{\frac{1}{2}}$, $d = s$. Assumption 30 implies

$$\frac{\left\|\mathbf{M}d - \mathbf{M}^{-1}y\right\|}{\left\|\mathbf{M}^{-1}y\right\|} \le \mu_1\left\|s\right\|^p \tag{2.107}$$

for some constant $\mu_1$. Moreover, since

$$\left\|s - \nabla^2 F(x^*)^{-1}y\right\|$$
$$\le K\left\|\nabla^2 F(x^*)^{-1}\right\|\max\{\left\|\overline{\mathbf{x}} - x^*\right\|^p, \left\|x - x^*\right\|\}\left\|s\right\|, \tag{2.108}$$

by (2.96) in Assumption 30, there exists some constant $\mu_2$ such that

$$\frac{\left\|s - \nabla^2 F(x^*)^{-1}y\right\|}{\left\|\mathbf{M}^{-1}\mathbf{y}\right\|} \le \mu_2\left\|s\right\|^p. \tag{2.109}$$

Since $\left\|s\right\| \le 2\max\{\left\|\overline{\mathbf{x}} - x^*\right\|, \left\|x - x^*\right\|\}$, (2.106), (2.107) and (2.109) implies

$$\left\|\overline{\mathbf{H}} - \nabla^2 F(x^*)^{-1}\right\|_M$$
$$\le \sqrt{1 - \alpha\theta^2}\left\|\mathbf{H} - \nabla^2 F(x^*)^{-1}\right\|_M$$
$$+ \max\{\left\|\overline{\mathbf{x}} - x^*\right\|^p, \left\|x - x^*\right\|\}^p(\alpha_1\left\|\mathbf{H} - \nabla^2 F(x^*)^{-1}\right\|_M + \alpha_2), \tag{2.110}$$

where $\alpha_1 = 5(1-\beta)^{-1}$, $\alpha_2 = 2\mu_2$. After retrieving subscripts in formula (2.110), when $k \to \infty$ it is easy to see that $\max\{\|\overline{\mathbf{x}}-x^*\|^p, \|x-x^*\|\}^p \to 0$. Therefore,

$$\left\|\mathbf{H}_{k+1}-\nabla^2 F(x^*)^{-1}\right\|_M \lesssim \sqrt{1-\alpha\theta_k^2}\left\|\mathbf{H}_k-\nabla^2 F(x^*)^{-1}\right\|_M$$
$$\leq (1-\frac{\alpha}{2}\theta_k^2)\left\|\mathbf{H}_k-\nabla^2 F(x^*)^{-1}\right\|_M,$$

and then summing over $k$ from 0 to $\infty$ gives

$$\frac{\alpha}{2}\sum_{k=1}^{\infty}\theta_k^2\left\|\mathbf{H}_k-\nabla^2 F(x^*)^{-1}\right\|_M$$
$$\lesssim \left\|\mathbf{H}_0-\nabla^2 F(x^*)^{-1}\right\|_M - \left\|\mathbf{H}_\infty-\nabla^2 F(x^*)^{-1}\right\|_M < \infty.$$

Therefore,

$$\lim_{k\to\infty}\frac{\left\|\left[\mathbf{H}_k-\nabla^2 F(x^*)^{-1}\right]y_k\right\|}{y_k} = 0 \tag{2.111}$$

By Dennis-Moré Theorem 28, we conclude that BFGS converges superlinearly. $\square$

### 2.3.3 Limited memory BFGS (LBFGS)

Previously we introduced BFGS which speeds up the second-order methods from cubic to quadratic time complexity in the dimensionality $d$. However, we still need to store the whole inverted approximate Hessian matrix $\mathbf{H}_k$. BFGS is therefore still both space and time consuming for large-scale optimization problems. It is desired to modify BFGS to make it suitable for large-scale optimization problems. This can be done by storing the most important part of the information in matrix $\mathbf{H}_k$. The resulting LBFGS method(Liu & Nocedal, 1989; Zhu et al., 1997) does not require knowledge of the sparsity structure of the Hessian and the following derivations show that is is simple to program. Let us derive the LBFGS method below.

$$\begin{aligned}
\mathbf{H}_{k+1} &= v_k^T\mathbf{H}_k v_k + \rho s_k s_k^T \qquad \text{where } v_k = \mathbf{I} - \rho y_k s_k^T \\
\mathbf{H}_1 &= v_0^T\mathbf{H}_0 v_0 + \rho_0 s_0 s_0^T \\
\mathbf{H}_2 &= v_1^T v_0^T\mathbf{H}_0 v_0 v_1 + v_1^T \rho_0 s_0 s_0^T v_1 + \rho_1 s_1 s_1^T \\
&\quad \dots \\
\mathbf{H}_{k+1} &= (v_k^T v_{k-1}^T\cdots v_0^T)\mathbf{H}_0(v_{0\,k-1}v_k) \\
&\quad + (v_k^T v_{k-1}^T\cdots v_0^T)\rho_0 s_0 s_0^T(v_{0\,k-1}v_k) \\
&\quad \dots \\
&\quad + (v_k^T v_{k-1}^T)\rho_{k-2}s_{k-2}s_{k-2}^T(v_{k-1}v_k) \\
&\quad + v_k^T\rho_{k-1}s_{k-1}s_{k-1}^T v_k \\
&\quad + \rho_k s_k s_k^T, \tag{2.112}
\end{aligned}$$

$\mathbf{H}_{k+1}$ requires the whole sequence $\{s_i, y_i\}_{i=0}^k$. We construct the following approximate formula with only $m$ data points.

$$\begin{aligned}
\mathbf{H}_k &= (v_{k-1\,k-m}^T{}^T)\mathbf{H}_k^0(v_{k-m\,k-1}) \\
&\quad + \rho_{k-m}(v_{k-1\,k-m+1}^T{}^T)s_{k-m}s_{k-m}^T(v_{k-m+1\,k-1}) \\
&\quad \dots \\
&\quad + \rho_{k-2}v_{k-1}^T s_{k-2}s_{k-2}^T v_{k-1} \\
&\quad + \rho_{k-1}s_{k-1}s_{k-1}^T. \tag{2.113}
\end{aligned}$$

The resulting pseudocode for LBFGS is provided below.

In Numerical Optimization (Nocedal & Wright, 2006) typically LBFGS is presented as a two-loop recursion algorithm (see Algorithm 2.1).

## 2.4 Convergence Analysis for LBFGS

In this section, we are going to prove that LBFGS is globally $\mathbb{R}$-linearly convergent (Definition 34) to the optimal point $x^*$ (Liu & Nocedal, 1989). In convergence analysis, we update the learning rate $\eta_k$ with line search. Moreover, we prefer to rewrite LBFGS algorithm 2.1 to directly update the approximate Hessian matrix $\mathbf{B}_k$ instead of $\mathbf{H}_k$, as shown is Algorithm 2.2.

**Definition 34** (R-linearly convergent). *Given an objective function $F$ and the sequence $\{x_k\}$, there exists a constant $0 \le R < 1$ such that $F(x_k) - F(x_*) \le R^k(F(x_0) - F(x_*))$.*

---

**Algorithm 2.1** LBFGS (Pseudocode 1: two-loop recursion)

---

1: **Input:** Init $x_0 \in \mathbb{R}^N$, $F{:}\mathbb{R}^N \to \mathbb{R}$, gradient $\nabla F{:}\mathbb{R}^N \to \mathbb{R}^N$, limited-memory size $m$, learning rate $\eta_k$
2: **Init:** $\mathbf{H}_0^0 = \mathbf{I}$, $r_0 = \nabla F(x_0)$, $k = 0$
3: **while** not converged **do**
4:      $x_{k+1} = x_k - \eta_k r_k$
5:      **if** $k > m$ **then** drop $(s_{k-m}, y_{k-m})$
6:      $s_{k+1} = x_{k+1} - x_k$, $y_{k+1} = \nabla F(x_{k+1}) - \nabla F(x_k)$
7:      $\mathbf{q} = \nabla F(x_{k+1})$, $\mathbf{H}_{k+1}^0 = \frac{s_{k+1}^T y_{k+1}}{y_{k+1}^T y_{k+1}} \mathbf{I}$, $\rho_{k+1} = \frac{1}{s_{k+1}^T y_{k+1}}$
8:      **for** i=1,...,m **do**
9:          $\alpha_i = \rho_{k+1-i} s_{k+1-i}^T \mathbf{q}$
10:         $\mathbf{q} = \mathbf{q} - \alpha_i y_{k+1-i}$
11:      **for** i=1,...,m **do**
12:         $\beta_i = \rho_{k+1-i} y_{k+1-i}^T r$
13:         $r_{k+1} = r_k + (\alpha_i - \beta_i) s_{k+1-i}$
14:      $k = k + 1$
15: **return** $x_k$

---

**Algorithm 2.2** LBFGS (Pseudocode 2 - line search learning rate)

---

1: **Input:** Init $x_0 \in \mathbb{R}^N$, $F : \mathbb{R}^N \to \mathbb{R}$, gradient $\nabla F : \mathbb{R}^N \to \mathbb{R}^N$, limited-memory size $m$, learning rate $\eta_k$ computed from line search $(0 < \beta' < \frac{1}{2}, \beta' < \beta < 1)$
2: **Init:** $\mathbf{B}_0^0 = \mathbf{I}$, $k = 0$
3: **while** not converged **do**
4:      $d_k = -\mathbf{B}_k^{-1} \nabla f(x_k)$
5:      $x_{k+1} = x_k + \eta_k d_k$
6:      Where $\eta_k$ satisfies the Wolfe conditions:
7:

$$F(x_k + \eta_k d_k) \le F(x_k) + \beta' \eta_k \nabla f(x_k) d_k \tag{2.114}$$
$$\nabla F(x_k + \eta_k d_k) \ge \beta \nabla F(x_k) d_k \tag{2.115}$$

8:      **if** $k > m$ **then** drop $(s_{k-m}, y_{k-m})$
9:      $s_{k+1} = x_{k+1} - x_k$, $y_{k+1} = \nabla f(x_{k+1}) - \nabla f(x_k)$
10:     Denote $\widetilde{s}_{kl} = s_{k+1-l}$, $\widetilde{y}_{kl} = y_{k+1-l}$
11:     **for** l=1,...,m-1 **do**

$$\mathbf{B}_k^{l+1} = \mathbf{B}_k^l - \frac{\mathbf{B}_k^l \widetilde{s}_{kl} \widetilde{s}_{kl}^T \mathbf{B}_k^l}{\widetilde{s}_{kl}^T \mathbf{B}_k^l \widetilde{s}_{kl}} + \frac{\widetilde{y}_{kl} \widetilde{y}_{kl}^T}{\widetilde{y}_{kl}^T \widetilde{s}_{kl}} \tag{2.116}$$

12:     $\mathbf{B}_{k+1} = \mathbf{B}_k^m$, $k = k + 1$
13: **return** $x_k$

---

We make the following assumption for the objective function $F$ to guarantee LBFGS is R-linear.

**Assumption 35** (Convex Setting)**.** *The objective function $F$ is twice continuously differentiable and level set $\mathcal{D} = \{x \in \mathbb{R}^n : F(x) \le F(x_0)\}$ is convex. The Hessian matrix of $F$ is denoted as $\mathbf{G}$. There exist positive constant $M_1$ and $M_2$ such that*

$$M_1 \|z\|^2 \le z^T \mathbf{G}(x) z \le M_2 \|z\|^2 \tag{2.117}$$

*for all $z \in \mathbb{R}^n$ and all $x \in \mathcal{D}$.*

**Theorem 36** (Convergence of LBFGS; Convex Setting, Theorem 6.1 in (Liu & Nocedal, 1989))**.** *Let $x_0$ be a starting point for LBFGS updates. Let $F$ satisfies Assumption 35 and assume that $\mathbf{B}_k^0 s$ are chosen so that $\left\|\mathbf{B}_k^0\right\|$ and $\left\|\mathbf{B}_k^{0^{-1}}\right\|$ are bounded. Then for all positive definite $\mathbf{B}_k^0$, Algorithm 2.2 generates a sequence $\{x_k\}$ that converges to $x^*$. Moreover there is a constant $R \in (0,1)$ such that*

$$F(x_k) - F(x^*) \le R^k [F(x_0) - F(x^*)], \tag{2.118}$$

*which implies that $\{x_k\}$ converges to $x^*$ $R$-linearly.*

*Proof.* If we define

$$\overline{\mathbf{G}}_k = \int_0^1 \mathbf{G}(x_k + \tau s_k) d\tau, \tag{2.119}$$

then

$$y_k = \overline{\mathbf{G}}_k s_k. \tag{2.120}$$

Thus (2.117) and (2.120) implies

$$M_1 \|s_k\|^2 \le y_k^T s_k \le M_2 \|s_k\|^2, \tag{2.121}$$

and

$$\frac{\|y_k\|^2}{y_k^T s_k} = \frac{s_k \overline{\mathbf{G}}_k^2 s_k}{s_k^T \overline{\mathbf{G}}_k s_k} \le M_2. \tag{2.122}$$

Then from (2.116), (2.122) and boundness of $\left\|\mathbf{B}_k^0\right\|$, the trace satisfies

$$tr(\mathbf{B}_{k+1}) \le tr(\mathbf{B}_k^0) + \sum_{l=0}^{m-1} \frac{\|\widetilde{y}_{kl}\|^2}{\widetilde{y}_{kl}^T \widetilde{s}_{kl}} \le tr(\mathbf{B}_k^0) + mM_2 \le M_3 \tag{2.123}$$

for some constant $M_3$. Similarly, the determinant satisfies

$$\begin{aligned} det(\mathbf{B}_{k+1}) &= det(\mathbf{B}_k^0) \prod_{l=0}^{m-1} \frac{\widetilde{y}_{kl}^T \widetilde{s}_{kl}}{\widetilde{s}_{kl}^T \mathbf{B}_k^l \widetilde{s}_{kl}} \\ &= det(\mathbf{B}_k^0) \prod_{l=0}^{m-1} \frac{\widetilde{y}_{kl}^T \widetilde{s}_{kl}}{\widetilde{s}_{kl}^T \widetilde{s}_{kl}} \frac{\widetilde{s}_{kl}^T \widetilde{s}_{kl}}{\widetilde{s}_{kl}^T \mathbf{B}_k^l \widetilde{s}_{kl}}. \end{aligned} \tag{2.124}$$

By (2.123), the largest eigenvalue of $\mathbf{B}_k^l$ is less than $M_3$ and together with (2.124) we get

$$det(\mathbf{B}_{k+1}) \ge det(\mathbf{B}_k^0)(M_1/M_3)^m \ge M_4, \tag{2.125}$$

for some positive constant $M_4$. Therefore, by combining (2.123) and (2.125) we conclude that there is a constant $\delta > 0$ such that

$$\cos \theta_k = \frac{s_k^T \mathbf{B}_k s_k}{\|s_k\| \|\mathbf{B}_k s_k\|} \ge \delta. \tag{2.126}$$

Then from the line search condition for the learning rate $\eta_k$ ((2.114)-(2.115) in Algorithm 2.2) and Assumption 35 we conclude that there exists $c > 0$ such that

$$F(x_{k+1}) - F(x^*) \le (1 - c\cos^2\theta_k)(F(x_k) - F(x^*)). \tag{2.127}$$

We have already proven (2.118), where $R = 1 - c\cos^2\theta_k$. From (2.117) we obtain

$$\frac{1}{2}M_1\|x_k - x^*\| \le F(x_k) - F(x_k).$$

We combine this with (2.118) to get $\|x_k - x^*\| \le R^{\frac{k}{2}}[2(F(x_0) - F(x^*))/M_1]^{\frac{1}{1}}$, so that the sequence $\{x_k\}$ is R-linearly convergent. □

## 3 Landscape-Aware Deep Learning Optimizers

This section first justifies the importance of studying the geometric properties of the landscape of the loss function of deep neural networks. Then we move to the theoretical analysis of specific DL optimization methods that adapt to the properties of the underlying loss landscape.

**Loss function in DL** Training a deep network requires finding network parameters that minimize the loss function $F(x)$ ($x$ denotes the set of parameters) that is defined as the sum of discrepancies between target data labels and their estimates obtained by the network. This sum is computed over the entire training data set. Due to the multi-layer structure of the network and non-linear nature of the network activation functions, $F(x)$ is a non-convex function of network parameters. Increasing the number of training data points complicates this function since it increases the number of its summands. Increasing the number of parameters (by adding more layers or expanding the existing ones) increases the complexity of each summand and results in the growth, which can be exponential (Anandkumar et al., 2015), of the number of critical points of $F(x)$. A typical use case for DL involves massive (i.e., high-dimensional and/or large) data sets and very large networks (i.e., with millions of parameters), which results in an optimization problem that is heavily non-convex and difficult to analyze.

The shape of the DL loss function is poorly understood. Most of what we know is either based on unrealistic assumptions and/or holds only for shallow (two-layer) networks. The existing literature emphasizes i) the proliferation of saddle points (Dauphin et al., 2014; Baldi & Hornik, 1989; Saxe et al., 2014) including degenerate or hard to escape ones (Ge et al., 2015; Anandkumar & Ge, 2016; Dauphin et al., 2014), ii) the equivalency of local minima to the global minimum (Chaudhari & Soatto, 2015; Haeffele & Vidal, 2015; Janzamin et al., 2015; Kawaguchi, 2016; Soudry & Carmon, 2016; Freeman & Bruna, 2017; Nguyen & Hein, 2017; Vidal et al., 2017; Du & Lee, 2018; Safran & Shamir, 2016; 2017; Hardt & Ma, 2017; Ge et al., 2017; Yun et al., 2018; Draxler et al., 2018; Trager et al., 2020), and iii) the existence of a large number of isolated minima and few dense regions with lots of minima close to each other(Baldassi et al., 2015; 2016a;b) (shown for shallow networks with discrete weights). (Choromanska et al., 2015a;b) has examined properties of the DL loss landscape using tools from statistical physics and show that as the network size increases the variance (and the mean) of the train and test loss values decreases and thus: i) the recovered minima become equivalent and ii) becoming stuck in poor minima is a major problem only for smaller networks (later confirmed in (Goodfellow & Vinyals, 2015)).

In order to design efficient optimization algorithms for DL we need to understand which regions of the DL loss landscape lead

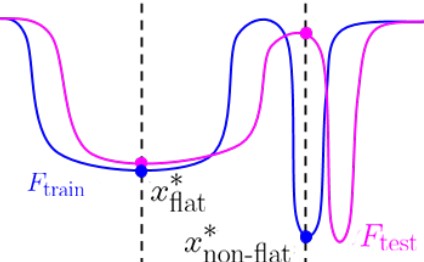

Figure 3: Consider the train and test loss (resp. $F_{\text{train}}$ and $F_{\text{test}}$), which have similar shape but are shifted with respect to each other. The local minimum that lies in the flat region of the landscape ($x^*_{\text{flat}}$) admits a similar value of the train and test loss, despite the shift between them, and thus generalizes well (in fact this property holds for network parameters that lie close enough to $x^*_{\text{flat}}$ as well). The local minimum that lies in the narrow region of the landscape (denoted as $x^*_{\text{non-flat}}$) admits a significantly larger value of the test loss compared to the train loss, due to the shift between them, and thus generalizes very poorly.

to good generalization and how to reach them. Existing work (Feng & Tu, 2020; Chaudhari et al., 2017; 2019; Hochreiter & Schmidhuber, 1997; Jastrzebski et al., 2018; Wen et al., 2018; Keskar et al., 2017a; Sagun et al., 2018; Simsekli et al., 2019; Jiang et al., 2019) shows that properly regularized SGD recovers solutions that generalize well and provides only a weak evidence that these solutions lie in wide valleys of the landscape. Prior studies (Chaudhari et al., 2017; 2019) demonstrated that the spectrum of the Hessian at a well-generalizing local minimum of the loss function is often almost flat, i.e., the majority of eigenvalues are close to zero. An intuitive illustration of why flatness potentially leads to better generalization is presented in Figure 3. (Bisla et al., 2022) provides a comprehensive study analyzing that the good generalization abilities of the DL model correlate with the properties of the loss landscape around recovered solutions.

There exists some approaches that aim at adapting the DL optimization strategy to the properties of the loss landscape and encouraging the recovery of flat optima. They can be summarized as i) regularization methods that regularize gradient descent strategy with sharpness measures such as the Minimum Description Length (Hochreiter & Schmidhuber, 1997), local entropy (Chaudhari et al., 2017), or variants of $\epsilon$-sharpness (Foret et al., 2021; Keskar et al., 2017b), and low-pass-filtering strategies (Bisla et al., 2022) ii) surrogate methods that evolve the objective function according to the diffusion equation (Mobahi, 2016), iii) averaging strategies that average model weights across training epochs (Izmailov et al., 2018; Cha et al., 2021), and iv) smoothing strategies that smoothens the loss landscape by introducing noise in the model weights and average model parameters across multiple workers run in parallel (Wen et al., 2018; Haruki et al., 2019; Lin et al., 2020) (such methods focus on distributed training of DL models with an extremely large batch size - a setting where SGD and its variants struggle).

The next section discusses in detail several popular optimization methods targeted at finding the flat minima in the DL optimization landscape. Let $\mathcal{S} = \{\xi_1, \cdots, \xi_n\}$ denote the set of $n$ samples drawn i.i.d. from an unknown distribution $\mathcal{D}$. Let $f(x; \xi)$ denote the loss of the model parametrized by $x$ for a specific example $\xi$. Denote the true loss as $F(x) \coloneqq \mathbb{E}_{\xi \sim D} f(x; \xi)$ and the empirical loss as $F_{\mathcal{S}}(x) \coloneqq \frac{1}{n} \sum_{i=1}^{n} f(x; \xi)$. We use the above notations for Section 3.

### 3.1 Sharpness-Aware Minimization (SAM)

Instead of seeking the optimal point for the original empirical training loss function $F_{\mathcal{S}}$, Sharpness-Aware Minimization (SAM) (Foret et al., 2021) targets parameters within a neighborhood where the loss is uniformly low. This formulation leads to a minmax optimization problem that underlines the SAM method and that allows for efficient implementation of gradient descent:

$$\min_{x} \quad F_{\mathcal{S}}^{SAM}(x) + \lambda \|x\|_2^2 \tag{3.1}$$
$$s.t. \quad F_{\mathcal{S}}^{SAM}(x) = \max_{\|\epsilon\|_p \leq \rho} F_{\mathcal{S}}(x + \epsilon)$$

where $\rho > 0$ is a hyperparameter and $p \in [1, \infty)$. (Foret et al., 2021) proposed default value 0.05 for the selection of hyperparameter $\rho$ and empirically showed that $p = 2$ (L-2 norm) is typically optimal among different norms.

SAM problem (3.1) could be restated as

$$\min_{x} F_{\mathcal{S}}(x) + \max_{\|\epsilon\|_p \leq \rho} \left[ F_{\mathcal{S}}(x + \epsilon) - F_{\mathcal{S}}(x) \right] + \lambda \|x\|_2^2. \tag{3.2}$$

The second term in square brackets captures the sharpness of the empirical loss $F_{\mathcal{S}}$, and we could treat it as a regularization term based on the sharpness measure. Therefore, SAM is indeed a regularization method that regularize gradient descent strategy with a sharpness measure.

The difficulty in solving the minimax problem 3.1 with gradient-base method lies in the computation of the derivative w.r.t parameter $x$ throughout the bilevel framework. (Foret et al., 2021) first approximate the

inner-level maximization problem via first-order Taylar expansion of $F_{\mathcal{S}}(x + \epsilon)$ w.r.t $\epsilon$ around 0:

$$
\begin{aligned}
\epsilon^*(x) &:= \underset{\|\epsilon\|_p \leq \rho}{\arg\max}\, F_{\mathcal{S}}(x + \epsilon) \\
&\approx \underset{\|\epsilon\|_p \leq \rho}{\arg\max}\, F_{\mathcal{S}}(x) + \epsilon^\top \nabla_x F_{\mathcal{S}}(x) \\
&= \underset{\|\epsilon\|_p \leq \rho}{\arg\max}\, \epsilon^\top \nabla_x F_{\mathcal{S}}(x).
\end{aligned}
\tag{3.3}
$$

The inner-level maximization problem can reformulated as a classical dual norm problem of the form:

$$
\|u\|_* = \sup\{u^\top v \mid \|u\|_p \leq 1\},
\tag{3.4}
$$

with $u = \epsilon/\rho$, $v = \rho \nabla_x F_{\mathcal{S}}(x)$. Since the dual norm of $l_p$ norm is $l_q$ norm, where $1/p + 1/q = 1$. we have $\|u\|_* = \|u\|_q$, and the solution for dual norm problem (3.4) is

$$
u^* = sign(v)\frac{|v|^{q-1}}{(\|v\|_q^q)^{1/p}}
$$

where $1/p + 1/q = 1$. Therefor the Solution for Problem (3.3) is:

$$
\hat{\epsilon}(x) = \rho\, sign(\nabla_x F_{\mathcal{S}}(x))\frac{|\nabla_x F_{\mathcal{S}}(x)|^{q-1}}{\left(\|\nabla_x F_{\mathcal{S}}(x)\|_q^q\right)^{1/p}},
\tag{3.5}
$$

where $1/p + 1/q = 1$. When $p = 2$, formula (3.5) is simplificated as

$$
\hat{\epsilon}(x) = \rho\frac{\nabla_x F_{\mathcal{S}}(x)}{\|\nabla_x F_{\mathcal{S}}(x)\|_2}.
\tag{3.6}
$$

After substituting (3.5) into (3.1), we compute the gradient $\nabla_x F_{\mathcal{S}}^{SAM}(x)$ as follows

$$
\begin{aligned}
\nabla_x F_{\mathcal{S}}^{SAM}(x) &\approx \nabla_x F_{\mathcal{S}}(x + \hat{\epsilon}(x)) = \frac{d(x + \hat{\epsilon}(x))}{dx}\nabla_x F_{\mathcal{S}}(x)|_{x+\hat{\epsilon}(x))} \\
&= \nabla_x F_{\mathcal{S}}(x)|_{x+\hat{\epsilon}(x))} + \frac{d\hat{\epsilon}(x)}{dx}\nabla_x F_{\mathcal{S}}(x)|_{x+\hat{\epsilon}(x))}
\end{aligned}
\tag{3.7}
$$

This approximation could be directly computed via chain rule. However, $\hat{\epsilon}(x)$ is the function of the first-order gradient $\nabla_x F_{\mathcal{S}}(x)$ and the computation of $\frac{d\hat{\epsilon}(x)}{dx}$ requires Hessian of $F_{\mathcal{S}}(x)$, which computation is time-comsuming. To further accelerate the gradient computation, (Foret et al., 2021) drops the second-order terms to obtain the final gradient approximation:

$$
\nabla_x F_{\mathcal{S}}^{SAM}(x) \approx \nabla_x F_{\mathcal{S}}(x)|_{x+\hat{\epsilon}}.
\tag{3.8}
$$

The approximation (3.8) without second-order information yields the following SAM algorithm (Algorithm 3.1). (Foret et al., 2021) empirically proves that this approximation is of good quality in a variety of settings.

---

**Algorithm 3.1** SAM

---

    **Inputs:** $x_k$: params **Hyperpar:** $\rho$: neighbourhood size, $\eta$: step size
    **Init** param $x_0$
    **while** not converged **do**
        Sample data batch $\mathcal{B} = (\eta_1 \cdots \eta_m)$
        Compute gradient $\nabla_x F_{\mathcal{B}}(x)$ of batch training loss
        Compute $\hat{\epsilon}(x)$ w.r.t formula (3.5)
        Compute gradient approximation for the SAM objective $g = \nabla_x F_{\mathcal{S}}(x)|_{x+\hat{\epsilon}}$
        $x_{k+1} = x_k - \eta * g$ // Update params

---

**Generalization Ability.** It should be noted that no theoretical generalization guarantees have been proven in the literature for the SAM method. However, the empirical results in (Foret et al., 2021; Kwon et al., 2021) verify that SAM really generalizes better than vanilla SGD.

## 3.2 Entropy-SGD

Motivated by the local geometry of the energy landscape (also interpreted as loss function) shown in Figure 4, (Chaudhari et al., 2017) proposed the Entropy-SGD algorithm that reshapes the problem of optimizing the loss function into the problem of optimizing the local entropy of the loss, arguing that it prioritizes the discovery of well-generalizing solutions within broad or flat regions of the energy landscape (also interpreted as loss function) and steers the optimizer away from poorly-generalizing solutions found in steep valleys of the loss landscape.

### 3.2.1 Local Entropy

The dicusson of local entropy builds upon (Baldassi et al., 2016a) and (Chaudhari et al., 2017). Formally, for a parameter vector $x$, consider a Gibbs distribution corresponding to a given energy landscape (also interpreted as loss function) $F(x)$:

$$P(x; \beta) = Z_\beta^{-1} \exp(-\beta F(x)) \qquad (3.9)$$

where $\beta$ is the inverse temperature and $Z_\beta$ is a normalizing constant, also called as partition function. When $\beta \to \infty$, the distribution concentrates on the global minimum of energy function (also interpreted as loss function) $F(x)$. However, the global minimum may in fact generalize very poorly. Using the toy energy landscape (also interpreted as loss function) illustrated in Figure 4 as an example, we identify $x_{\text{robust}}$ and $x_{\text{non-robust}}$ as two local minima in the energy landscape (also interpreted as loss function), situated in a flat and sharp valley,

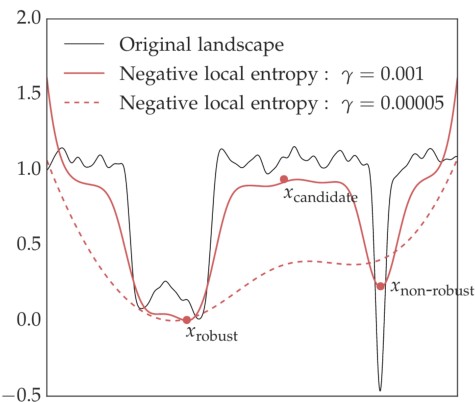

Figure 4: Local entropy concentrates on wide valleys in the energy landscape (also interpreted as loss function). This figure is taken from (Chaudhari et al., 2017) (Figure 2).

respectively. Despite the energy at $x_{\text{non-robust}}$ being significantly lower than that at $x_{\text{robust}}$, we prefer $x_{\text{robust}}$ due to its superior generalization. Consequently, this necessitates the introduction of a modified Gibbs distribution to emphasize the discovery of the flat regions in the loss landscape:

$$P(x'; x, \beta, \gamma) = Z_{x,\beta,\gamma}^{-1} \exp(-\beta F(x') - \beta \frac{\gamma}{2} \|x - x'\|_2^2), \qquad (3.10)$$

where $\beta$ is still the inverse temperature and $\gamma$ is the "scope" hyperparameter that controls the bias of modified distribution. When $\gamma \to 0$, the modified Gibbs distribution converges to the original Gibbs distribution. By increasing $\gamma$, the modified distribution concentrates on the neighbourhood around location $x$. When $\gamma \to \infty$, the modified Gibbs distribution converges to the uniform distribution. Without loss of generality, we could set the inverse temperature $\beta$ to 1 because $\gamma$ affords us similar control over the Gibbs distribution. (Chaudhari et al., 2017) proceeds by reshaping the original loss function to augment flat regions in the landscape that lie low and de-emphasize the sharp valleys. It is done through the local entropy defined with respect to the modified Gibbs distribution as:

**Definition 37** (Local Entropy). *Given the energy function (also interpreted as loss function) $F(x)$, the local entropy is defined as the log-partition function of modified Gibbs distribution ( 3.10), i.e*

$$F(x, \gamma) = \log \int_{x'} \exp(-F(x') - \frac{\gamma}{2} \|x - x'\|_2^2) dx', \qquad (3.11)$$

*where $\gamma$ is the "scope" hyperparameter that regulates the extent to which the neighborhood influences the location x.*

**Remark.** Figure 4 shows the negative local entropy $-F(x, \gamma)$ for two different values of $\gamma$. The lower the $\gamma$ is, the smoother the modified energy landscape (also interpreted as loss function) is compared to the original landscape.

### 3.2.2 Algorithm

Instead of minimizing the original loss function, Entropy-SGD minimizes the negative local entropy defined in Section 3.2.1. This section discusses how the algorithm is constructed via Langevin dynamics and several implementation details.

The optimization problem solved by Entropy-SGD is given as:

$$\min_x -F(x, \gamma; \mathcal{S}), \tag{3.12}$$

where the local entropy $F$ is dependent on the sampled dataset $\mathcal{S}$ explicitly:

$$F(x, \gamma; \mathcal{S}) = \log \int_{x'} \exp(-F_{\mathcal{S}}(x') - \frac{\gamma}{2} \|x - x'\|_2^2) dx'$$

$$= \log \int_{x'} \exp \left( -\frac{1}{n} \sum_{i=1}^{n} f(x'; \xi_i) - \frac{\gamma}{2} \|x - x'\|_2^2 \right) dx'. \tag{3.13}$$

The gradient for the negative local entropy is:

$$-\nabla F(x, \gamma; \mathcal{S}) = \gamma(x - \mathbb{E}_{x' \sim P}[x']), \tag{3.14}$$

where the latent variable $x'$ is satisfies the modified Gibbs Distribution:

$$P(x'; x, \gamma, \mathcal{S}) \propto \exp \left( -\frac{1}{n} \sum_{i=1}^{n} f(x'; \xi_i) - \frac{\gamma}{2} \|x - x'\|_2^2 \right). \tag{3.15}$$

The gradient requires the expectation of modified Gibbs distribution, which is hard to compute. Entropy-SGD however approximate its gradient using the stochastic gradient Langevin dynamics (SGLD) method, a Markov chain Monte-Carlo (MCMC) technique.

**Definition 38** (Stochastic Gradient Langevin Dynamics (SGLD)). *Given some parameter $x$, its prior distribution $p(x)$, and a set of datapoints $\mathcal{S} = \{\xi_1, ..., \xi_n\}$, the Langevin dynamics samples from the posterior distribution $p(x|\mathcal{S}) \propto p(x) \prod_{i=1}^{n} p(\xi_i|x)$ by updating the chain:*

$$\Delta x_t = \frac{\eta_t}{2} \left( \nabla \log p(x_t) + \sum_{i=1}^{n} \nabla \log p(\xi_i|x_t) \right) + \sqrt{\eta_t} \epsilon_t,$$

*where $\epsilon_t \sim N(0, \epsilon^2)$ is the Gaussian noise and $\eta_t$ is the learning rate. The parameter $\epsilon$ in SGLD is the thermal noise and was fixed this to $\epsilon \in [10^{-4}, 10^{-3}]$ in entropy-SGD. For a sample mini-batch $\mathcal{B} = \{\xi_{k_1}, ..., \xi_{k_m}\}$ from $\mathcal{S}$, the SGLD is given as:*

$$\Delta x_t = \frac{\eta_t}{2} \left( \nabla \log p(x_t) + \frac{m}{n} \sum_{i=1}^{m} \nabla \log p(\xi_{k_i}|x_t) \right) + \sqrt{\eta_t} \epsilon_t.$$

The modified Gibbs distribution defined in (3.15) could be decomposed as:

$$P(x'; x, \gamma, \mathcal{S}) \propto \exp \left( -\frac{1}{m} \sum_{i=1}^{m} f(x; \xi_{k_i}) - \frac{\gamma}{2} \|x - x'\|_2^2 \right)$$

$$= \exp(-\frac{\gamma}{2} \|x - x'\|_2^2) \prod_{i=1}^{m} \exp \left( -\frac{1}{m} f(x; \xi_{k_i}) \right). \tag{3.16}$$

One can combine (3.16) with Definition 38 and observe that given the sampled mini-batch $\mathcal{B} = \{\xi_{k_1}, ..., \xi_{k_m}\} \subset \mathcal{S}$ of size $m$, the update rule for sampling $x'$ with SGLD algorithm is:

$$x' = x' - \frac{\eta_t'}{2} \left( \frac{1}{m} \sum_{i=1}^{m} \nabla_{x'} f(x', \xi_{k_i}) - \gamma(x - x') \right) + \sqrt{\eta_t'} \epsilon_t. \tag{3.17}$$

Entropy-SGD uses moving average to compute the expectation (denoted as $\mu$) for latent variable $x'$:

$$\mu = (1-\alpha)\mu + \alpha x'.$$

This combined with (3.14) gives the approximated gradient update of the negative local entropy:

$$x = x - \eta\gamma(x-\mu).$$

The pseudocode for Entropy-SGD is shown in Algorithm 3.2.

---

**Algorithm 3.2** Entropy-SGD

---

**Inputs:** $x$: params, $K$: number of Langevin iterations
**Hyperpar:** $\gamma$: scope, $\eta$: step sizes, $\eta'$: step size for SGLD
//SGLD iterations
$x' = x$, $\mu = x$
**for** # of SGLD iterations $<$ K **do**
    Sample mini-batch $\mathcal{B} = (\xi_{k_1}\cdots\xi_{k_m})$
    Update latent variable $x'$ with equation (3.17)
    Compute the moving average $\mu = (1-\alpha)\mu + \alpha x'$ ($\alpha = 0.75$ suggested)

//Update param $x$
$x = x - \eta\gamma(x-\mu)$

---

### 3.2.3 Generalization Ability

This section discuss the generalization ability of Entropy-SGD. It shows that Entropy-SGD results in a smoother loss function and obtains better generalization error than when optimizing the original objective.

**Lemma 39.** *(Lemma 2 in (Chaudhari et al., 2017)) Assume the differentiable original loss function $f(x;\xi)$:* $\mathbb{R}^d \to \mathbb{R}$ *is $M$-Lipschitz continuous and $L$-smooth with respect to $l_2$-norm. Additionally assume that no eigenvalue of the Hessian $\nabla_{xx}^2 f(x;\xi)$ lies in the set $[-2\gamma - c, c]$ for some small $c > 0$. Then the local entropy $F(x,\gamma;\mathcal{S})$ defined in Definition 37 is $\frac{M}{1+\gamma^{-1}c}$-Lipschitz continuous and $\frac{L}{1+\gamma^{-1}c}$-smooth.*

*Proof.* Recall the gradient for the negative local entropy is:

$$-\nabla F(x,\gamma;\mathcal{S}) = \gamma(x - \mathbb{E}_{x'\sim P}[x']),$$

where

$$P(x';x,\gamma,\mathcal{S}) \propto \exp\left(-\frac{1}{n}\sum_{i=1}^{n} f(x';\xi_i) - \frac{\gamma}{2}\|x-x'\|_2^2\right).$$

Consider the term:

$$x - \mathbb{E}_{x'\sim P}[x']$$

$$= x - Z_{x,\gamma,\mathcal{S}}^{-1} \int_{x'} x' \exp\left(-\frac{1}{n}\sum_{i=1}^{n} f(x';\xi_i) - \frac{\gamma}{2}\|x-x'\|_2^2\right) dx'$$

$$\approx x - Z_{x,\gamma,\mathcal{S}}^{-1} \int_{s} (x+s) \exp\left(-\frac{1}{n}\sum_{i=1}^{n} f(x;\xi_i) - \frac{1}{n}\sum_{i=1}^{n} \nabla_x f(x;\xi_i)^\top s\right.$$

$$\left. -\frac{1}{2}s^\top\left(\gamma + \frac{1}{n}\sum_{i=1}^{n} \nabla_{xx}^2 f(x;\xi_i)\right)s\right) dx'$$

$$x - \mathbb{E}_{x' \sim P}[x']$$

$$= x \left[ 1 - Z_{x,\gamma,\mathcal{S}}^{-1} \int_s \exp\left( -\frac{1}{n} \sum_{i=1}^n f(x; \xi_i) - \frac{1}{n} \sum_{i=1}^n \nabla_x f(x; \xi_i)^\top s \right. \right.$$
$$\left. \left. -\frac{1}{2} s^\top \left( \gamma + \frac{1}{n} \sum_{i=1}^n \nabla_{xx}^2 f(x; \xi_i) \right) s \right) dx' \right]$$

$$- Z_{x,\gamma,\mathcal{S}}^{-1} \int_s s \exp\left( -\frac{1}{n} \sum_{i=1}^n f(x; \xi_i) - \frac{1}{n} \sum_{i=1}^n \nabla_x f(x; \xi_i)^\top s \right.$$
$$\left. -\frac{1}{2} s^\top \left( \gamma + \frac{1}{n} \sum_{i=1}^n \nabla_{xx}^2 f(x; \xi_i) \right) s \right) dx'$$

$$= - Z_{x,\gamma,\mathcal{S}}^{-1} \exp\left( -\frac{1}{n} \sum_{i=1}^n f(x; \xi_i) \right) \int_s s \exp\left( -\frac{1}{n} \sum_{i=1}^n f(x; \xi_i) - \frac{1}{n} \sum_{i=1}^n \nabla_x f(x; \xi_i)^\top s \right.$$
$$\left. -\frac{1}{2} s^\top \left( \gamma + \frac{1}{n} \sum_{i=1}^n \nabla_{xx}^2 f(x; \xi_i) \right) s \right) dx'.$$

The above expression is the mean of a distribution. We could approximate it using the saddle point method as the value of s that minimizes the exponent to get

$$x - \mathbb{E}_{x' \sim P}[x'] \approx \left( \gamma + \frac{1}{n} \sum_{i=1}^n \nabla_{xx}^2 f(x; \xi_i) \right)^{-1} \left( \frac{1}{n} \sum_{i=1}^n \nabla_x f(x; \xi_i) \right).$$

For notation simplicity, denote $f(x) = \frac{1}{n} \sum_{i=1}^n f(x; \xi_i)$. Obviously, the linear combination function $f$ is still $M$-Lipschitz continuous and $L$-smooth. And the formulation is simplified as:

$$x - \mathbb{E}_{x' \sim P}[x'] \approx \left( \gamma + \nabla_{xx}^2 f(x) \right)^{-1} \nabla f(x).$$

Denote $A(x) = \left( \gamma + \nabla_{xx}^2 f(x) \right)^{-1}$. For arbitrary $x, y$ we have:

$$\| \nabla F(x, \gamma; \mathcal{S}) - \nabla F(y, \gamma; \mathcal{S}) \| = \| A(x) \nabla f(x) - A(y) \nabla f(y) \| \leq L \sup_x \| A(x) \| \| x - y \|.$$

Since no eigenvalue of the Hessian $\nabla_{xx}^2 f(x; \xi)$ lies in the set $[-2\gamma - c, c]$ for some small $c > 0$, we have

$$\sup_x \| A(x) \| \leq \frac{1}{1 + \gamma^{-1} c}. \tag{3.18}$$

$\square$

**Theorem 40** (Section 4.4 in (Chaudhari et al., 2017)). *Assume the differentiable original loss function $f(x; \xi) : \mathbb{R}^d \to \mathbb{R}$ is $M$-Lipschitz continuous and $L$-smooth with respect to $l_2$-norm. Denote the stability gap of Entropy-SGD and SGD as $\epsilon_{Entropy\text{-}SGD}$ and $\epsilon_{SGD}$ respectively, we have*

$$\epsilon_{Entropy\text{-}SGD} \lesssim \left( \frac{M}{K} \right)^{\left( 1 - \frac{1}{1+\gamma^{-1}c} \right) L} \epsilon_{SGD}, \tag{3.19}$$

*where $K$ is the total number of iterations it runs.*

*Proof.* From Theorem 13 that bounds the stability of an optimization algorithm through the smoothness of its loss function and the number of iterations of SGD run on the training set, we know that the stability bound for SGD after $K$ iterations, denoted as $\epsilon_{\text{SGD}}$, is:

$$\epsilon_{\text{SGD}} \lesssim \frac{1}{n} M^{1/(1+L)} K^{1 - 1/(1+L)}. \tag{3.20}$$

This combined with Lemma 39 gives $\epsilon_{\text{Entropy-SGD}} \lesssim \left( \frac{M}{K} \right)^{\left( 1 - \frac{1}{1+\gamma^{-1}c} \right) L} \epsilon_{\text{SGD}}$. $\square$

**Remark.** Theorem 40 shows that Entropy-SGD generalizes better than SGD for all $K > M$ if they both converge after $K$ passes over the samples.

### 3.3 Low-pass filter SGD (LPF-SGD)

Similar to SAM, Low-pass filter SGD (LPF-SGD) (Bisla et al., 2022) algorithm also focuses on recovering the neighbourhood in the loss landscape with a uniformly low loss instead of finding the global minimum. The concept of LPF-SGD is to smooth the original loss with Gaussian kernel to formulate a new optimization problem. (Bisla et al., 2022) first introduces the definition of Low-pass filter (LPF) measure to quantify the sharpness of local minimum.

**Definition 41** (Low-pass filter (LPF) measure). *Let $\mathcal{K} \sim \mathcal{N}(0, \sigma^2 I)$ be a kernel of a Gaussian filter. LPF based sharpness measure at solution $x^*$ is defined as the convolution of the loss function with the Gaussian filter computed at $x^*$:*

$$(F \circledast \mathcal{K})(x^*) = \int F(x^* - \tau)\mathcal{K}(\tau)d\tau. \tag{3.21}$$

LPF-SGD incorporate the LPF sharpness measure into the DL optimization strategy to actively search for the flat regions in the DL loss landscape. Guiding the optimization process towards the well-generalizing flat regions of the DL loss landscape using LPF based measure can be done by solving the following optimization problem

$$\min_x F^{\text{LPF-SGD}}(x) = \min_x (F \circledast \mathcal{K})(x) = \min_x \int F(x - \tau)\mathcal{K}(\tau)d\tau, \tag{3.22}$$

where $\mathcal{K}$ is a Gaussian kernel and $F(x)$ is the training loss function. (Bisla et al., 2022) solve the problem given in Equation 3.22 using SGD. The gradient of the convolution between the loss function and the Gaussian kernel is

$$\nabla_x F^{\text{LPF-SGD}}(x) = \nabla_x (F \circledast \mathcal{K})(x)$$
$$= \nabla_x \int_{-\infty}^{\infty} F(x - \tau)\mathcal{K}(\tau)d\tau \tag{3.23}$$
$$\approx \frac{1}{M} \sum_{i=0}^{M} \nabla_x (F(x - \tau_i)), \tag{3.24}$$

where $\mathcal{K} \sim \mathcal{N}(0, \gamma\Sigma)$ and the matrix $\Sigma$ to is set to be proportional to the norm of the parameters in each filter of the network, i.e., we set $\Sigma = diag(\|x_1^t\|, \|x_2^t\| \cdots \|x_k^t\|)$, where $x_k^t$ is the weight matrix of the $k^{th}$ filter at iteration $t$ and $\gamma$ is the LPF radius. $\gamma$ in the Gaussian kernel controls the smoothness of loss landscape. Since the more we progress with the training, the more we care to recover flat regions in the loss landscape, $\gamma$ is progressively increasing during the training process:

$$\gamma_t = \gamma_0(\alpha/2(-\cos(t\pi/K) + 1) + 1), \tag{3.25}$$

where $K$ is total number of iterations (epochs * gradient updates per epoch) and $\alpha$ is set such that $\gamma_K = (\alpha + 1) * \gamma_0$.

The pseudocode for the LPF-SGD algorithm is shown in 3.3.

#### 3.3.1 Generalization Ability

(Bisla et al., 2022) theoretically shows that LPF-SGD converges to the optimal point with smaller generalization gap than SGD. The paper first formally confirms that indeed Gaussian LPF leads to a smoother objective function and then shows that SGD run on this smoother function recovers solution with smaller generalization error than in case of the original objective. The author first analyze the case when the variance of Gaussian kernel is identity $\Sigma = \sigma^2 I$ and then move to the analysis for non-scalar $\Sigma = \gamma diag(\|x_1\|, \|x_2\| \cdots \|x_k\|)$.

---

**Algorithm 3.3** LPF-SGD

---

**Inputs:** $x^t$: weights
**Hyperpar:** $\gamma$: filter radius, $M$: # MC iterations
**while** not converged **do**
    Sample data mini-batch $B = (\xi_1, \cdots, \xi_n)$
    Split batch into M splits $B = \{B_1 \cdots B_M\}$
    $g \leftarrow 0$
    **for** i=1 to M **do**
        $\Sigma = diag(\|x_i^t\|_{i=1}^k)$
        $g = g + \frac{1}{M}\nabla_x L(B_i, x^t + \mathcal{N}(0, \gamma\Sigma))$
    $x^{t+1} = x^t - \eta * g$ // Update weights

---

**Identical kernel covariance.** Assume $\mathcal{K} \sim \mathcal{N}(0, \sigma^2 I)$. Denote the distribution $\mathcal{N}(0, \sigma^2 I)$ as $\mu$. Define the convolution of original loss $f(x; \xi)$ with the Gaussian kernel K as

$$f_\mu(x; \xi) = (f(\cdot; \xi) \circledast \mathcal{K})(x) = \int_{\mathbb{R}^d} f(x - \tau; \xi)\mu(\tau)d\tau$$
$$= \mathbb{E}_{Z \sim \mu}[f(x + Z; \xi)] \tag{3.26}$$

where $d$ is the number of dimensions of the parameter and $Z$ is a random variable satisfying distribution $\mu$. The loss function smoothed by the Gaussian LPF, that we denote as $f_\mu$, satisfies the following theorem.

**Theorem 42.** *(Theorem 1 in (Bisla et al., 2022)) Let $\mu$ be the $\mathcal{N}(0, \sigma^2 I_{d \times d})$ distribution. Assume the differentiable loss function $f(x; \xi) : \mathbb{R}^d \to \mathbb{R}$ is $M$-Lipschitz continuous and $L$-smooth with respect to $l_2$-norm. The smoothed loss function $f_\mu(x; \xi)$ is defined as (3.26). Then the following properties hold:*

    *i) $f_\mu$ is $M$-Lipschitz continuous.*

    *ii) $f_\mu$ is continuously differentiable; moreover, its gradient is $\min\{\frac{M}{\sigma}, L\}$-Lipschitz continuous, i.e., $f_\mu$ is $\min\{\frac{M}{\sigma}, L\}$-smooth.*

    *iii) If $f$ is convex, $f(x; \xi) \le f_\mu(x; \xi) \le f(x; \xi) + \sigma M \sqrt{d}$.*

*In addition, for each bound i)-iii), there exists a function l such that the bound cannot be improved by more than a constant factor.*

Proof for Theorem 42 is deferred to Appendix C. Theorem 42 confirms that indeed $f_\mu$ is smoother than the original objective $f$. At the same time, if $\frac{M}{\sigma} < L$, increasing $\sigma$ leads to an increasingly smoother objective function, which is consistent with our intuition.

Recall Theorem 13 that bounds the stability of an optimization algorithm through the smoothness of its loss function and the number of iterations on the training set for SGD algorithm, we have already bound the stability gap for SGD and we denoted the gap as $\epsilon_{\text{SGD}}$. Then we will move into the stability gap $\epsilon_{\text{LPF-SGD}}$ for LPF-SGD algorithm. Let $\mu$ be distribution $\mathcal{N}(0, \sigma^2 I)$. By the definition of Gaussian LPF (Definition 41), the true loss and the empirical loss with respect to the Gaussian LPF smoothed function are

$$F^{\text{LPF-SGD}}(x) := (F \circledast \mathcal{K})(x) = \int_{-\infty}^{\infty} F(x - \tau)\mu(\tau)d\tau = \mathbb{E}_{Z \sim \mu}[F(x + Z)], \tag{3.27}$$

$$F_{\mathcal{S}}^{\text{LPF-SGD}}(x) \quad := (F_{\mathcal{S}} \circledast \mathcal{K})(x) = \int_{-\infty}^{\infty} F_{\mathcal{S}}(x - \tau)\mu(\tau)d\tau = \mathbb{E}_{Z \sim \mu}[F_{\mathcal{S}}(x + Z)], \tag{3.28}$$

where $\mathcal{K}$ is the Gaussian LPF kernel satisfies distribution $\mu$ and Z is a random variable satisfies distribution $\mu$. Since $F(x) \coloneqq \mathbb{E}_{\xi \sim D} f(x; \xi)$ and $F_{\mathcal{S}}(x) \coloneqq \frac{1}{m} \sum_{i=1}^{m} f(x; \xi)$, $L^{\text{LPF-SGD}}$ and $L_{\mathcal{S}}^{\text{LPF-SGD}}$ could be rewritten as

$$F^{\text{LPF-SGD}}(x) = \int_{-\infty}^{\infty} \mathbb{E}_{\xi \sim D}[f(x-\tau; \xi)]\mu(\tau)d\tau = \mathbb{E}_{\xi \sim D}\left[\int_{-\infty}^{\infty} f(x-\tau; \xi)\mu(\tau)d\tau\right] = \mathbb{E}_{\xi \sim D}\left[f_{\mu}(x; \xi)\right] \tag{3.29}$$

$$F_{\mathcal{S}}^{\text{LPF-SGD}}(x) = \int_{-\infty}^{\infty} \frac{1}{m} \sum_{i=1}^{m} f(x; \xi)\mu(\tau)d\tau = \frac{1}{m} \sum_{i=1}^{m} \left[\int_{-\infty}^{\infty} f(x-\tau; \xi_i)\mu(\tau)d\tau\right] = \frac{1}{m} \sum_{i=1}^{m} f_{\mu}(x; \xi_i). \tag{3.30}$$

Therefore, combine Theorem 42 with Theorem 13 we could conclude that the stability gap for LPF-SGD is

$$\epsilon_{\text{LPF-SGD}} \leq \frac{1 + 1/c\hat{L}}{n-1} (2cM^2)^{\frac{1}{c\hat{L}+1}} K^{\frac{c\hat{L}}{c\hat{L}+1}}.$$

where $\hat{L} = \min\{\frac{M}{\sigma}, L\}$, K is the number of total iterations. Then we could have the following Proposition 43 which supports that LPF-SGD generalize better than SGD.

**Theorem 43.** *(Generalization Guarantee of LPF-SGD for $\Sigma = \sigma^2 I$; Nonconvex Setting; Theorem 3 in (Bisla et al., 2022)) Let $\mu$ be the $\mathcal{N}(0, \sigma^2 I)$ distribution. Assume loss function $f(\theta; \xi): \mathbb{R}^d \to \mathbb{R}$ is $M$-Lipschitz and $L$-smooth. The smoothed loss function $l_\mu$ is defined as (3.26). Suppose that we run SGD and LPF-SGD for $K$ steps with non-increasing learning rate $\alpha_k \leq c/k$. Denote the stability gap of SGD and LPF-SGD as $\epsilon_{SGD}$ and $\epsilon_{LPF-SGD}$, respectively. Then the ratio of stability gap is*

$$\rho = \frac{\epsilon_{LPF-SGD}}{\epsilon_{SGD}} = \frac{1-p}{1-\hat{p}} \left(\frac{2cM}{K}\right)^{\hat{p}-p} = \mathcal{O}(\frac{1}{K^{\hat{p}-p}}), \tag{3.31}$$

*where $p = \frac{1}{cL+1}$, $\hat{p} = \frac{1}{c\min\{\frac{M}{\sigma}, L\}+1}$.*

*Finally, the following two properties hold:*

> *i) If $\sigma > \frac{M}{L}$ and $K \gg 2cM^2 \left(\frac{1-p}{1-\hat{p}}\right)^{\frac{1}{\hat{p}-p}}$, $\rho \ll 1$.*
>
> *ii) If $\sigma > \frac{M}{\beta}$ and $K > 2cM^2 \exp(\frac{2}{1-p})$, increasing $\sigma$ leads to a smaller $\rho$.*

*Proof.* For easy notation, denote $\hat{L} = \min\{\frac{M}{\sigma}, L\}$, $\epsilon_{\text{SGD}}$ and $\epsilon_{\text{LPF-SGD}}$ are stability gaps of SGD and LPF-SGD, respectively. From Theorem 13 and based on the facts that $f$ is $M$-Lipschitz continuous and $L$-smooth and that smoothed objective $f_\mu$ is $M$-Lipschitz continuous and $\min\{\frac{M}{\sigma}, L\}$-smooth, the upper bounds for the stability gaps are

$$\epsilon_{\text{SGD}} \leq \frac{1 + 1/cL}{n-1} (2cM^2)^{\frac{1}{cL+1}} K^{\frac{cL}{cL+1}},$$

$$\epsilon_{\text{LPF-SGD}} \leq \frac{1 + 1/\hat{\beta}c}{n-1} (2cM^2)^{\frac{1}{c\hat{L}+1}} K^{\frac{c\hat{L}}{c\hat{L}+1}}.$$

Denote $p = \frac{1}{cL+1}$, $\hat{p} = \frac{1}{c\hat{L}+1}$, the bound could be rewritten as

$$\epsilon_{\text{SGD}} \leq \frac{1}{(n-1)(1-p)} (2cM^2)^p K^{1-p},$$

$$\epsilon_{\text{LPF-SGD}} \leq \frac{1}{(n-1)(1-\hat{p})} (2cM^2)^{\hat{p}} K^{1-\hat{p}}.$$

Then the ratio of GE bound is

$$\rho = \frac{\epsilon_{\text{LPF-SGD}}}{\epsilon_{\text{SGD}}} = \frac{1-p}{1-\hat{p}} \left(\frac{2cM}{K}\right)^{\hat{p}-p} = O(\frac{1}{K^{\hat{p}-p}}).$$

> i) When $\sigma > \frac{M}{\beta}$ and $K > 2cM^2 \left(\frac{1-p}{1-\hat{p}}\right)^{\frac{1}{\hat{p}-p}}$, $\rho = \frac{1-p}{1-\hat{p}}(\frac{2cM^2}{K})^{\hat{p}-p} < 1$. Therefore, if $\sigma > \frac{M}{\beta}$ and $K \gg 2cM^2 \left(\frac{1-p}{1-\hat{p}}\right)^{\frac{1}{\hat{p}-p}}$, $\rho \ll 1$ and property i) holds.

ii) Denote $x := \hat{p} - p$, the reciprocal of approximated ratio could be re-written as

$$\frac{1}{\rho} \approx (1 - \frac{\hat{p} - p}{1 - p})(\frac{K}{2cM^2})^{\hat{p} - p} = (1 - \frac{x}{1 - p})(\frac{K}{2cM^2})^x$$

Define function $h(x) = (1 - ax)b^x$, where $a = \frac{1}{1-p}$ and $b = \frac{T}{2cM^2}$. Compute the derivative of function $h$:

$$h'(x) = (-ax \ln b - a + \ln b)b^x$$

$$h'(x_0) = 0 \iff x_0 = \frac{lnb - a}{a \ln b}$$

When $x \leq \frac{lnb-a}{a \ln b}$, $h'(x) \geq 0$. Otherwise $h'(x) < 0$. Since $0 < p < \hat{p} < 1$, obviously the domain of function $h$ is in the interval $[0, 1]$. If $\frac{lnb-a}{a \ln b} > 1$, the function $h$ is increasing in its domain. Which means that if the difference between $\hat{p}$ and $p$ increase, the reciprocal of approximated ratio of stability gap $\frac{1}{\rho}$ increase, which is equivalent to the approximate ratio $\rho$ of stability gap decrease. Because

$$\frac{\ln b - a}{a \ln b} > 1 \iff \ln b > \frac{a}{1 - a} \iff K > 2cM^2 e^{-p}.$$

In all, we could conclude if $T > 2cM^2 e^{-p}$, $\hat{p} - p$ increase leads to the approximate ratio of stability gap $\rho$ decrease.

What's more, we are going to analysis the relation between Gaussian filter factor $\sigma$ and the difference $\hat{p} - p$. Since

$$p = \frac{1}{cL + 1}, \quad \hat{p} = \frac{1}{c\hat{L} + 1},$$

where $\hat{L} = \min\{\frac{M}{\sigma}, L\}$. $\hat{p} - p$ increase is equivalent to $\hat{\beta}$ decrease. When the Gaussian factor $\sigma$ is large enough ($\frac{M}{\sigma} < L$), the smoother factor $\hat{\beta}$ for function $f_\mu$ is exactly $\frac{M}{\sigma}$. Moreover, increasing the factor $\sigma$ leads to the decrease of $\hat{L}$.

Due to the analysis above, if $K > 2cM^2 e^{-p}$ and $\frac{M}{\sigma} < \beta$, increasing $\sigma$ will cause the approximate ratio $\rho$ to decrease and the generalization error will be smaller. We finish the proof for condition ii).

$\square$

**Remark.** By point i) in Theorem 43, when number of iterations is large enough, stability gap of LPF-SGD is much smaller than that of SGD, which implies LPF-SGD converges to a better optimal point with lower generalization error than SGD. Moreover, point ii) in Theorem 43 indicates that increasing $\sigma$ leads to a smaller stability gap, otherwise lower generalization error.

**Non-scalar kernel covariance.** Assume $\mathcal{K} \sim \mathcal{N}(0, \Sigma)$. We next analyze the case when the covariance $\Sigma = \gamma * diag(\|\theta_1\|, \|\theta_2\| \cdots \|\theta_k\|)$ for Gaussian kernel K is no longer a scalar diagonal matrix. For easy notation, we denoted $\Sigma = diag(\sigma_1^2, \cdots, \sigma_d^2)$ where $\sigma_i^2 = \gamma * \|\theta_i\|$. The convolution of original loss $f(x; \xi)$ with the Gaussian kernel K is still defined as

$$f_\mu(x; \xi) = (f(\cdot; \xi) \circledast \mathcal{K})(x) = \int_{\mathbb{R}^d} f(x - \tau; \xi)\mu(\tau)d\tau$$
$$= \mathbb{E}_{Z \sim \mu}[f(x + Z; \xi)] \tag{3.32}$$

Then Theorem 42 could be modified into Theorem 44.

**Theorem 44.** *(Theorem 5 in (Bisla et al., 2022)) Let $\mu$ be the $\mathcal{N}(0, \Sigma)$ distribution, where $\Sigma = diag(\sigma_1^2, \cdots, \sigma_d^2) \in \mathbb{R}^{d \times d}$ is diagonal. Denote $\sigma_-^2 = \min\{\sigma_1^2, \cdots, \sigma_d^2\}$. Assume the differentiable loss function $f(\theta; \xi): \mathbb{R}^d \to \mathbb{R}$ is M-Lipschitz continuous and L-smooth with respect to $l_2$-norm. The smoothed loss function $f_\mu(\theta; \xi)$ is defined as (3.32). Then the following properties hold:*

*i) $f_\mu$ is M-Lipschitz continuous.*

    ii) $f_\mu$ *is continuously differentiable; moreover, its gradient is* $\min\{\frac{M}{\sigma_-}, L\}$-*Lipschitz continuous, i.e.* $f_\mu$ *is* $\min\{\frac{M}{\sigma_-}, L\}$-*smooth.*

    iii) *If $f$ is convex,* $f_\mu(\theta; \xi) = f(\theta; \xi) + M\sqrt{tr(\Sigma)} = f(\theta; \xi) + M\sqrt{\sum_{i=1}^{d} \sigma_i^2}$.

*In addition, for bound i) and iii), there exists a function $l$ such that the bound cannot be improved by more than a constant factor.*

Proof for Theorem 44 is quite similar to that of Theorem 42, and the detailed proof is deferred to Appendix C. Similar to the analysis for identical kernel covariance, we could conclude the stability gap for LPF-SGD for non-scalar kernel variance is

$$\epsilon_{\text{LPF-SGD}} \leq \frac{1 + 1/c\hat{L}}{n-1}(2cM^2)^{\frac{1}{c\hat{L}+1}} K^{\frac{c\hat{L}}{c\hat{L}+1}}.$$

where $\hat{L} = \min\{\frac{M}{\sigma_-}, L\}$, $\sigma_-^2 = \min\{\sigma_1^2, \cdots, \sigma_d^2\}$, $K$ is the total number of steps it runs. Then we can proceed with the following Theorem 45, which supports the claim that LPF-SGD generalizes better than SGD.

**Theorem 45.** *(Generalization Guarantee of LPF-SGD for $\Sigma = \gamma diag(\|x_1\|, \|x_2\| \cdots \|x_k\|)$; Nonconvex Setting) Let $\mu$ be the $\mathcal{N}(0, \Sigma)$ distribution, where $\Sigma = diag(\sigma_1^2, \cdots, \sigma_d^2) \in \mathbb{R}^{d \times d}$ is diagonal. Denote $\sigma_-^2 = \|\Sigma\|_\infty = \min\{\sigma_1^2, \cdots, \sigma_d^2\}$. Assume loss function $f(\theta; \xi) : \mathbb{R}^d \to \mathbb{R}$ is $M$-Lipschitz and $L$-smooth. The smoothed loss function $l_\mu$ is defined as (3.32). Suppose we execute SGD and LPF-SGD for $K$ steps with non-increasing learning rate $\alpha_k \leq c/k$. Then the ratio of stability gap is*

$$\rho = \frac{\epsilon_{LPF\text{-}SGD}}{\epsilon_{SGD}} = \frac{1-p}{1-\hat{p}}\left(\frac{2cM}{T}\right)^{\hat{p}-p} = \mathcal{O}(\frac{1}{K^{\hat{p}-p}}), \tag{3.33}$$

*where $p = \frac{1}{cL+1}$, $\hat{p} = \frac{1}{c\min\{\frac{\alpha}{\sigma_-}, L\}+1}$.*

*Finally, the following two properties hold:*

    i) *If $\sigma_- > \frac{M}{L}$ and $K \gg 2cM^2\left(\frac{1-p}{1-\hat{p}}\right)^{\frac{1}{\hat{p}-p}}$, $\rho \ll 1$.*

    ii) *If $\sigma_- > \frac{M}{L}$ and $K > 2cM^2 e^{-p}$, increasing $\sigma_-$ leads to a smaller $\rho$.*

*Proof.* By Theorem 44, the smoothed loss function $l_\mu$ is $\alpha$-Lipschitz continuous and $\min\{\frac{M}{\sigma_-}, L\}$-smooth. This gives as equivalency to Theorem 42 after substituting $\sigma_-$ for $\sigma$. Therefore, proof of Theorem 45 is exactly the same as that of Theorem 43 after performing this substitution and therefore will be omitted. $\qquad\square$

Theorem 45 indicates that generalization properties of LPF-SGD under the identical kernel covariance still hold for the non-scalar kernel covariance.

## 3.4 SmoothOut

Noise injection is a popular strategy used by practitioners to regularize the optimization process and encourage better generalization. SmoothOut (Wen et al., 2018), a method utilized in this context, involves perturbing multiple instances of the deep neural network (DNN) through noise injection and subsequently averaging these instances. Furthermore, one can provide an alternative interpretation of SmoothOut through the lenses of LPF-SGD. Analogous to the principle of LPF-SGD, where the loss function is smoothed via convolution with a Gaussian kernel, SmoothOut attenuates sharp minima by convolving the original loss with a uniform distribution.

SmoothOut method was first introduced in (Wen et al., 2018). Since sharp minima have large generalization gaps, the optimization goal is to encourage convergence to flat minima for more robust models. SmoothOut

intentionally inject noises into the model to smooth out sharp minima. Suppose the original loss is $F(x)$, where $x$ is model parameter. SmoothOut optimizes the following loss function with uniform perturbation:

$$\min_x L^{\text{SmoothOut}}(x) = \min_x \mathbb{E}_{\epsilon \sim U[-a,a]^d} F(x + \epsilon)$$

$$\approx \min_x \frac{1}{M} \sum_{i=1}^{M} F(x + \epsilon_i), \quad \text{where } \epsilon_i \overset{iid}{\sim} U[-a,a]^d \text{ and } \forall i = 1, 2, ..., M,$$

where $U[-a,a]^d$ is a uniform distribution within a range $[-a, a]$. If we assume $\mathcal{K} \sim U[-a,a]^d$ is a uniform distribution, the modified loss of SmoothOut could also be represented as the convolution of original loss $F(x)$ with kernal function $\mathcal{K}$:

$$\min_x F^{\text{SmoothOut}}(x) = \min_x \mathbb{E}_{\epsilon \sim U[-a,a]^d} F(x + \epsilon) = \min_x (F \circledast \mathcal{K})(x) = \min_x \int F(x - \tau)\mathcal{K}(\tau)d\tau.$$

The gradient of the modified loss is:

$$\nabla_x F^{\text{SmoothOut}}(x) = \nabla_x (F \circledast \mathcal{K})(x) = \nabla_x \int_{-\infty}^{\infty} F(x - \tau)\mathcal{K}(\tau)d\tau \tag{3.34}$$

$$\approx \frac{1}{M} \sum_{i=0}^{M} \nabla_x (F(x - \tau_i)), \tag{3.35}$$

This interpretation of SmoothOut method is novel and has not been shown in the literature. The pseudocode for SmoothOut is shown in Algorithm 3.4.

---

**Algorithm 3.4** SmoothOut

---

**Inputs:** $x^t$: weights
**Hyperpar:** $M$: # of samples for uniform pertubation
**while** not converged **do**
    Sample data mini-batch $B = (\xi_1, \cdots, \xi_n)$
    Pertubation: $x^t = x^t + \epsilon^t$, where $\epsilon_i^t \overset{iid}{\sim} U[-a,a]^d$.
    Backpropagation: $g = \nabla_x F(B, x^t)$.
    Denosing: $x^t = x^t - \epsilon^t$.
    Updating: $x^{t+1} = x^t - \eta * g$

---

### 3.4.1 Generalization Ability

We next introduce a novel technique that theoretically demonstrates the superior generalization capabilities of SmoothOut over SGD. This theoretical analysis draws inspiration from the proof employed for LPF-SGD, as the modified loss function for SmoothOut exhibits certain analogous properties to LPF-SGD. Specifically, both modified loss functions, denoted as $F^{\text{LPF-SGD}}$ and $F^{\text{SmoothOut}}$, operate by smoothing the original loss function $F$ through convolution with a kernel function. While $F^{\text{LPF-SGD}}$ utilizes a Gaussian kernel, $F^{\text{SmoothOut}}$ employs a uniform distribution for computing the convolution.

In this section, we first demonstrate how SmoothOut facilitates a smoother objective function. Building upon existing research (Hardt et al., 2016), which has consistently highlighted the strong correlation between the generalization error and both the Lipschitz continuity and smoothing characteristics of the objective function, we theoretically prove that SmoothOut leads to superior generalization performance.

We assume $\mathcal{K} \sim U[-a,a]^d$. Denote the uniform distribution $U[-a,a]^d$ on the l2-ball with radius $a$ as $\mu$. Define the convolution of the original loss $f(x; \xi)$ with the uniform density function K as:

$$f_\mu(x; \xi) = (f(\cdot; \xi) \circledast \mathcal{K})(x) = \int_{\mathbb{R}^d} f(x - \tau; \xi)\mu(\tau)d\tau$$

$$= \mathbb{E}_{Z \sim \mu}[f(x + Z; \xi)], \tag{3.36}$$

where $d$ is the number of dimensions of the parameter vector and $Z$ is a random variable satisfying distribution $\mu$. The loss function smoothed by the uniform distribution, that we denote as $l_\mu$, satisfies the following theorem.

**Theorem 46.** *Let $\mu$ be the the uniform distribution $U[-a, a]^d$. Assume the differentiable loss function $f(x; \xi) : \mathbb{R}^d \to \mathbb{R}$ is M-Lipschitz continuous and L-smooth with respect to $l_2$-norm. The smoothed loss function $f_\mu(x; \xi)$ is defined as (3.36). Then the following properties hold:*

*i)* $f_\mu$ *is M-Lipschitz continuous.*

*ii)* $f_\mu$ *is continuously differentiable; moreover, its gradient is* $\min\{\frac{M\sqrt{d}}{a}, \beta\}$*-Lipschitz continuous, i.e., $f_\mu$ is* $\min\{\frac{M\sqrt{d}}{a}, L\}$*-smooth.*

*iii)* *If $f$ is convex,* $f(x; \xi) \leq f_\mu(x; \xi) \leq f(x; \xi) + aM$.

*In addition, for each bound i)-iii), there exists a function $f$ such that the bound cannot be improved by more than a constant factor.*

Proof for Theorem 46 is quite similar to that of Theorem 42, and the detailed proof is deferred to Appendix D. Theorem 46 shows that $f_\mu$ is smoother than the original loss function $f$. The larger the radius $a$ of the uniform distribution $\mu$ is, the smoother the modified loss function $l_\mu$ is. Similar to the analysis for LPF-SGD, we could conclude the stability gap for SmoothOut is

$$\epsilon_{\text{SmoothOut}} \leq \frac{1 + 1/c\hat{L}}{n-1} (2cM^2)^{\frac{1}{c\hat{L}+1}} K^{\frac{c\hat{L}}{c\hat{L}+1}},$$

where $\hat{L} = \min\{\frac{M\sqrt{d}}{a}, L\}$, $K$ is the total number of steps it runs. Then we obtain the following Theorem 47 which supports the claim that SmoothOut generalizes better than SGD.

**Theorem 47** (Generalization Guarantee of SmoothOut; Nonconvex Setting)**.** *Let $\mu$ be the uniform distribution $U[-a, a]^d$ with radius of $a$. Assume loss function $f(\theta; \xi) : \mathbb{R}^d \to \mathbb{R}$ is M-Lipschitz and L-smooth. The smoothed loss function $f_\mu$ is defined as (3.36). Suppose we execute SGD and LPF-SGD for K steps with non-increasing learning rate $\alpha_k \leq c/k$. Then the ratio of stability gap is*

$$\rho = \frac{\epsilon_{SmoothOut}}{\epsilon_{SGD}} = \frac{1-p}{1-\hat{p}} \left( \frac{2cM}{T} \right)^{\hat{p}-p} = \mathcal{O}(\frac{1}{K^{\hat{p}-p}}), \tag{3.37}$$

*where $p = \frac{1}{cL+1}$, $\hat{p} = \frac{1}{\min\{\frac{M\sqrt{d}}{a}, L\}c+1}$.*

*Finally, the following two properties hold:*

*i)* *If $a > \frac{M\sqrt{d}}{L}$ and $K \gg 2c\alpha^2 \left( \frac{1-p}{1-\hat{p}} \right)^{\frac{1}{\hat{p}-p}}$, $\rho \ll 1$.*

*ii)* *If $a > \frac{M\sqrt{n}}{L}$ and $K > 2cM^2 e^{-p}$, increasing $a$ leads to a smaller $\rho$.*

We omit the proof for Theorem 47 since the proof is exactly the same as proof for Theorem 43 & 45 of LPF-SGD by changing some constant. By point i) in Theorem 47, when number of iterations is large enough, stability gap of SmoothOut is much smaller than that of SGD, which implies SmoothOut converges to a better optimal point with lower generalization error than SGD. Moreover, point ii) in Theorem 47 indicates that increasing the radius $a$ of the uniform distribution $\mu$ leads to a smaller stability gap, otherwise lower generalization error.

## 4 Distributed Optimization Methods

Distributed optimization has emerged as a critical approach in tackling large-scale optimization problems that arise in deep learning. As data size and model size continue to grow exponentially, traditional optimization

methods often become infeasible due to limitations in computational resources and memory. Distributed optimization enables the efficient processing of data across multiple machines or nodes, allowing for parallel computation and reduced communication overhead. This not only accelerates the convergence of algorithms but also enhances scalability and resilience against failures. As a result, developing robust distributed methods is essential for addressing the challenges posed by today's massive and dynamic data environments. While many distributed optimization methods have demonstrated remarkable performance across various tasks, there remains a gap in the literature regarding systematic discussions of the theoretical properties of these methods. This review focuses on bridging this gap by providing a comprehensive analysis of the convergence behaviors of key distributed optimization techniques. We aim to synthesize existing theoretical results and identify areas where further research is needed, highlighting the implications for practical applications.

Our review focuses on distributed optimization and do not cover federated learning (Kairouz et al., 2021; Wang et al., 2021; Banabilah et al., 2022; Wen et al., 2023). Distributed optimization and federated learning are two techniques that involve multiple computing units for model training but differ fundamentally in data handling and objectives. Distributed optimization operates on datasets shared among the computing nodes, where the same data is distributed across nodes for parallel processing. Thus all nodes have access to the same data or a shared copy. This type of optimization often requires frequent communication with a central server for gradient aggregation. In this setting it is reasonable to assume that the dataset on each node has the same distribution, just as what EASGD, LSGD and GRAWA require for their convergence guarantee. In contrast, federated learning focuses on training models in such a way that each node (device) has its own private data that is not shared with other devices, which is essential for maintaining privacy and security. Participants train local models and only send aggregated updates to a central server, minimizing data transfer and preserving user privacy. Each participant (e.g., mobile device) retains its local data, which is never shared, promoting privacy and security. The same proof technique for distributed optimization is no longer suitable for federated learning because we can not assume the dataset among nodes share the same distribution. Note that there already exist survey papers on federated learning for deep learning. A large body of work is purely empirical.

Distributed optimization could be categorized into two families: centralized and decentralized. Centralized techniques use a center server to coordinate the training process on workers, as opposed to the decentralized schemes. It could also be classified into synchronous and asynchronous. Asynchronous methods refers to operations or processes that occur time-wise independently of each other on each computational node. This means that different tasks can be executed on different nodes in parallel, and they are not timed with respect to each other (timing of their execution does not follow a predetermined sequence, but rather each node has its own clock). This way nodes do not wait for each other to perform computations. The synchronous method means otherwise.

In this section, we explore the theoretical guarantees of centralized and decentralized distributed optimization methods. We do not explore the difference between synchronous and asynchronous methods in our reviewbecause asynchronous methods often lack a well-defined mathematical update rule for the optimization system. This absence makes is challenging to establish convergence guarantees, which are essential for ensuring the reliability of the optimization process. Instead, for asynchronous methods like Downpour SGD, we focus on providing convergence guarantees specifically for synchronous versions of these methods.

### 4.1 Centralized Methods

Centralized distributed optimization methods represent a pivotal approach in addressing large-scale optimization problems across distributed computing environments. In these methods, a central entity coordinates the optimization process, leveraging information exchanged among distributed workers to collectively optimize a global objective function. In this section, we are going to discuss the convergence guarantee of some representative centralized methods.

#### 4.1.1 Downpour SGD

Introduced as part of Google's DistBelief framework, Downpour SGD (Dean et al., 2012), an asynchronous centralized stochastic gradient descent procedure, is proposed to efficiently train large-scale machine learning

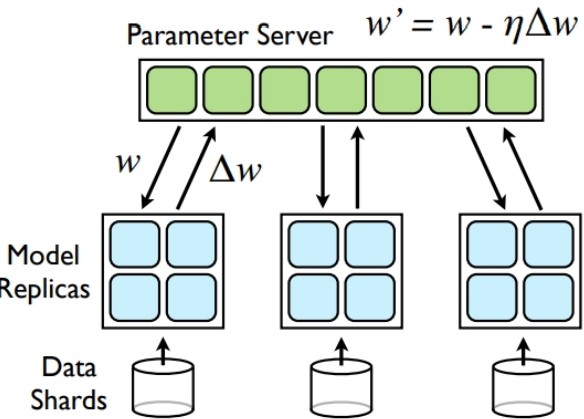

Figure 5: Dowpour SGD. Model replicas asynchronously fetch parameters and push gradients to the parameter server. The figure is taken from Figure 2 of (Dean et al., 2012)

models across distributed computing systems. In Downpour SGD, model parameters are aggregated through an asynchronous process where multiple workers independently compute gradients based on their mini-batches of data and send these gradients to a central parameter server. The server aggregates the received gradients, often by averaging or using weighted aggregation based on mini-batch sizes, and then updates the model parameters. After the update, the parameter server broadcasts the new model parameters back to all workers, allowing them to continue training with the most current values while managing the challenges of parameter staleness. Downpour SGD works as follows: the training data is segmented into multiple subsets, with each subset processed by an individual copy of a model (worker). The models exchange updates via a centralized parameter server, which maintains the current state of all model parameters, split across many machines (e.g., if we have 10 parameter server shards, each shard is responsible for storing and applying updates to $1/10^{\text{th}}$ of the model parameters). Figure 5 shows the framework of Downpour SGD.

The existing literature on Downpour SGD (Dean et al., 2012) primarily focuses on its implementation details and performance metrics, with limited emphasis on its theoretical guarantees. Our objective is to fill this gap by conducting a thorough theoretical analysis of the method and providing convergence proofs under both convex and non-convex assumptions. Since Downpour SGD serves as the basis for various centralized methods, its proof scheme holds the potential to transfer to other similar centralized methods.

Although Downpour SGD exhibits both data and model parallelism, as the dataset and model are distributed across multiple machines, it is important to note that model parallelism does not impact the convergence of the method, as the back-propagation process for gradient computation remains consistent with that of a single-machine setup. Hence, without loss of generality, we analyze the convergence guarantee of Downpour SGD under the assumption that all model parameters are kept on a single machine and only the dataset is partitioned.

In order to theoretically analyze Downpour SGD, we first mathematically formulate its underlying optimization problem. Assume that there are $m$ workers and a data subset processed by worker $i$ satisfies distribution $D_i$. The model parameter is denoted as $x$ where $x \in \mathbb{R}^d$. The purpose of distributed SGD is to train a model by minimizing the objective function $F(x)$ using $m$ workers. The optimization problem is formulated as follows:

$$
\begin{aligned}
\min_{x \in R^d} F(x) &= \min_{x \in R^d} \sum_{i=1}^{m} F_i(x) \\
&= \min_{x \in R^d} \sum_{i=1}^{m} \mathbb{E}_{\xi \sim D_i}[f(x; \xi)],
\end{aligned}
\tag{4.1}
$$

where $f(x; \xi)$ is the point-wise loss function and $F_i(x) = \mathbb{E}_{\xi \sim D_i}[f(x; \xi)]$ is the local objective function optimized by the $i$-th worker. The stochastic gradient update step for solving problem 4.1 is

$$x_{k+1} = x_k - \eta \sum_{i=1}^{m} g_i(x_k), \tag{4.2}$$

where $\eta$ is the learning rate and where $g_i(x_k)$ is a stochastic gradient of local objective $F_i(x)$ on worker $i$ at iteration $k$, such that $\mathbb{E}[g_i(x_k)] = \nabla F_i(x_k)$. The update rule specified in Equation 4.2 precisely reflects the methodology employed by Downpour.

We then move into the detailed theoretical analysis based on the mathematical formulation introduced above.

**Convex Setting.** We first show Downpour SGD reaches the same convergence rate as SGD under the stronly-convex assumption.

**Assumption 48** (Convex Setting). *We assume that the loss function $F(x) = \sum_{i=1}^{m} F_i(x)$ and stochastic gradient $g_i(x_k)$ satisfy the following conditions:*

*(1) Lipschitz gradient (L-smooth): $\|\nabla F_i(x) - \nabla F_i(y)\| \le L \|x - y\|$*

*(2) $\mu$-strongly convex: $F_i(y) \ge F_i(x) + \nabla F_i(x)^\top (x - y) + \frac{\mu}{2} \|x - y\|^2$*

*(3) Unbiased gradient: $\mathbb{E}_{\xi_i}[g_i(x_k)] = \nabla F_i(x)$*

*(4) Bounded variance: $\mathbb{E}[\|g_i(x_k) - \nabla F_i(x)\|^2] \le \sigma^2$*

**Theorem 49** (Convergence of Donwpour SGD; Convex Setting). *Under Assumption 48, when the learning rate $\eta$ satisefiles $\eta < \frac{1}{mL}$, the update rule 4.2 of Downpour SGD satisefies:*

$$\mathbb{E}[F(x_{k+1})] - F(x^*) \le (1 - \eta\mu)(F(x_k) - F(x^*)) + \frac{\eta^2 L \sigma^2}{2}.$$

*If the learning rate $\eta$ decreasse as $\eta_k = \frac{1}{\sqrt{k}}$, $\mathbb{E}[F(x_{k+1})] - F(x^*) \le \mathcal{O}(\frac{1}{k})$. In other words, Downpour SGD has sublinear convergence rate.*

*Proof.* Under the condition (1) in Assumption 48, we have

$$|F(x) - F(y)| = |\sum_{i=1}^{m} F_i(x) - \sum_{i=1}^{m} F_i(y)| \le \sum_{i=1}^{m} |F_i(x) - F_i(y)| \le mL \|x - y\|.$$

Therefore, we can conclude that the loss function $F$ is $mL$-smooth. From the update rule 4.2, we have:

$$\begin{aligned}
F(x_{k+1}) =&F(x_k - \eta \sum_{i=1}^{m} g_i(x_k)) \\
\le&F(x_k) - \eta \langle \nabla F(x_k), \sum_{i=1}^{m} g_i(x_k) \rangle + \frac{\eta^2 mL}{2} \left\| \sum_{i=1}^{m} g_i(x_k) \right\|^2 \\
\le&F(x_k) - \eta \langle \nabla F(x_k), \sum_{i=1}^{m} g_i(x_k) \rangle + \frac{\eta^2 mL}{2} \left[ \left\| \sum_{i=1}^{m} \nabla F_i(x_k) \right\|^2 + \sum_{i=1}^{m} \|g_i(x_k) - \nabla F_i(x_k)\|^2 \right] \\
=&F(x_k) - \eta \langle \nabla F(x_k), \sum_{i=1}^{m} g_i(x_k) \rangle + \frac{\eta^2 mL}{2} \left[ \|\nabla F(x_k)\|^2 + \sum_{i=1}^{m} \|g_i(x_k) - \nabla F_i(x_k)\|^2 \right].
\end{aligned}$$

By taking expectation on both sides, we have

$$\begin{aligned}
\mathbb{E}[F(x_{k+1})] \le&F(x_k) - \eta \|\nabla F(x_k)\|^2 + \frac{\eta^2 mL}{2} \|\nabla F(x_k)\|^2 + \frac{\eta^2 L \sigma^2}{2} \\
=&F(x_k) - \eta(1 - \frac{\eta mL}{2}) \|\nabla F(x_k)\|^2 + \frac{\eta^2 L \sigma^2}{2}. \tag{4.3}
\end{aligned}$$

Since the sum of $\mu$-strongly convex functions is still $\mu$-strongly convex, we could conclude that $F(x)$ is $\mu$-strong convex based on condition (2) in Assumption 48. Let $x^*$ be the minimizer of function $F$, by Polyak-Łojasiewicz inequality we have

$$F(x) - F(x^*) \leq \frac{1}{2\mu} \|\nabla F(x)\|^2, \forall x. \tag{4.4}$$

Combine formula (4.3) with (4.4) to obtain

$$\mathbb{E}[F(x_{k+1})] \leq F(x_k) - 2\eta\mu(1 - \frac{\eta mL}{2})(F(x_k) - F(x^*)) + \frac{\eta^2 L\sigma^2}{2}. \tag{4.5}$$

Since $\eta < \frac{1}{mL}$, we have

$$\mathbb{E}[F(x_{k+1})] \leq F(x_k) - \eta\mu(F(x_k) - F(x^*)) + \frac{\eta^2 L\sigma^2}{2}. \tag{4.6}$$

Substract $F(x^*)$ on both sides to obtain

$$\mathbb{E}[F(x_{k+1})] - F(x^*) \leq (1 - \eta\mu)(F(x_k) - F(x^*)) + \frac{\eta^2 L\sigma^2}{2}. \tag{4.7}$$

$\square$

**Nonconvex Setting.** We next show that Downpour SGD converges with sublinear convergence rate also under the nonconvex assumptions.

**Assumption 50** (Nonconvex Setting). *We assume that the loss function $F(x) = \sum_{i=1}^{m} F_i(x)$ and stochastic gradient $g_i(x_k)$ satisfy the following conditions:*

*(1) Lipschitz gradient (L-smooth): $\|\nabla F_i(x) - \nabla F_i(y)\| \leq L \|x - y\|$*

*(3) Unbiased gradient: $\mathbb{E}_{\xi_i}[g_i(x_k)] = \nabla F_i(x)$*

*(4) Bounded variance: $\mathbb{E}[\|g_i(x_k; \xi_i) - \nabla F_i(x)\|^2] \leq \sigma^2$*

**Theorem 51** (Convegrence of Donwpour SGD; Nonconvex Setting). *Under Assumption 50, when the learning rate $\eta$ satisefiles $\eta < \frac{1}{mL}$, the update rule 4.2 of Downpour SGD satisefies:*

$$\frac{1}{K} \sum_{i=0}^{K-1} \mathbb{E}[\|\nabla F(x_k)\|^2] \leq \frac{2\mathbb{E}[F(x_0) - F(x_K)]}{\eta K} + \eta L\sigma^2.$$

*If the learning rate $\eta$ decreasse as $\eta_k = \frac{1}{\sqrt{k}}$, $\frac{1}{K} \sum_{i=0}^{K-1} \|\nabla F(x_k)\|^2 \leq \mathcal{O}(\frac{1}{\sqrt{K}})$. In other words, Downpour SGD enjoys sublinear convergence rate.*

*Proof.* Similarly to the proof for convex setting, under condition (1) in Assumption 50, we have

$$\mathbb{E}[F(x_{k+1}) - F(x_k)] \leq -\eta(1 - \frac{\eta mL}{2})\mathbb{E}[\|\nabla F(x_k)\|^2] + \frac{\eta^2 L\sigma^2}{2}. \tag{4.8}$$

Since $\eta < \frac{1}{mL}$, we have:

$$\mathbb{E}[F(x_{k+1}) - F(x_k)] \leq -\frac{\eta}{2}\mathbb{E}[\|\nabla F(x_k)\|^2] + \frac{\eta^2 L\sigma^2}{2}. \tag{4.9}$$

Sum the inequality (4.9) from $k = 0, ...., K - 1$ and divide by $K$ to obtain:

$$\frac{1}{K} \sum_{i=0}^{K-1} \mathbb{E}[\|\nabla F(x_k)\|^2] \leq \frac{2\mathbb{E}[F(x_0) - F(x_K)]}{\eta K} + \eta L\sigma^2. \tag{4.10}$$

If $\eta = \frac{1}{\sqrt{K}}$, we have

$$\frac{1}{K} \sum_{i=0}^{K-1} \mathbb{E}[\|\nabla F(x_k)\|^2] \leq \mathcal{O}(\frac{1}{\sqrt{K}}). \tag{4.11}$$

$\square$

### 4.1.2 Elastic Averaging SGD (EASGD)

In the Downpour SGD, local workers receive parameters from a centralized parameter server and transmit the stochastic gradient term back to the central worker for subsequent update steps. However, this approach has two potential drawbacks: i) all local workers inherit identical parameters from the centralized parameter server, disregarding potential variations in problem settings arising from different data shards seen by different local workers. Consequently, this uniform parameter inheritance can limit exploration of local optima; ii) the computation of gradient updates for each worker relies on the parameters stored on the centralized parameter server. This necessitates frequent communication between the central worker and local workers to ensure the parameters on the local workers remain current. This, in turn, also counter-acts the exploration of the best possible local optima.

To address the limitations of Downpour SGD, (Zhang et al., 2015) introduced the Elastic Averaging SGD method (EASGD) along with its variants. EASGD enables each local worker to maintain its own set of parameters, while the center parameter is continuously updated as a moving average derived from the parameters computed by local workers. The coordination and communication among local workers are facilitated by an elastic force mechanism based on the quadratic penalty method, linking the local parameters with the center parameter stored in the centralized server. EASGD encourages exploration of the loss landscape by each local worker by allowing its parameters to fluctuate further from the center parameter, thereby reducing the need for extensive communication between local workers and the centralized server. The reduction in communication not only encourages exploration but also enhances computational efficiency.

The problem formulation for EASGD is provided next. Consider minimizing a function $F(x)$ in a parallel computing environment with $m$ local workers and a centralized parameter server. We focus on the stochastic optimization problem of the following form:

$$\min_x F(x) = \min_x \mathbb{E}_{\xi \sim D}[f(x, \xi)], \tag{4.12}$$

where $f(x, \xi)$ is the point-wise loss function, $x$ denotes the model parameters to be estimated, and $\xi$ is a random variable that follows the data distribution $D$. The stochastic optimization problem (4.12) can be reformulated as follow:

$$\min_{x^1,\ldots,x^n,\widetilde{x}} \sum_{i=1}^{n} \mathbb{E}[f(x^i, \xi^i)] + \frac{\rho}{2} \left\| x^i - \widetilde{x} \right\|^2, \tag{4.13}$$

where $x^i$ is the local parameter on worker $i$ and $\widetilde{x}$ is the center parameter, each $\xi^i$ satisfies the data distribution $D$ (EASGD assumes that each worker can sample the entire dataset). The problem of the equivalence of these two objectives is studied in the literature and is known as the augmentability or the global variable consensus problem (Hestenes, 1975; Boyd et al., 2011). The quadratic penalty term aims to prevent local workers from converging to distant attractors that deviate significantly from the central worker.

The gradient update rule for EASGD is then given as:

$$x_{k+1}^i = x_k^i - \eta(g_i(x_k^i) + \rho(x_k^i - \widetilde{x}_k)) \tag{4.14}$$

$$\widetilde{x}_{k+1} = \widetilde{x}_k + \eta \sum_{i=1}^{m} \rho(x_k^i - \widetilde{x}_k), \tag{4.15}$$

where $g_i(x_k^i)$ is a stochastic gradient of $F$ with respect to $x^i$ at iteration $k$, such that $\mathbb{E}[g_i(x_k^i)] = \nabla F(x_k^i)$, $\eta$ is the learning rate, $x_k^i$ and $\widetilde{x}_k$ are the parameters of the local worker $i$ and the centralized server at iteration $k$, respectively. Denote $\alpha = \eta\rho$ and $\beta = m\alpha$, the update rule in (4.14-4.15) could be rewitten as:

$$x_{k+1}^i = x_k^i - \eta g^i(x_k^i) - \alpha(x_k^i - \widetilde{x}_k) \tag{4.16}$$

$$\widetilde{x}_{k+1} = (1 - \beta)\widetilde{x}_k + \beta \left( \frac{1}{m} \sum_{i=1}^{m} x_k^i \right) \tag{4.17}$$

The term $-\alpha(x_k^i - \widetilde{x})$ in (4.16) is the elastic force between the local variable $x^i$ and center variable $\widetilde{x}$ that prevents the local worker to drift too far away from the center server. Equation (4.17) shows that the center

variable $\widetilde{x}$ is updated as the moving average of local workers $x^i$ where the average is taken both in time and space.

We then move into the detailed theoretical analysis of EASGD based on the mathematical formulation introduced above. The proof is motivated by the theoretical analysis in (Teng et al., 2019).

**Convex Setting.** We first show that EASGD reaches the same convergence rate as SGD under the stronly-convex assumption.

**Assumption 52** (Convex Setting). *We assume that the loss function $F = \mathbb{E}_{\xi \sim D}[f(x;\xi)]$ and stochastic gradient $g_i(x_k^i)$ satisfy the following conditions:*

(1) *Lipschitz gradient (L-smooth): $\|\nabla F(x) - \nabla F(y)\| \le L \|x - y\|$*

(2) *$\mu$-strongly convex: $F(y) \ge F(x) + \nabla F(x)^\top (x - y) + \frac{\mu}{2} \|x - y\|^2$*

(3) *Unbiased gradient: $\mathbb{E}[g_i(x_k^i)] = \nabla F(x_k^i)$*

(4) *Bounded variance: $\mathbb{E}[\|g_i(x_k^i) - \nabla F(x_k^i)^2\|] \le \sigma^2$.*

**Theorem 53** (Convergence of EASGD; Convex Setting). *Let $\delta_k^i = x_k^i - \widetilde{x}_k$. Under Assumption 52, when the learning rate satisfies $\eta < \frac{1}{2L}$ and $\eta\rho < \frac{\mu}{2L}$, the update rule (4.16-4.17) for EASGD satisefies:*

$$\mathbb{E}[F(x_{k+1}^i)] \le F(x_k^i) - \eta(1 - \eta L) \left\|\nabla F(x_k^i)\right\|^2 - \eta\rho(\frac{\mu}{2} - \eta\rho L) \left\|\delta_k^i\right\|^2$$

$$- \eta\rho(F(x_k^i) - F(\widetilde{x}_k)) + \frac{\eta^2}{2} L\sigma^2.$$

*Note the presence of the new term $-\eta\rho(F(x_k^i) - F(\widetilde{x}_k))$, which speeds up convergence when $F(\widetilde{x_k}) \le F(x_k^i)$, i.e., is better than local workers. If the center parameter $\widetilde{x_k}$ is always chosen so that $F(\widetilde{x_k}) \le F(x_k^i)$ at every step $k$ and the learning rate $\eta$ decrease as $\eta_k = \frac{1}{\sqrt{k}}$, then $\mathbb{E}[F(x_{k+1})] - F(x^*) \le \mathcal{O}(\frac{1}{k})$. In other words, EASGD converges sublinearly.*

*Proof.* Assume $\delta_k^i = x_k^i - \widetilde{x}_k$. From the update rule in (4.16-4.17) we have

$$F(x_{k+1}^i) = F(x_k^i) - \eta\langle\nabla F(x_k^i), g_i(x_k^i) + \rho\delta_k^i\rangle + \frac{\eta^2}{2} L \left\|g_i(x_k^i) + \rho\delta_k^i\right\|^2$$

$$= F(x_k^i) - \eta\langle\nabla F(x_k^i), g_i(x_k^i) + \rho\delta_k^i\rangle + \frac{\eta^2}{2} L \left\|\nabla F(x_k^i) + \rho\delta_k^i + g_i(x_k^i) - \nabla F(x_k^i)\right\|^2$$

$$\le F(x_k^i) - \eta\langle\nabla F(x_k^i), g_i(x_k^i) + \rho\delta_k^i\rangle + \frac{\eta^2}{2} L(\left\|\nabla F(x_k^i) + \rho\delta_k^i\right\|^2 + \left\|g_i(x_k^i) - \nabla F(x_k^i)\right\|^2).$$

Taking the expectation on both sides:

$$\mathbb{E}[F(x_{k+1}^i)] \le F(x_k^i) - \eta \left\|\nabla F(x_k^i)\right\|^2 - \eta\rho\nabla F(x_k^i)^\top \delta_k^i + \frac{\eta^2}{2} L \left\|\nabla F(x_k^i) + \rho\delta_k^i\right\|^2 + \frac{\eta^2}{2} L\sigma^2$$

$$\le F(x_k^i) - \eta(1 - \eta L) \left\|\nabla F(x_k^i)\right\|^2 + \frac{\eta^2}{2} L\sigma^2 - \eta\rho(\frac{\mu}{2} - \eta\rho L) \left\|\delta_k^i\right\|^2$$

$$- \eta\rho(F(x_k^i) - F(\widetilde{x}_k)).$$

Since $\eta < \frac{1}{2L}$ and $\eta\rho < \frac{\mu}{2L}$, we have:

$$\mathbb{E}[F(x_{k+1}^i)] \le F(x_k^i) - \frac{\eta}{2} \left\|\nabla F(x_k^i)\right\|^2 - \eta\rho(F(x_k^i) - F(\widetilde{x}_k)) + \frac{\eta^2}{2} L\sigma^2. \tag{4.18}$$

Let $x^*$ be the minimizer of function $F$, by Polyak-Łojasiewicz inequality for $\mu$-strongl convex function $F$ we have:

$$F(x) - F(x^*) \le \frac{1}{2\mu} \left\|\nabla F(x)\right\|^2, \forall x. \tag{4.19}$$

Combine (4.18) with (4.19) to obtain

$$\mathbb{E}[F(x_{k+1}^i)] \leq F(x_k^i) - \eta\mu(F(x) - F(x^*)) - \eta\rho(F(x_k^i) - F(\widetilde{x}_k)) + \frac{\eta^2}{2}L\sigma^2. \tag{4.20}$$

Substract $F(x^*)$ on both sides and proceed:

$$\mathbb{E}[F(x_{k+1}^i)] - F(x^*) \leq (1 - \eta\mu)(F(x) - F(x^*)) - \eta\rho(F(x_k^i) - F(\widetilde{x}_k)) + \frac{\eta^2}{2}L\sigma^2. \tag{4.21}$$

$\square$

**Nonconvex Setting.** We also show that EASGD converges sublinearly under the nonconvex assumptions.

**Assumption 54** (Nonconvex Setting.). *We assume that the loss function $F = \mathbb{E}_{\xi \sim D}[f(x;\xi)]$ and stochastic gradient $g_i(x_k^i)$ satisfy the following conditions:*

(1) *Lipschitz gradient (L-smooth):* $\|\nabla F(x) - \nabla F(y)\| \leq L\|x - y\|$

(2) *Unbiased gradient:* $\mathbb{E}[g_i(x_k^i)] = \nabla F(x_k^i)$

(3) *Bounded variance:* $\mathbb{E}[\|g_i(x_k^i) - \nabla F(x_k^i)^2\|] \leq \sigma^2$

(4) *Bounded domain:* $\|x_k^i - \widetilde{x}_k\|^2 \leq \delta^2$.

**Theorem 55** (Convergence of EASGD; Nonconvex Setting). *Under Assumption 54, when the learning rate $\eta$ and elastic force $\rho$ satisfy the condition: $2\eta L + \rho < 1$, the update rule 4.2 of EASGD satisfies:*

$$\frac{1}{K}\sum_{i=0}^{K-1}\mathbb{E}[\|\nabla F(x_k^i)\|^2] \leq \frac{2\mathbb{E}[F(x_0^i) - F(x_K^i)]}{\eta K} + 2\eta\rho L\delta^2 + \rho\delta^2 + \eta L\sigma^2.$$

*If the learning rate $\eta$ and elastic force $\rho$ decreasse as $\eta_k = \rho_k = \frac{1}{\sqrt{k}}$, $\frac{1}{K}\sum_{i=0}^{K-1}\|\nabla F(x_k^i)\|^2 \leq \mathcal{O}(\frac{1}{\sqrt{K}})$. In other words, Elastic SGD converges sublinearly.*

*Proof.* Similarly to the proof for convex setting we start as follows:

$$\mathbb{E}[F(x_{k+1}^i) - F(x_k^i)]$$

$$\leq -\eta(1 - \eta L)\mathbb{E}[\|\nabla F(x_k^i)\|^2] + \eta^2\rho L\|\delta_k^i\|^2 - \eta\rho\mathbb{E}[\nabla F(x_k^i)^\top\delta_k^i] + \frac{\eta^2}{2}L\sigma^2. \tag{4.22}$$

We are going to derive a bound for $-\mathbb{E}[\nabla F(x_k^i)^\top\delta_k^i]$:

$$-\mathbb{E}[\nabla F(x_k^i)^\top\delta_k^i] = \frac{1}{2}\mathbb{E}\left[\|\nabla F(x_k^i)\|^2 + \|\delta_k^i\|^2 - \|\nabla F(x_k^i) + \delta_k^i\|^2\right]$$

$$\leq \frac{1}{2}\mathbb{E}\left[\|\nabla F(x_k^i)\|^2 + \|\delta_k^i\|^2\right]$$

$$\leq \frac{1}{2}\mathbb{E}[\|\nabla F(x_k^i)\|]^2 + \frac{1}{2}\|\delta_k^i\|^2. \tag{4.23}$$

Therefore, combining (4.22) with (4.23) gives

$$\mathbb{E}[F(x_{k+1}^i) - F(x_k^i)] \leq -\eta(1 - \eta L - \frac{\rho}{2})\mathbb{E}[\|\nabla F(x_k^i)\|^2] + \frac{\eta\rho}{2}(2\eta L + 1)\|\delta_k^i\|^2 + \frac{\eta^2 L\sigma^2}{2}. \tag{4.24}$$

When $2\eta L + \rho < 1$, we have

$$\mathbb{E}[\|\nabla F(x_k^i)\|^2] \leq \frac{2\mathbb{E}[F(x_{k+1}^i) - F(x_k^i)]}{\eta} + 2\eta\rho L\delta^2 + \rho\delta^2 + \eta L\sigma^2. \tag{4.25}$$

Sum the inequality (4.25) from $k = 0, ...., K - 1$ and divide by $K$ to obtain:

$$\frac{1}{K} \sum_{i=0}^{K-1} \mathbb{E}[\|\nabla F(x_k^i)\|^2] \leq \frac{2\mathbb{E}[F(x_0^i) - F(x_K^i)]}{\eta K} + 2\eta\rho L\delta^2 + \rho\delta^2 + \eta L\sigma^2 \tag{4.26}$$

If $\eta = \frac{1}{\sqrt{K}}$ and $\rho = \frac{1}{\sqrt{K}}$, we have

$$\frac{1}{K} \sum_{i=0}^{K-1} \mathbb{E}[\|\nabla F(x_k^i)\|^2] \leq \mathcal{O}(\frac{1}{\sqrt{K}}). \tag{4.27}$$

$\square$

### 4.1.3 Leader Stochastic Gradient Descent (LSGD)

(Teng et al., 2019) improves EASGD and relies on the framework consisting of multiple leaders. $\widetilde{x}$ in EASGD is close to the average of the local workers. However, the average is a poor solution for the whole system when the local workers converges to different local optimizers. In order to overcome the disadvantage, $\widetilde{x}$ in LSGD is chosen to be the best performer among all local workers of current training step. Except for a single center parameter introduced by EASGD, LSGD introduces multiple local leaders for each group of local workers, and pulls all the local workers towards the current local leader as well as the global leader to ensure fast convergence. The multi-leader setting is well-aligned with the current hardware architecture, where the local workers forming a group lie within a single computational node and different groups correspond to different nodes.

The problem formulation for LSGD is next crystalized. Consider minimizing a function $F(x) = \mathbb{E}_{\xi \in D}[f(x, \xi)]$ in parallel computing environment, where $x$ corresponds to the model parameters and $\xi$ is a random variable sampled from distribution $D$. Firstly, recall the optimization problem formulation introduced in EASGD:

$$\min_{x^1, ..., x^l, \widetilde{x}} \sum_{i=1}^{l} \mathbb{E}[f(x^i, \xi^i)] + \frac{\lambda}{2} \|x^i - \widetilde{x}\|^2, \tag{4.28}$$

where $l$ is the number of workers, $\xi^i$s are samples from distribution $D$, $x^i$ is the local set of parameters processed by the $i^{\text{th}}$ worker, and $\widetilde{x}$ is the center parameter. In EASGD, $\widetilde{x}$ is the average of the local workers. However, the average is a poor solution for the whole system when the local workers converges to different local optimizers. In order to overcome the disadvantage, In LSGD, we define $\widetilde{x}$ as the best performing worker: $\widetilde{x} := \arg\min_{x^1, ..., x^l} \mathbb{E}[f(x^i, \xi^i)]$ and use the best performing worker $\widetilde{x}$ as the single leader of the system.

Formulation (4.28) could be extended to the multi-leader setting:

$$\min_{x^{1,1}, x^{1,2}, ..., x^{n,l}, \widetilde{x}} \sum_{j=1}^{n} \sum_{i=1}^{l} \mathbb{E}[f(x^{j,i}, \xi^{j,i})] + \frac{\lambda}{2} \|x^{j,i} - \widetilde{x}^j\|^2 + \frac{\lambda_G}{2} \|x^{j,i} - \widetilde{x}\|^2, \tag{4.29}$$

where $n$ is the number of groups, $l$ is the number of workers in each group, $\xi^{j,i}$s are samples from distribution $D$, and $\widetilde{x}^j$ and $\widetilde{x}$ is the local leader and global leader, respectively, i.e. $\widetilde{x}^j = \arg\min_{x^{j,1}, ..., x^{j,i}} \mathbb{E}[f(x^{j,i}, \xi^{j,i})]$ and $\widetilde{x} = \arg\min_{x^{1,1}, ..., x^{n,l}} \mathbb{E}[f(x^{j,i}, \xi^{j,i})]$.

The update rule for LSGD is:

$$x_{k+1}^{j,i} = x_t^{j,i} - \eta g_t^{j,i}(x_t^{j,i}) - \lambda(x_t^{j,i} - \widetilde{x}_t^j) - \lambda_G(x_t^{j,i} - \widetilde{x}_t), \tag{4.30}$$
$$\widetilde{x}^j = \arg\min_{x^{j,1}, ..., x^{j,i}} \mathbb{E}[f(x^{j,i}, \xi^{j,i})], \quad \widetilde{x} = \arg\min_{x^{1,1}, ..., x^{n,l}} \mathbb{E}[f(x^{j,i}, \xi^{j,i})],$$

where $g_t^{j,i}$ denotes the stochastic gradient of $\mathbb{E}[f(x^{j,i}, \xi^{j,i})]$.

From previous analysis, we know that the only difference between the formulation of EASGD and LSGD is that: EASGD only has a single center parameter $\widetilde{x}$ while LSGD has multiple leaders $\{\widetilde{x}^j\}$. We are going

to show The objective function of LSGD with multiple-leaders could be rewritten as the form of objective function of EASGD with a single center parameter.

From formulation 4.29, the objective function for multiple-leaders of LSGD is of the form:

$$f(x) + \frac{\lambda_1}{2} \|x - z_1\|^2 + \frac{\lambda_2}{2} \|x - z_2\|^2 + ... + \frac{\lambda_c}{2} \|x - z_c\|^2, \tag{4.31}$$

where $z_i$ represents the local leaders and the global leader in LSGD generally. Formula (4.31) could be rewritten as:

$$f(x) + \frac{\Lambda}{2} \|x - \widetilde{z}\|^2, \tag{4.32}$$

where $\Lambda = \sum_{i=1}^{c} \lambda_i$ and $\widetilde{z} = \frac{1}{\Lambda} \sum_{i=1}^{c} \lambda_i z_i$. By comparing the problem formulation of EASGD (Equation 4.28) with the reformulated objective function of LSGD (Equation 4.32), we observe that LSGD's objective function is structurally the same as that of EASGD, differing only in the choice of certain constants. Since we have already established the convergence guarantee for EASGD with a single center worker, this proof can be readily adapted to LSGD by modifying these constants. Therefore, we omit the proof of the convergence guarantee for LSGD in this manuscript.

### 4.1.4 Gradient-based Weighted Averaging (GRAWA)

Gradient-based Weighted Averaging (GRAWA) (Dimlioglu & Choromanska, 2024) considers the problem of minimizing a loss function $F$ with respect to model parameters $x$ over a large data set $D$. In a parallel computing environment with $m$ workers $(x^1, x^2, ..., x^m)$, this optimization problem can be written as:

$$\min_{x^1,...,x^m} \sum_{i=1}^{m} \mathbb{E}_{\xi \sim \mathcal{D}_i} f(x^i; \xi) + \frac{\lambda}{2} \|x^i - x^c\|^2, \tag{4.33}$$

where $D$ is partitioned and distributed across $m$ workers, $D_i$ is the data distribution exclusively seen by worker $i$. $\widetilde{x}$ is the center variable (for example, an average of all the workers), and $x^i$ stands for the parameters of model $i$.

Motivated by EASGD, GRAWA applies a pulling force on all workers towards the center worker. The penalty term $\frac{\lambda}{2} \|x^i - \widetilde{x}\|^2$ in the problem formulation (4.33) introduces the pulling force. However, differently from EASGD which applies the $1/m$ equal average, GRAWA takes the weighted average among local workers based on the gradients. Define the gradient vector for model $i$ as $A_i$. More specifically, the weights are calculated as follows ($\beta_i$ is the weight for worker $i$):

$$\beta_i \propto \frac{1}{\|A_i\|_2}, \quad \sum_{i=1}^{m} \beta^i = 1 \quad \Longrightarrow \quad \beta^i = \frac{\Theta}{\|\nabla f(x^i)\|},$$
$$\text{where} \quad \Theta = \frac{\prod_{i=1}^{m} \|\nabla f(x^i)\|}{\sum_{i=1}^{m} \frac{\prod_{j=1}^{m} \|\nabla f(x^j)\|}{\|\nabla f(x^i)\|}}. \tag{4.34}$$

The update rule for GRAWA is:

$$x_{k+1}^i = x_k^i - \eta(g^i(x_k^i) + \lambda(x_k^i - \widetilde{x_k})) \tag{4.35}$$

$$x_{k+1}^c = (1 - \beta_k^i)\widetilde{x_k} + \beta_k^i \left( \frac{1}{m} \sum_{i=1}^{m} x_k^i \right), \tag{4.36}$$

where $\beta_k^i$ is computed through Equation (4.34) defined above. We then move into the detailed theoretical analysis of GRAWA based on the mathematical formulation introduced above.

**Convex Setting.** We first show the GRAWA reaches sublinear convergene rate under the strongly-convex assumption.

**Assumption 56** (Convex Setting). *We assume that the loss function $F = \mathbb{E}_{\xi \sim D}[f(x; \xi)]$ with the minimum value $x^*$ and stochastic gradient $g_i(x_k^i)$ satisfy the following conditions:*

*(1) Lipschitz gradient (L-smooth): $\|\nabla F(x) - \nabla F(y)\| \leq L \|x - y\|$*

*(2) m-strongly convex: $F(y) \geq F(x) + \nabla F(x)^\top (x - y) + \frac{m}{2} \|x - y\|^2$*

*(3) v-cone: $|F(x) - F(x^*)| \geq v\|x - x^*\|$ i.e. lower bounded by a cone which has a tip at $x^*$ and a slope $k$*

*(4) $\mu$-sPL: $\|\nabla F(x)\| \geq \mu (F(x) - F(x^*))$*

*(5) Unbiased gradient: $\mathbb{E}[g_i(x_k^i)] = \nabla F(x_k^i)$*

*(6) Bounded variance: $\mathbb{E}[\|g_i(x_k^i) - \nabla F(x_k^i)^2\|] \leq \sigma^2$.*

**Theorem 57.** *(Theorem 1 in (Dimlioglu & Choromanska, 2024)) Let $x_k^c = \sum_{i=1}^m \beta_k^i x_k^i$ and $\beta_k^i$'s are calculated as in Equation 4.34. Under the Assumption 56 for the loss function $F$, the GRAWA center variable holds the following property: $F(x_k^c) \leq f(x_k^i)$ for all $i \in 1, 2, ..., m$ when $\mu v \geq L\sqrt{M}$.*

*Proof.* For notation simplicity, we omit the subscript $k$ in this proof. We denote $x_k^i$, $x_k^c$ and $\beta_k^i$ as $x^i$, $x^c$ and $\beta^i$ repectively.

Since $x^c = \sum_{i=1}^m \beta^i x^i$, where $\sum_{i=1}^m \beta^i = 1$, and $F$ is convex:

$$F(x^c) = F(\sum_{i=1}^m \beta^i x^i) \leq \sum_{i=1}^m \beta^i F(x^i). \tag{4.37}$$

By the definition of $\beta^i$ in Equations (4.34) we have

$$\|\nabla F(x^c)\|^2 \leq \sum_{i=1}^m \frac{\Theta^2}{\|\nabla F(x^i)\|^2} \|\nabla F(x^i)\|^2 \leq M\Theta^2. \tag{4.38}$$

Therefore, we have

$$\|\nabla F(x^c)\| \leq \sqrt{M}\Theta. \tag{4.39}$$

Since $\beta^i = \frac{\Theta}{\|\nabla F(x^i)\|} \leq 1$, we have $\Theta \leq \|\nabla F(x^i)\|$ for all $i = 1, 2, .., m$. Therefore:

$$\|\nabla F(x^c)\| \leq \sqrt{M} \|\nabla F(x^i)\|, \quad \forall i = 1, .., m. \tag{4.40}$$

Combine (4.40) with Assumption 56 for $F$ ($v$-cone and $L$-smooth, where $x^*$ is the minimum value):

$$F(x^i) - F(x^*) \geq v \|x^i - x^*\| \geq \frac{v}{L} \|\nabla F(x^i)\| \geq \frac{v}{\sqrt{M}L} \|\nabla F(x^c)\|.$$

Finally utilizing $\mu$-sPL assumption for $F$, we have

$$F(x^i) - F(x^*) \geq \frac{\mu v}{\sqrt{M}L}(x^c - x^*).$$

When $k\mu \geq L\sqrt{M}$, we can write: $F(x^i) \geq F(x^c)$. $\qquad \square$

**Theorem 58.** *(Convergence of GRAWA; Convex Setting; Theorem 2 in (Dimlioglu & Choromanska, 2024)) Let $\delta_k^i = x_k^i - \widetilde{x}_k$. Under the Assumption 56, when the learning rate satisfies $\eta < \frac{1}{2L}$ and $\eta\lambda < \frac{m}{2L}$, the update rule (4.35-4.36) for GRAWA satisefies:*

$$\mathbb{E}[F(x_{k+1}^i)] \leq F(x_k^i) - \eta(1 - \eta L) \|\nabla F(x_k^i)\|^2 - \eta\lambda(\frac{m}{2} - \eta\lambda L) \|\delta_k^i\|^2$$

$$- \eta\lambda(F(x_k^i) - F(x_k^c)) + \frac{\eta^2}{2} L\sigma^2.$$

*Note the presence of the new term $-\eta\lambda(F(x_k^i) - F(x_k^c))$, which speeds up convergence since $F(x_k^i) > F(x_k^c)$ is guaranteed by Theorem 57 when $\mu v \leq L\sqrt{M}$.*

*If the learning rate $\eta$ decreases as $\eta_k = \frac{1}{\sqrt{k}}$, then $\mathbb{E}[F(x_{k+1})] - F(x^\star) \leq \mathcal{O}(\frac{1}{k})$. In other words, GRAWA shows sublinear convergence rate.*

*Proof.* Since we already prove that $F(x_k^c) \leq f(x_k^i)$ for all $i \in 1, 2, ..., m$ under Assumption 56 in Theorem 57. The Proof scheme for Theorem 58 is exactly the same as that of EASGD based on the verified inequality. $\square$

**Nonconvex Setting.** The proof for GRAWA under nonconvex setting is exactly the same as that of EASGD if we assume the distance between local workers and the center worker is always bounded throughout the training process, i.e $\left\|x_k^i - x_k^c\right\| \leq \delta^2$.

## 4.2 Decentralized Methods

Differently from the centralized methods, decentralized methods eliminate the need for a central worker, distributing tasks among multiple workers that communicate according to a certain topology. The decentralized communication topology effectively addresses the communication bottleneck encountered in centralized methods, particularly in scenarios involving a large number of workers or limited network bandwidth within the framework, since we no longer need to send local parameters from all local workers to a single processing unit.

Before moving into the detailed discussion of different decentralized methods, we consider the problem formulation for decentralized distributed optimization. Assume that there are $m$ workers and each dataset satisfies distribution $D_i$. The model parameter is denoted as $x$ where $x \in \mathbb{R}^d$. The purpose of distributed optimization is to optimize the following problem:

$$\min_{x \in R^d} F(x) = \min_{x \in R^d} \sum_{i=1}^m F_i(x)$$
$$= \min_{x \in R^d} \frac{1}{m} \sum_{i=1}^m \mathbb{E}_{\xi \sim D_i}[f(x; \xi)], \tag{4.41}$$

where $f(x; \xi)$ is the point-wise loss function and $F_i(x) = \mathbb{E}_{\xi \sim D_i}[f(x; \xi)]$ is the local objective function optimized by the $i$-th worker. We are going to stick to this problem formulation in this section.

### 4.2.1 Decentralized Parallel SGD (D-PSGD)

Firstly, we introduce the Decentralized Parallel SGD (D-PSGD) method (Lian et al., 2017; 2018). The communication topology of D-PSGD is given by an undirected weighted graph $G = (V, \mathbf{W})$, where $V \coloneqq \{1, 2, ..., m\}$ denotes the set of $m$ workers, and $\mathbf{W} = (W_{ij})_{m \times m}$ represents the weight matrix of edges in the graph.

Within D-PSGD, each worker performs a local gradient update and subsequently averages its parameters with those of its neighboring workers, as determined by the weight matrix $W$. The weight $W_{ij}$ determines the influence of node $j$ on node $i$. The weight $W_{ij}$ should always be bounded in the range $[0, 1]$ and $\sum_{j=1}^m W_{ij} = 1$ for all $i$. $W_{ij} = 0$ indicates nodes $i$ and $j$ are not connected. The weight matrix $W \in \mathbb{R}^{m \times m}$ is a symmetric doubly stochastic matrix, which means: **(i)** $W_{ij} \in [0, 1], \forall i, j$; **(ii)** $W_{ij} = W_{ji}, \forall i, j$; **(iii)** $\sum_{j=1}^m W_{ij} = 1, \forall i$.

The gradient update rule for D-PSGD is:

$$x_{k+1,i} = \sum_{j=1}^m W_{ij} \left[x_{k,j} - \eta g_i(x_{k,i}; \xi_{k,i})\right], \tag{4.42}$$

where $\eta$ denotes the learning rate and $x_{k,i}$ denotes the model parameters of worker $i$ at iteration $k$. $g_i(x_{k,i}; \xi_{k,i})$ denotes the stochastic gradient of local loss $F_i$ with respect to the local parameters of worker $i$ at iteration $k$, such that $\mathbb{E}_{\xi_i}[g_i(x_{k,i}; \xi_{k,i})] = \nabla F_i(x_k)$.

We then move into the detailed proof of convergence based on the mathematical formulation introduced above. We focus on the non-convex setting. The proof is based on (Lian et al., 2017), it is however reorganized.

**Convergence Analysis.**  This section provides the analysis of the convergence of the D-PSGD algorithm. We define the average iterate at step $k$ as $\overline{x}_k = \frac{1}{m} \sum_{i=1}^{m} x_{k,i}$, and the minimum of the loss function as $F^*$. This section demonstrates that under certain assumptions, the averaged gradient norm $\frac{1}{K} \sum_{k=1}^{K} \mathbb{E}[\|\nabla F(\overline{x}_k)\|]$ converges to zero with sublinear convergence rate.

Before moving into the details, we first introduce the following Lemma 59 that the doubly stochastic matrix $\mathbf{W}$ satisfied. Lemma 59 will play a vital role in the convergence proof later on.

**Lemma 59.** *If $\mathbf{W}$ is a symmetric doubly stochastic matrix with the spectral norm $\rho = (\max\{|\lambda_2(\mathbf{W})|, |\lambda_m(\mathbf{W})|\})^2 < 1$, then for $\mathbf{J} = \mathbf{1}\mathbf{1}^\top/m$ and arbitrary matrix $\mathbf{B}$,*

$$\left\|\mathbf{B}\left(\mathbf{W}^k - \mathbf{J}\right)\right\|_F^2 \le \rho^k \|\mathbf{B}\|_F^2.$$

*Proof.* Since $\mathbf{W}$ is symmetric doubly stochastic and $\mathbf{J} = \mathbf{1}\mathbf{1}^\top/m$, we have $\mathbf{W}\mathbf{J} = \mathbf{J}$ and $\mathbf{J}^2 = \mathbf{J}$. Therefore,

$$\mathbf{W}^k - \mathbf{J} = \mathbf{W}(\mathbf{W}^{k-1} - \mathbf{J}). \tag{4.43}$$

Therefore, we have

$$\begin{aligned}
\left\|\mathbf{B}\left(\mathbf{W}^k - \mathbf{J}\right)\right\|_F^2 &= \sum_{i=1}^{d} \left\|\mathbf{b}_i^\top(\mathbf{W}^k - \mathbf{J})\right\|^2 \\
&= \sum_{i=1}^{d} \mathbf{b}_i^\top(\mathbf{W}^{k-1} - \mathbf{J})\mathbf{W}^2(\mathbf{W}^{k-1} - \mathbf{J})\mathbf{b}_i \\
&\le \sigma_{\max}(W^2) \sum_{i=1}^{d} \mathbf{b}_i^\top(\mathbf{W}^{k-1} - \mathbf{J})(\mathbf{W}^{k-1} - \mathbf{J})\mathbf{b}_i \\
&\le \rho \left\|\mathbf{B}\left(\mathbf{W}^{k-1} - \mathbf{J}\right)\right\|_F^2 \\
&\qquad \dots \\
&\le \rho^k \|\mathbf{B}\|_F^2.
\end{aligned}$$

$\qquad\qquad\qquad\qquad\qquad\qquad\qquad\qquad\qquad\qquad\qquad\qquad\qquad\qquad\qquad\qquad\qquad\qquad\qquad\qquad\qquad\qquad\qquad\square$

Then we are going to move to the main Theorem for the convergence analysis.

**Assumption 60** (Nonconvex Setting). *We assume that the loss function $F(x) = \sum_{i=1}^{m} F_i(x)$ satisfies the following conditions:*

*(1) Lipschitz gradient: $\|\nabla F_i(x) - \nabla F_i(y)\| \le L \|x - y\|$*

*(2) Unbiased gradient: $\mathbb{E}_{\xi_i}[g_i(x_{k,i}; \xi_{k,i})] = \nabla F_i(x_{k,i})$*

*(3) Bounded variance: $\mathbb{E}_{\xi_i}[\|g_i(x_{k,i}; \xi_{k,i}) - \nabla F_i(x_{k,i})\|^2] \le \sigma^2$*

*(4) Unified gradient: $\mathbb{E}_{\xi_i}[\|\nabla F_i(x) - \nabla F_i(x)\|^2] \le \zeta^2$.*

**Theorem 61** (Convergence of D-PSGD; Nonconvex Setting). *Suppose all local workers are initialized with the same point $\overline{x}_1$ and $\mathbf{W}$ is the symmetric doubly stochastic weight matrix used for the D-PSGD update rule. Define the spectral norm of $\mathbf{W}$ as $\rho = (\max\{|\lambda_2(\mathbf{W})|, |\lambda_m(\mathbf{W})|\})^2$. Under Assumption 60, if $\rho < 1$ and $\eta L \le \min\{1, (\sqrt{\rho^{-1} - 1})/4\}$, then after $K$ iterations:*

$$\frac{1}{K} \sum_{i=1}^{K} \mathbb{E}\left[\|\nabla F(\overline{x}_k)\|^2\right] \le \frac{8(F(\overline{x}_1) - F^*)}{\eta K} + \frac{4\eta L \sigma^2}{m} + \frac{8\eta^2 L^2 \rho}{1 - \sqrt{\rho}}\left(\frac{\sigma^2}{1 + \sqrt{\rho}} + \frac{3\zeta^2}{1 - \sqrt{\rho}}\right).$$

*When setting $\eta = \sqrt{\frac{m}{K}}$, we obtain sublinear convergence rate $\mathcal{O}(\frac{1}{\sqrt{mK}}) + \mathcal{O}(\frac{m}{K})$.*

*Proof.* For notation simplicity, we denote the stochastic gradient $g_i(x_{k,i}; \xi_{k,i})$ as $g_i(x_{k,i})$. We first introduce the following matrix forms for notation simplicity in the proof.

$$x^{(k)} = [x_{k,1}, x_{k,2}, ..., x_{k,m}];$$

$$\mathbf{G}^{(k)} = [g_1(x_{k,1}), g_2(x_{k,2}), ..., g_m(x_{k,m})];$$

$$\mathbf{F}^{(k)} = [\nabla F_1(x_{k,1}), \nabla F_2(x_{k,2}), ..., \nabla F_m(x_{k,m})].$$

The matrix update rule can be written as

$$x^{(k+1)} = (x^{(k)} - \eta \mathbf{G}^{(k)}) \mathbf{W}, \tag{4.44}$$

after taking the average, we have

$$\overline{x}_{k+1} = \overline{x}_k - \frac{\eta}{m} \mathbf{G}^{(k)} \mathbf{1}. \tag{4.45}$$

From Assumption 60, since function $F$ is Lipschitz smooth, we have

$$F(\overline{x}_{k+1}) - F(\overline{x}_k) \le \langle \nabla F(\overline{x}_k), \overline{x}_{k+1} - \overline{x}_k \rangle + \frac{L}{2} \|\overline{x}_{k+1} - \overline{x}_k\|^2. \tag{4.46}$$

Since $\overline{x}_{k+1} = \overline{x}_k - \frac{\eta}{m} \mathbf{G}^{(k)} \mathbf{1}$, we have

$$F(\overline{x}_{k+1}) - F(\overline{x}_k) \le -\eta \langle \nabla F(\overline{x}_k), \frac{\mathbf{G}^{(k)} \mathbf{1}}{m} \rangle + \frac{\eta^2 L}{2} \left\| \frac{\mathbf{G}^{(k)} \mathbf{1}}{m} \right\|^2. \tag{4.47}$$

Taking expectation gives

$$\mathbb{E}[F(\overline{x}_{k+1}) - F(\overline{x}_k)] \le -\eta \langle \nabla F(\overline{x}_k), \frac{\mathbf{F}^{(k)} \mathbf{1}}{m} \rangle + \frac{\eta^2 L}{2} \mathbb{E}\left[ \left\| \frac{\mathbf{G}^{(k)} \mathbf{1}}{m} \right\|^2 \right]. \tag{4.48}$$

We are going to bound the two terms in the RHS of formula 4.79 one by one.

(i) Firstly, we are going to bound $\langle \nabla F(\overline{x}_k), \frac{\mathbf{F}^{(k)} \mathbf{1}}{m} \rangle$.

$$\langle \nabla F(\overline{x}_k), \frac{\mathbf{F}^{(k)} \mathbf{1}}{m} \rangle = \langle \nabla F(\overline{x}_k), \frac{1}{m} \sum_{i=1}^m \nabla F_i(x_{k,i}) \rangle$$

$$= \frac{1}{2} \left[ \|\nabla F(\overline{x}_k)\|^2 + \left\| \frac{1}{m} \sum_{i=1}^m \nabla F_i(x_{k,i}) \right\|^2 - \left\| F(\overline{x}_k) - \frac{1}{m} \sum_{i=1}^m \nabla F_i(x_{k,i}) \right\|^2 \right]$$

$$= \frac{1}{2} \left[ \|\nabla F(\overline{x}_k)\|^2 + \left\| \frac{1}{m} \sum_{i=1}^m \nabla F_i(x_{k,i}) \right\|^2 - \left\| \frac{1}{m} \sum_{i=1}^m [\nabla F_i(\overline{x}_k) - \nabla F_i(x_{k,i})] \right\|^2 \right]$$

$$\ge \frac{1}{2} \left[ \|\nabla F(\overline{x}_k)\|^2 + \left\| \frac{1}{m} \sum_{i=1}^m \nabla F_i(x_{k,i}) \right\|^2 - \frac{1}{m} \sum_{i=1}^m \|\nabla F_i(\overline{x}_k) - \nabla F_i(x_{k,i})\|^2 \right]$$

(By Jensen's Inequlity.)

$$\ge \frac{1}{2} \left[ \|\nabla F(\overline{x}_k)\|^2 + \left\| \frac{1}{m} \sum_{i=1}^m \nabla F_i(x_{k,i}) \right\|^2 - \frac{L^2}{m} \sum_{i=1}^m \|\overline{x}_k - x_{k,i}\|^2 \right]$$

($F$ is L-Lipschitz smooth.)

Define

$$\mathbf{J} = \frac{\mathbf{1}_m}{m}.$$

We can reformulate the above bound as:

$$\langle \nabla F(\overline{x}_k), \frac{\mathbf{F}^{(k)} \mathbf{1}}{m} \rangle \ge \frac{1}{2} \|\nabla F(\overline{x}_k)\|^2 + \frac{1}{2} \left\| \frac{\mathbf{F}^{(k)} \mathbf{1}}{m} \right\|^2 - \frac{L^2}{2m} \|x_k(\mathbf{I} - \mathbf{J})\|_F^2. \tag{4.49}$$

(ii) Secondly, we are going to bound $\mathbb{E}\left[\left\|\frac{\mathbf{G}^{(k)}\mathbf{1}}{m}\right\|^2\right]$.

$$
\begin{aligned}
\mathbb{E}\left[\left\|\frac{\mathbf{G}^{(k)}\mathbf{1}}{m}\right\|^2\right] &= \mathbb{E}\left[\left\|\frac{1}{m}\sum_{i=1}^{m} g_i(x_{k,i})\right\|^2\right]\\
&= \mathbb{E}\left[\left\|\frac{1}{m}\sum_{i=1}^{m}[g_i(x_{k,i}) - \nabla F_i(x_{k,i}) + \nabla F_i(x_{k,i})]\right\|^2\right]\\
&\leq \frac{1}{m^2}\sum_{i=1}^{m}\mathbb{E}\left[\left\|\frac{1}{m}\sum_{i=1}^{m}[g_i(x_{k,i}) - \nabla F_i(x_{k,i})]\right\|^2\right] + \left\|\frac{1}{m}\sum_{i=1}^{m}\nabla F_i(x_{k,i})\right\|^2\\
&\leq \frac{\sigma^2}{m} + \left\|\frac{\mathbf{F}^{(k)}\mathbf{1}}{m}\right\|^2.
\end{aligned}
\tag{4.50}
$$

Combine formula (4.79), (4.80), and (4.81) to obtain

$$
\begin{aligned}
\mathbb{E}[F(\overline{x}_{k+1}) - F(\overline{x}_k)] \leq &-\frac{\eta}{2}\mathbb{E}[\|\nabla F(\overline{x}_k)\|^2] - \frac{\eta}{2}(1-\eta L)\left\|\frac{\mathbf{F}^{(k)}\mathbf{1}}{m}\right\|^2\\
&+ \frac{\eta L^2}{2m}\mathbb{E}\left[\|x_k(\mathbf{I}-\mathbf{J})\|_F^2\right] + \frac{\eta^2 L\sigma^2}{2m}.
\end{aligned}
\tag{4.51}
$$

Averaging over all iterations from $k = 1, ..., K$ we obtain

$$
\begin{aligned}
\frac{\mathbb{E}[F(\overline{x}_K) - F(\overline{x}_1)]}{K} \leq &-\frac{\eta}{2}\frac{1}{K}\sum_{k=1}^{K}\mathbb{E}[\|\nabla F(\overline{x}_k)\|^2] - \frac{\eta}{2}(1-\eta L)\frac{1}{K}\sum_{k=1}^{K}\left\|\frac{\mathbf{F}^{(k)}\mathbf{1}}{m}\right\|^2\\
&+ \frac{1}{K}\sum_{k=1}^{K}\frac{\eta L^2}{2m}\mathbb{E}\left[\|x_k(\mathbf{I}-\mathbf{J})\|_F^2\right] + \frac{\eta^2 L\sigma^2}{2m}.
\end{aligned}
\tag{4.52}
$$

By rearranging, we have

$$
\begin{aligned}
\frac{1}{K}\sum_{k=1}^{K}\mathbb{E}[\|\nabla F(\overline{x}_k)\|^2] \leq &\frac{2\mathbb{E}[F(\overline{x}_1) - F(\overline{x}_K)]}{\eta K} - \frac{1-\eta L}{m}\frac{1}{K}\sum_{k=1}^{K}\left\|\frac{\mathbf{F}^{(k)}\mathbf{1}}{m}\right\|^2\\
&+ \frac{L^2}{mK}\sum_{k=1}^{K}\mathbb{E}\left[\|x_k(\mathbf{I}-\mathbf{J})\|_F^2\right] + \frac{\eta L\sigma^2}{m}\\
\leq &\frac{2\mathbb{E}[F(\overline{x}_1) - F(\overline{x}_K)]}{\eta K} + \frac{L^2}{mK}\sum_{k=1}^{K}\mathbb{E}\left[\|x_k(\mathbf{I}-\mathbf{J})\|_F^2\right] + \frac{\eta L\sigma^2}{m}.
\end{aligned}
\tag{4.53}
$$

We now completed the first part of the proof. In the following section, we are going to bound $\mathbb{E}\left[\|x_k(\mathbf{I}-\mathbf{J})\|_F^2\right]$.

$$
\begin{aligned}
x_k(\mathbf{I}-\mathbf{J}) =&(x_{k-1} - \eta\mathbf{G}^{(k)})\mathbf{W}(\mathbf{I}-\mathbf{J})\\
=&x_{k-1}(\mathbf{I}-\mathbf{J})\mathbf{W} - \eta\mathbf{G}^{(k-1)}\mathbf{W}(\mathbf{I}-\mathbf{J})\\
=&...\\
=&x_1(\mathbf{I}-\mathbf{J})\mathbf{W}^{k-1} - \eta\sum_{q=1}^{k-1}\mathbf{G}^{(q)}\mathbf{W}^{k-q}(\mathbf{I}-\mathbf{J}).
\end{aligned}
\tag{4.54}
$$

Since all workers start from the same point, we have $x_1(\mathbf{I}-\mathbf{J}) = 0$. Moreover, since $\mathbf{W}$ is symmetric doubly stochastic and $\mathbf{J} = \mathbf{1}_m/m$, we know $\mathbf{W}\mathbf{J} = \mathbf{J}$. Thus:

$$\|x_k(\mathbf{I}-\mathbf{J})\|_F^2 = \eta^2 \left\|\sum_{q=1}^{k-1}\mathbf{G}^{(q)}(\mathbf{W}^{k-q}-\mathbf{J})\right\|_F^2$$

$$=\eta^2 \left\|\sum_{q=1}^{k-1}(\mathbf{G}^{(q)}-\nabla\mathbf{F}^{(q)}+\nabla\mathbf{F}^{(q)})(\mathbf{W}^{k-q}-\mathbf{J})\right\|_F^2$$

$$\leq 2\eta^2 \underbrace{\left\|\sum_{q=1}^{k-1}(\mathbf{G}^{(q)}-\nabla\mathbf{F}^{(q)})(\mathbf{W}^{k-q}-\mathbf{J})\right\|_F^2}_{:=I_1} + 2\eta^2 \underbrace{\left\|\sum_{q=1}^{k-1}\nabla\mathbf{F}^{(q)}(\mathbf{W}^{k-q}-\mathbf{J})\right\|_F^2}_{:=I_2}. \tag{4.55}$$

Then we are going to bound terms $(I_1)$ and $(I_2)$ in formula (4.86).

$$\mathbb{E}[I_1] \leq \sum_{q=1}^{k-1}\mathbb{E}\left[\left\|(\mathbf{G}^{(q)}-\nabla\mathbf{F}^{(q)})(\mathbf{W}^{k-q}-\mathbf{J})\right\|_F^2\right]$$

$$\leq \sum_{q=1}^{k-1}\rho^{k-q}\mathbb{E}\left[\left\|\mathbf{G}^{(q)}-\nabla\mathbf{F}^{(q)}\right\|_F^2\right] \quad \text{(Lemma 59)}$$

$$\leq \frac{m\sigma^2\rho}{1-\rho} \quad \text{(Assumption 60: variance bounded).} \tag{4.56}$$

$$\mathbb{E}[I_2] \leq \sum_{q=1}^{k-1}\mathbb{E}\left[\left\|\nabla\mathbf{F}^{(q)}(\mathbf{W}^{k-q}-\mathbf{J})\right\|_F^2\right]$$

$$+ \sum_{q=1}^{k}\sum_{p=1,p\neq q}^{k}\mathbb{E}\left[\left\|\nabla\mathbf{F}^{(q)}(\mathbf{W}^{k-q}-\mathbf{J})\right\|_F\left\|\nabla\mathbf{F}^{(p)}(\mathbf{W}^{k-p}-\mathbf{J})\right\|_F\right]$$

$$\leq \sum_{q=1}^{k-1}\rho^{k-q}\mathbb{E}\left[\left\|\nabla\mathbf{F}^{(q)}\right\|_F^2\right]$$

$$+ \sum_{q=1}^{k}\sum_{p=1,p\neq q}^{k}\mathbb{E}\left[\frac{1}{2\epsilon}\rho^{k-q}\left\|\nabla\mathbf{F}^{(q)}\right\|_F^2 + \frac{\epsilon}{2}\rho^{k-p}\left\|\nabla\mathbf{F}^{(p)}\right\|_F^2\right]$$

(Lemma 59 & Young's inequality).

If $\epsilon = \rho^{\frac{p-q}{2}}$ for the Young's inequality, we have

$$\mathbb{E}[I_2] \leq \sum_{q=1}^{k-1}\rho^{k-q}\mathbb{E}\left[\left\|\nabla\mathbf{F}^{(q)}\right\|_F^2\right]$$

$$+ \sum_{q=1}^{k}\sum_{p=1,p\neq q}^{k}\sqrt{\rho}^{2k-p-q}\mathbb{E}\left[\left\|\nabla\mathbf{F}^{(q)}\right\|_F^2 + \left\|\nabla\mathbf{F}^{(p)}\right\|_F^2\right]$$

$$= \sum_{q=1}^{k-1}\rho^{k-q}\mathbb{E}\left[\left\|\nabla\mathbf{F}^{(q)}\right\|_F^2\right] + \sum_{q=1}^{k}\sqrt{\rho}^{k-q}\mathbb{E}\left[\left\|\nabla\mathbf{F}^{(q)}\right\|_F^2\right]\sum_{p=1,p\neq q}^{k}\sqrt{\rho}^{k-p}$$

$$= \sum_{q=1}^{k-1}\rho^{k-q}\mathbb{E}\left[\left\|\nabla\mathbf{F}^{(q)}\right\|_F^2\right] + \sum_{q=1}^{k}\sqrt{\rho}^{k-q}\mathbb{E}\left[\left\|\nabla\mathbf{F}^{(q)}\right\|_F^2\right](\sum_{p=1}^{k}\sqrt{\rho}^{k-p}-\sqrt{\rho}^{k-q})$$

$$\leq \frac{\sqrt{\rho}}{1-\sqrt{\rho}}\sum_{q=1}^{k-1}\sqrt{\rho}^{k-q}\mathbb{E}\left[\left\|\nabla\mathbf{F}^{(q)}\right\|_F^2\right]. \tag{4.57}$$

After plugging (4.87) and (4.87) back into (4.86), summing the inequality for $k = 1, ..., K$, and then averaging by $mK$, we have

$$\frac{1}{mK} \sum_{k=1}^{K} \mathbb{E}\left[\|x_k(\mathbf{I} - \mathbf{J})\|\right] \leq \frac{2\eta^2 \sigma^2 \rho}{1 - \rho} + \frac{2\eta^2}{m} \frac{\sqrt{\rho}}{1 - \sqrt{\rho}} \frac{1}{K} \sum_{k=1}^{K} \sum_{q=1}^{k} \sqrt{\rho}^{k-q} \mathbb{E}\left[\left\|\nabla \mathbf{F}^{(q)}\right\|_F^2\right]$$

$$= \frac{2\eta^2 \sigma^2 \rho}{1 - \rho} + \frac{2\eta^2}{m} \frac{\sqrt{\rho}}{1 - \sqrt{\rho}} \frac{1}{K} \sum_{k=1}^{K} \mathbb{E}\left[\left\|\nabla \mathbf{F}^{(k)}\right\|_F^2\right] \sum_{q=1}^{k} \sqrt{\rho}^q$$

$$\leq \frac{2\eta^2 \sigma^2 \rho}{1 - \rho} + \frac{2\eta^2}{m} \frac{\rho}{(1 - \sqrt{\rho})^2} \frac{1}{K} \sum_{k=1}^{K} \mathbb{E}\left[\left\|\nabla \mathbf{F}^{(k)}\right\|_F^2\right]. \tag{4.58}$$

Note that:

$$\left\|\nabla \mathbf{F}^{(k)}\right\|_F^2 = \sum_{i=1}^{m} \|\nabla F_i(x_{i,k})\|^2$$

$$= \sum_{i=1}^{m} \|\nabla F_i(x_{i,k}) - \nabla F(x_{i,k}) + \nabla F(x_{i,k}) - \nabla F(\overline{x}_k) + \nabla F(\overline{x}_k)\|^2$$

$$\leq 3 \sum_{i=1}^{m} \left[\|\nabla F_i(x_{i,k}) - \nabla F(x_{i,k})\|^2 + \|\nabla F(x_{i,k}) - \nabla F(\overline{x}_k)\|^2 + \|\nabla F(\overline{x}_k)\|^2\right]$$

$$\leq 3m\zeta^2 + 3L^2 \|x_k(\mathbf{I} - \mathbf{J})\|_F^2 + 3m \|\nabla F(\overline{x}_k)\|^2. \tag{4.59}$$

Plugging (4.90) back into (4.89), we obtain:

$$\frac{1}{mK} \sum_{k=1}^{K} \mathbb{E}\left[\|x_k(\mathbf{I} - \mathbf{J})\|\right] \leq \frac{2\eta^2 \sigma^2 \rho}{1 - \rho} + \frac{6\eta^2 \zeta^2 \rho}{(1 - \sqrt{\rho})^2} + \frac{6\eta^2 L^2 \rho}{(1 - \sqrt{\rho})^2} \frac{1}{mK} \sum_{k=1}^{K} \mathbb{E}\left[\|x_k(\mathbf{I} - \mathbf{J})\|\right]$$

$$+ \frac{6\eta^2 \rho}{(1 - \sqrt{\rho})^2} \frac{1}{K} \sum_{k=1}^{K} \|\nabla F(\overline{x}_k)\|^2. \tag{4.60}$$

Define

$$D = \frac{6\eta^2 L^2 \rho}{(1 - \sqrt{\rho})^2}.$$

After rearranging, we have

$$\frac{1}{mK} \sum_{k=1}^{K} \mathbb{E}\left[\|x_k(\mathbf{I} - \mathbf{J})\|\right] \leq \frac{1}{1 - D} \left[\frac{2\eta^2 \sigma^2 \rho}{1 - \rho} + \frac{6\eta^2 \zeta^2 \rho}{(1 - \sqrt{\rho})^2} + \frac{6\eta^2 \rho}{(1 - \sqrt{\rho})^2} \frac{1}{K} \sum_{k=1}^{K} \|\nabla F(\overline{x}_k)\|^2\right]. \tag{4.61}$$

Plugging the bound (4.92) back into (4.84), we obtain

$$\frac{1}{K} \sum_{k=1}^{K} \mathbb{E}[\|\nabla F(\overline{x}_k)\|^2] \leq \frac{2\mathbb{E}[F(\overline{x}_1) - F(\overline{x}_K)]}{\eta K} + \frac{\eta L \sigma^2}{m}$$

$$+ \frac{1}{1 - D} \frac{2\eta^2 \sigma^2 \rho}{1 - \rho} + \frac{D\zeta^2}{1 - D} + \frac{D}{1 - D} \frac{1}{K} \sum_{k=1}^{K} \|\nabla F(\overline{x}_k)\|^2. \tag{4.62}$$

By rearranging, we have

$$
\begin{aligned}
\frac{1}{K} \sum_{k=1}^{K} \mathbb{E}[\|\nabla F(\overline{x}_k)\|^2] \leq & \frac{1-D}{1-2D} \left[ \frac{2\mathbb{E}[F(\overline{x}_1) - F(\overline{x}_K)]}{\eta K} + \frac{\eta L \sigma^2}{m} \right] \\
& + \frac{1}{1-2D} \frac{2\eta^2 L^2 \rho}{1 - \sqrt{\rho}} \left( \frac{\sigma^2}{1 + \sqrt{\rho}} + \frac{3\zeta^2}{1 - \sqrt{\rho}} \right) \\
\leq & \frac{1}{1-2D} \left[ \frac{2\mathbb{E}[F(\overline{x}_1) - F(\overline{x}_K)]}{\eta K} + \frac{\eta L \sigma^2}{m} \right] \\
& + \frac{1}{1-2D} \frac{2\eta^2 L^2 \rho}{1 - \sqrt{\rho}} \left( \frac{\sigma^2}{1 + \sqrt{\rho}} + \frac{3\zeta^2}{1 - \sqrt{\rho}} \right).
\end{aligned}
\tag{4.63}
$$

Since $\eta L \leq \frac{1-\sqrt{\rho}}{4\sqrt{\rho}}$, we have

$$
D = \frac{6\eta^2 L^2 \rho}{(1 - \sqrt{\rho})^2} \leq \frac{3}{8} < \frac{1}{2} \quad \Longrightarrow \quad \frac{1}{1-2D} \leq 4.
$$

Therefore (4.94) could be bounded as:

$$
\frac{1}{K} \sum_{k=1}^{K} \mathbb{E}[\|\nabla F(\overline{x}_k)\|^2] \leq \frac{8\mathbb{E}[F(\overline{x}_1) - F(\overline{x}_K)]}{\eta K} + \frac{4\eta L \sigma^2}{m} + \frac{8\eta^2 L^2 \rho}{1 - \sqrt{\rho}} \left( \frac{\sigma^2}{1 + \sqrt{\rho}} + \frac{3\zeta^2}{1 - \sqrt{\rho}} \right).
\tag{4.64}
$$

Since $F^*$ is the minimum value of the loss, we have:

$$
\frac{1}{K} \sum_{i=1}^{K} \mathbb{E}\left[ \|\nabla F(\overline{x}_k)\|^2 \right] \leq \frac{8(F(\overline{x}_1) - F^*)}{\eta K} + \frac{4\eta L \sigma^2}{m} + \frac{8\eta^2 L^2 \rho}{1 - \sqrt{\rho}} \left( \frac{\sigma^2}{1 + \sqrt{\rho}} + \frac{3\zeta^2}{1 - \sqrt{\rho}} \right).
\tag{4.65}
$$

If the learning rate is set to be $\eta = \sqrt{\frac{m}{K}}$, we have

$$
\begin{aligned}
\frac{1}{K} \sum_{i=1}^{K} \mathbb{E}\left[ \|\nabla F(\overline{x}_k)\|^2 \right] \leq & \frac{8(F(\overline{x}_1) - F^*) + 4L\sigma^2}{\sqrt{mK}} + \frac{8m}{K} \frac{L^2 \rho}{1 - \sqrt{\rho}} \left( \frac{\sigma^2}{1 + \sqrt{\rho}} + \frac{3\zeta^2}{1 - \sqrt{\rho}} \right) \\
= & \mathcal{O}\left( \frac{1}{\sqrt{mK}} \right) + \mathcal{O}\left( \frac{m}{K} \right).
\end{aligned}
\tag{4.66}
$$

$\square$

### 4.2.2 MATCHA

Since the communication topology in DP-SGD is fixed, the algorithm encounters an error-runtime trade-off issue. A dense topology requires significant communication time per iteration. Conversely, a sparse topology reduces communication time but results in slower convergence per iteration. To address this challenge, (Wang et al., 2019) introduced the MATCHA algorithm. This approach adopts a win-win strategy, enabling both rapid convergence and reduced communication time. MATCHA decomposes the topology into matching components, facilitating parallelization of inter-node communication.

In this section, we introduce the MATCHA technique. Consider a communication topology of $m$ worker nodes. The communication links connecting the nodes are represented by an undirected connected graph $G = (V, E)$, where $V = \{1, 2, ..., m\}$ are the vertices and $E$ is the set of edges ($E \subset V \times V$). The communication graph $G$ could be abstracted as a adjacency matrix $\mathbf{A}$, where $A_{ij} = 1$ means $(i, j) \in E$ and $A_{ij} = 0$ means $(i, j) \notin E$. The Laplacian matrix or $\mathbf{A}$ is defined as: $\mathbf{L} = diag(d_1, ..., d_m) - \mathbf{A}$, where $d_i$ denotes the degree of node $i$.

In order to reduce the frequency of inter-node communication without affecting the convergence speed, MATCHA decompose a dense topology with the matching decomposition and generate a new random

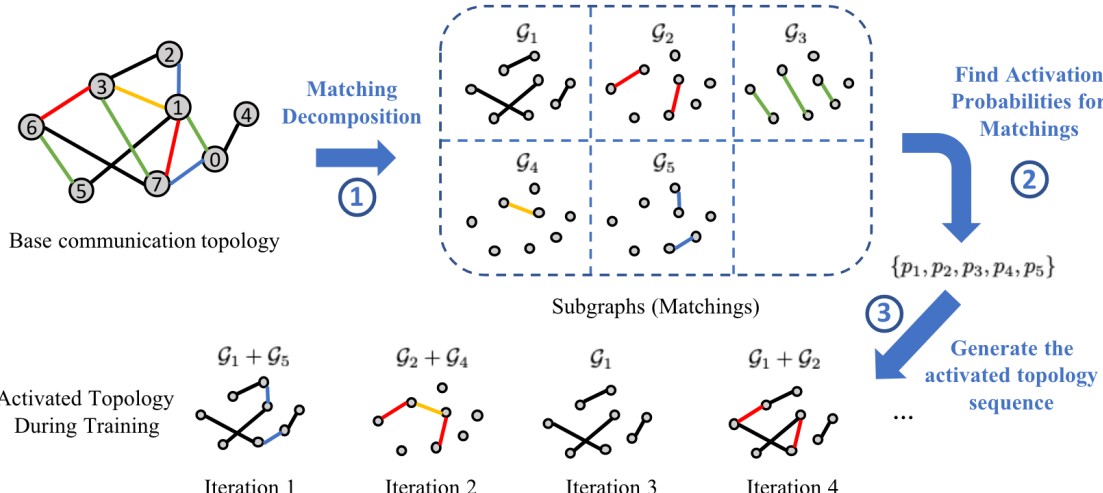

Figure 6: Illustration of MATCHA. This plot is taken from (Wang et al., 2019) (Figure 2).

temporary activated topology for each iteration based on this decomposition. The algorithm also follows the intuition that it is beneficial to communicate over critical links more frequently and less over other links. Figure 6 visualize the process of constructing the temporary activated topology for each iteration. We are going to briefly introduce three steps of MATCHA below.

*Step 1: Matching Decomposition.* The dense base communication topology is decomposed into a total of $M$ disjoint matchings, i.e., $G(V, E) = \bigcup_{j=1}^{M} G_j(V, E_i)$ and $E_i \cap E_j =, \forall i \neq j$. Each matching $G_j$ of graph $G$ is a subgraph where each vertex appears in at most one edge of this subgraph.

*Step 2: Computing Matching Activation Probabilities.* To manage the total communication time per iteration, MATCHA assigns an independent Bernoulli random variable $B_j \sim Bernoulli(p_j)$ to each matching, where $p_j$ denotes the activation probability for matching $G_j$. This approach ensures that each matching is activated with probability $p_j$ and deactivated with probability $1 - p_j$. As a result, the expected communication time for each iteration under this scheme can be expressed as:

$$\text{Expected Comm. Time} = \mathbb{E}\left[\sum_{j=1}^{M} B_j\right] = \sum_{j=1}^{M} p_j.$$

In order to reduce the total communication time, MATCHA defines a communication budget $C_b$ and sets a constraint on the activation probabilities $p_j$ based on the definition of the expected communication time: $\sum_{j=1}^{M} p_j \leq C_b M$.

In order to give more importance to critical links in the communication graph, MATCH maximizes the connectivity of the expected graph by solving the following optimization problem:

$$\begin{aligned}
\max_{p_1, \dots, p_M} \quad & \lambda_2(\sum_{j=1}^{M} p_j \mathbf{L}_j) \\
s.t. \quad & \sum_{j=1}^{M} p_j \leq C_b M \\
& 0 \leq p_j \leq 1, \forall j = 1, \dots, M,
\end{aligned} \tag{4.67}$$

where $\lambda_2$ is the second smallest eigenvalue of the graph Laplacian and formula (4.67) could be generated from the algebraic connectivity of the graph, as interpreted in (Bollobás, 2013).

*Step 3: Generating Random Topology Sequence.* Given the activation probability $p_j$ solved in step 2, MATCHA samples $B_j^{(k)}$ from the Bernoulli distribution $Bernoulli(p_j)$ before $k$-th iteration. $B_j^{(k)}$ controls whether the matching $j$ will be included in the current iteration. The temporary activated topology at $k$-th iteration is $\mathbf{G}^{(k)} = \bigcup_{j=1}^{M} B_j^{(k)} G_j$, which is sparse or even disconnected. The corresponding Laplacian matrix is $\mathbf{L}^{(k)} = \sum_{j=1}^{M} B_j^{(k)} \mathbf{L}_j$. The weight matrix $\mathbf{W}^{(k)}$ for MATCHA at iteration $k$ can be represented as:

$$\mathbf{W}^{(k)} = \mathbf{I} - \alpha \mathbf{L}^{(k)} = \mathbf{I} - \alpha \sum_{j=1}^{M} B_j^{(k)} \mathbf{L}_j. \tag{4.68}$$

The gradient update rule for MATCHA is:

$$x_{k+1,i} = \sum_{j=1}^{m} W_{ij}^{(k)} \left[ x_{k,j} - \eta g_i(x_{k,i}; \xi_{k,i}) \right], \tag{4.69}$$

where $\eta$ denotes the learning rate and $x_{k,i}$ denotes the model parameters of worker $i$ at iteration $k$. $g_i(x_{k,i}; \xi_{k,i})$ denotes the stochastic gradient of local loss $F_i$ with respect to the local parameters of worker $i$ at iteration $k$, such that $\mathbb{E}_{\xi_i}[g_i(x_{k,i}; \xi_{k,i})] = \nabla F_i(x_k)$.

**Convergence proof.** This section presents an analysis of the convergence rate of the MATCHA algorithm. The proof is structured similar to that of DP-SGD. We define the average iterate at step $k$ as $\overline{x}_k = \frac{1}{m} \sum_{i=1}^{m} x_{k,i}$ and the minimum of the loss function as $F^*$. This section demonstrates that under certain assumptions, the averaged gradient norm $\frac{1}{K} \sum_{k=1}^{K} \mathbb{E}[\|\nabla F(\overline{x}_k)\|]$ converges to zero with sublinear convergence rate.

Before moving into the detailed convergence guarantee, we first discuss the property of weight matrix $\mathbf{W}^{(k)}$ (Theorem 62), that will be used in the proof.

**Theorem 62.** *(Theorem 1 in (Wang et al., 2019)) Let $\{\mathbf{L}^{(k)}\}$ denote the sequence of Laplacian matrix generated by MATCHA algorithm with arbitrary communication budget $C_b > 0$ for the the base communication topology $G$. Let the weight matrix $\mathbf{W}^{(k)}$ be defined as in Equation (4.68). There exists a range of $\alpha$ such that the spectral norm $\rho = \left\| \mathbb{E}[\mathbf{W}^{(k)\top}\mathbf{W}^{(k)} - \mathbf{J}] \right\|_2 < 1$, where $\mathbf{J} = \mathbf{1}\mathbf{1}^\top/m$.*

*Proof.* Firstly, we are going to show that the expected activated topology $\sum_{j=1}^{M} p_j \mathbf{L}_j$ is connected. Let $p_0 = C_b$,

$$\lambda_2 \left( \sum_{j=1}^{M} p_j \mathbf{L}_j \right) \geq p_0 \lambda_2 \left( p_0 \sum_{j=1}^{M} \mathbf{L}_j \right) = p_0 \lambda_2 \left( \sum_{j=1}^{M} \mathbf{L}_j \right) > 0.$$

where $p_j$ is the activation probability of matching $G_j$ ($p_0$ is the activation probability of first matching $G_0$). $\lambda_i$ is the i-th smallest eigenvalue of the graph Laplacian $\mathbf{L}$. Therefore, the base communication topology is connected.

$$\rho = \left\| \mathbb{E}[\mathbf{W}^{(k)\top}\mathbf{W}^{(k)} - \mathbf{J}] \right\|_2 = \left\| \mathbb{E}\left[ (\mathbf{I} - \alpha \mathbf{L}^{(k)})^\top (\mathbf{I} - \alpha \mathbf{L}^{(k)}) - \mathbf{J} \right] \right\|$$
$$= \left\| \mathbb{E}\left[ \mathbf{I} - 2\alpha \mathbb{E}\left[ \mathbf{L}^{(k)} \right] + \alpha^2 \mathbb{E}\left[ \mathbf{L}^{(k)\top}\mathbf{L}^{(k)} \right] - \mathbf{J} \right] \right\|, \tag{4.70}$$

where $\mathbf{L}^{(k)} = \sum_{j=1}^{M} B_j^{(k)} \mathbf{L}_j$. Since $B_j^{(k)}$ are i.i.d. across all subgraphs and iterations,

$$\mathbb{E}\left[\mathbf{L}^{(k)}\right] = \sum_{j=1}^{m} p_j \mathbf{L}_j \tag{4.71}$$

$$\begin{aligned}
\mathbb{E}\left[\mathbf{L}^{(k)\top}\mathbf{L}^{(k)}\right] &= \sum_{j=1}^{M} p_j^2 \mathbf{L}_j^2 + \sum_{j=1}^{M} \sum_{t=1, t\neq j}^{M} p_j p_t \mathbf{L}_j^{\top} \mathbf{L}_t + \sum_{j=1}^{M} p_j(1-p_j)\mathbf{L}_j^2 \\
&= \left(\sum_{j=1}^{M} p_j \mathbf{L}_j\right)^2 + \sum_{j=1}^{M} p_j(1-p_j)\mathbf{L}_j^2 \\
&= \left(\sum_{j=1}^{M} p_j \mathbf{L}_j\right)^2 + 2\sum_{j=1}^{M} p_j(1-p_j)\mathbf{L}_j.
\end{aligned} \tag{4.72}$$

Plugging (4.71) and (4.72) back to (4.70), we have

$$\begin{aligned}
\left\|\mathbb{E}\left[\mathbf{W}^{(k)\top}\mathbf{W}^{(k)}\right] - J\right\| &\leq \left\|\left(I - \alpha \sum_{j=1}^{m} p_j \mathbf{L}_j\right)^2 - \mathbf{J}\right\| + 2\alpha^2 \left\|\sum_{j=1}^{M} p_j(1-p_j)\mathbf{L}_j\right\| \\
&= \max\{(1-\alpha\lambda_2)^2, (1-\alpha\lambda_m)^2\} + 2\alpha^2 \zeta,
\end{aligned} \tag{4.73}$$

where $\lambda_l$ denotes the $l$-th smallest eigenvalue of matrix $\sum_{j=1}^{m} p_j \mathbf{L}_j$ and $\zeta > 0$ denotes the spectral norm of matrix $\sum_{j=1}^{m} p_j(1-p_j)\mathbf{L}_j$. Suppose $h_\lambda(\alpha) = (1-\alpha\lambda)^2 + 2\alpha^2\zeta$. Then we have:

$$\frac{\partial h}{\partial \alpha} = -2\lambda(1-\alpha\lambda) + 4\alpha\zeta,$$

$$\frac{\partial^2 h}{\partial \alpha^2} = 2\lambda^2 + 4\zeta > 0.$$

Therefore, $h_\lambda(\alpha)$ is a convex function. By setting its derivative to zero, we can get the minimal value:

$$\alpha^* = \frac{\lambda}{\lambda^2 + 2\zeta},$$

$$h_\lambda(\alpha^*) = \frac{4\zeta^2}{(\lambda^2+2\zeta)^2} + \frac{2\lambda^2\zeta}{(\lambda^2+2\zeta)^2} = \frac{2\zeta}{\lambda^2+2\zeta}. \tag{4.74}$$

We already prove that $\lambda_m \geq \lambda_2 > 0$. Therefore, we have $\alpha^* > 0$ and $h_\lambda(\alpha^*) < 1$. Note that $h_\lambda(0) = 1$ and $h_\lambda(\alpha)$ is a quadratic function. Therefore, if $\alpha \in (0, 2\alpha^*)$, $h_\lambda(\alpha^*) \leq h_\lambda(\alpha) < 1$. Thus when $\alpha < (0, \min\{\frac{2\lambda_2}{(\lambda_m^2+2\zeta)}, \frac{2\lambda_m}{(\lambda_m^2+2\zeta)}\})$, we have

$$\left\|\mathbb{E}\left[\mathbf{W}^{(k)\top}\mathbf{W}^{(k)}\right] - J\right\| \leq \max\{h_{\lambda_2}(\alpha), h_{\lambda_m}(\alpha)\} < 1.$$

$\square$

Theorem 62 demonstrates that for MATCHA with an arbitrary communication budget $C_b > 0$, there exists a value of $\alpha$ such that the resulting spectral norm $\rho < 1$. This spectral norm is crucial for ensuring the convergence of MATCHA.

**Lemma 63.** *(Lemma 1 in (Wang et al., 2019)) Let $\{\mathbf{W}^{(k)}\}_{k=1}^{\infty}$ be an i.i.d. symmetric and doubly stochastic matrices sequence. Then for arbitrary matrix $\mathbf{B}$*

$$\mathbb{E}\left[\left\|\mathbf{B}\left(\prod_{l=1}^{n} \mathbf{W}^{(l)} - \mathbf{J}\right)\right\|_F^2\right] \leq \rho^n \|B\|_F^2.$$

*Proof.* Define $\vec{A}_{q,n} = \prod_{l=q}^{n} \mathbf{W}^{(l)} - \mathbf{J}$, $\mathbf{b}_i^\top$ denote the $i$-th row vector of $\mathbf{B}$. Since for all $k$, $\mathbf{W}^{(k)\top} = \mathbf{W}^{(k)}$ and $\mathbf{W}^{(k)}\mathbf{J} = \mathbf{J}\mathbf{W}^{(k)} = \mathbf{W}^{(k)}$. Thus, we have

$$\mathbf{A}_{1,n} = \prod_{l=1}^{n} \left( \mathbf{W}^{(l)} - \mathbf{J} \right) = A_{1,n-1}(\widetilde{\mathbf{W}}^{(n)} - \mathbf{J}).$$

Thus, we have

$$\mathbb{E}_{\mathbf{W}^{(n)}} \left[ \|\mathbf{B}\mathbf{A}_{1,n}\|_F^2 \right] \leq \sum_{i=1}^{d} \mathbb{E}_{\mathbf{W}^{(n)}} \left[ \|\mathbf{b}_i^\top \mathbf{A}_{1,n}\|^2 \right]$$

$$= \sum_{i=1}^{d} \mathbf{b}_i^\top \mathbf{A}_{1,n-1} \mathbb{E}_{\mathbf{W}^{(n)}} \left[ \mathbf{W}^{(n)\top}\mathbf{W}^{(n)} - \mathbf{J} \right] \mathbf{A}_{1,n-1}^\top \mathbf{b}_i$$

Let $\mathbf{C} = \mathbb{E}_{\mathbf{W}^{(n)}} \left[ \mathbf{W}^{(n)\top}\mathbf{W}^{(n)} - \mathbf{J} \right]$, $v_i = \mathbf{A}_{1,n-1}^\top \mathbf{b}_i$, we have:

$$\mathbb{E}_{\mathbf{W}^{(n-1)}} \left[ \|\mathbf{B}\mathbf{A}_{1,n-1}\|_F^2 \right] = \sum_{i=1}^{d} v_i^\top \mathbf{C} v$$

$$\leq \sigma_{\max}(\mathbf{C}) \sum_{i=1}^{d} v_i$$

$$\leq \rho \|\mathbf{B}\mathbf{A}_{1,n-1}\|.$$

Repeat the previous procedure, since $\mathbf{W}^{(k)}$ are all i.i.d, we could get the final results. $\qquad\square$

We are next going to move to the main Theorem that captures the convergence of MATCHA. Lemma 63 is used to prove the theorem.

**Assumption 64** (Nonconvex Setting). *We assume that the loss function $F(x) = \sum_{i=1}^{m} F_i(x)$ satisfy the following conditions:*

(1) *Lipschitz gradient:* $\|\nabla F_i(x) - \nabla F_i(y)\| \leq L \|x - y\|$

(2) *Unbiased gradient:* $\mathbb{E}_{\xi_i}[g_i(x_{k,i}; \xi_{k,i})] = \nabla F_i(x_{k,i})$

(3) *Bounded variance:* $\mathbb{E}_{\xi_i}[\|g_i(x_{k,i}; \xi_{k,i}) - \nabla F_i(x_{k,i})\|^2] \leq \sigma^2$

(4) *Unified gradient:* $\mathbb{E}_{\xi_i}[\|\nabla F_i(x) - \nabla F_i(x)\|^2] \leq \zeta^2$.

**Theorem 65** (Convergence of MATCHA; Nonconvex Setting). *(Theorem 2 in (Wang et al., 2019)) Suppose all local workers are initialized with the same point $\overline{x}_1$ and $\{\mathbf{W}^k\}_{k=1}^{K}$ is the i.i.d. weight matrix generated by the MATCHA algorithm. Define the spectral norm of $\mathbf{W}$ as $\rho = \left\| \mathbb{E}[\mathbf{W}^{(k)\top}\mathbf{W}^{(k)} - \mathbf{J}] \right\|$. Under Assumption 64, if learning rate $\eta L \leq \min 1, (\sqrt{\rho^{-1} - 1})/4$, then after $K$ iterations:*

$$\frac{1}{K} \sum_{i=1}^{K} \mathbb{E}\left[\|\nabla F(\overline{x}_k)\|^2\right] \leq \frac{8(F(\overline{x}_1) - F^*)}{\eta K} + \frac{4\eta L \sigma^2}{m} + \frac{8\eta^2 L^2 \rho}{1 - \sqrt{\rho}} \left( \frac{\sigma^2}{1 + \sqrt{\rho}} + \frac{3\zeta^2}{1 - \sqrt{\rho}} \right).$$

*When setting $\eta = \sqrt{\frac{m}{K}}$, we obtain sublinear convergence rate $\mathcal{O}(\frac{1}{\sqrt{mK}}) + \mathcal{O}(\frac{m}{K})$.*

*Proof.* For notation simplicity, we denote the stochastic gradient $g_i(x_{k,i}; \xi_{k,i})$ as $g_i(x_{k,i})$. We first introduce the following matrix forms for notation simplicity in the proof.

$$x^{(k)} = [x_{k,1}, x_{k,2}, ..., x_{k,m}];$$
$$\mathbf{G}^{(k)} = [g_1(x_{k,1}), g_2(x_{k,2}), ..., g_m(x_{k,m})];$$
$$\mathbf{F}^{(k)} = [\nabla F_1(x_{k,1}), \nabla F_2(x_{k,2}), ..., \nabla F_m(x_{k,m})].$$

The matrix update rule can be written as

$$x^{(k+1)} = (x^{(k)} - \eta \mathbf{G}^{(k)})\mathbf{W}^{(k)}, \tag{4.75}$$

after taking the average, we have

$$\overline{x}_{k+1} = \overline{x}_k - \frac{\eta}{m}\mathbf{G}^{(k)}\mathbf{1}. \tag{4.76}$$

From Assumption 60, since function $F$ is Lipschitz smooth, we have

$$F(\overline{x}_{k+1}) - F(\overline{x}_k) \le \langle \nabla F(\overline{x}_k), \overline{x}_{k+1} - \overline{x}_k \rangle + \frac{L}{2}\|\overline{x}_{k+1} - \overline{x}_k\|^2. \tag{4.77}$$

Since $\overline{x}_{k+1} = \overline{x}_k - \frac{\eta}{m}\mathbf{G}^{(k)}\mathbf{1}$, we have

$$F(\overline{x}_{k+1}) - F(\overline{x}_k) \le -\eta\langle \nabla F(\overline{x}_k), \frac{\mathbf{G}^{(k)}\mathbf{1}}{m}\rangle + \frac{\eta^2 L}{2}\left\|\frac{\mathbf{G}^{(k)}\mathbf{1}}{m}\right\|^2. \tag{4.78}$$

Taking expectation, we have

$$\mathbb{E}[F(\overline{x}_{k+1}) - F(\overline{x}_k)] \le -\eta\langle \nabla F(\overline{x}_k), \frac{\mathbf{F}^{(k)}\mathbf{1}}{m}\rangle + \frac{\eta^2 L}{2}\mathbb{E}\left[\left\|\frac{\mathbf{G}^{(k)}\mathbf{1}}{m}\right\|^2\right]. \tag{4.79}$$

We are going to bound the 2 terms in the RHS of formula 4.79 on by one.

(i) Firstly, we are going to bound $\langle \nabla F(\overline{x}_k), \frac{\mathbf{F}^{(k)}\mathbf{1}}{m}\rangle$.

$$\langle \nabla F(\overline{x}_k), \frac{\mathbf{F}^{(k)}\mathbf{1}}{m}\rangle = \langle \nabla F(\overline{x}_k), \frac{1}{m}\sum_{i=1}^m \nabla F_i(x_{k,i})\rangle$$

$$= \frac{1}{2}\left[\|\nabla F(\overline{x}_k)\|^2 + \left\|\frac{1}{m}\sum_{i=1}^m \nabla F_i(x_{k,i})\right\|^2 - \left\|F(\overline{x}_k) - \frac{1}{m}\sum_{i=1}^m \nabla F_i(x_{k,i})\right\|^2\right]$$

$$= \frac{1}{2}\left[\|\nabla F(\overline{x}_k)\|^2 + \left\|\frac{1}{m}\sum_{i=1}^m \nabla F_i(x_{k,i})\right\|^2 - \left\|\frac{1}{m}\sum_{i=1}^m \left[\nabla F_i(\overline{x}_k) - \nabla F_i(x_{k,i})\right]\right\|^2\right]$$

$$\ge \frac{1}{2}\left[\|\nabla F(\overline{x}_k)\|^2 + \left\|\frac{1}{m}\sum_{i=1}^m \nabla F_i(x_{k,i})\right\|^2 - \frac{1}{m}\sum_{i=1}^m \|\nabla F_i(\overline{x}_k) - \nabla F_i(x_{k,i})\|^2\right]$$

(By Jensen's Inequlity.)

$$\ge \frac{1}{2}\left[\|\nabla F(\overline{x}_k)\|^2 + \left\|\frac{1}{m}\sum_{i=1}^m \nabla F_i(x_{k,i})\right\|^2 - \frac{L^2}{m}\sum_{i=1}^m \|\overline{x}_k - x_{k,i}\|^2\right]$$

($F$ is L-Lipschitz smooth.)

Define

$$\mathbf{J} = \frac{\mathbf{1}_m}{m},$$

we could reformulate the above bound as:

$$\langle \nabla F(\overline{x}_k), \frac{\mathbf{F}^{(k)}\mathbf{1}}{m}\rangle \ge \frac{1}{2}\|\nabla F(\overline{x}_k)\|^2 + \frac{1}{2}\left\|\frac{\mathbf{F}^{(k)}\mathbf{1}}{m}\right\|^2 - \frac{L^2}{2m}\|x_k(\mathbf{I} - \mathbf{J})\|_F^2 \tag{4.80}$$

(ii) Secondly, we are going to bound $\mathbb{E}\left[\left\|\frac{\mathbf{G}^{(k)}\mathbf{1}}{m}\right\|^2\right]$.

$$
\begin{aligned}
\mathbb{E}\left[\left\|\frac{\mathbf{G}^{(k)}\mathbf{1}}{m}\right\|^2\right] &= \mathbb{E}\left[\left\|\frac{1}{m}\sum_{i=1}^{m}g_i(x_{k,i})\right\|^2\right] \\
&= \mathbb{E}\left[\left\|\frac{1}{m}\sum_{i=1}^{m}[g_i(x_{k,i})-\nabla F_i(x_{k,i})+\nabla F_i(x_{k,i})]\right\|^2\right] \\
&\leq \frac{1}{m^2}\sum_{i=1}^{m}\mathbb{E}\left[\left\|\frac{1}{m}\sum_{i=1}^{m}[g_i(x_{k,i})-\nabla F_i(x_{k,i})]\right\|^2\right] + \left\|\frac{1}{m}\sum_{i=1}^{m}\nabla F_i(x_{k,i})\right\|^2 \\
&\leq \frac{\sigma^2}{m} + \left\|\frac{\mathbf{F}^{(k)}\mathbf{1}}{m}\right\|^2
\end{aligned}
\tag{4.81}
$$

Combine formula (4.79), (4.80) and (4.81), we have

$$
\begin{aligned}
\mathbb{E}[F(\overline{x}_{k+1})-F(\overline{x}_k)] \leq &- \frac{\eta}{2}\mathbb{E}[\|\nabla F(\overline{x}_k)\|^2] - \frac{\eta}{2}(1-\eta L)\left\|\frac{\mathbf{F}^{(k)}\mathbf{1}}{m}\right\|^2 \\
&+ \frac{\eta L^2}{2m}\mathbb{E}\left[\|x_k(\mathbf{I}-\mathbf{J})\|_F^2\right] + \frac{\eta^2 L\sigma^2}{2m}
\end{aligned}
\tag{4.82}
$$

Averaging over all iterations from $k=1,...,K$, we have

$$
\begin{aligned}
\frac{\mathbb{E}[F(\overline{x}_K)-F(\overline{x}_1)]}{K} \leq &- \frac{\eta}{2}\frac{1}{K}\sum_{k=1}^{K}\mathbb{E}[\|\nabla F(\overline{x}_k)\|^2] - \frac{\eta}{2}(1-\eta L)\frac{1}{K}\sum_{k=1}^{K}\left\|\frac{\mathbf{F}^{(k)}\mathbf{1}}{m}\right\|^2 \\
&+ \frac{1}{K}\sum_{k=1}^{K}\frac{\eta L^2}{2m}\mathbb{E}\left[\|x_k(\mathbf{I}-\mathbf{J})\|_F^2\right] + \frac{\eta^2 L\sigma^2}{2m}
\end{aligned}
\tag{4.83}
$$

By rearranging, we have

$$
\begin{aligned}
\frac{1}{K}\sum_{k=1}^{K}\mathbb{E}[\|\nabla F(\overline{x}_k)\|^2] \leq &\frac{2\mathbb{E}[F(\overline{x}_1)-F(\overline{x}_K)]}{\eta K} - \frac{1-\eta L}{m}\frac{1}{K}\sum_{k=1}^{K}\left\|\frac{\mathbf{F}^{(k)}\mathbf{1}}{m}\right\|^2 \\
&+ \frac{L^2}{mK}\sum_{k=1}^{K}\mathbb{E}\left[\|x_k(\mathbf{I}-\mathbf{J})\|_F^2\right] + \frac{\eta L\sigma^2}{m} \\
\leq &\frac{2\mathbb{E}[F(\overline{x}_1)-F(\overline{x}_K)]}{\eta K} + \frac{L^2}{mK}\sum_{k=1}^{K}\mathbb{E}\left[\|x_k(\mathbf{I}-\mathbf{J})\|_F^2\right] + \frac{\eta L\sigma^2}{m}
\end{aligned}
\tag{4.84}
$$

We now complete the first part of the proof. In the following section, we are going to bound $\mathbb{E}\left[\|x_k(\mathbf{I}-\mathbf{J})\|_F^2\right]$.

$$
\begin{aligned}
x_k(\mathbf{I}-\mathbf{J}) &= (x_{k-1}-\eta\mathbf{G}^{(k)})\mathbf{W}^{(k)}(\mathbf{I}-\mathbf{J}) \\
&= x_{k-1}(\mathbf{I}-\mathbf{J})\mathbf{W}^{(k)} - \eta\mathbf{G}^{(k-1)}\mathbf{W}^{(k-1)}(\mathbf{I}-\mathbf{J}) \\
&= ... \\
&= x_1(\mathbf{I}-\mathbf{J})\prod_{q=1}^{k-1}\mathbf{W}^{(q)} - \eta\sum_{q=1}^{k-1}\mathbf{G}^{(q)}(\prod_{q=1}^{k-1}\mathbf{W}^{(q)}-\mathbf{J})
\end{aligned}
\tag{4.85}
$$

Since all workers start from the same point, we have $x_1(\mathbf{I}-\mathbf{J})=0$. Moreover, since $\mathbf{W}$ is symmetric doubly stochastic and $\mathbf{J}=\mathbf{1}_m/m$, we know $\mathbf{WJ}=\mathbf{J}$. In all:

$$\|x_k(\mathbf{I} - \mathbf{J})\|_F^2 = \eta^2 \left\| \sum_{q=1}^{k-1} \mathbf{G}^{(q)} (\prod_{l=q}^{k-1} \mathbf{W}^{(l)} - \mathbf{J}) \right\|_F^2$$

$$= \eta^2 \left\| \sum_{q=1}^{k-1} (\mathbf{G}^{(q)} - \nabla\mathbf{F}^{(q)} + \nabla\mathbf{F}^{(q)}) (\prod_{l=q}^{k-1} \mathbf{W}^{(l)} - \mathbf{J}) \right\|_F^2$$

$$\leq 2\eta^2 \underbrace{\left\| \sum_{q=1}^{k-1} (\mathbf{G}^{(q)} - \nabla\mathbf{F}^{(q)}) (\prod_{l=q}^{k-1} \mathbf{W}^{(l)} - \mathbf{J}) \right\|_F^2}_{:=I_1} + 2\eta^2 \underbrace{\left\| \sum_{q=1}^{k-1} \nabla\mathbf{F}^{(q)} (\prod_{l=q}^{k-1} \mathbf{W}^{(l)} - \mathbf{J}) \right\|_F^2}_{:=I_2} \qquad (4.86)$$

Then we are going to bound terms $(I_1)$ and $(I_2)$ in formula (4.86).

$$\mathbb{E}[I_1] \leq \sum_{q=1}^{k-1} \mathbb{E}\left[ \left\| (\mathbf{G}^{(q)} - \nabla\mathbf{F}^{(q)}) (\prod_{l=q}^{k-1} \mathbf{W}^{(l)} - \mathbf{J}) \right\|_F^2 \right]$$

$$\leq \sum_{q=1}^{k-1} \rho^{k-q} \mathbb{E}\left[ \left\| \mathbf{G}^{(q)} - \nabla\mathbf{F}^{(q)} \right\|_F^2 \right] \quad \text{(Lemma 59)}$$

$$\leq \frac{m\sigma^2 \rho}{1-\rho} \quad \text{(Assumption 60: variance bounded)} \qquad (4.87)$$

$$\mathbb{E}[I_2] \leq \sum_{q=1}^{k-1} \mathbb{E}\left[ \left\| \nabla\mathbf{F}^{(q)} (\prod_{l=q}^{k-1} \mathbf{W}^{(l)} - \mathbf{J}) \right\|_F^2 \right]$$

$$+ \sum_{q=1}^{k} \sum_{p=1, p\neq q}^{k} \mathbb{E}\left[ \left\| \nabla\mathbf{F}^{(q)} (\prod_{l=q}^{k-1} \mathbf{W}^{(l)} - \mathbf{J}) \right\|_F \left\| \nabla\mathbf{F}^{(p)} (\mathbf{W}^{k-p} - \mathbf{J}) \right\|_F \right]$$

$$\leq \sum_{q=1}^{k-1} \rho^{k-q} \mathbb{E}\left[ \left\| \nabla\mathbf{F}^{(q)} \right\|_F^2 \right]$$

$$+ \sum_{q=1}^{k} \sum_{p=1, p\neq q}^{k} \mathbb{E}\left[ \frac{1}{2\epsilon} \rho^{k-q} \left\| \nabla\mathbf{F}^{(q)} \right\|_F^2 + \frac{\epsilon}{2} \rho^{k-p} \left\| \nabla\mathbf{F}^{(p)} \right\|_F^2 \right]$$

(Lemma 59 & Young's inequality)

If $\epsilon = \rho^{\frac{p-q}{2}}$ for the Young's inequality, we have

$$\mathbb{E}[I_2] \leq \sum_{q=1}^{k-1} \rho^{k-q} \mathbb{E}\left[ \left\| \nabla\mathbf{F}^{(q)} \right\|_F^2 \right]$$

$$+ \sum_{q=1}^{k} \sum_{p=1, p\neq q}^{k} \sqrt{\rho}^{2k-p-q} \mathbb{E}\left[ \left\| \nabla\mathbf{F}^{(q)} \right\|_F^2 + \left\| \nabla\mathbf{F}^{(p)} \right\|_F^2 \right]$$

$$= \sum_{q=1}^{k-1} \rho^{k-q} \mathbb{E}\left[ \left\| \nabla\mathbf{F}^{(q)} \right\|_F^2 \right] + \sum_{q=1}^{k} \sqrt{\rho}^{k-q} \mathbb{E}\left[ \left\| \nabla\mathbf{F}^{(q)} \right\|_F^2 \right] \sum_{p=1, p\neq q}^{k} \sqrt{\rho}^{k-p}$$

$$= \sum_{q=1}^{k-1} \rho^{k-q} \mathbb{E}\left[ \left\| \nabla\mathbf{F}^{(q)} \right\|_F^2 \right] + \sum_{q=1}^{k} \sqrt{\rho}^{k-q} \mathbb{E}\left[ \left\| \nabla\mathbf{F}^{(q)} \right\|_F^2 \right] (\sum_{p=1}^{k} \sqrt{\rho}^{k-p} - \sqrt{\rho}^{k-q})$$

$$\leq \frac{\sqrt{\rho}}{1-\sqrt{\rho}} \sum_{q=1}^{k-1} \sqrt{\rho}^{k-q} \mathbb{E}\left[ \left\| \nabla\mathbf{F}^{(q)} \right\|_F^2 \right] \qquad (4.88)$$

Plugging (4.87) and (4.87) back to (4.86), and summing the inequality for $k = 1, ..., K$ then averaging by $mK$, we have

$$
\begin{aligned}
\frac{1}{mK} \sum_{k=1}^{K} \mathbb{E}\left[\|x_k(\mathbf{I}-\mathbf{J})\|\right] &\leq \frac{2\eta^2\sigma^2\rho}{1-\rho} + \frac{2\eta^2}{m}\frac{\sqrt{\rho}}{1-\sqrt{\rho}}\frac{1}{K}\sum_{k=1}^{K}\sum_{q=1}^{k}\sqrt{\rho}^{k-q}\mathbb{E}\left[\left\|\nabla\mathbf{F}^{(q)}\right\|_F^2\right] \\
&= \frac{2\eta^2\sigma^2\rho}{1-\rho} + \frac{2\eta^2}{m}\frac{\sqrt{\rho}}{1-\sqrt{\rho}}\frac{1}{K}\sum_{k=1}^{K}\mathbb{E}\left[\left\|\nabla\mathbf{F}^{(k)}\right\|_F^2\right]\sum_{q=1}^{k}\sqrt{\rho}^q \\
&\leq \frac{2\eta^2\sigma^2\rho}{1-\rho} + \frac{2\eta^2}{m}\frac{\rho}{(1-\sqrt{\rho})^2}\frac{1}{K}\sum_{k=1}^{K}\mathbb{E}\left[\left\|\nabla\mathbf{F}^{(k)}\right\|_F^2\right]
\end{aligned}
\tag{4.89}
$$

Note that:

$$
\begin{aligned}
\left\|\nabla\mathbf{F}^{(k)}\right\|_F^2 &= \sum_{i=1}^{m}\|\nabla F_i(x_{i,k})\|^2 \\
&= \sum_{i=1}^{m}\|\nabla F_i(x_{i,k}) - \nabla F(x_{i,k}) + \nabla F(x_{i,k}) - \nabla F(\overline{x}_k) + \nabla F(\overline{x}_k)\|^2 \\
&\leq 3\sum_{i=1}^{m}\left[\|\nabla F_i(x_{i,k}) - \nabla F(x_{i,k})\|^2 + \|\nabla F(x_{i,k}) - \nabla F(\overline{x}_k)\|^2 + \|\nabla F(\overline{x}_k)\|^2\right] \\
&\leq 3m\zeta^2 + 3L^2\|x_k(\mathbf{I}-\mathbf{J})\|_F^2 + 3m\|\nabla F(\overline{x}_k)\|^2.
\end{aligned}
\tag{4.90}
$$

Plugging (4.90) back to (4.89), we have:

$$
\begin{aligned}
\frac{1}{mK}\sum_{k=1}^{K}\mathbb{E}\left[\|x_k(\mathbf{I}-\mathbf{J})\|\right] &\leq \frac{2\eta^2\sigma^2\rho}{1-\rho} + \frac{6\eta^2\zeta^2\rho}{(1-\sqrt{\rho})^2} + \frac{6\eta^2 L^2\rho}{(1-\sqrt{\rho})^2}\frac{1}{mK}\sum_{k=1}^{K}\mathbb{E}\left[\|x_k(\mathbf{I}-\mathbf{J})\|\right] \\
&\quad + \frac{6\eta^2\rho}{(1-\sqrt{\rho})^2}\frac{1}{K}\sum_{k=1}^{K}\|\nabla F(\overline{x}_k)\|^2
\end{aligned}
\tag{4.91}
$$

Define

$$
D = \frac{6\eta^2 L^2\rho}{(1-\sqrt{\rho})^2},
$$

After rearranging, we have

$$
\frac{1}{mK}\sum_{k=1}^{K}\mathbb{E}\left[\|x_k(\mathbf{I}-\mathbf{J})\|\right] \leq \frac{1}{1-D}\left[\frac{2\eta^2\sigma^2\rho}{1-\rho} + \frac{6\eta^2\zeta^2\rho}{(1-\sqrt{\rho})^2} + \frac{6\eta^2\rho}{(1-\sqrt{\rho})^2}\frac{1}{K}\sum_{k=1}^{K}\|\nabla F(\overline{x}_k)\|^2\right].
\tag{4.92}
$$

Plugging the bound (4.92) back to (4.84), we have

$$
\begin{aligned}
\frac{1}{K}\sum_{k=1}^{K}\mathbb{E}[\|\nabla F(\overline{x}_k)\|^2] &\leq \frac{2\mathbb{E}[F(\overline{x}_1)-F(\overline{x}_K)]}{\eta K} + \frac{\eta L\sigma^2}{m} \\
&\quad + \frac{1}{1-D}\frac{2\eta^2\sigma^2\rho}{1-\rho} + \frac{D\zeta^2}{1-D} + \frac{D}{1-D}\frac{1}{K}\sum_{k=1}^{K}\|\nabla F(\overline{x}_k)\|^2
\end{aligned}
\tag{4.93}
$$

By rearranging, we have

$$
\begin{aligned}
\frac{1}{K}\sum_{k=1}^{K}\mathbb{E}[\|\nabla F(\overline{x}_k)\|^2] &\le \frac{1-D}{1-2D}\left[\frac{2\mathbb{E}[F(\overline{x}_1)-F(\overline{x}_K)]}{\eta K}+\frac{\eta L\sigma^2}{m}\right] \\
&\quad + \frac{1}{1-2D}\frac{2\eta^2 L^2\rho}{1-\sqrt{\rho}}\left(\frac{\sigma^2}{1+\sqrt{\rho}}+\frac{3\zeta^2}{1-\sqrt{\rho}}\right) \\
&\le \frac{1}{1-2D}\left[\frac{2\mathbb{E}[F(\overline{x}_1)-F(\overline{x}_K)]}{\eta K}+\frac{\eta L\sigma^2}{m}\right] \\
&\quad + \frac{1}{1-2D}\frac{2\eta^2 L^2\rho}{1-\sqrt{\rho}}\left(\frac{\sigma^2}{1+\sqrt{\rho}}+\frac{3\zeta^2}{1-\sqrt{\rho}}\right)
\end{aligned}
\tag{4.94}
$$

Since $\eta L \le \frac{1-\sqrt{\rho}}{4\sqrt{\rho}}$, we have

$$
D = \frac{6\eta^2 L^2\rho}{(1-\sqrt{\rho})^2}\le\frac{3}{8}<\frac{1}{2}\quad\Longrightarrow\quad\frac{1}{1-2D}\le 4
$$

Therefore (4.94) could be bounded as:

$$
\frac{1}{K}\sum_{k=1}^{K}\mathbb{E}[\|\nabla F(\overline{x}_k)\|^2]\le\frac{8\mathbb{E}[F(\overline{x}_1)-F(\overline{x}_K)]}{\eta K}+\frac{4\eta L\sigma^2}{m}+\frac{8\eta^2 L^2\rho}{1-\sqrt{\rho}}\left(\frac{\sigma^2}{1+\sqrt{\rho}}+\frac{3\zeta^2}{1-\sqrt{\rho}}\right)
\tag{4.95}
$$

Since $F^*$ is the minimum value of the loss, we could have:

$$
\frac{1}{K}\sum_{i=1}^{K}\mathbb{E}\left[\|\nabla F(\overline{x}_k)\|^2\right]\le\frac{8(F(\overline{x}_1)-F^*)}{\eta K}+\frac{4\eta L\sigma^2}{m}+\frac{8\eta^2 L^2\rho}{1-\sqrt{\rho}}\left(\frac{\sigma^2}{1+\sqrt{\rho}}+\frac{3\zeta^2}{1-\sqrt{\rho}}\right).
\tag{4.96}
$$

If the learning rate is set to be $\eta=\sqrt{\frac{m}{K}}$, we have

$$
\begin{aligned}
\frac{1}{K}\sum_{i=1}^{K}\mathbb{E}\left[\|\nabla F(\overline{x}_k)\|^2\right]&\le\frac{8(F(\overline{x}_1)-F^*)+4L\sigma^2}{\sqrt{mK}}+\frac{8m}{K}\frac{L^2\rho}{1-\sqrt{\rho}}\left(\frac{\sigma^2}{1+\sqrt{\rho}}+\frac{3\zeta^2}{1-\sqrt{\rho}}\right) \\
&=\mathcal{O}\left(\frac{1}{\sqrt{mK}}\right)+\mathcal{O}\left(\frac{m}{K}\right).
\end{aligned}
\tag{4.97}
$$

$\square$

### 4.2.3 Adjacent Leader Decentralized SGD (AL-DSGD)

The Adjacent Leader Decentralized SGD (AL-DSGD) algorithm, introduced by (He et al., 2024), aims to enhance the final model performance, accelerate convergence, and minimize communication load in decentralized DL optimizers. The algorithm operates by assigning weights to neighboring workers based on their performance and the degree before averaging them. Additionally, it exerts a corrective force on workers determined by both the best-performing neighbor and the neighbor with the highest degree. To address the issue of reduced convergence speed and performance in nodes with lower degrees, AL-DSGD employs dynamic communication graphs. These graphs enable workers to interact with more nodes while maintaining low node degrees. Algorithm 4.1 shows the pseudocode for AL-DSGD algorithm. Let $k$ denote the iteration index ($k=1,2,\ldots,K$) and $i$ denote the index of the worker ($i=1,2,\ldots,m$). Let $x_{k,i}^N$ denote the best performing worker from among the workers adjacent to node $i$ at iteration $k$ and let $x_{k,i}^\tau$ denote the maximum degree worker from among the workers adjacent to node $i$ at iteration $k$. Let $\{G_{(i)}\}_{i=1}^n$ denote the dynamic communication graphs, each communication graph $\{G_{(i)}\}$ has its own weight matrices sequence $\{\mathbf{W}^{(k)}\}$. There are mainly three steps needed to execute AL-DSGD algorithm and we discuss them below.

*Step 1: The Corrective Force.* The workers with lower degree in the communication topology have worse performance than that with higher degree. In order to augment the influence of the best-performing worker

---

**Algorithm 4.1** Proposed AL-DSGD algorithm

1: **Initialization:** initialize local models $\{x_0^i\}_{i=1}^m$ with the **different** initialization, learning rate $\gamma$, weight matrices sequence $\{\mathbf{W}^{(k)}\}$, and the total number of iterations $K$. Initialize communication graphs set $\{G_{(i)}\}_{i=1}^n$, each communication graph $\{G_{(i)}\}$ has its own weight matrices sequence $\{\mathbf{W}^{(k)}\}$. Pulling coefficients $\lambda_N$ and $\lambda_\tau$. Model weights coefficients $w_N$ and $w_\tau$. Split the original dataset into subsets.

2: **while** k=0, 1, 2, ... K-1 ≤ K **do**

3:      Compute the local stochastic gradient $\nabla F_i(x_{k,i}; \xi_{k,i})$ on all workers

4:      For each worker, fetch neighboring models and determine the adjacent best worker $x_{k,i}^N$ and the adjacent maximum degree worker: $x_{k,i}^\tau$.

5:      Update the local model with corrective force:
$$x_{k+\frac{1}{2},i} = x_{k,i} - \gamma\nabla F_i(x_{k,i}; \xi_{k,i}) - \gamma\lambda_N(x_{k,i} - x_{k,i}^N)$$
$$- \gamma\lambda_\tau(x_{k,i} - x_{k,i}^\tau)$$

6:      Average the model with neighbors, give additional weights to worker $x_{k,i}^N$ and $x_{k,i}^\tau$:
$$x_{k+1,i} = (1 - w_N - w_\tau) \cdot \big( \sum_{j=1, j\neq i}^m W_{ij}^{(k)} x_{k,j} + W_{ii}^{(k)} x_{k+\frac{1}{2},i} \big) + w_N \cdot x_{k,i}^N + w_\tau \cdot x_{k,i}^\tau$$

7:      Switch to new communication graph.

8: **Output:** the average of all workers $\frac{1}{m}\sum_{i=1}^m x_{k,i}$

---

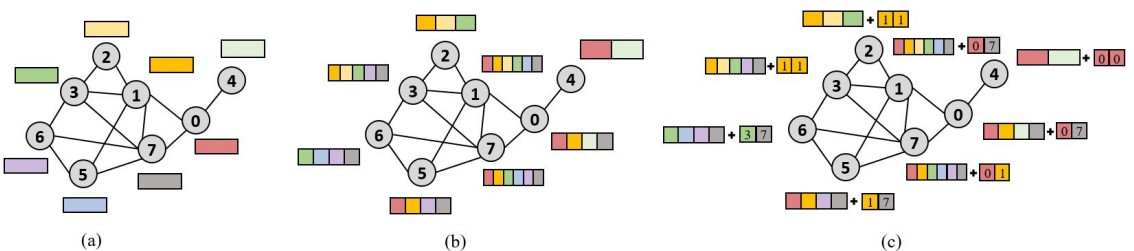

Figure 7: (a) The weights before communication are represented as colored blocks, where different colors correspond to different workers. (b) Previous methods simply average the training model with neighbors. Each colored block denotes the identity of workers whose parameters were taken to compute the average. (c) To illustrate AL-DSGD, we assume that the higher is the index of the worker, the worse is its performance in this iteration. For each node, in addition to averaging with neighboring models, AL-DSGD assigns additional weights to the best performing adjacent model and the maximum degree adjacent model. This is depicted as the sum, where the additional block has two pieces (the left corresponds to the best performing adjacent model and the right corresponds to the maximum degree adjacent model; the indexes of these models are also provided). For example, in the case of model 2, both the best-performing adjacent model and the maximum degree adjacent model is model 1.

on others, AL-DSGD introduces the corrective force between workers to pull current local worker to their adjacent nodes with the largest degrees and the lowest train loss. For each current worker $x_{k,i}$, denote the best-performing adjacent worker based on the train loss and the largest degree adjacent work as $x_{k,i}^N$ and $x_{k,i}^\tau$, respectively. Adding the corrective force is done as follows:

$$x_{k+\frac{1}{2},i} = x_{k,i} - \gamma\nabla F_i(x_{k,i}; \xi_{k,i}) - \gamma\lambda_N(x_{k,i} - x_{k,i}^N) - \gamma\lambda_\tau(x_{k,i} - x_{k,i}^\tau)$$

where $\lambda_N$ and $\lambda_\tau$ are pulling coefficients, $\gamma$ is the learning rate, and $\nabla F_i(x_{k,i}; \xi_{k,i})$ is the gradient of the loss function for worker $i$ computed on parameters $x_{k,i}$ and local data sample $\xi_{k,i}$ that is seen by worker $i$ at iteration $k$.

*Step 2: The Averaging Step.* When averaging workers, AL-DSGD weight them according to their degree and performance. Figure 7 visualizes the process and the update step is shown below.

$$x_{k+1,i} = (1 - w_N - w_\tau) \cdot \big( W_{ii}^{(k)} x_{k+\frac{1}{2},i} + \sum_{j=1, j\neq i}^m W_{ij}^{(k)} x_{k,j} \big) + w_N \cdot x_{k,i}^N + w_\tau \cdot x_{k,i}^\tau.$$

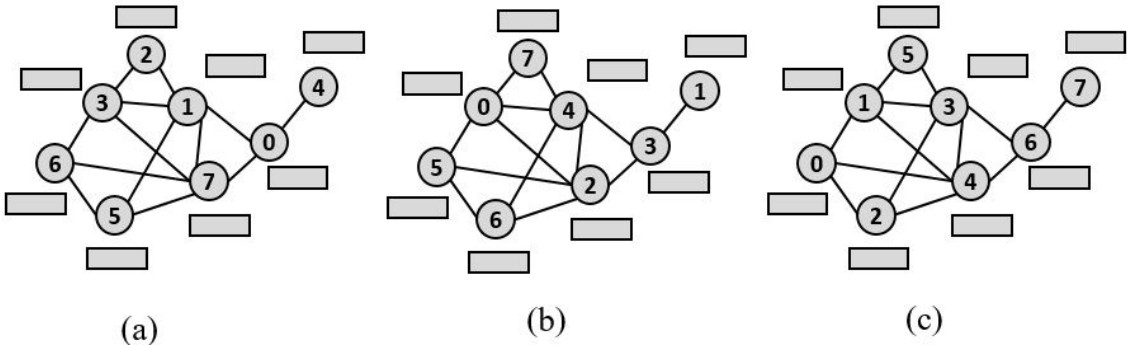

Figure 8: AL-DSGD with three Laplacian matrices rotates workers locations between (a), (b), and (c).

*Step 3: The Dynamic Communication Graph.* Instead of being restricted to a single communication topology, AL-DSGD introduces $n$ different communication topologies and switches between them. Figure 8 visualizes the process.

The gradient update step for AL-DSGD is:

$$x_{k+1,i} = \overbrace{(1 - \omega_N - \omega_\tau) \sum_{j=1}^{m} W_{ij}^{(k)} x_{k,j} + \omega_N x_{k,i}^n + \omega_\tau x_{k,i}^\tau}^{(I)}$$

$$- \gamma (1 - \omega_N - \omega_\tau) W_{ii}^{(k)} \left[ \nabla F_i(x_{k,i}; \xi_{k,i}) + \lambda_N (x_{k,i} - x_{k,i}^N) + \lambda_\tau (x_{k,i} - x_{k,i}^\tau) \right] \qquad (4.98)$$

**Convergence proof.** This section presents an analysis of the convergence rate of the AL-DSGD algorithm. The proof is structured similarly to that of DP-SGD and MATCHA. We define the average iterate at step $k$ as $\overline{x}_k = \frac{1}{m} \sum_{i=1}^{m} x_{k,i}$ and the minimum of the loss function as $F^*$. This section demonstrates that under certain assumptions, the averaged gradient norm $\frac{1}{K} \sum_{k=1}^{K} \mathbb{E}[\|\nabla F(\overline{x}_k)\|]$ converges to zero with sublinear convergence rate.

We first reformulate the update rule (4.98) of AL-DSGD into matrix representation for easy notaion. We first consider part (I) in formula (4.98) without the gradient update step. We define $\widetilde{x}_{k+\frac{1}{2},i} := (I)$, and denote

$$x_k = [x_{k,1}, x_{k,2}, ..., x_{k,m}],$$
$$\widetilde{x}_{k+\frac{1}{2}} = [\widetilde{x}_{k+\frac{1}{2},1} \widetilde{x}_{k+\frac{1}{2},2}, ..., \widetilde{x}_{k+\frac{1}{2},m}],$$
$$x_k^N = [x_{k,1}^N, x_{k,2}^N, ..., x_{k,m}^N],$$
$$x_k^\tau = [x_{k,1}^\tau, x_{k,2}^\tau, ..., x_{k,m}^\tau].$$

We have:

$$\widetilde{x}_{k+\frac{1}{2}} = X_k \widetilde{\mathbf{W}}^{(k)}$$
$$\widetilde{\mathbf{W}}^{(k)} = (1 - \omega_N - \omega_\tau) \mathbf{W}^{(k)} + \omega_N \mathbf{A}_k^N + \omega_\tau \mathbf{A}_k^\tau$$
$$\mathbf{W}^{(k)} = 1 - \alpha \mathbf{L}^{(k)}, \qquad (4.99)$$

where $\mathbf{L}^{(k)}$ denotes the graph Laplacian matrix at the $k^{\text{th}}$ iteration, $x_k^N$ and $x_k^\tau$ are the model parameter matrix of the adjacent best workers and adjacent maximum degree workers at the $k^{\text{th}}$ iteration. Assume $x_k^N = x_k \mathbf{A}_k^N$, $x_k^\tau = x_k \mathbf{A}_k^\tau$. Since every row in $x_k^N$ and $x_k^\tau$ is also a row of the model parameter matrix $x_k$, we could conclude that the transformation matrices $\mathbf{A}_k^N$ and $\mathbf{A}_k^\tau$ must be the left stochastic matrices.

AL-DSGD switches between $n$ communication graphs $\{G_{(i)}\}_{i=1}^n$. Let $\{\mathbf{L}_{(i),j}\}_{j=1}^m$ be the Laplacian matrices set as matching decompositions of graph $G_{(i)}$. Led by MATCHA approach, to each matching of $\mathbf{L}_{(i),j}$ to graph $G_{(i)}$ we assign an independent Bernoulli random variable $B_{(i),j}$ with probability $p_{(i),j}$ based on the communication budget $C_b$.

Then the graph Laplacian matrix at the $k^{\text{th}}$ iteration $\mathbf{L}^{(k)}$ can be written as:

$$\mathbf{L}^{(k)} = \begin{cases} \sum_{j=1}^m B_{(1),j}^{(k)} \mathbf{L}_{(1),j} & \text{if } k \bmod n = 1 \\[2mm] \sum_{j=1}^m B_{(2),j}^{(k)} \mathbf{L}_{(2),j} & \text{if } k \bmod n = 2 \\[2mm] \quad\quad ... \\[2mm] \sum_{j=1}^m B_{(n),j}^{(k)} \mathbf{L}_{(n),j} & \text{if } k \bmod n = 0. \end{cases}$$

The convergence of AL-DSGD algorithm requires $\rho = \max\{\left\|\mathbb{E}\left[\widetilde{\mathbf{W}}^{(k)}(I-J)\widetilde{\mathbf{W}}^{(k)\top}\right]\right\|, \left\|\mathbb{E}\left[\widetilde{\mathbf{W}}^{(k)}\widetilde{\mathbf{W}}^{(k)\top}\right]\right\|\} < 1$, where $\mathbf{J} = \mathbf{1}\mathbf{1}^\top/m$. The following Theorem 66 illustrated that for arbitrary communication budget $C_b$ there exists some $\alpha$, $\omega_N$ and $\omega_\tau$ such that the spectral norm $\rho < 1$.

**Theorem 66.** *(Theorem 1 in (He et al., 2024)) Let $\{\mathbf{L}^{(k)}\}$ denote the sequence of Laplacian matrices generated by AL-DSGD algorithm with arbitrary communication budget $C_b > 0$ for the dynamic communication graph set $\{G_{(i)}\}_{i=1}^n$. The mixing matrix $\widetilde{\mathbf{W}}^{(k)}$ is defined as (4.99). There exists a range of $\alpha$ and a range of average parameters $\omega_N = \omega_\tau \in (0, \omega(\alpha))$, whose bound is dictated by $\alpha$, such that the spectral norm $\rho = \max\{\left\|\mathbb{E}\left[\widetilde{\mathbf{W}}^{(k)}(\mathbf{I}-\mathbf{J})\widetilde{\mathbf{W}}^{(k)\top}\right]\right\|, \left\|\mathbb{E}\left[\widetilde{\mathbf{W}}^{(k)}\widetilde{\mathbf{W}}^{(k)\top}\right]\right\|\} < 1.$*

*Proof.* See Appendix 4.1. □

**Lemma 67.** *(Lemma 3 in (He et al., 2024)) Let $\{\widetilde{\mathbf{W}}^{(k)}\}_{k=1}^\infty$ be i.i.d matrix generated from AL-DSGD algorithm and*

$$\rho = \max\{\left\|\mathbb{E}\left[\widetilde{\mathbf{W}}^{(k)}(\mathbf{I}-\mathbf{J})\widetilde{\mathbf{W}}^{(k)\top}\right]\right\|, \left\|\mathbb{E}\left[\widetilde{\mathbf{W}}^{(k)}\widetilde{\mathbf{W}}^{(k)\top}\right]\right\|\} < 1.$$

*Then for arbitrary $\mathbf{B}$*

$$\mathbb{E}\left[\left\|\mathbf{B}\left(\prod_{l=1}^n \widetilde{\mathbf{W}}^{(l)}\right)(\mathbf{I}-\mathbf{J})\right\|_F^2\right] \le \rho^n \|\mathbf{B}\|_F^2.$$

*Proof.* See Appendix E □

Next we are going to move to the main theorem capturing the convergence of AL-DSGD. The proof is based on Lemma 67.

**Assumption 68** (Nonconvex Setting). *We assume that the loss function $F(x) = \sum_{i=1}^m F_i(x)$ satisfies the following conditions:*

(1) *Lipschitz continuous:* $\|F_i(x) - F_i(y)\| \le \beta \|x - y\|$

(2) *Lipschitz gradient:* $\|\nabla F_i(x) - \nabla F_i(y)\| \le L \|x - y\|$

(3) *Unbiased gradient:* $\mathbb{E}_{\xi_i}[g_i(x_{k,i};\xi_{k,i})] = \nabla F_i(x_{k,i})$

(4) *Bounded variance:* $\mathbb{E}_{\xi_i}[\|g_i(x_{k,i};\xi_{k,i}) - \nabla F_i(x_{k,i})\|^2] \le \sigma^2$

(5) *Unified gradient:* $\mathbb{E}_{\xi_i}[\|\nabla F_i(x) - \nabla F_i(x)\|^2] \le \zeta^2$

(6) *Bounded domain:* $\max\{\|x_{k,i} - x_{k,i}^N\|, \|x_{k,i} - x_{k,i}^\tau\|\} \le \Delta^2$.

**Theorem 69.** *(Convergence of AL-DSGD; Nonconvex Setting; Theorem 2 in (He et al., 2024)) Suppose all local workers are initialized with $\overline{x}^{(1)} = 0$ and $\{\widetilde{\mathbf{W}}^{(k)}\}_{k=1}^K$ is an i.i.d. matrix sequence generated by AL-DSGD algorithm which satisfies the spectral norm condition $\rho < 1$ ($\rho$ is defined in Theorem 66). Under Assumption 14, if $\lambda = 2\lambda_N = 2\lambda_\tau$ and $(1-\alpha)(1-\omega)\gamma L \le \min\{1, (\sqrt{\rho^{-1}} - 1)\}$, then after K iterations:*

$$\frac{1}{K}\sum_{i=1}^K \mathbb{E}\left[\|\nabla F(\overline{x}_k)\|^2\right] \le \frac{8(F(\overline{x}_1) - F^*)}{\eta K} + \frac{8M}{\eta} + \frac{8\eta^2 L^2 \rho}{1 - \sqrt{\rho}}\left(\frac{m\sigma^2 + \lambda^2\Delta^2}{m(1+\sqrt{\rho})} + \frac{3\zeta^2}{1-\sqrt{\rho}}\right),$$

*where $\eta = (1-\alpha)(1-\omega)\gamma$ and $M = \frac{\eta^2 L\sigma^2}{2m} + \lambda\eta\beta\Delta + \lambda\eta^2 L\beta\Delta + \frac{\lambda^2\eta^2 L\Delta^2}{2}$. When setting $\lambda = \sqrt{\frac{m}{K}}$, $\gamma = \sqrt{\frac{m}{(1-\omega)(1-\alpha)K}}$, we obtain sublinear convergence rate $\mathcal{O}(\frac{1}{\sqrt{mK}}) + \mathcal{O}(\sqrt{\frac{m}{K}}) + \mathcal{O}(\sqrt{\frac{m}{K^3}})$.*

*Proof.* Recall the update rule for AL-DSGD algorithm:

$$x_{k+1} = \widetilde{\mathbf{W}}^k x_k - \gamma(1-\omega)Diag\left(\mathbf{W}^{(k)}\right)\left[\mathbf{G}^{(k)} + \lambda_N(x_k - x_k^N) + \lambda_\tau(x_k - x_k^\tau)\right], \qquad (4.100)$$

where

$$x_k = [x_{1,k}, ..., x_{m,k}],$$
$$\mathbf{G}^{(k)} = [g_1(x_{1,k}), ..., g_m(x_{m,k})],$$
$$\nabla\mathbf{F}^{(k)} = [\nabla F_1(x_{1,k}), ..., \nabla F_m(x_{m,k})]$$

Let $\lambda = 2\lambda_N = 2\lambda_\tau$, and $\mathbf{C}_k = \frac{1}{2}(X_k^N + X_k^\tau)$. Then we have

$$X_{k+1} = \widetilde{\mathbf{W}}^k x_k - \gamma(1-\omega)Diag\left(\mathbf{W}^{(k)}\right)\left[\mathbf{G}^{(k)} + \lambda(x_k - \mathbf{C}_k)\right] \qquad (4.101)$$

By the construction of $\mathbf{W}^{(k)}$, the diagonal term in $\mathbf{W}^{(k)}$ are all $1 - \alpha$, we have

$$X_{k+1} = \widetilde{\mathbf{W}}^k x_k - \gamma(1-\omega)(1-\alpha)\left[\mathbf{G}^{(k)} + \lambda(x_k - \mathbf{C}_k)\right] \qquad (4.102)$$

After taking the average and define $\eta = \gamma(1-\omega)(1-\alpha)$, we have

$$\overline{x}_{k+1} = \overline{x}_k - \eta\left[\frac{\mathbf{G}^{(k)}\mathbf{1}}{m} + \lambda(\overline{x}_{k+\frac{1}{2}} - \overline{c}_k)\right] = \overline{x}_k - \eta\left[\frac{\mathbf{G}^{(k)}\mathbf{1}}{m} + \lambda\Delta^{(k)}\right]. \qquad (4.103)$$

Denote $\Delta^{(k)} = \overline{x}_{k+\frac{1}{2}} - \overline{c}_k$. from Assumption 14 (5), we could conclude $\left\|\Delta^{(k)}\right\|^2 \le \Delta^2$.
Then we have

$$\begin{aligned}
F(\overline{x}_{k+1}) - F(\overline{x}_k) \le & \langle\nabla F(\overline{x}_k), \overline{x}_{k+1} - \overline{x}_k\rangle + \frac{L}{2}\|\overline{x}_{k+1} - \overline{x}_k\| \\
\le & -\eta\langle\nabla F(\overline{x}_k), \frac{\mathbf{G}^{(k)}\mathbf{1}}{m} + \lambda\Delta^{(k)}\rangle + \frac{\eta^2 L}{2}\left\|\frac{\mathbf{G}^{(k)}\mathbf{1}}{m} + \lambda\Delta^{(k)}\right\| \\
\le & -\eta\langle\nabla F(\overline{x}_k), \frac{\mathbf{G}^{(k)}\mathbf{1}}{m}\rangle + \frac{\eta^2 L}{2}\left\|\frac{\mathbf{G}^{(k)}\mathbf{1}}{m}\right\| \\
& -\lambda\eta\langle\nabla F(\overline{x}_k), \Delta^{(k)}\rangle + \frac{\lambda^2\eta^2 L}{2}\left\|\Delta^{(k)}\right\|^2 + \lambda\eta^2 L\langle\frac{\mathbf{G}^{(k)}\mathbf{1}}{m}, \Delta^{(k)}\rangle \\
\le & -\eta\langle\nabla F(\overline{x}_k), \frac{\mathbf{G}^{(k)}\mathbf{1}}{m}\rangle + \frac{\eta^2 L}{2}\left\|\frac{\mathbf{G}^{(k)}\mathbf{1}}{m}\right\| + \lambda\eta\beta\Delta + \lambda\eta^2 L\beta\Delta + \frac{\lambda^2\eta^2 L\Delta^2}{2} \qquad (4.104)
\end{aligned}$$

Inspired by the proof in form MATCHA, we have:

$$\mathbb{E}\left[F(\overline{x}_{k+1}) - F(\overline{x}_k)\right]$$

$$\leq -\frac{\eta}{2}\mathbb{E}\left[\|\nabla F(\overline{x}_k)\|^2\right] - \frac{\eta}{2}(1 - \eta L)\mathbb{E}\left[\left\|\frac{\nabla \mathbf{F}^{(k)}\mathbf{1}}{m}\right\|^2\right] + \frac{\eta L^2}{2m}\mathbb{E}\left[\|x_k(\mathbf{I} - \mathbf{J})\|_F^2\right]$$

$$+ \frac{\eta^2 L \sigma^2}{2m} + \lambda\eta\beta\Delta + \lambda\eta^2 L\beta\Delta + \frac{\lambda^2\eta^2 L\Delta^2}{2} \tag{4.105}$$

Denote $M = \frac{\eta^2 L\sigma^2}{2m} + \lambda\eta\beta\Delta + \lambda\eta^2 L\beta\Delta + \frac{\lambda^2\eta^2 L\Delta^2}{2}$, the bound could be simplified as

$$\mathbb{E}\left[F(\overline{x}_{k+1}) - F(\overline{x}_k)\right] \leq -\frac{\eta}{2}\mathbb{E}\left[\|\nabla F(\overline{x}_k)\|^2\right] - \frac{\eta}{2}(1 - \eta L)\mathbb{E}\left[\left\|\frac{\nabla \mathbf{F}^{(k)}\mathbf{1}}{m}\right\|^2\right]$$

$$+ \frac{\eta L^2}{2m}\mathbb{E}\left[\|x_k(\mathbf{I} - \mathbf{J})\|_F^2\right] + M. \tag{4.106}$$

Summing over all iterations and then take the average, we have

$$\frac{\mathbb{E}\left[F(\overline{x}_K) - F(\overline{x}_1)\right]}{K} \leq -\frac{\eta}{2}\frac{1}{K}\sum_{k=1}^{k}\mathbb{E}\left[\|\nabla F(\overline{x}_k)\|^2\right] - \frac{\eta}{2}(1 - \eta L)\frac{1}{K}\sum_{k=1}^{k}\mathbb{E}\left[\left\|\frac{\nabla \mathbf{F}^{(k)}\mathbf{1}}{m}\right\|^2\right]$$

$$+ \frac{\eta L^2}{2mK}\sum_{k=1}^{k}\mathbb{E}\left[\|x_k(\mathbf{I} - \mathbf{J})\|_F^2\right] + M. \tag{4.107}$$

By rearranging the inequality, we have

$$\frac{1}{K}\sum_{k=1}^{k}\mathbb{E}\left[\|\nabla F(\overline{x}_k)\|^2\right] \leq \frac{2(F(\overline{x}_1) - F^*)}{\eta K} - (1 - \eta L)\frac{1}{K}\sum_{k=1}^{k}\mathbb{E}\left[\left\|\frac{\nabla \mathbf{F}^{(k)}\mathbf{1}}{m}\right\|^2\right]$$

$$+ \frac{L^2}{mK}\sum_{k=1}^{k}\mathbb{E}\left[\|x_k(\mathbf{I} - \mathbf{J})\|_F^2\right] + \frac{2M}{\eta}$$

$$\leq \frac{2(F(\overline{x}_1) - F^*)}{\eta K} + \frac{L^2}{mK}\sum_{k=1}^{k}\mathbb{E}\left[\|x_k(\mathbf{I} - \mathbf{J})\|_F^2\right] + \frac{2M}{\eta}. \tag{4.108}$$

Then we are goint to bound $\mathbb{E}\left[\|x_k(\mathbf{I} - \mathbf{J})\|_F^2\right]$. By the property of matrix $J$, we have

$$x_k(\mathbf{I} - \mathbf{J}) = (x_{k-1} - \eta(\mathbf{G}^{(k-1)} + \lambda\Delta_{k-1})\widetilde{\mathbf{W}}^{(k-1)}(\mathbf{I} - \mathbf{J})$$

$$= X_{k-1}\widetilde{\mathbf{W}}^{(k-1)}(\mathbf{I} - \mathbf{J}) - \eta(\mathbf{G}^{(k-1)} + \lambda\Delta^{(k-1)})\widetilde{\mathbf{W}}^{(k-1)}(\mathbf{I} - \mathbf{J})$$

$$= \dots$$

$$= X_1\prod_{q=1}^{k-1}\widetilde{\mathbf{W}}^{(q)}(\mathbf{I} - \mathbf{J}) - \eta\sum_{q=1}^{k-1}(\mathbf{G}^{(k-1)} + \lambda\Delta^{(k)})\left(\prod_{l=q}^{k-1}\widetilde{\mathbf{W}}^{(l)}\right)(\mathbf{I} - \mathbf{J}) \tag{4.109}$$

Without loss of generalizty, assume $X_1 = 0$. Therefore, by Assumption (14) and Lemma 75 we have

$$\|x_k(\mathbf{I} - \mathbf{J})\|_F^2$$

$$= \eta^2\left\|\sum_{q=1}^{k-1}(\mathbf{G}^{(k-1)} + \lambda\Delta^{(k)})\left(\prod_{l=q}^{k-1}\widetilde{\mathbf{W}}^{(l)}\right)(\mathbf{I} - \mathbf{J})\right\|_F^2$$

$$\leq 2\eta^2\left\|\sum_{q=1}^{k-1}\mathbf{G}^{(k-1)}\left(\prod_{l=q}^{k-1}\widetilde{\mathbf{W}}^{(l)}\right)(\mathbf{I} - \mathbf{J})\right\|_F^2 + 2\eta^2\lambda^2\left\|\sum_{q=1}^{k-1}\Delta^{(k)}\left(\prod_{l=q}^{k-1}\widetilde{\mathbf{W}}^{(l)}\right)(\mathbf{I} - \mathbf{J})\right\|_F^2. \tag{4.110}$$

From Assumption 14 (5), we could conclude $\left\|\Delta^{(k)}\right\|^2 \le \Delta^2$ for all $k$. Combine with Lemma 75, we have

$$\|x_k(\mathbf{I}-\mathbf{J})\|_F^2$$

$$\le 2\eta^2 \left\|\sum_{q=1}^{k-1} \mathbf{G}^{(k-1)}\left(\prod_{l=q}^{k-1} \widetilde{\mathbf{W}}^{(l)}\right)(\mathbf{I}-\mathbf{J})\right\|_F^2 + 2\eta^2\lambda^2\Delta^2\sum_{q=1}^{k}\rho^q$$

$$\le 2\eta^2 \left\|\sum_{q=1}^{k-1} \mathbf{G}^{(k-1)}\left(\prod_{l=q}^{k-1} \widetilde{\mathbf{W}}^{(l)}\right)(\mathbf{I}-\mathbf{J})\right\|_F^2 + \frac{2\eta^2\lambda^2\Delta^2\rho}{1-\rho}$$

$$\le 2\eta^2 \left\|\sum_{q=1}^{k-1} \left(\mathbf{G}^{(k-1)}-\triangledown\mathbf{F}^{(q)}\right)\left(\prod_{l=q}^{k-1} \widetilde{\mathbf{W}}^{(l)}\right)(\mathbf{I}-\mathbf{J})\right\|_F^2$$

$$+ 2\eta^2 \left\|\sum_{q=1}^{k-1} \nabla\mathbf{F}^{(q)}\left(\prod_{l=q}^{k-1} \widetilde{\mathbf{W}}^{(l)}\right)(\mathbf{I}-\mathbf{J})\right\|_F^2 + \frac{2\eta^2\lambda^2\Delta^2\rho}{1-\rho}. \tag{4.111}$$

Taking expectation, we have:

$$\mathbb{E}\left[\|x_k(\mathbf{I}-\mathbf{J})\|_F^2\right] \le 2\eta^2\mathbb{E}\left[\left\|\sum_{q=1}^{k-1} \nabla\mathbf{F}^{(q)}\left(\prod_{l=q}^{k-1} \widetilde{\mathbf{W}}^{(l)}\right)(\mathbf{I}-\mathbf{J})\right\|_F^2\right] + \frac{2m\eta^2\sigma^2\rho}{1-\rho} + \frac{2\eta^2\lambda^2\Delta^2\rho}{1-\rho} \tag{4.112}$$

For notation simplicity, let $\mathbf{B}_{q,p} = \left(\prod_{l=q}^{p} \widetilde{\mathbf{W}}^{(l)}\right)(\mathbf{I}-\mathbf{J})$, then we have

$$\mathbb{E}\left[\left\|\sum_{q=1}^{k-1} \nabla\mathbf{F}^{(q)}\left(\prod_{l=q}^{k-1} \widetilde{\mathbf{W}}^{(l)}\right)(\mathbf{I}-\mathbf{J})\right\|_F^2\right]$$

$$= \sum_{q=1}^{k-1}\mathbb{E}\left[\left\|\nabla\mathbf{F}^{(q)}\mathbf{B}_{q,k-1}\right\|_F^2\right] + \sum_{q=1}^{k-1}\sum_{p=1,p\neq q}^{k-1}\mathbb{E}\left[Tr\{\mathbf{B}_{q,k-1}^{\top}\nabla\mathbf{F}^{(q)\top}\nabla\mathbf{F}^{(p)}\mathbf{B}_{p,k-1}\}\right]$$

$$\le \sum_{q=1}^{k-1}\mathbb{E}\left[\left\|\nabla\mathbf{F}^{(q)}\mathbf{B}_{q,k-1}\right\|_F^2\right] + \sum_{q=1}^{k-1}\sum_{p=1,p\neq q}^{k-1}\mathbb{E}\left[\left\|\nabla\mathbf{F}^{(q)}\mathbf{B}_{q,k-1}\right\|_F^2\left\|\nabla\mathbf{F}^{(p)}\mathbf{B}_{p,k-1}\right\|_F^2\right]$$

$$\le \sum_{q=1}^{k-1}\mathbb{E}\left[\left\|\nabla\mathbf{F}^{(q)}\mathbf{B}_{q,k-1}\right\|_F^2\right] + \sum_{q=1}^{k-1}\sum_{p=1,p\neq q}^{k-1}\mathbb{E}\left[\frac{1}{2\epsilon}\left\|\nabla\mathbf{F}^{(q)}\mathbf{B}_{q,k-1}\right\|_F^2 + \frac{\epsilon}{2}\left\|\nabla\mathbf{F}^{(p)}\mathbf{B}_{p,k-1}\right\|_F^2\right]$$

$$\le \sum_{q=1}^{k-1}\mathbb{E}\left[\left\|\nabla\mathbf{F}^{(q)}\mathbf{B}_{q,k-1}\right\|_F^2\right] + \sum_{q=1}^{k-1}\sum_{p=1,p\neq q}^{k-1}\mathbb{E}\left[\frac{\rho^{k-q}}{2\epsilon}\left\|\nabla\mathbf{F}^{(q)}\right\|_F^2 + \frac{\rho^{k-p}\epsilon}{2}\left\|\nabla\mathbf{F}^{(p)}\right\|_F^2\right]. \tag{4.113}$$

Taking $\epsilon = \rho^{\frac{p-q}{2}}$, we have

$$\mathbb{E}\left[\left\|\sum_{q=1}^{k-1} \nabla\mathbf{F}^{(q)}\left(\prod_{l=q}^{k-1} \widetilde{\mathbf{W}}^{(l)}\right)(\mathbf{I}-\mathbf{J})\right\|_F^2\right]$$

$$\le \sum_{q=1}^{k-1}\mathbb{E}\left[\left\|\nabla\mathbf{F}^{(q)}\mathbf{B}_{q,k-1}\right\|_F^2\right] + \frac{1}{2}\sum_{q=1}^{k-1}\sum_{p=1,p\neq q}^{k-1}\sqrt{\rho}^{2k-p-q}\mathbb{E}\left[\left\|\nabla\mathbf{F}^{(q)}\right\|_F^2 + \left\|\nabla\mathbf{F}^{(p)}\right\|_F^2\right]$$

$$= \sum_{q=1}^{k-1}\mathbb{E}\left[\left\|\nabla\mathbf{F}^{(q)}\mathbf{B}_{q,k-1}\right\|_F^2\right] + \sum_{q=1}^{k-1}\sqrt{\rho}^{k-q}\mathbb{E}\left[\left\|\nabla\mathbf{F}^{(q)}\right\|_F^2\right]\sum_{p=1,p\neq q}^{k-1}\sqrt{\rho}^{k-p}$$

$$= \sum_{q=1}^{k-1}\mathbb{E}\left[\left\|\nabla\mathbf{F}^{(q)}\mathbf{B}_{q,k-1}\right\|_F^2\right] + \sum_{q=1}^{k-1}\sqrt{\rho}^{k-q}\mathbb{E}\left[\left\|\nabla\mathbf{F}^{(q)}\right\|_F^2\right]\left(\sum_{p=1}^{k-1}\sqrt{\rho}^{k-p} - \sqrt{\rho}^{k-q}\right)$$

$$\le \frac{\sqrt{\rho}}{1-\sqrt{\rho}}\sum_{q=1}^{k-1}\sqrt{\rho}^{k-q}\mathbb{E}\left[\left\|\nabla\mathbf{F}^{(q)}\right\|_F^2\right].$$

Therefore,

$$\mathbb{E}\left[\|x_k(\mathbf{I}-\mathbf{J})\|_F^2\right] \leq \frac{2\eta^2\sqrt{\rho}}{1-\sqrt{\rho}}\sum_{q=1}^{k-1}\sqrt{\rho}^{k-q}\mathbb{E}\left[\left\|\nabla\mathbf{F}^{(q)}\right\|_F^2\right] + \frac{2\eta^2\rho(m\sigma^2+\lambda^2\Delta^2)}{1-\rho}. \tag{4.114}$$

Therefore

$$\begin{aligned}
\frac{1}{mK}\sum_{k=1}^{K}\mathbb{E}\left[\|x_k(\mathbf{I}-\mathbf{J})\|_F^2\right] &\leq \frac{2\eta^2\sqrt{\rho}}{mK(1-\sqrt{\rho})}\sum_{k=1}^{K}\sum_{q=1}^{k-1}\sqrt{\rho}^{k-q}\mathbb{E}\left[\left\|\nabla\mathbf{F}^{(q)}\right\|_F^2\right] + \frac{2\eta^2\rho(m\sigma^2+\lambda^2\Delta^2)}{m(1-\rho)} \\
&\leq \frac{2\eta^2\sqrt{\rho}}{mK(1-\sqrt{\rho})}\sum_{k=1}^{K}\frac{\sqrt{\rho}}{1-\sqrt{\rho}}\mathbb{E}\left[\left\|\nabla\mathbf{F}^{(q)}\right\|_F^2\right] + \frac{2\eta^2\rho(m\sigma^2+\lambda^2\Delta^2)}{m(1-\rho)} \\
&= \frac{2\eta^2\rho}{mK(1-\sqrt{\rho})^2}\sum_{k=1}^{K}\mathbb{E}\left[\left\|\nabla\mathbf{F}^{(q)}\right\|_F^2\right] + \frac{2\eta^2\rho(m\sigma^2+\lambda^2\Delta^2)}{m(1-\rho)}
\end{aligned} \tag{4.115}$$

Note that

$$\begin{aligned}
\left\|\nabla\mathbf{F}^{(q)}\right\|_F^2 &= \sum_{i=1}^{m}\|\nabla F_i(x_{i,k})\|^2 \\
&= \sum_{i=1}^{m}\|\nabla F_i(x_{i,k})-\nabla F(x_{i,k})+\nabla F(x_{i,k})-\nabla F(\overline{x}_k)+\nabla F(\overline{x}_k)\|^2 \\
&\leq 3\sum_{i=1}^{m}\left[\|\nabla F_i(x_{i,k})-\nabla F(x_{i,k})\|^2+\|\nabla F(x_{i,k})-\nabla F(\overline{x}_k)\|^2+\|\nabla F(\overline{x}_k)\|^2\right] \\
&\leq 3m\zeta^2+3L^2\|x_k(\mathbf{I}-\mathbf{J})\|_F^2+3m\|\nabla F(\overline{x}_k)\|^2.
\end{aligned} \tag{4.116}$$

Therefore, we have

$$\begin{aligned}
\frac{1}{mK}\sum_{k=1}^{K}\mathbb{E}\left[\|x_k(\mathbf{I}-\mathbf{J})\|_F^2\right] &\leq \frac{2\eta^2\rho(m\sigma^2+\lambda^2\Delta^2)}{m(1-\rho)}+\frac{6\eta^2\zeta^2\rho}{(1-\sqrt{\rho})^2}+\frac{6\eta^2\zeta^2\rho}{(1-\sqrt{\rho})^2}\frac{1}{mK}\mathbb{E}\left[\|x_k(\mathbf{I}-\mathbf{J})\|_F^2\right] \\
&\quad + \frac{6\eta^2\rho}{(1-\sqrt{\rho})^2}\frac{1}{K}\sum_{i=1}^{K}\mathbb{E}\left[\|\nabla F(\overline{x}_k)\|^2\right].
\end{aligned} \tag{4.117}$$

Define $D = \frac{6\eta^2 L^2\rho}{(1-\sqrt{\rho})^2}$, by rearranging we have

$$\begin{aligned}
\frac{1}{mK}\sum_{k=1}^{K}\mathbb{E}\left[\|x_k(\mathbf{I}-\mathbf{J})\|_F^2\right] &\leq \frac{1}{1-2D}\Big[\frac{2\eta^2\rho(m\sigma^2+\lambda^2\Delta^2)}{m(1-\rho)}+\frac{6\eta^2\zeta^2\rho}{(1-\sqrt{\rho})^2} \\
&\quad + \frac{6\eta^2\rho}{(1-\sqrt{\rho})^2}\frac{1}{K}\sum_{i=1}^{K}\mathbb{E}\left[\|\nabla F(\overline{x}_k)\|^2\right]\Big]
\end{aligned} \tag{4.118}$$

Plugging (4.118) back to (4.108), we have

$$\begin{aligned}
\frac{1}{K}\sum_{i=1}^{K}\mathbb{E}\left[\|\nabla F(\overline{x}_k)\|^2\right] &\leq \frac{2(F(\overline{x}_1)-F^*)}{\eta K}+\frac{2M}{\eta}+\frac{1}{1-D}\frac{2\eta^2 L^2\rho(m\sigma^2+\lambda^2\Delta^2)}{m(1-\rho)}+\frac{D\zeta^2}{1-D} \\
&\quad + \frac{D}{1-D}\frac{1}{K}\sum_{i=1}^{K}\mathbb{E}\left[\|\nabla F(\overline{x}_k)\|^2\right].
\end{aligned} \tag{4.119}$$

Therefore,

$$\begin{aligned}
\sum_{i=1}^{K}\mathbb{E}\left[\|\nabla F(\overline{x}_k)\|^2\right] &\leq \left(\frac{2(F(\overline{x}_1)-F^*)}{\eta K}+\frac{2M}{\eta}\right)\frac{1-D}{1-2D}+\left(\frac{2\eta^2 L^2\rho(m\sigma^2+\lambda^2\Delta^2)}{m(1-\rho)}+\frac{6\eta^2 L^2\zeta^2\rho}{(1-\sqrt{\rho})^2}\right)\frac{1}{1-2D} \\
&\leq \left(\frac{2(F(\overline{x}_1)-F^*)}{\eta K}+\frac{2M}{\eta}\right)\frac{1}{1-2D}+\frac{2\eta^2 L^2\rho}{1-\sqrt{\rho}}\left(\frac{m\sigma^2+\lambda^2\Delta^2}{m(1+\sqrt{\rho})}+\frac{3\zeta^2}{1-\sqrt{\rho}}\right)\frac{1}{1-2D}
\end{aligned} \tag{4.120}$$

Recall that $\eta L \le (1 - \sqrt{\rho})/4\sqrt{\rho}$, we could know that $\frac{1}{1-2D} \le 4$. Therefore the bound could be simplified as

$$\sum_{i=1}^{K} \mathbb{E}\left[\left\|\nabla F(\overline{x}_k)\right\|^2\right] \le \frac{8(F(\overline{x}_1) - F^*)}{\eta K} + \frac{8M}{\eta} + \frac{8\eta^2 L^2 \rho}{1 - \sqrt{\rho}}\left(\frac{m\sigma^2 + \lambda^2 \Delta^2}{m(1 + \sqrt{\rho})} + \frac{3\zeta^2}{1 - \sqrt{\rho}}\right), \quad (4.121)$$

where $M = \frac{\eta^2 L \sigma^2}{2m} + \lambda\eta\beta\Delta + \lambda\eta^2 L\beta\Delta + \frac{\lambda^2 \eta^2 L \Delta^2}{2}$. When $\eta = \lambda = \sqrt{\frac{m}{K}}$,

$$
\begin{aligned}
\sum_{i=1}^{K} \mathbb{E}\left[\left\|\nabla F(\overline{x}_k)\right\|^2\right] \le{}& \frac{8(F(\overline{x}_1) - F^*)}{\sqrt{mK}} + \frac{4L\sigma^2}{\sqrt{mk}} + 8\beta\Delta\sqrt{\frac{m}{K}} + 8\lambda L\beta\Delta\sqrt{\frac{m}{K}} + 4\lambda^2 L \Delta^2 \sqrt{\frac{m}{K}} \\
&+ \frac{8\sqrt{m}L^2\rho}{(1 - \sqrt{\rho})\sqrt{K}}\left(\frac{\sigma^2}{1 + \sqrt{\rho}} + \frac{\Delta^2}{K(1 + \sqrt{\rho})} + \frac{3\zeta^2}{1 - \sqrt{\rho}}\right) \\
={}& \mathcal{O}(\frac{1}{\sqrt{mK}}) + \mathcal{O}(\sqrt{\frac{m}{K}}) + + \mathcal{O}(\sqrt{\frac{m}{K^3}})
\end{aligned}
\quad (4.122)
$$

$\square$

## 5 Conclusion

This paper offers a comprehensive exploration of the theoretical foundations of optimization methods in DL, emphasizing methodologies, convergence analyses, and generalization abilities. While the field has seen rapid advancements in DL models and the growth of available data, the optimization of these models remains a focal point for researchers aiming to enhance their performance. Despite the plethora of optimization techniques available, many existing survey papers tend to focus on summarizing methodologies, often neglecting the theoretical analyses of these methods.

Our study delves deep into the theoretical analysis of popular gradient-based first-order and second-order methods, shedding light on their intricacies and potential applications. We also discuss the theoretical analysis of optimization techniques that adapt to the geometrical properties of the DL loss landscapes with a goal of identifying optimal points that minimize the generalization error. Furthermore, our analysis extends to distributed optimization methods that enable parallel computations and encompasses both centralized and decentralized approaches.

By consolidating all these various insights, this paper serves as a comprehensive theoretical handbook of optimization methods for DL. We aim to bridge the gap between theory and practice in DL, offering valuable lesson to the readers and building understanding of the DL optimization field, thereby facilitating further advancements and innovations in DL optimization.

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

# A    Proof for Adagrad and Adam.

In this section, we detailed discuss the Lemma introduced in the convergence proof for Adagrad and Adam.

**Lemma 70** (Adaptive update with momentum approximately follows a descent direction)**.** *Given $x_0 \in \mathbb{R}^d$, the iterates defined by the (2.50) that is non-decreasing, and under the Assumptions 24, as well as $0 \le \beta_1 < \beta_2 \le 1$, we have for all iterations $k \in \mathbb{N}^*$,*

$$\mathbb{E}\left[\sum_{i\in[d]} G_{k,i}\frac{m_{k,i}}{\sqrt{\epsilon+v_{k,i}}}\right] \ge \frac{1}{2}\left(\sum_{i\in[d]}\sum_{l=0}^{k-1}\beta_1^l\mathbb{E}\left[\frac{G_{k-l,i}^2}{\sqrt{\epsilon+\tilde{v}_{k,l+1,i}}}\right]\right)$$
$$-\frac{\alpha_k^2 L^2}{4R}\sqrt{1-\beta_1}\left(\sum_{l=1}^{k-1}\|u_{k-l}\|_2^2\sum_{p=l}^{k-1}\beta_1^p\sqrt{p}\right) - \frac{3R}{\sqrt{1-\beta_1}}\left(\sum_{l=0}^{k-1}\left(\frac{\beta_1}{\beta_2}\right)^l\sqrt{l+1}\|U_{k-l}\|_2^2\right). \tag{A.1}$$

*where $G_{k,i} = \nabla_i F(x_{k-1})$, $g_{k,i} = \nabla_i f_k(x_{k-1})$, $u_{k,i} = \frac{m_{k,i}}{\sqrt{\epsilon+v_{k,i}}}$, and $U_{k,i} = \frac{g_{k,i}}{\sqrt{\epsilon+v_{k,i}}}$.*

*Proof.* Following (Ward et al., 2020), we can use the following inequality in this proof, which we repeat here for convenience,

$$\forall \lambda > 0,\, x, y \in \mathbb{R}, xy \le \frac{\lambda}{2}x^2 + \frac{y^2}{2\lambda}. \tag{A.2}$$

Let us take an iteration $k \in \mathbb{N}^*$ for the duration of the proof. We have

$$\sum_{i\in[d]} G_{k,i}\frac{m_{k,i}}{\sqrt{\epsilon+v_{k,i}}} = \sum_{i\in[d]}\sum_{l=0}^{k-1}\beta_1^l G_{k,i}\frac{g_{k-l,i}}{\sqrt{\epsilon+v_{k,i}}}$$
$$= \underbrace{\sum_{i\in[d]}\sum_{k=0}^{k-1}\beta_1^l G_{k-l,i}\frac{g_{k-l,i}}{\sqrt{\epsilon+v_{k,i}}}}_{A} + \underbrace{\sum_{i\in[d]}\sum_{l=0}^{n-1}\beta_1^l\left(G_{k,i}-G_{k-l,i}\right)\frac{g_{k-l,i}}{\sqrt{\epsilon+v_{k,i}}}}_{B}, \tag{A.3}$$

Let us now take an index $0 \le l \le k-1$. We show that the contribution of past gradients $G_{k-l}$ and $g_{k-l}$ due to the heavy-ball momentum can be controlled thanks to the decay term $\beta_1^l$. Let us first have a look at $B$. Using equation A.2 with

$$\lambda = \frac{\sqrt{1-\beta_1}}{2R\sqrt{l+1}},\, x = |G_{k,i}-G_{k-l,i}|,\, y = \frac{|g_{k-l,i}|}{\sqrt{\epsilon+v_{n,i}}},$$

we have

$$|B| \le \sum_{i\in[d]}\sum_{l=0}^{k-1}\beta_1^l\left(\frac{\sqrt{1-\beta_1}}{4R\sqrt{l+1}}\left(G_{k,i}-G_{k-l,i}\right)^2 + \frac{R\sqrt{k+1}}{\sqrt{1-\beta_1}}\frac{g_{k-l,i}^2}{\epsilon+v_{k,i}}\right). \tag{A.4}$$

Notice first that for any dimension $i \in [d]$, $\epsilon + v_{k,i} \ge \epsilon + \beta_2^k v_{k-l,i} \ge \beta_2^k(\epsilon+v_{k-l,i})$, so that

$$\frac{g_{k-l,i}^2}{\epsilon+v_{k,i}} \le \frac{1}{\beta_2^l}U_{k-l,i}^2 \tag{A.5}$$

Besides, using the L-smoothness of $L$ given by Assumption 24, we have

$$\|G_k - G_{k-l}\|_2^2 \le L^2 \|x_{k-1}-x_{k-l-1}\|_2^2$$
$$= L^2\left\|\sum_{p=1}^{l}\alpha_{k-p}u_{k-p}\right\|_2^2$$
$$\le \alpha_n^2 L^2 k\sum_{p=1}^{l}\|u_{k-p}\|_2^2, \tag{A.6}$$

using Jensen inequality and the fact that $\alpha_k$ is non-decreasing. Injecting equation A.5 and equation A.6 into equation A.4, we obtain

$$|B| \leq \left( \sum_{l=0}^{k-1} \frac{\alpha_k^2 L^2}{4R} \sqrt{1-\beta_1} \beta_1^l \sqrt{l} \sum_{p=1}^{l} \|u_{k-p}\|_2^2 \right) + \left( \sum_{l=0}^{k-1} \frac{R}{\sqrt{1-\beta_1}} \left( \frac{\beta_1}{\beta_2} \right)^l \sqrt{l+1} \|U_{k-l}\|_2^2 \right)$$

$$= \sqrt{1-\beta_1} \frac{\alpha_k^2 L^2}{4R} \left( \sum_{l=1}^{k-1} \|u_{k-l}\|_2^2 \sum_{p=l}^{k-1} \beta_1^p \sqrt{p} \right) + \frac{R}{\sqrt{1-\beta_1}} \left( \sum_{l=0}^{k-1} \left( \frac{\beta_1}{\beta_2} \right)^l \sqrt{l+1} \|U_{k-l}\|_2^2 \right). \quad (A.7)$$

Now going back to the $A$ term in equation A.3, we will study the main term of the summation, i.e. for $i \in [d]$ and $l < k$

$$\mathbb{E}\left[ G_{k-l,i} \frac{g_{k-l,i}}{\sqrt{\epsilon + v_{k,i}}} \right] = \mathbb{E}\left[ \nabla_i F(x_{k-l-1}) \frac{\nabla_i f_{k-l}(x_{k-l-1})}{\sqrt{\epsilon + v_{k,i}}} \right]. \quad (A.8)$$

Notice that we could almost apply Lemma 71 to it, except that we have $v_{k,i}$ in the denominator instead of $v_{k-l,i}$. Thus we will need to extend the proof to decorrelate more terms. We will further drop indices in the rest of the proof, noting $G = G_{k-l,i}$, $g = g_{k-l,i}$, $\tilde{v} = \tilde{v}_{k,l+1,i}$ and $v = v_{k,i}$. Finally, let us note

$$\delta^2 = \sum_{j=k-l}^{k} \beta_2^{k-j} g_{j,i}^2 \qquad \text{and} \qquad r^2 = \mathbb{E}_{k-l-1}\left[ \delta^2 \right]. \quad (A.9)$$

In particular we have $\tilde{v} - v = r^2 - \delta^2$. With our new notations, we can rewrite equation A.8 as

$$\mathbb{E}\left[ G \frac{g}{\sqrt{\epsilon + v}} \right] = \mathbb{E}\left[ G \frac{g}{\sqrt{\epsilon + \tilde{v}}} + Gg \left( \frac{1}{\sqrt{\epsilon + v}} - \frac{1}{\sqrt{\epsilon + \tilde{v}}} \right) \right]$$

$$= \mathbb{E}\left[ \mathbb{E}_{k-l-1}\left[ G \frac{g}{\sqrt{\epsilon + \tilde{v}}} \right] + Gg \frac{r^2 - \delta^2}{\sqrt{\epsilon + v}\sqrt{\epsilon + \tilde{v}}(\sqrt{\epsilon + v} + \sqrt{\epsilon + \tilde{v}})} \right]$$

$$= \mathbb{E}\left[ \frac{G^2}{\sqrt{\epsilon + \tilde{v}}} \right] + \mathbb{E}\left[ \underbrace{Gg \frac{r^2 - \delta^2}{\sqrt{\epsilon + v}\sqrt{\epsilon + \tilde{v}}(\sqrt{\epsilon + v} + \sqrt{\epsilon + \tilde{v}})}}_{C} \right]. \quad (A.10)$$

We first focus on $C$:

$$|C| \leq \underbrace{|Gg| \frac{r^2}{\sqrt{\epsilon + v}(\epsilon + \tilde{v})}}_{\kappa} + \underbrace{|Gg| \frac{\delta^2}{(\epsilon + v)\sqrt{\epsilon + \tilde{v}}}}_{\rho},$$

due to the fact that $\sqrt{\epsilon + v} + \sqrt{\epsilon + \tilde{v}} \geq \max(\sqrt{\epsilon + v}, \sqrt{\epsilon + \tilde{v}})$ and $\left| r^2 - \delta^2 \right| \leq r^2 + \delta^2$.

Applying equation A.2 to $\kappa$ with

$$\lambda = \frac{\sqrt{1-\beta_1}\sqrt{\epsilon + \tilde{v}}}{2}, \quad x = \frac{|G|}{\sqrt{\epsilon + \tilde{v}}}, \quad y = \frac{|g| r^2}{\sqrt{\epsilon + \tilde{v}}\sqrt{\epsilon + v}},$$

we obtain

$$\kappa \leq \frac{G^2}{4\sqrt{\epsilon + \tilde{v}}} + \frac{1}{\sqrt{1-\beta_1}} \frac{g^2 r^4}{(\epsilon + \tilde{v})^{3/2}(\epsilon + v)}.$$

Given that $\epsilon + \tilde{v} \geq r^2$ and taking the conditional expectation, we can simplify as

$$\mathbb{E}_{k-l-1}\left[ \kappa \right] \leq \frac{G^2}{4\sqrt{\epsilon + \tilde{v}}} + \frac{1}{\sqrt{1-\beta_1}} \frac{r^2}{\sqrt{\epsilon + \tilde{v}}} \mathbb{E}_{k-l-1}\left[ \frac{g^2}{\epsilon + v} \right]. \quad (A.11)$$

Now turning to $\rho$, we use equation A.2 with

$$\lambda = \frac{\sqrt{1-\beta_1}\sqrt{\epsilon+\tilde{v}}}{2r^2}, \; x = \frac{|G\delta|}{\sqrt{\epsilon+\tilde{v}}}, \; y = \frac{|\delta g|}{\epsilon+v},$$

we obtain

$$\rho \leq \frac{G^2}{4\sqrt{\epsilon+\tilde{v}}}\frac{\delta^2}{r^2} + \frac{1}{\sqrt{1-\beta_1}}\frac{r^2}{\sqrt{\epsilon+\tilde{v}}}\frac{g^2\delta^2}{(\epsilon+v)^2}. \tag{A.12}$$

Given that $\epsilon+v \geq \delta^2$, and $\mathbb{E}_{k-l-1}\left[\frac{\delta^2}{r^2}\right] = 1$, we obtain after taking the conditional expectation,

$$\mathbb{E}_{k-l-1}\left[\rho\right] \leq \frac{G^2}{4\sqrt{\epsilon+\tilde{v}}} + \frac{1}{\sqrt{1-\beta_1}}\frac{r^2}{\sqrt{\epsilon+\tilde{v}}}\mathbb{E}_{k-l-1}\left[\frac{g^2}{\epsilon+v}\right]. \tag{A.13}$$

Notice that in A.12, we possibly divide by zero. It suffice to notice that if $r^2 = 0$ then $\delta^2 = 0$ a.s. so that $\rho = 0$ and equation A.13 is still verified. Summing equation A.11 and equation A.13, we get

$$\mathbb{E}_{k-l-1}\left[|C|\right] \leq \frac{G^2}{2\sqrt{\epsilon+\tilde{v}}} + \frac{2}{\sqrt{1-\beta_1}}\frac{r^2}{\sqrt{\epsilon+\tilde{v}}}\mathbb{E}_{k-l-1}\left[\frac{g^2}{\epsilon+v}\right]. \tag{A.14}$$

Given that $r \leq \sqrt{\epsilon+\tilde{v}}$ by definition of $\tilde{v}$, and that using Assumption 24, $r \leq \sqrt{k+1}R$, we have[1], reintroducing the indices we had dropped

$$\mathbb{E}_{k-l-1}\left[|C|\right] \leq \frac{G_{k-l,i}^2}{2\sqrt{\epsilon+\tilde{v}_{k,l+1,i}}} + \frac{2R}{\sqrt{1-\beta_1}}\sqrt{l+1}\mathbb{E}_{k-l-1}\left[\frac{g_{k-l,i}^2}{\epsilon+v_{k,i}}\right]. \tag{A.15}$$

Taking the complete expectation and using that by definition $\epsilon+v_{k,i} \geq \epsilon+\beta_2^k v_{k-l,i} \geq \beta_2^k(\epsilon+v_{k-l,i})$ we get

$$\mathbb{E}\left[|C|\right] \leq \frac{1}{2}\mathbb{E}\left[\frac{G_{k-l,i}^2}{\sqrt{\epsilon+\tilde{v}_{k,l+1,i}}}\right] + \frac{2R}{\sqrt{1-\beta_1}\beta_2^k}\sqrt{l+1}\mathbb{E}\left[\frac{g_{k-l,i}^2}{\epsilon+v_{k-l,i}}\right]. \tag{A.16}$$

Injecting equation A.16 into equation A.10 gives us

$$\mathbb{E}\left[A\right] \geq \sum_{i\in[d]}\sum_{l=0}^{k-1}\beta_1^l\left(\mathbb{E}\left[\frac{G_{k-l,i}^2}{\sqrt{\epsilon+\tilde{v}_{k,l+1,i}}}\right] - \left(\frac{1}{2}\mathbb{E}\left[\frac{G_{k-l,i}^2}{\sqrt{\epsilon+\tilde{v}_{k,l,i}}}\right] + \frac{2R}{\sqrt{1-\beta_1}\beta_2^k}\sqrt{l+1}\mathbb{E}\left[\frac{g_{k-l,i}^2}{\epsilon+v_{k-l,i}}\right]\right)\right)$$
$$= \frac{1}{2}\left(\sum_{i\in[d]}\sum_{l=0}^{k-1}\beta_1^l\mathbb{E}\left[\frac{G_{k-l,i}^2}{\sqrt{\epsilon+\tilde{v}_{k,l+1,i}}}\right]\right) - \frac{2R}{\sqrt{1-\beta_1}}\left(\sum_{i\in[d]}\sum_{l=0}^{k-1}\left(\frac{\beta_1}{\beta_2}\right)^l\sqrt{l+1}\mathbb{E}\left[\|U_{k-l}\|_2^2\right]\right). \tag{A.17}$$

Injecting equation A.17 and equation A.7 into equation A.3 finishes the proof.

$$\square$$

**Lemma 71** (adaptive update approximately follow a descent direction). *For all $n \in \mathbb{N}^*$ and $i \in [d]$, we have:*

$$\mathbb{E}_{n-1}\left[\nabla_i F(x_{n-1})\frac{\nabla_i f_n(x_{n-1})}{\sqrt{\epsilon+v_{n,i}}}\right] \geq \frac{(\nabla_i F(x_{n-1}))^2}{2\sqrt{\epsilon+\tilde{v}_{n,i}}} - 2R\mathbb{E}_{n-1}\left[\frac{(\nabla_i f_n(x_{n-1}))^2}{\epsilon+v_{n,i}}\right]. \tag{A.18}$$

*Proof.* We take $i \in [d]$ and note $G = \nabla_i F(x_{n-1})$, $g = \nabla_i f_n(x_{n-1})$, $v = v_{n,i}$ and $\tilde{v} = \tilde{v}_{n,i}$.

$$\mathbb{E}_{n-1}\left[\frac{Gg}{\sqrt{\epsilon+v}}\right] = \mathbb{E}_{n-1}\left[\frac{Gg}{\sqrt{\epsilon+\tilde{v}}}\right] + \mathbb{E}_{n-1}\left[\underbrace{Gg\left(\frac{1}{\sqrt{\epsilon+v}} - \frac{1}{\sqrt{\epsilon+\tilde{v}}}\right)}_{A}\right]. \tag{A.19}$$

---
[1]Note that we do not need the almost sure bound on the gradient, and a bound on $\mathbb{E}\left[\|\nabla f(x)\|_\infty^2\right]$ would be sufficient.

Given that $g$ and $\tilde{v}$ are independent knowing $f_1, \ldots, f_{n-1}$, we immediately have

$$\mathbb{E}_{n-1}\left[\frac{Gg}{\sqrt{\epsilon + \tilde{v}}}\right] = \frac{G^2}{\sqrt{\epsilon + \tilde{v}}}. \tag{A.20}$$

Now we need to control the size of the second term $A$,

$$A = Gg\frac{\tilde{v} - v}{\sqrt{\epsilon + v}\sqrt{\epsilon + \tilde{v}}(\sqrt{\epsilon + v} + \sqrt{\epsilon + \tilde{v}})}$$

$$= Gg\frac{\mathbb{E}_{n-1}\left[g^2\right] - g^2}{\sqrt{\epsilon + v}\sqrt{\epsilon + \tilde{v}}(\sqrt{\epsilon + v} + \sqrt{\epsilon + \tilde{v}})}$$

$$|A| \le \underbrace{|Gg|\frac{\mathbb{E}_{n-1}\left[g^2\right]}{\sqrt{\epsilon + v}(\epsilon + \tilde{v})}}_{\kappa} + \underbrace{|Gg|\frac{g^2}{(\epsilon + v)\sqrt{\epsilon + \tilde{v}}}}_{\rho}.$$

The last inequality comes from the fact that $\sqrt{\epsilon + v} + \sqrt{\epsilon + \tilde{v}} \ge \max(\sqrt{\epsilon + v}, \sqrt{\epsilon + \tilde{v}})$ and $\left|\mathbb{E}_{n-1}\left[g^2\right] - g^2\right| \le \mathbb{E}_{n-1}\left[g^2\right] + g^2$. Following (Ward et al., 2019), we can use the following inequality to bound $\kappa$ and $\rho$,

$$\forall \lambda > 0, \, x, y \in \mathbb{R}, xy \le \frac{\lambda}{2}x^2 + \frac{y^2}{2\lambda}. \tag{A.21}$$

First applying equation A.21 to $\kappa$ with

$$\lambda = \frac{\sqrt{\epsilon + \tilde{v}}}{2}, \; x = \frac{|G|}{\sqrt{\epsilon + \tilde{v}}}, \; y = \frac{|g|\,\mathbb{E}_{n-1}\left[g^2\right]}{\sqrt{\epsilon + \tilde{v}}\sqrt{\epsilon + v}},$$

we obtain

$$\kappa \le \frac{G^2}{4\sqrt{\epsilon + \tilde{v}}} + \frac{g^2\mathbb{E}_{n-1}\left[g^2\right]^2}{(\epsilon + \tilde{v})^{3/2}(\epsilon + v)}.$$

Given that $\epsilon + \tilde{v} \ge \mathbb{E}_{n-1}\left[g^2\right]$ and taking the conditional expectation, we can simplify as

$$\mathbb{E}_{n-1}\left[\kappa\right] \le \frac{G^2}{4\sqrt{\epsilon + \tilde{v}}} + \frac{\mathbb{E}_{n-1}\left[g^2\right]}{\sqrt{\epsilon + \tilde{v}}}\mathbb{E}_{n-1}\left[\frac{g^2}{\epsilon + v}\right]. \tag{A.22}$$

Given that $\sqrt{\mathbb{E}_{n-1}\left[g^2\right]} \le \sqrt{\epsilon + \tilde{v}}$ and $\sqrt{\mathbb{E}_{n-1}\left[g^2\right]} \le R$, we can simplify equation A.22 as

$$\mathbb{E}_{n-1}\left[\kappa\right] \le \frac{G^2}{4\sqrt{\epsilon + \tilde{v}}} + R\mathbb{E}_{n-1}\left[\frac{g^2}{\epsilon + v}\right]. \tag{A.23}$$

Now turning to $\rho$, we use equation A.21 with

$$\lambda = \frac{\sqrt{\epsilon + \tilde{v}}}{2\mathbb{E}_{n-1}\left[g^2\right]}, \; x = \frac{|Gg|}{\sqrt{\epsilon + \tilde{v}}}, \; y = \frac{g^2}{\epsilon + v},$$

we obtain

$$\rho \le \frac{G^2}{4\sqrt{\epsilon + \tilde{v}}}\frac{g^2}{\mathbb{E}_{n-1}\left[g^2\right]} + \frac{\mathbb{E}_{n-1}\left[g^2\right]}{\sqrt{\epsilon + \tilde{v}}}\frac{g^4}{(\epsilon + v)^2}, \tag{A.24}$$

Given that $\epsilon + v \ge g^2$ and taking the conditional expectation we obtain

$$\mathbb{E}_{n-1}\left[\rho\right] \le \frac{G^2}{4\sqrt{\epsilon + \tilde{v}}} + \frac{\mathbb{E}_{n-1}\left[g^2\right]}{\sqrt{\epsilon + \tilde{v}}}\mathbb{E}_{n-1}\left[\frac{g^2}{\epsilon + v}\right], \tag{A.25}$$

which we simplify using the same argument as for equation A.23 into

$$\mathbb{E}_{n-1}\left[\rho\right] \le \frac{G^2}{4\sqrt{\epsilon + \tilde{v}}} + R\mathbb{E}_{n-1}\left[\frac{g^2}{\epsilon + v}\right]. \tag{A.26}$$

Notice that in equation A.24, we possibly divide by zero. It suffice to notice that if $\mathbb{E}_{n-1}\left[g^2\right] = 0$ then $g^2 = 0$ a.s. so that $\rho = 0$ and equation A.26 is still verified. Summing equation A.23 and equation A.26 we can bound

$$\mathbb{E}_{n-1}\left[|A|\right] \le \frac{G^2}{2\sqrt{\epsilon + \tilde{v}}} + 2R\mathbb{E}_{n-1}\left[\frac{g^2}{\epsilon + v}\right]. \tag{A.27}$$

Injecting equation A.27 and equation A.20 into equation A.19 finishes the proof. $\square$

**Lemma 72** (sum of ratios of the square of a decayed sum and a decayed sum of square). *We assume we have $0 < \beta_2 \le 1$ and $0 < \beta_1 < \beta_2$, and a sequence of real numbers $(a_k)_{k \in \mathbb{N}^*}$. We define $b_k = \sum_{j=1}^{k} \beta_2^{k-j} a_j^2$ and $c_k = \sum_{j=1}^{k} \beta_1^{k-j} a_j$. Then we have*

$$\sum_{j=1}^{k} \frac{c_j^2}{\epsilon + b_j} \le \frac{1}{(1-\beta_1)(1-\beta_1/\beta_2)}\left(\ln\left(1 + \frac{b_k}{\epsilon}\right) - n\ln(\beta_2)\right). \tag{A.28}$$

*Proof.* Now let us take $j \in \mathbb{N}^*$, $j \le n$, we have using Jensen inequality

$$c_j^2 \le \frac{1}{1-\beta_1}\sum_{l=1}^{j} \beta_1^{j-l} a_l^2,$$

so that

$$\frac{c_j^2}{\epsilon + b_j} \le \frac{1}{1-\beta_1}\sum_{l=1}^{j} \beta_1^{j-l} \frac{a_l^2}{\epsilon + b_j}.$$

Given that for $l \in [j]$, we have by definition $\epsilon + b_j \ge \epsilon + \beta_2^{j-l} b_l \ge \beta_2^{j-l}(\epsilon + b_l)$, we get

$$\frac{c_j^2}{\epsilon + b_j} \le \frac{1}{1-\beta_1}\sum_{l=1}^{j} \left(\frac{\beta_1}{\beta_2}\right)^{j-l} \frac{a_l^2}{\epsilon + b_l}. \tag{A.29}$$

Thus, when summing over all $j \in [l]$, we get

$$\sum_{j=1}^{n} \frac{c_j^2}{\epsilon + b_j} \le \frac{1}{1-\beta_1}\sum_{j=1}^{n}\sum_{l=1}^{j} \left(\frac{\beta_1}{\beta_2}\right)^{j-l} \frac{a_l^2}{\epsilon + b_l}$$

$$= \frac{1}{1-\beta_1}\sum_{l=1}^{k} \frac{a_l^2}{\epsilon + b_l}\sum_{j=l}^{k} \left(\frac{\beta_1}{\beta_2}\right)^{j-l}$$

$$\le \frac{1}{(1-\beta_1)(1-\beta_1/\beta_2)}\sum_{l=1}^{k} \frac{a_l^2}{\epsilon + b_l}. \tag{A.30}$$

Given that ln is increasing, and the fact that $b_l > a_l^2 \ge 0$, we have for all $j \in \mathbb{N}^*$,

$$\frac{a_l^2}{\epsilon + b_l} \le \ln(\epsilon + b_l) - \ln(\epsilon + b_l - a_l^2)$$

$$= \ln(\epsilon + b_l) - \ln(\epsilon + \beta_2 b_{l-1})$$

$$= \ln\left(\frac{\epsilon + b_l}{\epsilon + b_{l-1}}\right) + \ln\left(\frac{\epsilon + b_{l-1}}{\epsilon + \beta_2 b_{l-1}}\right).$$

The first term forms a telescoping series, while the second one is bounded by $-\ln(\beta_2)$. Summing over all $j \in [K]$ gives the desired result.

$\square$

We also need two technical lemmas on the sum of series.

**Lemma 73** (sum of a geometric term times a square root). *Given $0 < a < 1$ and $Q \in \mathbb{N}$, we have,*

$$\sum_{q=0}^{Q-1} a^q \sqrt{q+1} \le \frac{1}{1-a}\left(1 + \frac{\sqrt{\pi}}{2\sqrt{-\ln(a)}}\right) \le \frac{2}{(1-a)^{3/2}}. \tag{A.31}$$

*Proof.* We first need to study the following integral:

$$\int_0^\infty \frac{a^x}{2\sqrt{x}}\mathrm{d}x = \int_0^\infty \frac{\mathrm{e}^{\ln(a)x}}{2\sqrt{x}}\mathrm{d}x \quad \text{, then introducing } y = \sqrt{x},$$

$$= \int_0^\infty \mathrm{e}^{\ln(a)y^2}\mathrm{d}y \quad \text{, then introducing } u = \sqrt{-2\ln(a)}y,$$

$$= \frac{1}{\sqrt{-2\ln(a)}}\int_0^\infty \mathrm{e}^{-u^2/2}\mathrm{d}u$$

$$\int_0^\infty \frac{a^x}{2\sqrt{x}}\mathrm{d}x = \frac{\sqrt{\pi}}{2\sqrt{-\ln(a)}}, \tag{A.32}$$

where we used the classical integral of the standard Gaussian density function.

Let us now introduce $A_Q$:

$$A_Q = \sum_{q=0}^{Q-1} a^q \sqrt{q+1},$$

then we have

$$A_Q - aA_Q = \sum_{q=0}^{Q-1} a^q\sqrt{q+1} - \sum_{q=1}^{Q} a^q\sqrt{q} \quad \text{, then using the concavity of } \sqrt{\cdot},$$

$$\le 1 - a^Q\sqrt{Q} + \sum_{q=1}^{Q-1} \frac{a^q}{2\sqrt{q}}$$

$$\le 1 + \int_0^\infty \frac{a^x}{2\sqrt{x}}\mathrm{d}x$$

$$(1-a)A_Q \le 1 + \frac{\sqrt{\pi}}{2\sqrt{-\ln(a)}},$$

where we used equation A.32. Given that $\sqrt{-\ln(a)} \ge \sqrt{1-a}$ we obtain equation A.31. $\square$

**Lemma 74** (sum of a geometric term times roughly a power 3/2). *Given $0 < a < 1$ and $Q \in \mathbb{N}$, we have,*

$$\sum_{q=0}^{Q-1} a^q\sqrt{q}(q+1) \le \frac{4a}{(1-a)^{5/2}}. \tag{A.33}$$

*Proof.* Let us introduce $A_Q$:

$$A_Q = \sum_{q=0}^{Q-1} a^q\sqrt{q}(q+1),$$

then we have

$$
\begin{aligned}
A_Q - aA_Q &= \sum_{q=0}^{Q-1} a^q \sqrt{q}(q+1) - \sum_{q=1}^{Q} a^q \sqrt{q-1}\,q \\
&\leq \sum_{q=1}^{Q-1} a^q \sqrt{q}\left((q+1) - \sqrt{q}\sqrt{q-1}\right) \\
&\leq \sum_{q=1}^{Q-1} a^q \sqrt{q}\left((q+1) - (q-1)\right) \\
&\leq 2\sum_{q=1}^{Q-1} a^q \sqrt{q} \\
&= 2a \sum_{q=0}^{Q-2} a^q \sqrt{q+1} \quad \text{,then using Lemma 73,} \\
(1-a)A_Q &\leq \frac{4a}{(1-a)^{3/2}}.
\end{aligned}
$$

$\square$

# B  Proof for BFGS

In this section, we prove the two lemmas introduced in the main body of the paper used for the convergence proof of BFGS.

**Lemma 32.** *If* $\overline{\mathbf{H}}, \mathbf{H} \in L(\mathbb{R}^n)$, $s, y, d \in \mathbb{R}^n$ *satisfy*

$$
\overline{\mathbf{H}} = \mathbf{H} + \frac{(s-\mathbf{H}y)d^T + d(s-\mathbf{H}y)^T}{d^T y} - \frac{y^T(s-\mathbf{H}y)dd^T}{(d^T y)^2} \tag{2.100}
$$

*and* $\mathbf{M} \in \mathbf{\mathit{L}}(\mathbb{R}^n)$ *is a non-singular symmetric matrix, then for all* $\mathbf{A} \in L(\mathbb{R}^n)$ *it follows that*

$$
\overline{\mathbf{E}} = \mathbf{P}^T \mathbf{E} \mathbf{P} + \frac{\mathbf{M}(s-\mathbf{A}y)(\mathbf{M}d)^T}{d^T y} + \frac{\mathbf{M}d(s-\mathbf{A}y)^T \mathbf{M} \mathbf{P}}{d^T y},
$$

$$
\mathbf{P} = \mathbf{I} - \frac{(\mathbf{M}^{-1}y)(\mathbf{M}d)^T}{d^T y},
$$

*where* $\mathbf{E} = \mathbf{M}(\mathbf{H} - \mathbf{A})\mathbf{M}$, $\overline{\mathbf{E}} = \mathbf{M}(\overline{\mathbf{H}} - \mathbf{A})\mathbf{M}$.

*Proof.*

$$
\overline{\mathbf{E}} = \mathbf{E} + \mathbf{M}\left[\frac{(\mathbf{s}-\mathbf{H}\mathbf{y})\mathbf{d}^T + \mathbf{d}(\mathbf{s}-\mathbf{H}\mathbf{y})^T}{\mathbf{d}^T\mathbf{y}} - \frac{\mathbf{y}^T(\mathbf{s}-\mathbf{H}\mathbf{y})\mathbf{d}\mathbf{d}^T}{(\mathbf{d}^T\mathbf{y})^2}\right]\mathbf{M} \tag{B.1}
$$

Since $\mathbf{E} = \mathbf{M}(\mathbf{H}-\mathbf{A})\mathbf{M}$, we have $\mathbf{H} = \mathbf{M}^{-1}\mathbf{E}\mathbf{M}^{-1} + \mathbf{A}$, substitute it into (B.1):

$$
\begin{aligned}
\overline{\mathbf{E}} = \mathbf{E} + \mathbf{M}&\left[-\frac{\mathbf{y}^T(\mathbf{s}-\mathbf{A}\mathbf{y}-\mathbf{M}^{-1}\mathbf{E}\mathbf{M}^{-1})\mathbf{d}\mathbf{d}^T}{(\mathbf{d}^T\mathbf{y})^2}\right. \\
&\left.+\frac{(\mathbf{s}-\mathbf{A}\mathbf{y}-\mathbf{M}^{-1}\mathbf{E}\mathbf{M}^{-1})\mathbf{d}^T + \mathbf{d}(\mathbf{s}-\mathbf{A}\mathbf{y}-\mathbf{M}^{-1}\mathbf{E}\mathbf{M}^{-1})^T}{\mathbf{d}^T\mathbf{y}}\right]\mathbf{M} \\
= \mathbf{E} &+ \frac{\mathbf{M}(\mathbf{s}-\mathbf{A}\mathbf{y})\mathbf{d}^T\mathbf{M}}{\mathbf{d}^T\mathbf{y}} + \frac{\mathbf{M}\mathbf{d}(\mathbf{s}-\mathbf{A}\mathbf{y})\mathbf{M}}{\mathbf{d}^T\mathbf{y}} \\
&- \frac{\mathbf{M}\mathbf{y}^T(\mathbf{s}-\mathbf{A}\mathbf{y})\mathbf{d}\mathbf{d}^T\mathbf{M}}{\mathbf{d}^T\mathbf{y}} - \frac{\mathbf{E}\mathbf{M}^{-1}\mathbf{y}\mathbf{d}^T\mathbf{M}}{\mathbf{d}^T\mathbf{y}} \\
&+ \frac{\mathbf{M}\mathbf{d}\mathbf{y}^T\mathbf{M}^{-1}\mathbf{E}}{\mathbf{d}^T\mathbf{y}} + \frac{(\mathbf{y}^T\mathbf{M}^{-1}\mathbf{E}\mathbf{M}^{-1}\mathbf{y})\mathbf{M}\mathbf{d}\mathbf{d}^T\mathbf{M}}{(\mathbf{d}^T\mathbf{y})^2},
\end{aligned} \tag{B.2}
$$

since

$$\mathbf{P}^T\mathbf{EP} = \left(\mathbf{I} - \frac{(\mathbf{M}^{-1}\mathbf{y})(\mathbf{Md})^T}{\mathbf{d}^T\mathbf{y}}\right)\mathbf{E}\left(\mathbf{I} - \frac{(\mathbf{M}^{-1}\mathbf{y})(\mathbf{Md})^T}{\mathbf{d}^T\mathbf{y}}\right)$$

$$= \mathbf{E} - \frac{\mathbf{EM}^{-1}\mathbf{yd}^T\mathbf{M}}{\mathbf{d}^T\mathbf{y}} + \frac{\mathbf{Mdy}^T\mathbf{M}^{-1}\mathbf{E}}{\mathbf{d}^T\mathbf{y}}$$

$$+ \frac{(\mathbf{y}^T\mathbf{M}^{-1}\mathbf{EM}^{-1}\mathbf{y})\mathbf{Mdd}^T\mathbf{M}}{(\mathbf{d}^T\mathbf{y})^2}, \tag{B.3}$$

combine (B.2) and (B.3), we could conclude

$$\overline{\mathbf{E}} = \mathbf{P}^T\mathbf{EP} + \frac{\mathbf{M}(\mathbf{s}-\mathbf{Ay})(\mathbf{Md})^T}{\mathbf{d}^T\mathbf{y}} + \frac{\mathbf{Md}(\mathbf{s}-\mathbf{Ay})^T\mathbf{MP}}{\mathbf{d}^T\mathbf{y}}.$$

$\square$

**Lemma 33.** *Let* $\mathbf{M} \in \text{Ł}(\mathbb{R}^n)$ *be a non-singular symmetric matrix such that*

$$\left\|\mathbf{M}d - \mathbf{M}^{-1}y\right\| \le \beta \left\|\mathbf{M}^{-1}y\right\| \tag{2.101}$$

*for some* $\beta \in [0, \frac{1}{3}]$ *and* $d, y \in \mathbb{R}^n$ *with* $y \neq 0$. *Then*

    *a)* $(1-\beta)\left\|\mathbf{M}^{-1}y\right\|^2 \le d^T y \le (1+\beta)\left\|\mathbf{M}^{-1}y\right\|^2$,

*for non-zero* $\mathbf{E} \in L(\mathbb{R}^n)$

    *b)* $\left\|\mathbf{E}\left(\mathbf{I} - \frac{(\mathbf{M}^{-1}y)(\mathbf{M}^{-1}y)^T}{d^T y}\right)\right\|_F \le \sqrt{1 - \alpha\theta^2}\left\|\mathbf{E}\right\|_F$,

    *c)* $\left\|\mathbf{EP}\right\|_F \le \left[\sqrt{1-\alpha\theta^2} + (1-\beta)^{-1}\frac{\left\|\mathbf{M}d - \mathbf{M}^{-1}y\right\|}{\left\|\mathbf{M}^{-1}y\right\|}\right]\left\|\mathbf{E}\right\|_F$,

*where*

$$\mathbf{P} = \mathbf{I} - \frac{(\mathbf{M}^{-1}y)(\mathbf{M}d)^T}{d^T y}, \alpha = \frac{1-2\beta}{1-\beta^2}, \quad \theta = \frac{\left\|\mathbf{EM}^{-1}\right\|y}{\left\|\mathbf{E}\right\|_F\left\|\mathbf{M}^{-1}y\right\|},$$

*moreover, for* $\forall s \in \mathbb{R}^n$ *and* $\forall \mathbf{A} \in L(\mathbb{R}^n)$

    *d)* $\left\|\frac{(s-\mathbf{A}y)(\mathbf{M}d)^T}{d^T y}\right\|_F \le 2\frac{\left\|s-\mathbf{A}y\right\|}{\left\|\mathbf{M}^{-1}y\right\|}$.

*Proof.*

$$\mathbf{d}^T\mathbf{y} = (\mathbf{Md})^T\mathbf{M}^{-1}\mathbf{y} = (\mathbf{Md} - \mathbf{M}^{-1}\mathbf{y})^T\mathbf{M}^{-1}\mathbf{y} + \left\|\mathbf{M}^{-1}\mathbf{y}\right\|^2,$$

by (2.101), $\left\|(\mathbf{Md}-\mathbf{M}^{-1}\mathbf{y})^T\mathbf{M}^{-1}\mathbf{y}\right\| \le \beta\left\|\mathbf{M}^{-1}\mathbf{y}\right\|^2$, so

$$(1-\beta)\left\|\mathbf{M}^{-1}\mathbf{y}\right\|^2 \le \mathbf{d}^T\mathbf{y} \le (1+\beta)\left\|\mathbf{M}^{-1}\mathbf{y}\right\|^2. \tag{$a$}$$

Because

$$\left\|\mathbf{E}\left[\mathbf{I} - \mathbf{uv}^T\right]\right\|_F = \|E\|_F^2 - 2\mathbf{v}^T\mathbf{E}^T\mathbf{Eu} + \|\mathbf{Eu}\|^2\|\mathbf{v}\|^2,$$

let $\mathbf{u} = \frac{\mathbf{M}^{-1}\mathbf{y}}{\mathbf{d}^T\mathbf{y}}$, $\mathbf{v} = \mathbf{M}^{-1}\mathbf{y}$,

$$\left\| \mathbf{E}^2 \left( \mathbf{I} - \frac{(\mathbf{M}^{-1}\mathbf{y})(\mathbf{M}^{-1}\mathbf{y})^T}{\mathbf{d}^T\mathbf{y}} \right) \right\|_F$$

$$= \|\mathbf{E}\|_F^2 - 2\mathbf{y}\mathbf{M}^{-1}\mathbf{E}^T\mathbf{E}\frac{\mathbf{M}^{-1}\mathbf{y}}{\mathbf{d}^T\mathbf{y}} + \left\| \mathbf{E}\frac{\mathbf{M}^{-1}\mathbf{y}}{\mathbf{d}^T\mathbf{y}} \right\|_F^2 \|\mathbf{M}^{-1}\mathbf{y}\|$$

$$= \|\mathbf{E}\|_F^2 + (\|\mathbf{M}^{-1}\mathbf{y}\| - 2\mathbf{d}^T\mathbf{y})\frac{\|\mathbf{E}\mathbf{M}^{-1}\mathbf{y}\|^2}{(\mathbf{d}^T\mathbf{y})^2}$$

$$\leq \|\mathbf{E}\|_F^2 + (\frac{\|\mathbf{M}^{-1}\mathbf{y}\|}{\mathbf{d}^T\mathbf{y}} - 2)\frac{\|\mathbf{E}\mathbf{M}^{-1}\mathbf{y}\|^2}{\mathbf{d}^T\mathbf{y}}$$

$$\leq \|\mathbf{E}\|_F^2 + (\frac{1}{1-\beta} - 2)\frac{\|\mathbf{E}\mathbf{M}^{-1}\mathbf{y}\|^2}{\mathbf{d}^T\mathbf{y}}$$

$$\leq \|\mathbf{E}\|_F^2 - \frac{1-2\beta}{1-\beta^2}\left( frac\|\mathbf{E}\mathbf{M}^{-1}\mathbf{y}\|\mathbf{d}^T\mathbf{y}\right)^2$$

$$= \|\mathbf{E}\|_F^2 - \alpha \left( \frac{\|\mathbf{E}\mathbf{M}^{-1}\mathbf{y}\|}{\mathbf{d}^T\mathbf{y}} \right)^2$$

$$= (1 - \alpha\theta^2)\|\mathbf{E}\|_F^2 , \qquad\qquad (b)$$

where $\alpha = \frac{1-2\beta}{1-\beta^2}$, $\theta = \frac{\|\mathbf{E}\mathbf{M}^{-1}\|\mathbf{y}}{\|\mathbf{E}\|_F\|\mathbf{M}^{-1}\mathbf{y}\|}$.

$$\left\| \mathbf{E} \left( \mathbf{I} - \frac{(\mathbf{M}^{-1}\mathbf{y})(\mathbf{M}\mathbf{d})^T}{\mathbf{d}^T\mathbf{y}} \right) \right\|_F$$

$$= \left\| \mathbf{E} \left( \mathbf{I} - \frac{\mathbf{M}^{-1}\mathbf{y}(\mathbf{M}^{-1}\mathbf{y})^T}{\mathbf{d}^T\mathbf{y}} \right) + \frac{\mathbf{M}^{-1}\mathbf{y}(\mathbf{M}^{-1}\mathbf{y} - \mathbf{M}^{-1}\mathbf{d}))^T}{\mathbf{d}^T\mathbf{y}} \right\|_F$$

$$\leq \sqrt{1 - \alpha\theta^2}\|\mathbf{E}\|_F + \frac{\|\mathbf{M}^{-1}\mathbf{y}\|^2\|\mathbf{M}^{-1}\mathbf{y} - \mathbf{M}^{-1}\mathbf{d}\|}{\mathbf{d}^T\mathbf{y}\|\mathbf{M}^{-1}\mathbf{y}\|}\|\mathbf{E}\|_F$$

$$= \left[ \sqrt{1 - \alpha\theta^2} + \frac{1}{1-\beta}\frac{\|\mathbf{M}^{-1}\mathbf{y} - \mathbf{M}^{-1}\mathbf{d}\|}{\|\mathbf{M}^{-1}\mathbf{y}\|} \right]\|\mathbf{E}\|_F . \qquad (c)$$

We already prove condition (a-c). For condition (d):

$$\left\| \frac{(\mathbf{s} - \mathbf{A}\mathbf{y})(\mathbf{M}\mathbf{d})^T}{\mathbf{d}^T\mathbf{y}} \right\|_F$$

$$\leq \frac{\|\mathbf{s} - \mathbf{A}\mathbf{y}\|}{\mathbf{d}^T\mathbf{y}}(\|\mathbf{M}^{-1}\mathbf{y} - \mathbf{M}^{-1}\mathbf{d})\| + \|\mathbf{M}^{-1}\mathbf{y}\|)$$

$$\leq (1 + \beta)\|\mathbf{s} - \mathbf{A}\mathbf{y}\|\frac{\|\mathbf{M}^{-1}\mathbf{y}\|}{\mathbf{d}^T\mathbf{y}}$$

$$\leq \frac{1 + \beta}{1 - \beta}\frac{\|\mathbf{s} - \mathbf{A}\mathbf{y}\|}{\|\mathbf{M}^{-1}\mathbf{y}\|} \leq 2\frac{\|\mathbf{s} - \mathbf{A}\mathbf{y}\|}{\|\mathbf{M}^{-1}\mathbf{y}\|} \qquad\qquad (d)$$

$\square$

# C    Proof For Generalization ability of LPF-SGD.

## C.1    Proof for Theorem 42

Proof in this section is inspired by the analysis in (Duchi et al., 2012).

**Lemma 75.** *Let $f(x;\xi)$ be $M$-Lipschitz continuous with respect to $l_2$-norm. Let variable $Z$ be distributed according to the distribution $\mu$. Then*

$$\|\nabla f_\mu(u;\xi) - \nabla f_\mu(v;\xi)\| = \mathbb{E}_{Z\sim\mu}[\nabla f_\mu(u+Z;\xi) - \nabla f_\mu(v+Z;\xi)] \tag{C.1}$$

$$\leq M \int |\mu(z-u) - \mu(z-v)|dz.$$

*If distribution $\mu$ is rotationally symmetric and non-increasing, the bound is tight and can be attained by the function*

$$f(x;\xi) = M\frac{\|u\|^2 + \|v\|^2}{\|u-v\|}\left|< \frac{u-v}{\|u\|^2 + \|v\|^2}, x > -\frac{1}{2}\right|.$$

*Proof.* Let $Z$ be the random variable satisfies distribution $\mu$.

$$\mathbb{E}_{Z\sim\mu}[\nabla f_\mu(u+Z;\xi) - \nabla f_\mu(v+Z;\xi)]$$
$$= \int \nabla f_\mu(u+Z;\xi)\mu(z)dz - \int \nabla f_\mu(v;\xi)\mu(z)dz$$
$$= \int \nabla f_\mu(u;\xi)\mu(z)dz - \int \nabla f_\mu(v;\xi)\mu(z)dz$$
$$= \int_{I_>} \nabla f(z)[\mu(z-u) - \mu(z-v)]dz - \int_{I_<} \nabla F(x_0)(z)[\mu(z-v) - \mu(z-u)]dz$$

where

$$I_> = \{z \in \mathbb{R}^d | \mu(z-u) > \mu(z-v)\},$$
$$I_< = \{z \in \mathbb{R}^d | \mu(z-u) < \mu(z-v)\}.$$

Obviously,

$$\|\mathbb{E}_{Z\sim\mu}[\nabla f_\mu(u+Z;\xi) - \nabla f_\mu(v+Z;\xi)]\|$$
$$\leq \sup_{z\in I_>\cup I_<}\|\nabla f(z)\|\left|\int_{I_>}[\mu(z-u) - \mu(z-v)]dz - \int_{I_<} f(z)[\mu(z-v) - \mu(z-u)]dz\right|$$
$$\leq M\left|\int_{I_>}[\mu(z-u) - \mu(z-v)]dz - \int_{I_<} f(z)[\mu(z-v) - \mu(z-u)]dz\right|$$
$$= M\int |\mu(z-u) - \mu(z-v)|dz.$$

We already prove the inequality C.1. We are going to show that the bound is tight and could be attained. Since $\mu$ is an rotationally symmetric and non-increasing, the set $I_>$ could be rewritten as

$$I_> = \{z \in \mathbb{R}^d | \mu(z-u) > \mu(z-v)\}$$
$$= \{z \in \mathbb{R}^d | \|z-u\|^2 > \|z-v\|^2\}$$
$$= \{z \in \mathbb{R}^d | \langle z, u-v \rangle > \frac{1}{2}(\|u\|^2 + \|v\|^2)\},$$

similarly,

$$I_< = \{z \in \mathbb{R}^d | \langle z, u-v \rangle < \frac{1}{2}(\|u\|^2 + \|v\|^2)\}.$$

For given $u, v$, define function $f$ as

$$f(x;\xi) = M\frac{\|u\|^2 + \|v\|^2}{\|u-v\|}\left|< \frac{u-v}{\|u\|^2 + \|v\|^2}, x > -\frac{1}{2}\right|.$$

Therefore, the gradient of function $f$ is

$$\nabla f(x;\xi) = \begin{cases} M\dfrac{u-v}{\|u-v\|} & \text{if}\langle x, u-v\rangle > \dfrac{1}{2}(\|u\|^2 + \|v\|^2) \\ -M\dfrac{u-v}{\|u-v\|} & \text{if}\langle x, u-v\rangle < \dfrac{1}{2}(\|u\|^2 + \|v\|^2) \end{cases} \tag{C.2}$$

Hence,

$$\|\mathbb{E}_{Z\sim\mu}[\nabla f_\mu(u+Z;\xi) - \nabla f_\mu(v+Z;\xi)]\|$$

$$= \left\|\int_{I_>} \nabla f(z)[\mu(z-u) - \mu(z-v)]dz - \int_{I_<} \nabla f(z)[\mu(z-v) - \mu(z-u)]dz\right\|$$

$$= \left\|\int_{I_>} M\frac{u-v}{\|u-v\|}[\mu(z-u) - \mu(z-v)]dz + \int_{I_<} M\frac{u-v}{\|u-v\|}[\mu(z-v) - \mu(z-u)]dz\right\|$$

$$= \left\|M\frac{u-v}{\|u-v\|}\int |\mu(z-u) - \mu(z-v)|dz\right\|$$

$$= M\int |\mu(z-u) - \mu(z-v)|dz\left\|\frac{u-v}{\|u-v\|}\right\|$$

$$= M\int |\mu(z-u) - \mu(z-v)|dz$$

We already show that the equality holds for given function $f$. Therefore the bound is tight. □

**Theorem 42.** *(Theorem 1 in (Bisla et al., 2022)) Let $\mu$ be the $\mathcal{N}(0, \sigma^2 I_{d\times d})$ distribution. Assume the differentiable loss function $f(x;\xi) : \mathbb{R}^d \to \mathbb{R}$ is $M$-Lipschitz continuous and $L$-smooth with respect to $l_2$-norm. The smoothed loss function $f_\mu(x;\xi)$ is defined as (3.26). Then the following properties hold:*

*i) $f_\mu$ is $M$-Lipschitz continuous.*

*ii) $f_\mu$ is continuously differentiable; moreover, its gradient is $\min\{\frac{M}{\sigma}, L\}$-Lipschitz continuous, i.e., $f_\mu$ is $\min\{\frac{M}{\sigma}, L\}$-smooth.*

*iii) If $f$ is convex, $f(x;\xi) \le f_\mu(x;\xi) \le f(x;\xi) + \sigma M\sqrt{d}$.*

*In addition, for each bound i)-iii), there exists a function $l$ such that the bound cannot be improved by more than a constant factor.*

*Proof.* We are going the prove the properties one by one.

i) Since $\nabla f_\mu(x;\xi) = \mathbb{E}_{Z\sim\mu}[\nabla f(x+Z;\xi)]$, we have

$$\|\nabla f_\mu(x;\xi)\| = \|\mathbb{E}_{Z\sim\mu}[\nabla f(x+Z;\xi)]\| \le \mathbb{E}_{Z\sim\mu}[\|\nabla f(x+Z;\xi)\|] \le M.$$

Therefore, $f_\mu$ is $M$-Lipschitz continuous. To prove the bound is tight, we define

$$f(x;\xi) = \frac{1}{2}v^T x,$$

where $v \in \mathbb{R}^d$ is a scalar. Hence, we have

$$f_\mu(x;\xi) = \mathbb{E}_{Z\sim\mu}[f(x+Z;\xi)] = \mathbb{E}_{Z\sim\mu}[\frac{1}{2}v^T(x-Z)] = \frac{1}{2}v^T x = f(x;\xi).$$

Both $f$ and smoothed $f_\mu$ have the gradient $v$ and $f_\mu$ is exactly $M$-Lipschitz.

ii) The proof scheme for this part is organized as follow: Firstly we show that $f_\mu$ is $\frac{M}{\sigma}$-smooth and the bound can not be improved by more than a constant factor. Then we show that $f_\mu$ is $L$-smooth and the bound can not be improved by more than a constant factor as well. In all, we could draw the conclusion that $f_\mu$ is $\min\{\frac{M}{\sigma}, L\}$-smooth and the bound is tight.

By Lemma 75, for $\forall u, v \in \mathbb{R}^n$,

$$\|\nabla f_\mu(u;\xi) - \nabla f_\mu(v;\xi)\| \le M \underbrace{\int |\mu(z-u) - \mu(z-v)| dz}_{I_2}. \tag{C.3}$$

Denote the integral as $I_2$. We follow a technique used in (Lakshmanan & Pucci De Farias, 2008; Duchi et al., 2012). Since $\mu(z-u) \ge \mu(z-v)$ is equivalent to $\|z-u\| \ge \|z-v\|$,

$$
\begin{aligned}
I_2 &= \int |\mu(z-u) - \mu(z-v)| dz \\
&= \int_{z:\|z-u\|\ge\|z-v\|} [\mu(z-u) - \mu(z-v)] dz + \int_{z:\|z-u\|\le\|z-v\|} [\mu(z-v) - \mu(z-u)] dz \\
&= 2 \int_{z:\|z-u\|\le\|z-v\|} [\mu(z-u) - \mu(z-v)] dz \\
&= 2 \int_{z:\|z-u\|\le\|z-v\|} \mu(z-u) dz - 2 \int_{z:\|z-u\|\le\|z-v\|} \mu(z-v) dz.
\end{aligned}
$$

Denote $w = z - u$ for $\mu(z-u)$ term and $w = z - v$ for $\mu(z-v)$ term, we have

$$
\begin{aligned}
I_2 &= 2 \int_{z:\|w\|\le\|w-(u-v)\|} \mu(w) dz - 2 \int_{z:\|w\|\ge\|w-(u-v)\|} \mu(w) dz \\
&= 2\mathbb{P}_{Z\sim\mu}(\|Z\| \le \|Z-(u-v)\|) - 2\mathbb{P}_{Z\sim\mu}(\|Z\| \ge \|Z-(u-v)\|).
\end{aligned}
$$

Obviously,

$$
\begin{aligned}
&\mathbb{P}_{Z\sim\mu}(\|Z\| \le \|Z-(u-v)\|) \\
&= \mathbb{P}_{Z\sim\mu}(\|Z\|^2 \le \|Z-(u-v)\|^2) \\
&= \mathbb{P}_{Z\sim\mu}(2\langle z, u-v\rangle \le \|u-v\|^2) \\
&= \mathbb{P}_{Z\sim\mu}(2\langle z, \frac{u-v}{\|u-v\|}\rangle \le \|u-v\|),
\end{aligned}
$$

$\frac{u-v}{\|u-v\|}$ has norm 1 and $Z \sim \mathcal{N}(0, \sigma^2 I)$ implies $\langle z, \frac{u-v}{\|u-v\|}\rangle \sim \mathcal{N}(0, \sigma^2 I)$. Hence, we have

$$
\begin{aligned}
&\mathbb{P}_{Z\sim\mu}(\|Z\| \le \|Z-(u-v)\|) \\
&= \mathbb{P}_{Z\sim\mu}(\langle z, \frac{u-v}{\|u-v\|}\rangle \le \frac{\|u-v\|}{2}) \\
&= \int_{-\infty}^{\frac{\|u-v\|}{2}} \frac{1}{\sqrt{2\pi\sigma^2}} \exp(-\frac{w^2}{2\sigma^2}) dw.
\end{aligned}
$$

Similarly,

$$
\begin{aligned}
&\mathbb{P}_{Z\sim\mu}(\|Z\| \ge \|Z-(u-v)\|) \\
&= \mathbb{P}_{Z\sim\mu}(\langle z, \frac{u-v}{\|u-v\|}\rangle \ge \frac{\|u-v\|}{2}) \\
&= \int_{\frac{\|u-v\|}{2}}^{+\infty} \frac{1}{\sqrt{2\pi\sigma^2}} \exp(-\frac{w^2}{2\sigma^2}) dw.
\end{aligned}
$$

Therefore,

$$
\begin{aligned}
I_2 &= 2 \int_{-\infty}^{\frac{\|u-v\|}{2}} \frac{1}{\sqrt{2\pi\sigma^2}} \exp(-\frac{w^2}{2\sigma^2}) dw - 2 \int_{\frac{\|u-v\|}{2}}^{+\infty} \frac{1}{\sqrt{2\pi\sigma^2}} \exp(-\frac{w^2}{2\sigma^2}) dw \\
&= 2 \int_{-\frac{\|u-v\|}{2}}^{\frac{\|u-v\|}{2}} \frac{1}{\sqrt{2\pi\sigma^2}} \exp(-\frac{w^2}{2\sigma^2}) dw \\
&\leq \frac{\sqrt{2}\|u-v\|}{\sigma\sqrt{\pi}}
\end{aligned}
\tag{C.4}
$$

In conclusion, combine formula (C.3) and (C.4) we have

$$
\|\nabla f_\mu(u;\xi) - \nabla f_\mu(v;\xi)\| \leq M \frac{\sqrt{2}\|u-v\|}{\sigma\sqrt{\pi}} \leq \frac{M}{\sigma}\|u-v\|.
$$

We finish proving that $f_\mu$ is $\frac{M}{\sigma}$-smooth. We are going to show the bound is tight. For any given $x, y$, define function $f$ as

$$
f(x;\xi) = M \frac{\|x\|^2 + \|y\|^2}{\|u-v\|} \left| < \frac{u-v}{\|x\|^2 + \|y\|^2}, x > -\frac{1}{2} \right|,
$$

Uniform Lemma 75 and former proof, we know that

$$
\begin{aligned}
\|\nabla f_\mu(u;\xi) - \nabla f_\mu(v;\xi)\| &= M \int |\mu(z-u) - \mu(z-v)| dz \\
&= \frac{\sqrt{2}M}{\sigma\sqrt{\pi}} \int_{-\frac{\|u-v\|}{2}}^{\frac{\|u-v\|}{2}} \exp(-\frac{w^2}{2\sigma^2}) dw
\end{aligned}
\tag{C.5}
$$

Because

$$
\frac{\sqrt{2}M}{\sigma\sqrt{\pi}} \exp(-\frac{\|u-v\|^2}{8\sigma^2}) \|u-v\| \leq \frac{\sqrt{2}M}{\sigma\sqrt{\pi}} \int_{-\frac{\|u-v\|}{2}}^{\frac{\|u-v\|}{2}} \exp(-\frac{w^2}{2\sigma^2}) dw \leq \frac{\sqrt{2}M}{\sigma\sqrt{\pi}} \|u-v\|
$$

Obviously, taking $x, y$ such that $\|u-v\| \leq 2\sqrt{2}\sigma$,

$$
\frac{\sqrt{2}M}{e\sigma\sqrt{\pi}} \|u-v\| \leq \|\nabla f_\mu(u;\xi) - \nabla f_\mu(v;\xi)\| \leq \frac{\sqrt{2}M}{\sigma\sqrt{\pi}} \|u-v\|
$$

we could conclude the Lipschitz bound for $\nabla f_\mu$ cannot be improved by more than a constant factor. Then we are going to show smooth objective $f_\mu$ is $L$ smooth and the bound is tight.

$$
\begin{aligned}
\|\nabla f_\mu(u;\xi) - \nabla f_\mu(v;\xi)\| &= \|\nabla \mathbb{E}_{Z\sim\mu}[f(u+Z)] - \nabla \mathbb{E}_{Z\sim\mu}[f(u+Z)]\| \\
&= \|\mathbb{E}_{Z\sim\mu}[\nabla f(u+Z) - \nabla f(v+Z)]\| \\
&= \left\| \int [\nabla f(u+Z) - \nabla f(v+Z)]\mu(z)dz \right\| \\
&\leq \int \|\nabla f(u+Z) - \nabla f(v+Z)\| \mu(z)dz \\
&\leq \int L \|(u+Z) - (v+Z)\| \mu(z)dz \\
&= L\|u-v\| \int \mu(z)dz \\
&= L\|u-v\|
\end{aligned}
$$

Therefore, $f_\mu$ is $L$-smooth. Then we are going to show the bound is tight and cannot be improved. Define $M$ Lipschitz continuous and $L$-smooth function $f : \mathbb{R}^d \to \mathbb{R}$ as

$$
f(x;\xi) = \frac{1}{2}L\|x\|^2 \qquad x \in B(0, \frac{M}{L}).
$$

Hence, we have

$$
\begin{aligned}
\|\nabla f_\mu(u;\xi) - \nabla f_\mu(v;\xi)\| &= \left\| \int (Lu - Lv)\mu(z)dz \right\| \\
&= \left\| L(u-v) \int \mu(z)dz \right\| \\
&= L\|u-v\|.
\end{aligned}
$$

Therefore. $f_\mu$ is exactly $L$-smooth.

iii) By Jensen's inequality, for left hand side:

$$
f_\mu(u;\xi) = \mathbb{E}_{Z\sim\mu}[f(x+Z;\xi)] \geq f(x + \mathbb{E}_{Z\sim\mu}[Z];\xi) = f(u;\xi).
$$

For the tightness proof, defining $f(u;\xi) = \frac{1}{2}v^T x$ for $v \in \mathbb{R}^d$ leads to $f_\mu = f$.

For right hand side:

$$
\begin{aligned}
f_\mu(u;\xi) &= \mathbb{E}_{Z\sim\mu}[f(x+Z;\xi)] \\
&\leq f(u;\xi) + M\mathbb{E}_{Z\sim\mu}[\|Z\|] && (M\text{-Lipchitz continuous}) \\
&\leq f(u;\xi) + M\sqrt{\mathbb{E}[\|Z\|^2]} && (\frac{\|Z\|^2}{\sigma^2} \sim \mathcal{X}^2(d)) \\
&= f(u;\xi) + \sigma M\sqrt{d}.
\end{aligned}
$$

For the tightness proof, taking $f(u;\xi) = M\|x\|$. Since $f_\mu(u;\xi) \geq c\sigma M\sqrt{d}$ for some constant $c$. Therefore, the bound cannot be improved by more than a constant factor.

$\square$

## C.2 Proof of Theorem 44

*Proof.* We are going to prove the properties one by one.

i) The proof for properties i) is exactly the same as Theorem 42.

ii) As is shown in the proof for Theorem 42, firstly, we need to first address that $f_\mu$ is $\frac{M}{\sigma}$-smooth. then show that $f_\mu$ is $L$-smooth. Since, the proof for second part remains the same as what in Theorem 42. We will focus on demonstrating $f_\mu$ is $\frac{M}{\sigma}$-smooth.

By Lemma 75, for $\forall x, y \in \mathbb{R}^n$,

$$
\|\nabla f_\mu(u;\xi) - \nabla f_\mu(v;\xi)\| \leq M \underbrace{\int |\mu(z-u) - \mu(z-v)|dz}_{I_2}. \tag{C.6}
$$

Denoted the integral as $I_2$. We follow the technique in (Lakshmanan & Pucci De Farias, 2008) and (Duchi et al., 2012). Since $\mu(z-u) \geq \mu(z-v)$ is equivalent to $\|z-u\| \geq \|z-v\|$,

$$
\begin{aligned}
I_2 &= \int |\mu(z-u) - \mu(z-v)|dz \\
&= \int_{z:\|z-u\|\geq\|z-v\|} [\mu(z-u) - \mu(z-v)]dz + \int_{z:\|z-u\|\leq\|z-v\|} [\mu(z-v) - \mu(z-u)]dz \\
&= 2\int_{z:\|z-u\|\leq\|z-v\|} [\mu(z-u) - \mu(z-v)]dz \\
&= 2\int_{z:\|z-u\|\leq\|z-v\|} \mu(z-u)dz - 2\int_{z:\|z-u\|\leq\|z-v\|} \mu(z-v)dz.
\end{aligned}
$$

Denote $w = z - u$ for $\mu(z - u)$ term and $w = z - v$ for $\mu(z - v)$ term, we have

$$I_2 = 2 \int_{z:\|w\|\le\|w-(u-v)\|} \mu(w)dz - 2 \int_{z:\|w\|\ge\|w-(u-v)\|} \mu(w)dz$$
$$= 2\mathbb{P}_{Z\sim\mu}(\|Z\| \le \|Z - (u-v)\|) - 2\mathbb{P}_{Z\sim\mu}(\|Z\| \ge \|Z - (u-v)\|).$$

Obviously,

$$\mathbb{P}_{Z\sim\mu}(\|Z\| \le \|Z - (u-v)\|)$$
$$= \mathbb{P}_{Z\sim\mu}(\|Z\|^2 \le \|Z - (u-v)\|^2)$$
$$= \mathbb{P}_{Z\sim\mu}(2\langle z, u-v\rangle \le \|u-v\|^2)$$
$$= \mathbb{P}_{Z\sim\mu}(2\langle z, \frac{u-v}{\|u-v\|}\rangle \le \|u-v\|),$$

Denote $p = \frac{u-v}{\|u-v\|} \in \mathbb{R}^{d\times d}$, $\frac{u-v}{\|u-v\|}$ has norm 1 implies $\sum_{i=1}^d p_i^2 = 1$. Since $Z \sim \mathcal{N}(0, \Sigma)$ and $\Sigma = diag(\sigma_1^2, \cdots, \sigma_d^2)$, each element in vector $Z$ satisfies $z_i \sim \mathcal{N}(0, \sigma_i^2)$. Hence, we have

$$\langle z, \frac{u-v}{\|u-v\|}\rangle = \sum_{i=1}^d p_i z_i \sim \mathcal{N}(0, \sum_{i=1}^d p_i^2 \sigma_i^2).$$

Denote $\sigma^2 = \sum_{i=1}^d p_i^2 \sigma_i^2$, $\sigma_+^2 = \max\{\sigma_1^2, \cdots, \sigma_d^2\}$ and $\sigma_-^2 = \min\{\sigma_1^2, \cdots, \sigma_d^2\}$. Because $\sum_{i=1}^d p_i^2 = 1$, it is easy to know

$$\langle z, \frac{u-v}{\|u-v\|}\rangle \sim \mathcal{N}(0, \sigma^2), \quad \text{where } \sigma_-^2 \le \sigma^2 \le \sigma_+^2. \tag{C.7}$$

Hence, we have

$$\mathbb{P}_{Z\sim\mu}(\|Z\| \le \|Z - (u-v)\|)$$
$$= \mathbb{P}_{Z\sim\mu}(\langle z, \frac{u-v}{\|u-v\|}\rangle \le \frac{\|u-v\|}{2})$$
$$= \int_{-\infty}^{\frac{\|u-v\|}{2}} \frac{1}{\sqrt{2\pi\sigma^2}} \exp(-\frac{w^2}{2\sigma^2})dw.$$

Similarly,

$$\mathbb{P}_{Z\sim\mu}(\|Z\| \ge \|Z - (u-v)\|)$$
$$= \mathbb{P}_{Z\sim\mu}(\langle z, \frac{u-v}{\|u-v\|}\rangle \ge \frac{\|u-v\|}{2})$$
$$= \int_{\frac{\|u-v\|}{2}}^{+\infty} \frac{1}{\sqrt{2\pi\sigma^2}} \exp(-\frac{w^2}{2\sigma^2})dw.$$

Therefore,

$$I_2 = 2 \int_{-\infty}^{\frac{\|u-v\|}{2}} \frac{1}{\sqrt{2\pi\sigma^2}} \exp(-\frac{w^2}{2\sigma^2})dw - 2 \int_{\frac{\|u-v\|}{2}}^{+\infty} \frac{1}{\sqrt{2\pi\sigma^2}} \exp(-\frac{w^2}{2\sigma^2})dw$$
$$= 2 \int_{-\frac{\|u-v\|}{2}}^{\frac{\|u-v\|}{2}} \frac{1}{\sqrt{2\pi\sigma^2}} \exp(-\frac{w^2}{2\sigma^2})dw$$
$$\le \frac{\sqrt{2}\|u-v\|}{\sigma\sqrt{\pi}} \le \frac{\sqrt{2}\|u-v\|}{\sigma_-\sqrt{\pi}} \tag{C.8}$$

In conclusion, combine formula (D.1) and (C.8) we have

$$\|\nabla f_\mu(u;\xi) - \nabla f_\mu(v;\xi)\| \le M\frac{\sqrt{2}\|u-v\|}{\sigma_-\sqrt{\pi}} \le \frac{M}{\sigma_-}\|u-v\|.$$

We finish proving that $f_\mu$ is $\frac{M}{\sigma_-}$-smooth. Since covariance matrix $\Sigma$ for distribution $\mu$ is no longer a scalar matix and $\mu$ is not rotationally symmetric, the bound can no longer be achieved.

iii) By Jensen's inequality, for left hand side:

$$f_\mu(x;\xi) = \mathbb{E}_{Z\sim\mu}[f(x+Z;\xi)] \geq f(x+\mathbb{E}_{Z\sim\mu}[Z];\xi) = f(x;\xi).$$

For the tightness proof, defining $f(x;\xi) = \frac{1}{2}v^T\theta$ for $v \in \mathbb{R}^d$ leads to $f_\mu = f$.

For right hand side:

$$\begin{aligned}
f_\mu(x;\xi) &= \mathbb{E}_{Z\sim\mu}[f(x+Z;\xi)] \\
&\leq f(x;\xi) + M\mathbb{E}_{Z\sim\mu}[\|Z\|] \qquad (M\text{-Lipchitz continuous}) \\
&\leq f(x;\xi) + M\sqrt{\mathbb{E}[\|Z\|^2]}.
\end{aligned}$$

Letting $C^TC = \Sigma$ and $V \sim \mathcal{N}(0,I)$, because $Z \sim \mathcal{N}(0,\Sigma)$, we have

$$\mathbb{E}[\|Z\|^2] = \mathbb{E}[\|CV\|^2] = \mathbb{E}[V^TC^TCV] = tr(C^TC\mathbb{E}[V^TV]) = tr(\Sigma).$$

Therefore,

$$f_\mu(x;\xi) = f(x;\xi) + M\sqrt{tr(\Sigma)} = f(x;\xi) + M\sqrt{\sum_{i=1}^{d}\sigma_i^2}.$$

For the tightness proof, taking $f(x;\xi) = M\|x\|$. Since $f_\mu(x;\xi) \geq cM\sqrt{tr(\Sigma)}$ for some constant $c$. Therefore, the bound cannot be improved by more than a constant factor.

$\square$

# D  Proof for SmoothOut.

## D.1  Proof for Theorem 46.

*Proof.* We are going to prove the properties one by one.

i) The proof for Property i) is the same as that of Theorem 42.

ii) We first need to address that $f_\mu$ is $\frac{M\sqrt{d}}{a}$-smooth then need to address that $f_\mu$ is $\beta$-smooth. The proof for $\beta$-smooth remains the same as that in Theorem 42. We will focus on showing $f_\mu$ is $\frac{M\sqrt{d}}{a}$-smooth in this part. By Lemma 75, for $\forall x,y \in \mathbb{R}^n$,

$$\|\nabla f_\mu(u;\xi) - \nabla f_\mu(v;\xi)\| \leq M \underbrace{\int |\mu(z-u) - \mu(z-v)|dz}_{I_2}. \tag{D.1}$$

Denoted the integral as $I_3$. We follow the technique in (Lakshmanan & Pucci De Farias, 2008) and (Duchi et al., 2012).

We are going to discuss the bound of $I_3$ in 2 separate cases: $\|u-v\| > 2a$ and $\|u-v\| \leq 2a$.

a) If $\|u-v\| > 2a$: For every $z$ with $\|z-u\| \leq a$, we have $\|z-v\| > a$, which implies $\mu(z-v) = 0$. Therefore:

$$\int_{\|z-u\|\leq a} |\mu(z-u) - \mu(z-v)|dz = 1. \tag{D.2}$$

Similarly, we have:

$$\int_{\|z-v\|\leq a} |\mu(z-u) - \mu(z-v)|dz = 1. \tag{D.3}$$

In all, we could conclude:

$$
\begin{aligned}
I_3 &= \int |\mu(z-u) - \mu(z-v)| dz \\
&= \int_{z:\|z-u\|\le a} |\mu(z-u) - \mu(z-v)| dz + \int_{z:\|z-v\|\le a} |\mu(z-u) - \mu(z-v)| dz \\
&= 2
\end{aligned}
\tag{D.4}
$$

Since $\frac{\|u-v\|}{a} > 2$, we have

$$
I_3 \le \frac{\|u-v\|}{a}.
$$

b) If $\|u-v\| \le 2a$:

$$
\begin{aligned}
I_3 &= \int_{\|z-u\|\le a, \|z-v\|\le a} |\mu(z-u) - \mu(z-v)| dz \\
&\quad + \int_{\|z-u\|\le a, \|z-v\|>a} |\mu(z-u) - \mu(z-v)| dz \\
&\quad + \int_{\|z-u\|>a, \|z-v\|\le a} |\mu(z-u) - \mu(z-v)| dz \\
&\quad + \int_{\|z-u\|>a, \|z-v\|>a} |\mu(z-u) - \mu(z-v)| dz \\
&= 2 \int_{\|z-u\|\le a, \|z-v\|>a} |\mu(z-u) - \mu(z-v)| dz
\end{aligned}
$$

Denote set $S = \{z \in \mathbb{R}^d \mid \|z-u\| \le a \text{ and } \|z-v\| > a\}$, we obtain:

$$
I_3 = \frac{2}{C_d a^d} V_S,
$$

where

$$
C_d = \frac{\pi^{\frac{d}{2}}}{\Gamma(\frac{d}{2}+1)}, \quad \Gamma(\frac{d}{2}+1) =
\begin{cases}
(\frac{d}{2})! & \text{if d is even} \\
\sqrt{\pi}\,\dfrac{d!!}{2^{(d+1)/2}} & \text{if d is odd}
\end{cases}
\tag{D.5}
$$

$$
V_S = \text{Volumn of the set } S. \tag{D.6}
$$

We would like to upper bound the volume $V_S$. Let $V_{\text{cap}}(r)$ denote the column of the spherical cap with distance $r$ from the center of the sphere, therefore:

$$
V_S = C_d a^d - 2 V_{\text{cap}}\left(\frac{\|u-v\|}{2}\right),
$$

$$
V_{\text{cap}}(r) = \int r^a C_{d-1}(\sqrt{a^2 - \rho^2})^{d-1} r\rho, \quad \text{for } r \in [0, a].
$$

We have for $r \in [0, a]$:

$$
V'_{\text{cap}}(r) = -C_{d-1}(a^2 - r^2)^{\frac{d-1}{2}} \le 0,
$$

$$
V''_{\text{cap}}(r) = (d-1)C_{d-1}d(a^2 - r^2)^{\frac{d-3}{2}} \ge 0.
$$

Therefore for $r \in [0, a]$:

$$
V_{\text{cap}}(r) \ge V_{\text{cap}}(0) + V'_{\text{cap}}(0)r.
$$

Since $V_{\text{cap}}(0) = \frac{1}{2}C_d a^d$ and $V'_{\text{cap}}(0) = -C_{d-1}a^{d-1}$, we have:

$$
V_{\text{cap}}(r) \ge \frac{1}{2}C_d a^d - C_{d-1}a^{d-1}r.
$$

Therefore, we have

$$V_S = C_d a^d - 2V_{\text{cap}}\left(\frac{\|u-v\|}{2}\right)$$
$$\leq 2C_{d-1}a^{d-1}\frac{\|u-v\|}{2}$$
$$= C_{d-1}a^{d-1}\|u-v\|.$$

Therefore:

$$I_3 \leq \frac{2}{C_d a^d}C_{d-1}a^{d-1}\|u-v\|$$
$$= \frac{2C_{d-1}}{C_d}\frac{\|u-v\|}{a}$$
$$= \kappa\frac{d!!}{(d-1)!!}\frac{\|u-v\|}{a},$$

where

$$\kappa = \begin{cases} \dfrac{2}{\pi} & \text{if } d \text{ is even,} \\ 1 & \text{if } d \text{ is odd.} \end{cases}$$

Since $\lim_{d\to\infty}\frac{\kappa\frac{d!!}{(d-1)!!}}{\sqrt{d}} = \sqrt{\frac{\pi}{2}}$, we could conclude that,

$$I_3 \leq \frac{\sqrt{d}\|u-v\|}{a}.$$

Combine a) and b), we know that $f_\mu$ is $\frac{M\sqrt{d}}{a}$-smooth.

iii) By Jensen's inequality, for left hand side:

$$f_\mu(x;\xi) = \mathbb{E}_{Z\sim\mu}[f(x+Z;\xi)] \geq f(x+\mathbb{E}_{Z\sim\mu}[Z];\xi) = f(x;\xi).$$

For the tightness proof, defining $f(x;\xi) = \frac{1}{2}v^T\theta$ for $v \in \mathbb{R}^d$ leads to $f_\mu = f$.

For right hand side, $Z \sim U[-a,a]$:

$$f_\mu(x;\xi) = \mathbb{E}_{Z\sim\mu}[f(x+Z;\xi)]$$
$$\leq f(x;\xi) + \alpha\mathbb{E}_{Z\sim\mu}[\|Z\|] \qquad (\alpha\text{-Lipchitz continuous})$$
$$\leq f(x;\xi) + \alpha\sqrt{\mathbb{E}[\|Z\|^2]}$$
$$\leq f(x;\xi) + \alpha a$$

The bound is tight if $Z$ is sampled one the edge of the area.

$\square$

# E  Proof for AL-DSGD.

## E.1  Proof for Theorem 66.

*Proof.* In this section, we are going to find a range of $\alpha$, and some averaging hyperparameter $\omega$, such that the spectral norm $\rho = \max\{\left\|\mathbb{E}\left[\widetilde{\mathbf{W}}^{(k)}(\mathbf{I}-\mathbf{J})\widetilde{\mathbf{W}}^{(k)\top}\right]\right\|, \left\|\mathbb{E}\left[\widetilde{\mathbf{W}}^{(k)}\widetilde{\mathbf{W}}^{(k)\top}\right]\right\|\}$ is smaller than 1. Recall the formula of mixing matrix $\widetilde{\mathbf{W}}^{(k)}$:

$$\widetilde{\mathbf{W}}^{(k)} = (1 - \omega_N - \omega_\tau)\mathbf{W}^{(k)} + \omega_N\mathbf{A}_k^N + \omega_\tau\mathbf{A}_k^\tau. \tag{E.1}$$

Let $\mathbf{A}^{(k)} = \frac{\mathbf{A}_k^N + \mathbf{A}_k^\tau}{2}$, $\omega = 2\omega_N = 2\omega_\tau$, we have

$$\widetilde{\mathbf{W}}^{(k)} = (1-\omega)\mathbf{W}^{(k)} + \omega\mathbf{A}^{(k)}, \tag{E.2}$$

where $\mathbf{A}^{(k)}$ is still a left stochastic matrix. Therefore:

$$\widetilde{\mathbf{W}}^{(k)}(\mathbf{I}-\mathbf{J})\widetilde{\mathbf{W}}^{(k)\top} = \widetilde{\mathbf{W}}^{(k)}\widetilde{\mathbf{W}}^{(k)\top} - \widetilde{\mathbf{W}}^{(k)}\mathbf{J}\widetilde{\mathbf{W}}^{(k)\top} \tag{E.3}$$

Since

$$\widetilde{\mathbf{W}}^{(k)}\widetilde{\mathbf{W}}^{(k)\top} = \left[(1-\omega)\mathbf{W}^{(k)} + \omega\mathbf{A}^{(k)}\right]\left[(1-\omega)\mathbf{W}^{(k)} + \omega\mathbf{A}^{(k)}\right]^\top$$
$$= (1-\omega)^2\mathbf{W}^{(k)}\mathbf{W}^{(k)\top} + \omega(1-\omega)\mathbf{W}^{(k)}\mathbf{A}^{(k)\top} + \omega(1-\omega)\mathbf{A}^{(k)}\mathbf{W}^{(k)\top}$$
$$+ \omega^2 + \omega(1-\omega)\mathbf{A}^{(k)}\mathbf{A}^{(k)\top}$$
$$\widetilde{\mathbf{W}}^{(k)}\mathbf{J}\widetilde{\mathbf{W}}^{(k)\top} = \left[(1-\omega)\mathbf{W}^{(k)}\mathbf{J} + \omega\mathbf{A}^{(k)}\mathbf{J}\right]\left[(1-\omega)\mathbf{W}^{(k)} + \omega\mathbf{A}^{(k)}\right]^\top$$
$$= (1-\omega)^2\mathbf{W}^{(k)}\mathbf{J}\mathbf{W}^{(k)\top} + \omega(1-\omega)\mathbf{W}^{(k)}\mathbf{J}\mathbf{A}^{(k)\top} + \omega(1-\omega)\mathbf{A}^{(k)}\mathbf{J}\mathbf{W}^{(k)\top}$$
$$+ \omega^2 + \omega(1-\omega)\mathbf{A}^{(k)}\mathbf{J}\mathbf{A}^{(k)\top}$$

Since we know that $\mathbf{W}^{(k)}$ is symmetric doubly stochastic matrix and $\mathbf{A}^{(k)}$ is the left stochastic matrix, we know that $\mathbf{J} = \mathbf{W}^{(k)}\mathbf{J} = \mathbf{J}\mathbf{W}^{(k)}$ and $\mathbf{J} = \mathbf{J}\mathbf{A}^{(k)} \neq \mathbf{A}^{(k)}\mathbf{J}$. Putting (E.4) back to (E.3), we could get

$$\widetilde{\mathbf{W}}^{(k)}(\mathbf{I}-\mathbf{J})\widetilde{\mathbf{W}}^{(k)\top} \tag{E.4}$$
$$= (1-\omega)^2\left[\mathbf{W}^{(k)\top}\mathbf{W}^{(k)} - \mathbf{J}\right] + \omega(1-\omega)\left[\mathbf{W}^{(k)}\mathbf{A}^{(k)\top} - \mathbf{J}\mathbf{A}^{(k)\top}\right]$$
$$+ \omega(1-\omega)\left[\mathbf{A}^{(k)}\mathbf{W}^{(k)} - \mathbf{A}^{(k)}\mathbf{J}\right] + \omega^2\left[\mathbf{A}^{(k)}\mathbf{A}^{(k)\top} - \mathbf{A}^{(k)}\mathbf{J}\mathbf{A}^{(k)\top}\right] \tag{E.5}$$

Therefore, we have

$$\left\|\mathbb{E}\left[\widetilde{\mathbf{W}}^{(k)}(\mathbf{I}-\mathbf{J})\widetilde{\mathbf{W}}^{(k)\top}\right]\right\| \leq (1-\omega)^2\left\|\mathbb{E}\left[\mathbf{W}^{(k)\top}\mathbf{W}^{(k)}\right]-\mathbf{J}\right\| + \omega^2\left\|\mathbb{E}\left[\mathbf{A}^{(k)}(\mathbf{I}-\mathbf{J})\mathbf{A}^{(k)\top}\right]\right\|$$
$$+ 2\omega(1-\omega)\left\|\mathbb{E}\left[\mathbf{W}^{(k)}\mathbf{A}^{(k)\top} - \mathbf{J}\mathbf{A}^{(k)\top}\right]\right\| \tag{E.6}$$
$$\left\|\mathbb{E}\left[\widetilde{\mathbf{W}}^{(k)}\widetilde{\mathbf{W}}^{(k)\top}\right]\right\| \leq (1-\omega)^2\left\|\mathbb{E}\left[\mathbf{W}^{(k)\top}\mathbf{W}^{(k)}\right]\right\| + 2\omega(1-\omega)\left\|\mathbb{E}\left[\mathbf{W}^{(k)}\mathbf{A}^{(k)\top}\right]\right\|$$
$$+ \omega^2\left\|\mathbb{E}\left[\mathbf{A}^{(k)}\mathbf{A}^{(k)\top}\right]\right\| \tag{E.7}$$

Firstly, We are going to bound each term in iequality (E.6) one by one.

**(1) Bound $\left\|\mathbb{E}\left[\mathbf{W}^{(k)\top}\mathbf{W}^{(k)}\right]-\mathbf{J}\right\|$.**

$$\left\|\mathbb{E}\left[\mathbf{W}^{(k)\top}\mathbf{W}^{(k)}\right]-\mathbf{J}\right\| = \left\|\mathbb{E}\left[\left(I-\alpha\mathbf{L}^{(k)}\right)^\top\left(I-\alpha\mathbf{L}^{(k)}\right)\right]-\mathbf{J}\right\|$$
$$= \left\|I - 2\alpha\mathbb{E}\left[\mathbf{L}^{(k)}\right] + \alpha^2\mathbb{E}\left[\mathbf{L}^{(k)\top}\mathbf{L}^{(k)}\right]-\mathbf{J}\right\|. \tag{E.8}$$

Recall there are two communication graph and $\mathbf{L}^{(k)}$ is periodically switched between them:

$$\mathbf{L}^{(k)} = \begin{cases} \sum_{j=1}^{m} B_{(1),j}^{(k)}\mathbf{L}_{(1),j} & \text{if } k \bmod n = 1 \\[2ex] \sum_{j=1}^{m} B_{(2),j}^{(k)}\mathbf{L}_{(2),j} & \text{if } k \bmod n = 2 \\[1ex] \cdots \\[1ex] \sum_{j=1}^{m} B_{(n),j}^{(k)}\mathbf{L}_{(n),j} & \text{if } k \bmod n = 0 \end{cases}$$

We analysis the case for $k \bmod n = i$, where $i = 1, 2, ..., n-1, 0$. For notation convenience, we use $k \bmod n = n$ instead of $k \bmod n = 0$ without loss of generality. Then the condition could be rewritten as $k \bmod n = i$, where $i = 1, 2, ..., n-1, n$.

If $k \bmod n = i$, then from Appendix B in (Wang et al., 2019) we have

$$\mathbb{E}\left[\mathbf{L}^{(k)}\right] = \sum_{j=1}^{m} p_{(i),j} \mathbf{L}_{(i),j}$$

$$\mathbb{E}\left[\mathbf{L}^{(k)\top}\mathbf{L}^{(k)}\right] = \left(\sum_{j=1}^{m} p_{(i),j} \mathbf{L}_{(i),j}\right)^2 + 2\sum_{j=1}^{M} p_{(i),j}(1 - p_{(i),j})\mathbf{L}_{(i),j}.$$

And

$$\left\|\mathbb{E}\left[\mathbf{W}^{(k)\top}\mathbf{W}^{(k)}\right] - J\right\| \leq \left\|\left(I - \alpha\sum_{j=1}^{m} p_{(i),j}\mathbf{L}_{(i),j}\right)^2 - J\right\| + 2\alpha^2 \left\|2\sum_{j=1}^{M} p_{(i),j}(1 - p_{(i),j})\mathbf{L}_{(i),j}\right\|$$

$$= \max\{(1 - \alpha\lambda_{(i),2})^2, (1 - \alpha\lambda_{i1),m})^2\} + 2\alpha^2\zeta_{(i)}, \tag{E.9}$$

where $\lambda_{(i),l}$ denote the $l$-th smallest eigenvalue of matrix $\sum_{j=1}^{m} p_{(i),j}\mathbf{L}_{(i),j}$ and $\zeta_{(i)} > 0$ denote the spectral norm of matrix $\sum_{j=1}^{m} p_{(i),j}(1 - p_{(i),j})\mathbf{L}_{(i),j}$.

In all, generalized all $k \bmod n = i(i = 1, ..., n)$ we could conclude

$$\left\|\mathbb{E}\left[\mathbf{W}^{(k)\top}\mathbf{W}^{(k)}\right] - \mathbf{J}\right\| \leq \max\{(1 - \alpha\lambda_{(1),2})^2, (1 - \alpha\lambda_{(1),m})^2, ..., (1 - \alpha\lambda_{(n),2})^2, (1 - \alpha\lambda_{(n),m})^2\}$$

$$+ 2\alpha^2\max\{\zeta_{(1)}, \zeta_{(2)}, ..., \zeta_{(n)}\}. \tag{E.10}$$

Assume $\lambda$ represents the eigenvalue such that $|1 - \alpha\lambda| = \max\{|1 - \alpha\lambda_{(1),2}|, |1 - \alpha\lambda_{(1),m}|, |1 - \alpha\lambda_{(2),2}|, |1 - \alpha\lambda_{(2),m}|, ..., |1 - \alpha\lambda_{(n),2}|, |1 - \alpha\lambda_{(n),m}|\}$, and $\zeta = \max\{\zeta_{(1)}, \zeta_{(2)}, ..., \zeta_{(n)}\}$, we could have

$$\left\|\mathbb{E}\left[\mathbf{W}^{(k)\top}\mathbf{W}^{(k)}\right] - \mathbf{J}\right\| \leq |1 - \alpha\lambda|^2 + 2\alpha^2\zeta. \tag{E.11}$$

**(2) Bound** $\left\|\mathbb{E}\left[\mathbf{W}^{(k)}\mathbf{A}^{(k)\top} - \mathbf{J}\mathbf{A}^{(k)\top}\right]\right\|$.

Because $\mathbf{A}^{(k)}$ is a left stochastic matrix and $\mathbf{W}^{(k)}$ is doubly stochastic matrix, from the property of spectrum nor $\|\cdot\| \leq \|\cdot\|_1 \|\cdot\|_\infty$, we could know that for all $k$

$$\left\|A^k\right\| \leq \sqrt{m}, \quad \left\|\mathbf{W}^{(k)}\right\| \leq 1.$$

Moreover, we could easy check $\|\mathbf{J}\| = 1$. Therefore,

$$\left\|\mathbb{E}\left[\mathbf{W}^{(k)}\mathbf{A}^{(k)\top} - \mathbf{J}\mathbf{A}^{(k)\top}\right]\right\| \leq \left\|\mathbb{E}\left[(\mathbf{W}^{(k)} - \mathbf{J})\mathbf{A}^{(k)\top}\right]\right\| \leq \mathbb{E}\left[\left\|(\mathbf{W}^{(k)} - \mathbf{J})\mathbf{A}^{(k)\top}\right\|\right]$$

$$\leq \mathbb{E}\left[\left\|(\mathbf{W}^{(k)} - \mathbf{J})\right\|\left\|\mathbf{A}^{(k)\top}\right\|\right] \leq \mathbb{E}\left[\left(\left\|\mathbf{W}^{(k)}\right\| + \|\mathbf{J}\|\right)\left\|\mathbf{A}^{(k)\top}\right\|\right] \leq 2\sqrt{m} \tag{E.12}$$

**(3) Bound** $\left\|\mathbb{E}\left[\mathbf{A}^{(k)}(\mathbf{I} - \mathbf{J})\mathbf{A}^{(k)\top}\right]\right\|$.

$$\left\|\mathbb{E}\left[\mathbf{A}^{(k)}(\mathbf{I} - \mathbf{J})\mathbf{A}^{(k)\top}\right]\right\| \leq \mathbb{E}\left[\left\|\mathbf{A}^{(k)}(\mathbf{I} - \mathbf{J})\mathbf{A}^{(k)\top}\right\|\right] \leq \mathbb{E}\left[\left\|\mathbf{A}^{(k)}\right\|^2 \|(\mathbf{I} - \mathbf{J})\|\right] \leq m$$

Combine **(1)-(3)** and (E.6), we have

$$\left\|\mathbb{E}\left[\widetilde{\mathbf{W}}^{(k)}(\mathbf{I} - \mathbf{J})\widetilde{\mathbf{W}}^{(k)\top}\right]\right\| \leq (1 - \omega)^2(1 - \alpha\lambda)^2 + 4\omega(1 - \omega)\sqrt{m} + 2\omega^2 m \tag{E.13}$$

Similarly, we are going to bound each term in iequality (E.7) one by one as well.

**(4) Bound** $\left\| \mathbb{E}\left[ \mathbf{W}^{(k)\top}\mathbf{W}^{(k)} \right] \right\|$.

Similar to proof in **(1)**, if $k \bmod n = i$ $(i = 1, ..., n)$, we have

$$\left\| \mathbb{E}\left[ \mathbf{W}^{(k)\top}\mathbf{W}^{(k)} \right] \right\| \leq \left\| \left( I - \alpha \sum_{j=1}^{m} p_{(i),j}\mathbf{L}_{(i),j} \right)^2 \right\| + 2\alpha^2 \left\| 2 \sum_{j=1}^{M} p_{(i),j}(1 - p_{(i),j})\mathbf{L}_{(i),j} \right\|$$

$$= \max\{(1 - \alpha\lambda_{(i),2})^2, (1 - \alpha\lambda_{(i),m})^2\} + 2\alpha^2\zeta_{(i)}, \tag{E.14}$$

where $\lambda_{(i),l}$ denote the $l$-th smallest eigenvalue of matrix $\sum_{j=1}^{m} p_{(i),j}\mathbf{L}_{(i),j}$ and $\zeta_{(1)} > 0$ denote the spectral norm of matrix $\sum_{j=1}^{m} p_{(i),j}(1 - p_{(i),j})\mathbf{L}_{(i),j}$.

In all, generalize all $k \bmod n = i(i = 1, ..., n)$ we could conclude

$$\left\| \mathbb{E}\left[ \mathbf{W}^{(k)\top}\mathbf{W}^{(k)} \right] \right\| \leq \max\{(1 - \alpha\lambda_{(1),2})^2, (1 - \alpha\lambda_{(1),m})^2, ..., (1 - \alpha\lambda_{(n),2})^2, (1 - \alpha\lambda_{(n),m})^2\}$$

$$+ 2\alpha^2 \max\{\zeta_{(1)}, \zeta_{(2)}, ..., \zeta_{(n)}\}. \tag{E.15}$$

Assume $\lambda$ represents the eigenvalue such that $|1 - \alpha\lambda| = \max\{|1 - \alpha\lambda_{(1),2}|, |1 - \alpha\lambda_{(1),m}|, |1 - \alpha\lambda_{(2),2}|, |1 - \alpha\lambda_{(2),m}|, ..., |1 - \alpha\lambda_{(n),2}|, |1 - \alpha\lambda_{(n),m}|\}$, and $\zeta = \max\{\zeta_{(1)}, \zeta_{(2)}, ..., \zeta_{(n)}\}$, we could have

$$\left\| \mathbb{E}\left[ \mathbf{W}^{(k)\top}\mathbf{W}^{(k)} \right] \right\| \leq |1 - \alpha\lambda|^2 + 2\alpha^2\zeta. \tag{E.16}$$

**(2) Bound** $\left\| \mathbb{E}\left[ \mathbf{W}^{(k)}\mathbf{A}^{(k)\top} \right] \right\|$.

Because $\mathbf{A}^{(k)}$ is a left stochastic matrix and $\mathbf{W}^{(k)}$ is doubly stochastic matrix, from the property of spectrum nor $\|\cdot\| \leq \|\cdot\|_1 \|\cdot\|_\infty$, we could know that for all $k$

$$\left\| \mathbf{A}^k \right\| \leq \sqrt{m}, \quad \left\| \mathbf{W}^{(k)} \right\| \leq 1.$$

Moreover, we could easy check $\|J\| = 1$. Therefore,

$$\left\| \mathbb{E}\left[ \mathbf{W}^{(k)}\mathbf{A}^{(k)\top} \right] \right\| \leq \mathbb{E}\left[ \left\| \mathbf{W}^{(k)} \right\| \left\| \mathbf{A}^{(k)\top} \right\| \right] \leq \sqrt{m} \tag{E.17}$$

**(3) Bound** $\left\| \mathbb{E}\left[ \mathbf{A}^{(k)}\mathbf{A}^{(k)\top} \right] \right\|$.

$$\left\| \mathbb{E}\left[ \mathbf{A}^{(k)}\mathbf{A}^{(k)\top} \right] \right\| \leq \mathbb{E}\left[ \left\| \mathbf{A}^{(k)} \right\|^2 \right] \leq m$$

Combine **(4)-(5)** and (E.7), we have

$$\left\| \mathbb{E}\left[ \widetilde{\mathbf{W}}^{(k)}\widetilde{\mathbf{W}}^{(k)\top} \right] \right\| \leq (1 - \omega)^2(1 - \alpha\lambda)^2 + 2\omega(1 - \omega)\sqrt{m} + \omega^2 m \tag{E.18}$$

From the proof in Appendix B of (Wang et al., 2019), we know that $\lambda > 0$. We assume $0 < \alpha < \frac{1}{\lambda}$, and $\omega \in (0,1)$, combine (E.13) and (E.18) we have

$$
\begin{aligned}
\rho &= \max\left\{\left\|\mathbb{E}\left[\widetilde{\mathbf{W}}^{(k)}(\mathbf{I}-\mathbf{J})\widetilde{\mathbf{W}}^{(k)\top}\right]\right\|, \left\|\mathbb{E}\left[\widetilde{\mathbf{W}}^{(k)}\widetilde{\mathbf{W}}^{(k)\top}\right]\right\|\right\}\\
&\leq (1-\omega)^2(1-\alpha\lambda)^2 + 4\omega(1-\omega)\sqrt{m} + 2\omega^2 m + 2\alpha^2\zeta\\
&\leq (1-\omega)^2(1-\alpha\lambda)^2 + 4\omega(1-\omega)\sqrt{m} + 4\omega^2 m + 2\alpha^2\zeta\\
&= (1-\omega)^2(1-\alpha\lambda)^2 + 4\omega(1-\omega)\sqrt{m}\left[(1-\alpha\lambda)+\alpha\lambda\right] + \omega^2 m + 2\alpha^2\zeta\\
&\leq (1-\omega)^2(1-\alpha\lambda)^2 + 4\omega(1-\omega)\sqrt{m}(1-\alpha\lambda) + 4\omega^2 m + 4\alpha\lambda\omega(1-\omega)\sqrt{m} + 2\alpha^2\zeta\\
&\leq \left[(1-\omega)(1-\alpha\lambda)+2\sqrt{m}\omega\right]^2 + 4\alpha\lambda\omega(1-\omega)\sqrt{m} + 2\alpha^2\zeta\\
&\leq \left[(1-\omega)(1-\alpha\lambda)+2\sqrt{m}\omega\right]^2 + 4\omega\sqrt{m} + 2\alpha^2\zeta
\end{aligned}
\tag{E.19}
$$

Define $f_{\lambda,\alpha}(\omega) = \left[(1-\omega)(1-\alpha\lambda)+2\sqrt{m}\omega\right]^2 + 4\omega\sqrt{m} + 2\alpha^2\zeta$, we have

$$
f'_{\lambda,\alpha}(\omega) = 2\left[(1-\omega)(1-\alpha\lambda)+2\sqrt{m}\omega\right]\left[2\sqrt{m}\omega - (1-\alpha\lambda)\right] + 4\sqrt{m}.
$$

$m$ is the number of the worker and it must satisfy $m > 1$. Together with $\alpha\lambda \in (0,1)$, we could conclude $f'_{\lambda,\alpha}(\omega) > 0$ for all $\omega \in (0,1)$. Then take

$$
\omega_0 = \frac{1-\alpha\lambda}{2k\sqrt{m}},
$$

where $k > 1$. We know that for all $\omega \in (0, \omega_0)$,

$$
\begin{aligned}
f_{\lambda,\alpha}(\omega) \leq f_{\lambda,\alpha}(\omega_0) &= \left[\left(1-\frac{1-\alpha\lambda}{2k\sqrt{m}}\right)(1-\alpha\lambda) + \frac{1-\alpha\lambda}{k}\right]^2 + \frac{2(1-\alpha\lambda)}{k} + 2\alpha^2\zeta\\
&\leq \frac{(k+1)^2}{k^2}(1-\alpha\lambda)^2 + \frac{2(1-\alpha\lambda)}{k} + 2\alpha^2\zeta
\end{aligned}
\tag{E.20}
$$

Define $h_\lambda(\alpha) = \frac{(k+1)^2}{k^2}(1-\alpha\lambda)^2 + \frac{2(1-\alpha\lambda)}{k} + 2\alpha^2\zeta$, then we have

$$
\begin{aligned}
h'_\lambda(\alpha) &= -\frac{2(k+1)^2}{k^2}\lambda(1-\alpha\lambda) - \frac{2\lambda}{k} + 4\alpha\zeta,\\
h''_\lambda(\alpha) &= \frac{2(k+1)^2}{k^2}\lambda^2 + 4\zeta.
\end{aligned}
$$

Since $h''_\lambda(\alpha) > 0$, $h_\lambda(\alpha)$ is convex quadratic fucntion. Let $h'_\lambda(\alpha) = 0$, we could get the minimun point is:

$$
\alpha^* = \frac{\left[(k+1)^2 + k\right]\lambda}{(k+1)^2\lambda^2 + 2k^2\zeta}.
$$

We take $\widetilde{\alpha} = \frac{(k+1)^2\lambda}{(k+1)^2\lambda^2 + 2k^2\zeta}$, it is easy to know $0 < \widetilde{\alpha} < \alpha^*$.

$$
h_\lambda(\widetilde{\alpha}) = \frac{(k+1)^2}{k^2}(1-\widetilde{\alpha}\lambda)^2 + \frac{2}{k}(1-\widetilde{\alpha}\lambda) + 2\alpha^2\zeta = \frac{4k\zeta + 2(k+1)^2\zeta}{(k+1)^2\lambda^2 + 2k^2\zeta}.
$$

It is obvious that $\widetilde{\alpha}\lambda \in (0,1)$. Then, we are going to compute the bound for $k$ to ensure $h_\lambda(\widetilde{\alpha}) < 1$. When $k > \max\{1, \frac{8\zeta}{\lambda^2} - 1\}$, we have:

$$
\begin{aligned}
&\frac{k+1}{8} > \frac{\zeta}{\lambda^2} \Rightarrow \frac{(k+1)^2}{8K+8} > \frac{\zeta}{\lambda^2} \Rightarrow \frac{(k+1)^2}{8K+4} > \frac{\zeta}{\lambda^2}\\
&\Rightarrow 2(k+1)^2\zeta + 4k\zeta < (k+1)^2\lambda^2 + 2k^2\zeta\\
&\Rightarrow \frac{4k\zeta + 2(k+1)^2\zeta}{(k+1)^2\lambda^2 + 2k^2\zeta} < 1
\end{aligned}
$$

For any $k > \max\{1, \frac{8\zeta}{\lambda^2} - 1\}$, by the convex property of $h_\lambda(\alpha)$, we know when $\alpha \in (\alpha_{min}, \alpha_{max})$, where:

$$\alpha_{min} = \frac{(k+1)^2\lambda}{(k+1)^2\lambda^2 + 2k^2\zeta}, \quad \alpha_{max} = \min\{\frac{1}{\lambda}, \frac{[(k+1)^2 + k]\lambda}{(k+1)^2\lambda^2 + 2k^2\zeta}\}.$$

There exists a range of averaging parameter $\omega \in (0, \frac{1-\alpha\lambda}{2k\sqrt{m}})$, such that

$$\rho = \max\{\left\|\mathbb{E}\left[\widetilde{\mathbf{W}}^{(k)}(\mathbf{I} - \mathbf{J})\widetilde{\mathbf{W}}^{(k)\top}\right]\right\|, \left\|\mathbb{E}\left[\widetilde{\mathbf{W}}^{(k)}\widetilde{\mathbf{W}}^{(k)\top}\right]\right\|\} \leq h_\lambda(\alpha) < 1$$

.

Furthermor, $k > \frac{\lambda^2}{2\zeta} \Rightarrow \frac{1}{\lambda} > \frac{[(k+1)^2 + k]\lambda}{(k+1)^2\lambda^2 + 2k^2\zeta}$. Therefore, for any $k > \max\{1, \frac{8\zeta}{\lambda^2} - 1, \frac{\lambda^2}{2\zeta}\}$, when $\alpha \in (\alpha_{min}, \alpha_{max})$, where:

$$\alpha_{min} = \frac{(k+1)^2\lambda}{(k+1)^2\lambda^2 + 2k^2\zeta}, \quad \alpha_{max} = \frac{[(k+1)^2 + k]\lambda}{(k+1)^2\lambda^2 + 2k^2\zeta}.$$

Going back to the assumption for $\lambda$, when $1 - \alpha\lambda \in (0, 1)$ always holds for sufficient small $\alpha$, $\lambda$ represents the eigenvalue such that $|1 - \alpha\lambda| = \max\{|1 - \alpha\lambda_{(1),2}|, |1 - \alpha\lambda_{(1),m}|, |1 - \alpha\lambda_{(2),2}|, |1 - \alpha\lambda_{(2),m}|, ..., |1 - \alpha\lambda_{(n),2}|, |1 - \alpha\lambda_{(n),m}|\}$ should be exactly $\lambda = \min\{\lambda_{(1),2}, \lambda_{(2),2}, ..., \lambda_{(n),2}\}$.

We generalized the above analysis of the construction for $\alpha$ and $\omega$ as the following. Assume $\lambda_{min} = \min\{\lambda_{(i),2} : i = 1, ..., n\}$, $\lambda_{max} = \min\{\lambda_{(i),m} | i = 1, ..., n\}$ and $\zeta = \max\{\zeta_{(1)}, \zeta_{(2)}, ..., \zeta_{(n)}\}$. For any $k > \max\{1, \frac{8\zeta}{\lambda_{min}^2} - 1, \frac{\lambda_{max}^2}{2\zeta}\}$, there exists a range $\alpha \in (\frac{(k+1)^2\lambda_{min}}{(k+1)^2\lambda_{min}^2 + 2k^2\zeta}, \frac{[(k+1)^2 + k]\lambda_{min}}{(k+1)^2\lambda_{min}^2 + 2k^2\zeta})$, such that for any $\alpha$ in this range, we could find a range $\omega \in (0, \frac{1-\alpha\lambda}{2k\sqrt{m}})$ such that the spectral norm

$$\rho = \max\{\left\|\mathbb{E}\left[\widetilde{\mathbf{W}}^{(k)}(\mathbf{I} - \mathbf{J})\widetilde{\mathbf{W}}^{(k)\top}\right]\right\|, \left\|\mathbb{E}\left[\widetilde{\mathbf{W}}^{(k)}\widetilde{\mathbf{W}}^{(k)\top}\right]\right\|\} < 1$$

. $\qquad\qquad\qquad\qquad\qquad\qquad\qquad\qquad\qquad\qquad\qquad\qquad\qquad\qquad\qquad\qquad\qquad\qquad\quad \square$

### E.2    Proof for Lemma 67.

*Proof.* Define $\mathbf{A}_{q,n} = \prod_{k=q}^{n} \widetilde{\mathbf{W}}^{(k)}$, $\mathbf{b}_i^\top$ denote the $i$-th row vector of $B$. Thus, we have

$$\mathbf{A}_{1,n} = \mathbf{A}_{1,n-1}\widetilde{\mathbf{W}}^{(n)}.$$

Thus, we have

$$\mathbb{E}_{\widetilde{\mathbf{W}}^{(n)}}\left[\|\mathbf{B}\mathbf{A}_{1,n}(\mathbf{I} - \mathbf{J})\|_F^2\right] \leq \sum_{i=1}^{d} \mathbb{E}_{\widetilde{\mathbf{W}}^{(n)}}\left[\left\|b_i^\top \mathbf{A}_{1,n}(\mathbf{I} - \mathbf{J})\right\|^2\right]$$

$$= \sum_{i=1}^{d} \mathbf{b}_i^\top \mathbf{A}_{1,n-1}\mathbb{E}_{\widetilde{\mathbf{W}}^{(n)}}\left[\widetilde{\mathbf{W}}^{(n)}(\mathbf{I} - \mathbf{J})^2\widetilde{\mathbf{W}}^{(n)\top}\right]\mathbf{A}_{1,n-1}^\top \mathbf{b}_i$$

$$= \sum_{i=1}^{d} \mathbf{b}_i^\top \mathbf{A}_{1,n-1}\mathbb{E}_{\widetilde{\mathbf{W}}^{(n)}}\left[\widetilde{\mathbf{W}}^{(n)}(\mathbf{I} - \mathbf{J})\widetilde{\mathbf{W}}^{(n)\top}\right]\mathbf{A}_{1,n-1}^\top \mathbf{b}_i$$

Let $\mathbf{C} = \mathbb{E}_{\widetilde{\mathbf{W}}^{(n)}}\left[\widetilde{\mathbf{W}}^{(n)}(\mathbf{I} - J)\widetilde{\mathbf{W}}^{(n)\top}\right]$, $v_i = \mathbf{A}_{1,n-1}^\top \mathbf{b}_i$, we have:

$$\mathbb{E}_{\widetilde{\mathbf{W}}^{(n)}}\left[\|\mathbf{B}\mathbf{A}_{1,n}(\mathbf{I} - \mathbf{J})\|_F^2\right] \leq \sum_{i=1}^{d} v_i^\top \mathbf{C} v_i \leq \sigma_{max}(\mathbf{C}) \sum_{i=1}^{d} v_i^\top v_i = \rho \|\mathbf{B}\mathbf{A}_{1,n-1}\|_F^2$$

Similarly, we could have

$$
\begin{aligned}
\mathbb{E}_{\widetilde{\mathbf{W}}^{(n-1)}}\left[\left\|\mathbf{B}\mathbf{A}_{1,n-1}\right\|_F^2\right] &\leq \sum_{i=1}^{d}\mathbb{E}_{\widetilde{\mathbf{W}}^{(n-1)}}\left[\left\|\mathbf{b}_i^\top\mathbf{A}_{1,n-1}\right\|^2\right] \\
&= \sum_{i=1}^{d}\mathbf{b}_i^\top\mathbf{A}_{1,n-2}\mathbb{E}_{\widetilde{\mathbf{W}}^{(n-1)}}\left[\widetilde{\mathbf{W}}^{(n-1)}\widetilde{\mathbf{W}}^{(n-1)\top}\right]\mathbf{A}_{1,n-2}^\top\mathbf{b}_i \\
&\leq \rho\left\|\mathbf{B}\mathbf{A}_{1,n-2}\right\|_F^2
\end{aligned}
$$

$$
\mathbb{E}_{\widetilde{\mathbf{W}}^{(1)},\dots,\widetilde{\mathbf{W}}^{(n)}}\left[\left\|B\left(\prod_{k=1}^{n}\widetilde{\mathbf{W}}^{(k)}\right)(\mathbf{I}-\mathbf{J})\right\|_F^2\right] \leq \rho^n\left\|\mathbf{B}\right\|_F^2.
$$

$\square$

