# OpenReview forum: "A Survey of Optimization Methods for Training DL Models: Theoretical Perspective on Convergence and Generalization"
_TMLR — Rejected by TMLR_

### Review · Reviewer_HTdu · 2024-09-21

**Summary Of Contributions:**

This submission is a review survey to summarize first order algorithms and second order algorithms for training deep learning models with corresponding convergence proofs. This survey paper also covers some technique used in distributed optimization settings, where both centralized and de-centralized scenarios are covered.

**Audience:**

Yes

**Claims And Evidence:**

Yes

**Requested Changes:**

(1) Regarding classic methods:

- for first-order methods, which are most widely deployed in implementation, can authors talk about the theory-experiment gap and give hyperparameter setting examples within existing work with respect to algorithm discussed?

- With respect to assumptions made for various algorithms, can authors remark on whether or not practically involved training loss functions satisfy those assumptions?

- Authors can add in a few sentences (in the introduction section or in the first-order algorithm section) about the following two threads of work on first-order algorithm, for instance. [1-2] focus on leveraging stochastic perturbation to imbue the first-order optimization methods to have probabilistic guarantee to converge to a local optimal point, which used to be an exclusive privilege of second-order algorithms. [3-5] focus on the format of Langevin dynamics for optimization.

  Authors do _not_ have to add a whole new section about them. Instead, two or three sentences to mention them with citation should suffice well.

- Theorem 35 does not seem clear to me. What is the conclusion of that statement?

- Can authors point out the acceleration effect when comparing adaptive learning rate methods with momentum to normal SGD?
This comparison analysis helps readers to understand why acceleration is achieved.

  To make my question more specific, on Page 24, Remark gives $\mathcal{O}(\frac{\ln K}{\sqrt{K}})$ sublinear convergence rate for Adam. Compared with Remark on Page 8 that SGD has sublinear convergence rate $\mathcal{O}(\frac{1}{\sqrt{K}})$, why is Adam even an acceleration compared to SGD?

- Any comments on what each convergence bound term represents? Their interpretations and how practical setting of various hyperparameters is going to impact on those bound terms?

- For second-order algorithms, [James Martens](https://scholar.google.com/citations?user=LlK_saMAAAAJ&hl=en) has a thread of work on Hessian-free style of methods, in particular [6]. It is an implicit methods leveraging conjugate gradients. This is an area authors seem to miss out.

[1] On Nonconvex Optimization for Machine Learning: Gradients, Stochasticity, and Saddle Points, Chi Jin, Praneeth Netrapalli, Rong Ge, Sham M. Kakade, Michael I. Jordan, ICML 2017

[2] Perturbed proximal descent to escape saddle points for non-convex and non-smooth objective functions, Huang and Becker, INND BDDL, 2019

[3] Non-Convex Learning via Stochastic Gradient Langevin Dynamics: A Nonasymptotic Analysis, Raginsky, Rakhlin, Telgarsky, 2017

[4] User-friendly guarantees for the Langevin Monte Carlo with inaccurate gradient, Dalalyan, Karagulyan, 2017

[5] Stochastic Gradient Langevin Dynamics with Variance Reduction, Huang, Becker, IJCNN, 2021

[6] Deep learning via Hessian-free optimization, Martens, ICML 2010

*************************************************************************************************************************************************************************

(2) Regarding distributed optimization methods:

- On page 48 of the first submitted edition, authors mention "_we do not cover ... federated learning, ..._". It is not evident to me why methods covered in section 4.1 are not federated learning algorithms. Can authors elaborate this difference clearer?

- define __asynchronous__ clearly. This is usually a more challenging case.

- How are model parameters aggregated for "Downpour SGD"?

- Besides clearly clarifying the connection and difference between the algorithms discussed in section 4.1 and federated learning algorithms, it may be helpful to cite [1-2]

- [3-4] also discusses the technique of gradient tracking. [4], in particular, provides the method of induction to show the convergence of distributed SG tracking methods that distinguishes it from the cited work in the submission. For the thread of distributed optimization methods, variants on time-varying graph settings, incorporation of implicit optimization methods such as push-pull, asynchronous setting, I suggest authors dig into [Angelia Nedic's work](https://scholar.google.com/citations?user=86PxxsoAAAAJ&hl=en) a bit.

- Define $\mathbf{1}$ clearly. Any notation meaning conflict between Lemma 56 on page 59 and Theorem 59 on page 66?

-  in theorem 59 proof:
  -- What is $p_0$ ?
  -- Define $\lambda_i$ clearly.

[1] Advances and Open Problems in Federated Learning, Kairouz et al., 2021

[2] A Field Guide to Federated Optimization, Wang et al., 2021

[3] Harnessing Smothness to Accelerate Distributed Optimization, Qu and Li, 2017

[4] Distributed Stochastic Gradient Tracking Methods, Pu and Nedic, 2020

*************************************************************************************************************************************************************************

(3) Other general comments:

- if authors really want to mention the recently heated progress of large language models such as Llamas, Gemini and GPTs, why not briefly mentioning on high level the algorithms in which those models are trained? Otherwise, I do not see why it is needed to mention these LLMs in this survey.

- typo: equation (2.50), should be $v_{k,i}$ not $v_k, i$
- typo: parenthesis missing in theorem 58 on page 59\
- typo: equation (4.70), (4.72), $\top$ symbols are off

**Strengths And Weaknesses:**

I read the section concerning distributed optimization in detail while only scanning through the earlier sections, the content of which is more standard.

__Strengths__:

The structure of the survey is clear and the narrative flows naturally. This survey can be useful for graduate students who start to work in the field of optimization and they can quickly pick up conventional techniques that every researcher is expected to know.

__Weaknesses__:

The style of this submission is to state algorithms and then present proofs. What is lacking is the analysis about comparisons between algorithms.

For instance, there can be a table to summarize the needed assumptions for convergence bounds to hold, convergence rate, pros and cons.

What are the differences in main key techniques used in respective proofs?

The point is that it is expected that a survey can provide not only the factual information, but also authors' understanding on the differences between different methods. The analysis can consist of two parts: explanation of key difference points about pros and cons of algorithms; a table listing quantitative differences.

A good example for a table comparing algorithms are in these two works: Table 1 in [NEON2: Finding Local Minima
via First-Order Oracles, NeurIPS 2018](https://proceedings.neurips.cc/paper_files/paper/2018/file/d4b2aeb2453bdadaa45cbe9882ffefcf-Paper.pdf) and Table 1 in
[Natasha 2: Faster Non-Convex Optimization Than, SGD, Allen-Zhu, NeurIPS 2018](https://proceedings.neurips.cc/paper_files/paper/2018/file/79a49b3e3762632813f9e35f4ba53d6c-Paper.pdf)

To achieve the claim authors made in the Abstract that "_this paper aims to serve as a comprehensive theoretical handbook on optimization methods for DL_", making requested changes can be directly helpful. My questions and polishing suggestions are detailed in the following section.

---

> ### Author Response · Authors · 2024-10-04
> **Rebuttal - Part 1**
>
> We thank the Reviewer for his/her feedback. We address specific comments below. We have also already incorporated specific changes into our updated revision. All editions are marked with blue.
>
> ### **Comparisons of different optimization methods:**
>
> We now added Table 1&2 to summarize the convergence results for the first and second-order optimization methods and distributed (centralized and decentralized) optimization methods, and Table 3 to summarize the generalization results for the first-order methods and landscape-aware optimization methods. You could reach the tables in the introduction section of the updated revision. The tables guidance as to the difference between the methods and comments regarding their practical performance.
>
> ### **Regarding classic methods:**
> 1. **Theory-experiment gap:** For first-order methods, such as SGD, SGD-momentum, Adagrad, Adam, etc, that are most widely deployed in implementations, the theory-experiment gap often arises because theoretical guarantees are based on idealized conditions that may not hold in real-world scenarios. For example, in the proof of SGD and SGD-momentum under the nonconvex setting, we assume that the learning rate (step size) decreases inversely proportionally ($\alpha_t=1/t$). However, in empirical settings, decreasing the learning rate too quickly leads to terminating the learning process too quickly and consequently recovering suboptimal solution. Instead of the learning rate that is inversely proportional to the iteration counter dictated by the theoretical guarantees, practitioners typically use step-wise learning rate, linear learning rate, cosine annealing learning rate, etc. Those learning rates drop slower.
>
>     Hyperparameter settings vary widely, but common choices include using a learning rate of 0.001 for Adam and a learning rate of 0.01 or 0.1 for SGD. Other parameters, like momentum (typically set around 0.9) for SGD, can also significantly affect performance. Recent studies emphasize the need for careful tuning in practical applications.
>
> 2. **Assumptions for loss functions:** We are going to discuss whether the training loss functions commonly encountered in practice satisfy the following assumptions one by one.
>
>         1) Convexity: Deep learning often deals with non-convex loss landscapes. Regarding Table 1, all methods we mention except for BFGS have the convergence guarantee under nonconvex assumption.
>
>         2) Smoothness: All methods we listed require the smoothness of the loss function. In practice, loss functions may not be smooth, especially in the presence of noise or irregular data distributions. To solve this problem, deep learning researchers use techniques that improve the smoothness of the loss, among them regularization, dropout, batch normalization and so on.
>
>         3) Variance bounded: All the stochastic methods assume that the variance of the gradient estimates is bounded. In practice, training loss functions may not always satisfy this assumption, particularly in scenarios involving noisy or heterogeneous data. For instance, datasets with outliers or imbalanced classes can lead to higher gradient variance, complicating the training process. In order to solve this problem, we could implement mini-batch training instead of point-wise stochastic training to average out noise and stabilize gradient estimates. This is indeed done in practice.
>
>     We can approach the assumptions for convergence guarantees from a different perspective than viewing soft assumptions as the ones implying more useful bounds and strict assumptions as the ones implying worse bounds. Instead, these assumptions should be viewed as indicators of the specific conditions under which an algorithm is likely to perform optimally. For example, the smoothness assumption is crucial for convergence guarantees of all methods, it implies we could achieve better performance by smoothing the loss landscape of deep learning models. This is also the motivation of landscape-aware optimization methods we introduced in this paper (e.g. SAM, Entropy-SGD, etc) which could reach better generalization ability.
>
> 3. **Extra citations for first-order methods:** This is now added. Thank you.
>
> 4. **Clarification for Thm. 35:** Theorem 35 should be called Definition 35, since this is not a theorem. This is a typo. This change has been incorporated in the revised version. Definition 35 presents the update rule for the Stochastic Gradient Langevin Dynamics (SGLD) optimization algorithm, which is used to derive the update rule for sampling the latent variable x′ in the Entropy-SGD algorithm.

---

> ### Author Response · Authors · 2024-10-04
> **Rebuttal - Part 2**
>
> 5. **Convergence rate of SGD vs Adaptive methods:** We would like to clarify that the convergence rate $O(\frac{\ln K}{\sqrt K})$ is slower than $O(\frac{1}{\sqrt K})$  as $K$ increase. The term $\ln K$ grows without bound as $K$ increases, albeit slowly. So, while $\sqrt{K}​$ also increases, the presence of the logarithmic term in the first expression makes it grow slower than the linear rate in the second expression.
> We want to emphasize that the assumptions on step size between SGD and adaptive learning rate methods (Adagrad and Adam) are different. SGD assumes that the learning rate decreases inverse proportionally ($\alpha_t=1/\sqrt{t}$) while Adaptive methods assume the learning rate is constant or bounded. We therefore cannot conclude that Adaptive methods theoretically converge faster or slower than vanilla SGD since the convergence results rely on different assumptions. However, several studies [1,2,3] have empirically shown that Adaptive methods converge faster and achieve lower training loss in fewer iterations than SGD in various applications. It is because Adaptive methods compute individual learning rates for each parameter based on their historical gradients. This allows for larger updates for infrequent features and smaller updates for frequent ones, promoting efficient learning.
>
>     [1] Adam: A Method for Stochastic Optimization
>
>     [2] On the Convergence of Adam and Beyond
>
>     [3] Descending through a crowded valley-benchmarking deep learning optimizers
> 6. **Comments on convergence bounds:** We now have tables summarizing the convergence results (showing the bounds and their corresponding assumptions). Also, read our comment in point 2.
> 7. **Hessain-free methods:** We now added appropriate citations in the introduction section. Note however that these methods come with no theoretical guarantees, they are purely empirical.
>
> ### **Regarding distributed optimization methods:**
> 1. **Difference between distributed optimization and federated learning:** Distributed optimization and federated learning are two techniques that involve multiple computing units for model training but differ fundamentally in data handling and objectives. Distributed optimization operates on datasets shared among the computing nodes, where the same data is distributed across nodes for parallel processing. Thus all nodes have access to the same data or a shared copy. This type of optimization often requires frequent communication with a central server for gradient aggregation. In this setting it is reasonable to assume that the dataset on each node has the same distribution, just as what EASGD, LSGD and GRAWA require for their convergence guarantee. In contrast, federated learning focuses on training models in such a way that each node (device) has its own private data that is not shared with other devices, which is essential for maintaining privacy and security. Participants train local models and only send aggregated updates to a central server, minimizing data transfer and preserving user privacy.  Each participant (e.g., mobile device) retains its local data, which is never shared, promoting privacy and security. The same proof technique for distributed optimization is no longer suitable for federated learning because we can not assume the dataset among nodes share the same distribution. In our paper we focus on distributed optimization. Note that there already exist survey papers on federated learning for deep learning. A large body of work is purely empirical. We now clarify their difference in the main body of our paper.
>
> 2. **Definition of asynchronous:** In the context of optimization, "asynchronous distributed optimization" refers to operations or processes that occur time-wise independently of each other on each computational node. This means that different tasks can be executed on different nodes in parallel, and they are not timed with respect to each other (timing of their execution does not follow a predetermined sequence, but rather each node has its own clock). This way nodes do not wait for each other to perform computations.
>
> 3. **Clarification for Downpour SGD:** In Downpour SGD, model parameters are aggregated through an asynchronous process where multiple workers independently compute gradients based on their mini-batches of data and send these gradients to a central parameter server. The server aggregates the received gradients, often by averaging or using weighted aggregation based on mini-batch sizes, and then updates the model parameters. After the update, the parameter server broadcasts the new model parameters back to all workers, allowing them to continue training with the most current values while managing the challenges of parameter staleness.

---

> ### Author Response · Authors · 2024-10-04
> **Rebuttal -Part 3**
>
> 4. **Discussion on gradient tracking:** Note that papers [1,2] on gradient tracking focus on optimization over a network, rather than deep learning. Network problems focus on optimizing and analyzing structures like communication and transportation networks, while deep learning involves using neural networks to learn patterns from data for tasks like image and language processing. These works therefore are not directly relevant to our paper and it is unclear how to extend these methods to distributed optimization in deep learning.
>
>     [1] Distributed Stochastic Gradient Tracking Methods
>
>     [2] Harnessing Smothness to Accelerate Distributed Optimization
>
> 5. **Typo on $\mathbf{1}$:** $\mathbf{1}$ represents a vector where all elements are equal to 1. There is a typo in the definition of J in Lemma 56, it should be “$J=\mathbf{1}\mathbf{1}^T/m$”, just as in Theorem 59, not “$J=\mathbf{1}^T/m$”. J represents an $m\times m$ matrix filled with $1/m$.
>
> 6. **Notations for Thm. 59:** $p_0$ and $\lambda_i$ are defined in the previous paragraph in section 4.2.2. MATCHA assigns an independent Bernoulli random variable $B_j ≥ Bernoulli(p_j)$ for each matching, where $p_j$ is the activation probability of matching $G_j$, and $p_0$ is the activation probability of the first matching $G_0$. $\lambda_i$ is the $i$-th smallest eigenvalue of the graph Laplacian $\mathbf{L}$. We re-explained the notation in the proof to make sure it is easy to read.
>
> 7. **Contexts regarding LLMs:** We agree with the reviewer that our review focuses on the theoretical analysis of optimizations methods and mentioning LLMs is not our priority. We have already softened our statement on the LLMs in the main body of our paper.

---

> ### Comment · Reviewer_HTdu · 2024-12-09
> **Further comments on authors' revision as of December 9, 2024**
>
> I thank authors' kind attention to my review opinion. I have two main comments:
>
> (1) I personally do not find the explanation about adaptive algorithms satisfying. Specifically, based on the convergence order conclusion, the "accelerated algorithms" do not accelerate.
>
> Therefore, there should be more detailed articulation on why adaptive algorithms perform better than SGD. It deserves a subsection to summarize the reasons from established work, and I believe the entire community can quite benefit from this polished and organized content. If clearly explained and well-written, it can even become the standarized text for most optimization classes.
>
> (2) Respectfully, I disagree with authors' comment on the "Difference between distributed optimization and federated learning" and on "gradient tracking". The central characteristics of federated learning are clients' partial participation and the existence of a centralized server. Distributed optimization in a general sense means _a centralized server can be absent_.
>
> Angelica Nedic's work is highly relevant to distributed optimization, and in particular the gradient tracking.
>
> Regarding authors' words "_Note that papers [1,2] on gradient tracking focus on optimization over a network, rather than deep learning_", I believe this comment is not constructive.
>
> We are currently talking about optimization, and all cited work are focusing on optimization problems under distributed setting. The phrase "_deep learning_" really does not mean anything concrete in the optimization setting.
>
> Specifically, Pu and Nedic pointed out in [Distributed Stochastic Gradient Tracking Methods] that "_where each agent possesses a local cost function that is smooth and strongly convex._" This is a well-defined distributed optimization problem.
>
> Unless authors very explicitly define about the meaning of the specific sub-domain they want to cover in this review paper, using "distributed optimization" to mean exactly what the current text in the manuscript covers in section 4 is not acceptable to me.
>
> (Additional Information on Dec 10) It will be helpful to also use a table to quantify what authors mean by "distributed" and "federated". The settings should be clear and the corresponding titles for each setting should be accurate, in my opinion.

---

> ### Author Response · Authors · 2025-01-20
>
> 1. Thank you for your thoughtful feedback. We understand that the convergence order conclusion might not fully capture the intuitive advantages of adaptive algorithms over SGD, especially with respect to their empirical performance. Note however that this theoretical order is consistent with the literature. For example, Robbins and Monro (1951) established an $O(1/\sqrt{K})$ convergence rate for stochastic gradient descent (SGD) under certain conditions. Momentum-based SGD methods also achieve a similar $O(1/\sqrt{K})$ rate, as shown in [2]. Adaptive learning rate algorithms, such as Adam and Adagrad, have gained popularity due to their empirical acceleration over vanilla SGD. Theoretical analyses, including [3], prove an $O(\ln K / \sqrt{K})$ convergence rate for Adam and Adagrad, while AMSGrad achieves the same rate as demonstrated in [4]. To clarify, while the convergence rate $O(\ln K / \sqrt K)$ does indeed have a slower asymptotic rate compared to $O(1 / \sqrt{K})$, the presence of the logarithmic term becomes significant primarily for very large values of K. As I stated before, since the assumptions used for SGD and Adaptive methods are different, we could not conclude which algorithm converges faster based on the theoretical results. Importantly, the real-world performance of adaptive algorithms often transcends theoretical convergence rates due to their ability to adjust learning rates based on the data's geometry and the gradient history.
> In terms of practical performance, adaptive algorithms (such as Adagrad, Adam, etc.) have been shown to perform better than SGD in several scenarios due to the following key reasons, which we now add a paragraph in section 2.2 and expanded upon in the revised manuscript:
>
>         i) Individualized Learning Rates: Adaptive methods assign different learning rates to each parameter based on its historical gradients. This enables the method to give larger updates to infrequent parameters (which often carry useful information) and smaller updates to frequent parameters, improving overall efficiency.
>
>         ii) Efficient Use of Gradient Information: Adaptive methods often utilize gradient scaling to control the impact of noisy or sparse gradients, which allows them to avoid the slow progress SGD might experience in such scenarios, especially when dealing with sparse data or noisy gradients.
>
>         iii) Robustness to Hyperparameter Choices: Adaptive algorithms tend to be more robust to the choice of initial learning rate, as the learning rates are adapted throughout the training process. This contrasts with SGD, where choosing the right learning rate can be crucial for convergence and performance, requiring more manual tuning.
>
>         iv) Empirical Evidence: Several studies (e.g., [4,5,6]) have empirically shown that adaptive methods converge faster and require fewer iterations to achieve lower loss compared to SGD, particularly for complex, non-convex optimization problems. These results are often seen in practical deep learning applications, where adaptive methods have become the default choice due to their superior performance in terms of training efficiency.
>
>
>     Empirical solutions described in the points above however significantly complicate the theoretical analysis. To the best of our knowledge no theoretical work includes those in their analysis.
>
>     The Reviewer writes “it can even become the standarized text for most optimization classes.” We could not agree more with the Reviewer. Indeed our goal is for our survey paper to become a textbook of Optimization for Deep Learning. It should also help new students (PhD-level but also Master’s interested in conducting research), to more quickly learn how to derive convergence and generalization guarantees for different optimization approaches in DL and guide them in deriving theoretical guarantees for their own methods. Currently, there are no such standardized texts and the literature is vast, hard to penetrate, and heavily dominated by experimental papers. The first author, who was a PhD student and has now graduated, has met many peers during her PhD studies who were struggling to learn theoretical concepts related to optimization or prove theoretical results for their own DL algorithms, due to the lack of standardized textbooks. This mainly motivated the writing of this survey. It is also important for the authors to popularize the perception of DL discipline as not a purely empirical discipline.
>
>     [1]  A stochastic approximation method.
>
>     [2] Unified convergence analysis of stochastic momentum methods for convex and non-convex optimization.
>
>     [3] A Simple Convergence Proof of Adam and Adagrad
>
>     [4] On the Convergence of Adam and Beyond
>
>     [5] Adam: A Method for Stochastic Optimization
>
>     [6] Descending through a crowded valley-benchmarking deep learning optimizers

---

> > ### Author Response · Authors · 2025-01-20
> >
> > 2. Thank you for your insightful comments. We would like to clarify that the primary focus of our survey is on optimization methods applied in training DL models, as indicated by the title: “A Survey of Optimization Methods for Training Deep Learning Models: Theoretical Perspective on Convergence and Generalization”. Our goal is to review optimization methods that are widely used for training deep learning models, and as such, we do not cover all distributed optimization techniques. While we acknowledge that gradient tracking is an important topic in distributed optimization, it is not directly relevant to our survey because it is not typically applied in the context of deep learning training.
> >
> >     Regarding the comment on federated learning, we agree that federated learning and general distributed optimization are related but distinct concepts. Section 4 of our paper specifically focuses on distributed optimization methods, and we opted not to delve deeply into federated learning, as its characteristics—such as clients’ partial participation and the presence of a centralized server—introduce unique considerations that differ from distributed optimization. This unique considerations come from the fact that federated learning addresses privacy and security constraints when training deep learning models - so federated learning is dedicated to particular applications. Our paper however considers general optimization algorithms for deep learning, rather than application-specific. This distinction is why we have not focused heavily on federated learning in our review. So to put it in other words and define the sub-domain we want to cover when referring to distributed optimization according to the Reviewer’s request, in our survey paper we consider data-unconstrained optimizers. This is now clarified in our paper.
> >
> >     We appreciate your reference to Angelica Nedic’s work, which is indeed highly relevant to distributed optimization. However, as the techniques discussed in the cited works [1, 2] on gradient tracking primarily address optimization in networked settings rather than deep learning models, they fall outside the scope of our review. In our context, “deep learning” refers to the specific application of optimization methods to the training of deep neural networks, where the optimization problems typically involve high-dimensional, non-convex functions that are distinct from other forms of distributed optimization. Thus, "deep learning" in optimization settings has a concrete meaning to us.
> >
> >     [1] Distributed Stochastic Gradient Tracking Methods
> >
> >     [2] Harnessing Smothness to Accelerate Distributed Optimization

---

### Review · Reviewer_SP72 · 2024-12-04

**Summary Of Contributions:**

This article provides a technical review of existing optimization methods for training deep learning models,
from the perspective of algorithm convergence and generalization. Several categories of methods are considered,
ranging from first-order methods with/without adaptivity, second-order methods, and landscape-aware methods
to distributed methods. State-of-the-art results about these methods are given, as well as their detailed proofs.

**Audience:**

Yes

**Claims And Evidence:**

No

**Requested Changes:**

- The definition of the stability in Def. 5 is not so clear, where is your variable z in eq. 2.4?
- Some statements of the theorem are not consistent, e.g
	* Theorem 8: the assumption that t>0 is a typo, should be K>0? What does that mean that F(x) is L-smooth? Is it defined later in Assumption 13?
	* Theorem 12: what is the k in eq. 2.28? Should it be K?
	* Definition of UA: in eq. 2.50, the square root should include the epsilon or not? Otherwise, how can it capture the AdaGrad method in eq. 2.51?
	* Theorem 24: what is the x_t in the statement?
	* Definition 34: there is dx’ missing in eq. 3.11
	* Definition 35: the choice of the noise level epsilon for the random variable epsilon_t, should that be set to 1 or not, i.e. epsilon=1?
	* Assumption 49: what is the x in (3) and (4)?
	* Theorem 50: the statement makes a strong assumption that F(tilde x_k) <= F(x_k^i) . It is unclear when this is satisfied.
	* Theorem 52: what is the x_k ? Should it be x_k^i?
	* Theorem 66. (Convergence of AL-DSGD; Nonconvex Setting; Theorem 2 in ()). Missing citation about Theorem 2.
- Some proofs in the article should be double-checked.
	* Theorem 7: choice of alpha_k involves a constant s, where is this constant used in the proof?
	* Theorem 8: where is the constant G used in the proof? Should that appear in eq. 2.19? What is the eta in eq. 2.18?
	* Theorem 24: there is some error in the Taylor expansion step.
-	Regarding your remark about Adam with momentum after Theorem 23, it seems in practice beta_1 = 0.99 is often used. When you say that increasing beta_1 always deteriorates the bound, does that mean one should use a smaller beta_1 in practice?
-	The description of the Downpour SGD method in section 4 is not complete, as it can involve asynchronous multistep local gradient computations. This should be mentioned even though it is not analyzed.
-	Missing sub-index k in eq. 4.14. and eq. 4.16.
-	Is it unclear why the update rule of GRAWA in eq. 4.36 in correct. Which i are you using?


Above all, the reviewer feels that the submission is not serious enough. As a consequence, a careful check of the proof is not possible. Therefore a rejection of the article is recommended.

**Strengths And Weaknesses:**

Strength:
- The article covers a wide range of optimization methods for deep learning, which are very interesting.
- It gives an analysis of their training and test performance, in particular in the non-convex setting.

Weakness:
- The writing should still be improved. In particular, several mathematical parts of the article are not very clear / consistent and therefore not so sound.
- The LBFGS method part is less interesting compared to the rest. It is better to includes recent proposed methods which involve stochastic gradients, i.e. the family of stochastic LBFGS.
- The article summarizes the results in Table 1 in terms of convergence rate. However, they are not compared on the same stopping criteria therefore it is not informative.

---

> ### Author Response · Authors · 2024-12-09
>
> We thank you for your review. We are currently working on improving the paper according to your suggestions as well as in-detail verifying the consistency of the notation and re-checking the proofs.

---

> ### Author Response · Authors · 2024-12-20
> **Rebuttal - part 1**
>
> 1. **LSGB:** We choose to focus on the batch case for second-order quasi-Newton methods because these methods are primarily studied and well-established in batch optimization, where theoretical results are clearer and easier to interpret. Although stochastic versions of these methods are primarily applied in deep learning, the theoretical guarantees for stochastic second order methods are less developed and more difficult to generalize, especially in the deep learning context. While stochastic second-order often show faster convergence rate per iteration in practical experiments compared to first-order stochastic methods, we can currently only provide sub-linear convergence rate for the stochastic second-order methods – similar to the first-order methods – due to the variance in the stochastic approximation.
> 2. **Stopping Criterion:** The stopping criterion for convex and nonconvex settings are aligned with the characteristics of each problem type. For convex functions, the stopping criterion is based on the condition that the difference between the current function value $F(x)$ and the optimal value $F(x^*)$ becomes sufficiently small. In the case of nonconvex setting, the stopping criterion is determinated by the gradient norm $||\nabla F(x)||$, which should be sufficiently small.
> 3. **Def.5:** We make a typo here in the definition. The variable should be $\xi$ not $z$ in the formula. Where $\xi$ represents a specific data example sampled from the space $Z$ with the data distribution $D$, and $f(x;\xi)$ denotes the loss of the model for the data sample $\xi$. The correct formula is $\sup_{\xi}E[f(A(\mathcal{S};\xi))-f(A(\mathcal{S}’;\xi))]\leq\epsilon_s.$
> 4. **Thm.8:** Yes, it should be $K>0$.
> 5. **Thm12:** We now change it to the capital K.
> 6. **Def of UA:** The square root should include $\epsilon$ and we change it.
> 7. **Thm 24:** It is $x_k$ not $x_t$, we change it.
> 8. **Def. 34:** We already add the missing $dx’$.
> 9. **Def. 35:**  Regarding to the update rule of Entropy-SGD introduced in [1], the parameter $\epsilon$ in SGLD is the level of  thermal noise and was fixed to $\epsilon\in[10^{-4}, 10^{-3}]$. We have add the claim in the main body of the survey.
>
>     [1] Entropy-sgd: Biasing gradient descent into wide valleys
>
> 10. **Asumption 49:** Here is typo in the Assumption 49, $x$ in condition (3) and (4)  should be $x_k^i$. $E[g_i(x_k^i)]=\nabla F(x_k^i)$ and $E[||g_i(x_k^i)-\nabla F(x_k^i)||^2]\leq\sigma^2$ in (3) and (4) mean that for the parameter $x_k^i$ of model on worker i at iteration k, the gradient is unbaised and variance of gradient approximation is bounded respectively.
> 11. **Thm 50:** The assumption for Theorem 50 is motivated from the convergence proof of Leader Stochastic Gradient Descent (LSGD) method introduced in [2]. In the proof of EASGD, we assume that the center parameter always wins the local workers, so that we could assume $F(\widetilde{x}_k)\leq F(xk^i)$, where $\widetilde{x}_k$ is the parameter of center worker at step k and $x_k^i$ is the parameter of worker i at step k. The same assumption is used in GRAWA algorithm [3].
>
>     [2] Leader stochastic gradient descent for distributed training of deep learning models
>
>     [3] GRAWA: Gradient-based Weighted Averaging for Distributed Training of Deep Learning Models
> 12. **Thm 52:** Yes, it should be $x_k^i$ and already corrected.
> 13. **Thm 66:** Already add the missing citation.
> 14. **Thm 7:** It is a typo here. The definition of learning rate is $\alpha_k=\alpha_0/(k+1)$ not $\alpha_k=\alpha_0/(ks+1)$. The following proof of Theorem 7 is based on the definition of learning rate as $\alpha_k=\alpha_0/(k+1)$。
> 15. **Thm 8:** $||\nabla F(x)||\leq G$ means that function $F$ is $G$-Lipschitz continuous (bounded gradient). The Lipschitz continuous assumption is not used in the convergence proof of SGD and could be removed. We have already update that. $\eta$ here represents the learning rate, which is denoted as $\alpha$ in this proof. We already correct that.
> 16. **Thm 24:** We use the second-order Taylor expansion of $f(x)$ at point $x_k$. To be precise, we should use $f(x)\approx f(x’)+\nabla f(x) (x-x’)+(x-x’)^T\nabla^2f(x)(x-x’)$ instead of  $f(x)= f(x’)+\nabla f(x) (x-x’)+(x-x’)^T\nabla^2f(x)(x-x’)$. However, we would like to emphasize that this approximation will not change the final conclusion, since the bias term of second-order Taylor expansion is $O(||x-x’||^3)$, which is smaller than $O(||x-x’||^2)$.

---

> ### Author Response · Authors · 2024-12-20
> **Rebuttal - part 2**
>
> 17. **Hyperparam in Adam and Adagrad:** $\beta_1$ is the momentum parameter. In particular, it is often chosen as $\beta_1=0.9$, not 0.99. We want to emphasize that it is normal that there is a gap between the practical experiments and theoretical resulys. In the experimental section of [4], which provides the convergence proof of Adagrad and Adam, the author also report the gap. In the toy example, $\beta_1$ has limited influence on the final results. And in the deep learning task of CIFAR10, they observe a sweet spot for the momentum $\beta_1$, not predicted by the theory.
> 18. **Downpour SGD:** We already add it in the paper.In Downpour SGD, model parameters are aggregated through an asynchronous process where multiple workers independently compute gradients based on their mini-batches of data and send these gradients to a central parameter server. The server aggregates the received gradients, often by averaging or using weighted aggregation based on mini-batch sizes, and then updates the model parameters. After the update, the parameter server broadcasts the new model parameters back to all workers, allowing them to continue training with the most current values while managing the challenges of parameter staleness.
>
> 19: **Update rule for GRAWA:** $x_k^c$ in GRAWA denotes the same thing as $\widetilde{x}_k$ in Downpour SGD and LSGD, which all represents the center variable. In order to keep consistency among the notations in the survey,we have already change all $x_k^c$ into $\widetilde{x}_k$.

---

### Review · Reviewer_Vr6h · 2024-12-09

**Summary Of Contributions:**

This survey paper summarizes existing results in optimization for deep learning training, covering first-order, second-order, quasi-Newton, landscape-aware, and distributed optimization methods. It features inclusion of rigorous proofs for convergence rates and generalization error bounds, which is based on a stability argument.

**Audience:**

Yes

**Broader Impact Concerns:**

N/A.

**Claims And Evidence:**

No

**Requested Changes:**

Please consider providing unified, new insights in the revision or significantly shortening the paper to an annotated bibliography. In addition, please ensure consistency and precision in the presentation.

**Strengths And Weaknesses:**

**Strengths.**
- I appreciate that this paper discusses generalization error, an aspect often overlooked in many papers on optimization for machine learning.
- Some of the cited papers are informative to me.

**Weaknesses.**
- In my opinion, the main issue is the *necessity of publishing such a paper*. It does not present new insights or results but merely collects existing results. Then, an annotated bibliography would suffice.
- Even as a survey paper compiling existing results, this work does not effectively achieve its goal:
    - The sections on second-order and quasi-Newton methods consider only the batch case and do not cite any paper for deep learning training. This choice requires justification.
    - It is unclear why the paper devoted a section on leader stochastic gradient descent. I am not an expert in this field, but according to Google Scholar, the associated paper (Teng et al., 2019) only has less than 20 citations. There is another section on adjacent leader decentralized SGD. The associated paper (He et al., 2024) was uploaded to arXiv this year and has received 0 citations. These choices need justifications.
    - It is clearly stated that Figure 6 is taken from Wang et al. (2019). I am not sure if one is allowed to do this in academic writing.
    - The beginning paragraphs of Section 3 only summarizes results, without any comments or conclusions.
    - The definitions of convergence rates in Section 2.1.1 are given with respect to distance to the minimizer. However, these definitions do not appear to be consistent with those used in the rest of the paper, which are stated in terms of the gradient norm.
    - When presenting the so-called adaptive unified (UA) algorithm, Défossez et al. (2022) should be cited. The current presentation may mislead readers into thinking that the formulation of UA is a novel contribution of this survey paper.
    - The choice of citations is weird. For example, Section 1 cites Robbins & Monro (1951) for stochastic optimization methods, Nesterov (1983) for accelerated gradient methods, but Liu et al. (2020) for momentum. It is also weird to cite Ramezani-Kebrya et al. (2018) for the notion of stability.
    - It is stated that the provided interpretation of SmoothOut is novel. But I believe it is common sense and can be traced back to, for example, Chapter 9 of the book by Nemirovsky and Yudin.
    - Theorem 24, Theorem 33, and Theorem 37 lack citations.
    - Definition 5: The alternative dataset $\mathcal{S}'$ does not appear in (2.4). There is no need to introduce $\xi'$. The notation $z$ is not defined, though its meaning can be inferred for those who are familiar with stability.
    - It seems that the definition of $L$-smoothness is not provided.
    - The term "energy" is used frequently in Section 3.2 but not in other sections. Please keep consistency of presentation.
    - There are several imprecise statements. For example, it is stated that "Newton’s method achieves a quadratic convergence rate, which is much faster that the linear convergence rate of first-order gradient descent (GD) method." Whether GD can achieve a linear rate depends on properties of the objective function. I do not get what "model performance" on p. 6 means. In Section 3.4, it is stated that $\epsilon_i$ follows a uniform distribution on $[-a, a]$ for some $a > 0$, but $\epsilon_i$ should be a vector.
    - There are a number of typos. Some papers should have been cited using the \citet command instead of \citep (or \cite).
    - Please use the \operatorname command when typing the $\operatorname{sign}$ operator.

---

> ### Author Response · Authors · 2024-12-20
> **Rebuttal - part 1**
>
> 1. **Main contribition:** We would like to clarify several points regarding our manuscripts. Our submission aims to serve as a comprehensive “theoretical” handbook on optimization methods for deep learning (DL), designed to provide valuable insights and understanding to both novice and seasoned researchers in this field. Beyond simply collecting results, we compare various approaches and provide theoretical guarantees for techniques that were previously lacking such guarantees, such as SmoothOut, Downpour SGD and EASGD. These methods did not have convergence proof in their original paper proposed these methods, and we address that. To facilitate comparison, we include Table 1 and 2 to summarize the convergence result for first/second-order optimization methods, as well as distributed optimization. Additionally, Table 3 summarizes the generalization results for first-order and landscape-aware optimization methods.
> 2. **Batch-version of quasi-newton methods:** We choose to focus on the batch case for second-order quasi-Newton methods because these methods are primarily studied and well-established in batch optimization, where theoretical results are clearer and easier to interpret. Although stochastic versions of these methods are primarily applied in deep learning, the theoretical guarantees for stochastic second order methods are less developed and more difficult to generalize, especially in the deep learning context. While stochastic second-order often show faster convergence rate per iteration in practical experiments compared to first-order stochastic methods, we can currently only provide sub-linear convergence rate for the stochastic second-order methods – similar to the first-order methods – due to the variance in the stochastic approximation.
> 3. **Clarification for citations:** As stated earlier, our goal is to provide a theoretical handbook for optimization methods in deep learning. We include the papers on Leader Stochastic SGD (Teng et al., 2019) and adjacent leader decentralized SGD (He et al., 2024) because they offer detailed theoretical analysis that can be broadly applied to other centralized and decentralized methods. For example, the convergence proof scheme for LSGD can be adapted to prove the convergence of EASGD and GRAWA with light adjustments on assumptions, while maintaining the same proof structure. The proof for AL-DSGD is valuable for understanding how to provide convergence guarantee for distributed optimization methods with many hyperparameters, since AL-DSGD has up to 4 hyperparameters ($\lambda_N, \lambda_\tau, w_N, w_\tau$). This introduces additional complexity in bounding each hyperparameter to establish the overall convergence guarantee.
>
> 4. **Beginning of section 3:** The beginning paragraphs of section 3 provide a summary of the key results in landscape-aware optimization methods, in order to establish a clear foundation for the subsequent analysis and discussion. We aimed to present these result concisely in order to give the readers ad quick overview before moving on to the detailed comment.
> 5. **Clarification for convergence rate:** In deep learning optimization, under convex settings, convergence guarantees are often proved with respect to the minimizer $x^*$. This is possible because the convexity assumption implies that $f(x)-f(y)\geq\nabla f(x)^T (x-y)$, which allows mapping the distance to minimal value $f(x)-f(x^*)$ to the distance to the minimizer $||x-x^*||$. However, in nonconvex settings, this mapping no longer holds, making it impossible to prove convergence in terms of distance to the minimizer. As a results, researchers typically focus on proving the gradient norm converges to zero, as $\nabla f(x^*)=0$ implies that.
> 6. **Def. for UA:** We now cite Défossez et al. (2022) when introducing the definition.
> 7. **Change citations:**  We already change the citation. We now cite [1] for momentum method.
>
>     [1] Some methods of speeding up the convergence of iteration methods
> 8. **SmoothOut:** We now soften the claim about the novelty of the interpretation of SmoothOut. While the general idea behind the interpretation may be straightforward, our contribution lies in applying this interpretation specifically to the landscape-aware optimization methods. We believe that it adds values by enhancing our understanding to SmoothOut method.
> 9. **Lacking citations:** We already add the missing citations for Theorems.
> 10. **Def. 5:**  We make a typo here in the definition. The variable should be $\xi$ not $z$ in the formula. Where $\xi$ represents a specific data example sampled from the space $Z$ with the data distribution $D$, and $f(x;\xi)$ denotes the loss of the model for the data sample $\xi$. The correct formula is $\sup_{\xi}E[f(A(\mathcal{S};\xi))-f(A(\mathcal{S}’;\xi))]\leq\epsilon_s.$

---

> ### Author Response · Authors · 2024-12-20
> **Rebuttal - part 2**
>
> 11. **Def for L-smooth:** We did not specifically introduce the definition of L-smoothness in our survey, as we consider it is a standard concept in optimization, just as strongly-convex and L-Lipschitz continuous. Now we add the definition of strongly-convex, L-Lipschitz, and L-smooth in section 2.1 to help people understand.
> 12. **Definition of 'energy' in Entropy-SGD:** The term "energy" used in Section 3.2 refers to the same concept as "loss" in other sections. We chose to use "energy" instead of "loss" to align with the terminology from the paper by [2], which introduced the Entropy-SGD algorithm. In their work, the term "energy" is used to describe the loss function and its landscape. To maintain consistency with their approach, we adopted this terminology. We will make this clearer in the revised version to avoid confusion.
>
>     [2] Entropy-sgd: Biasing gradient descent into wide valleys
> 13. **Imprecise statements:**  We have already fixed these imprecise statements.
>
>     1)"Popular Newton’s method achieves a quadratic convergence rate under nonconvex assumptions, which is much faster that the linear convergence rate of first-order gradient descent (GD) method under the same assumptions."
>
>     2)We would like to clarify that if the topology of the communication graph in decentralized optimization methods is too sparse, the performance of the model trained on each worker will degrade.
>
>     3) We change $U(-a, a)$ into $U[-a, a]^d$ to represents the d-dimentional uniform distribution.

---

### Review · Reviewer_1VbJ · 2024-12-10

**Summary Of Contributions:**

This paper surveys optimization methods for training deep neural networks, focusing on theoretical convergence and generalization properties. The authors analyze various stochastic first-order (SGD, momentum, Adam, Adagrad), second-order (Newton, BFGS, L-BFGS), landscape-aware (SAM, Entropy-SGD, LPF-SGD, SmoothOut), distributed centralized (Downpour SGD, EASGD, LSGD, GRAWA), and decentralized (D-PSGD, MATCHA, AL-DSGD) methods. Convergence rates and step-size parameters are summarized and compared across these methods in several tables.

**Audience:**

Yes

**Broader Impact Concerns:**

No concerns

**Claims And Evidence:**

No

**Requested Changes:**

- Thoroughly correct all typos, mathematical errors, and inconsistencies in notation and proofs. This is paramount for credibility.

- Justify the inclusion of each optimization method, especially those less widely used in modern deep learning. Provide a clear rationale for the choice of methods included in the survey.

- Remove results that depend on overly restrictive assumptions.

- Reduce the scope of the survey to a smaller, more manageable set of optimization methods, allowing for a much deeper and more rigorous analysis.

- Provide complete and detailed proofs, clearly explaining all steps and assumptions.

- Replace outdated results with the most recent findings and cite appropriate contemporary work. This includes revisiting Adam [1] and quasi-Newton methods [2], as well as asynchronous SGD [4].

- Address the specific issues raised regarding Theorem 7, Theorem 8, Theorem 10, Theorem 12, Assumption 32, and other theorems where issues have been identified.

- Improve the overall readability, especially using math commands for "sign" and "diag", as well as clarifying all notations.

**Strengths And Weaknesses:**

## Strengths

- The paper addresses a crucial topic in deep learning: the theoretical understanding of optimization algorithms. This is a valuable contribution, particularly given the lack of rigorous theoretical analysis in many existing surveys.

- The survey encompasses a wide range of optimization methods, providing a diverse perspective.

## Weaknesses

- The selection of methods is not well-justified. While some popular (in practice and theory) methods (SGD with momentum, Adam, Distributed SGD) are included, others (e.g., L-BFGS, EASGD, LSGD, GRAWA) lack sufficient justification for their inclusion, especially considering the availability of more widely used and theoretically well-understood alternatives in modern deep learning.

- The emphasis on generalization guarantees is unclear. The paper's focus seems primarily on optimization, and the reliance on the uniform stability framework, especially in non-convex settings with Lipschitz assumptions, is problematic due to poor theory-practice correspondence.

- The manuscript contains numerous typos, mathematical errors, and inconsistencies in notation, notably in the convergence proofs. These errors significantly hinder the paper's credibility.

- Several theoretical results are vacuous because they depend on overly strong assumptions (e.g., strong convexity and bounded gradient norm simultaneously) that rarely hold in practice, rendering the findings largely inapplicable. Some proofs seem to ignore stated assumptions, such as the bounded gradient norm in Theorems 7 and 8.

- Some results are based on outdated papers and techniques, lacking references to more recent, improved analyses (e.g., see [1, 2] for recent advances on Adam and quasi-Newton methods, respectively). Theorem 12, for instance, relies on a 2016 that is likely outdated.

- Several proofs lack crucial steps and explanations (e.g., Theorem 10) and contain unclear statements ("approximately minimized"). The notation $k_0$ in Theorem 10 also requires clarification.

- The statement that
>We first show that GRAWA reaches the same convergence rate as SGD
is incorrect as the analysis requires stronger assumptions like $v$-cone. Moreover, it seems problematic as it probably implies a lower bound on the gradient norm analogously to function Lipschitzness. Thus, it contradicts the PL condition, allowing an arbitrarily large gradient norm. Thus, the result holds for an empty class of functions, which allows arbitrary results to be proved.

- Feasibility/limitation of assumption of 32 is unclear. Are there any practical function examples that satisfy this?

- The links to the Appendix do not function correctly.

___

[1] Li, Haochuan, Alexander Rakhlin, and Ali Jadbabaie. "Convergence of Adam under relaxed assumptions." Advances in Neural Information Processing Systems 36 (2024).

[2] Rodomanov, Anton, and Yurii Nesterov. "New results on superlinear convergence of classical quasi-Newton methods." Journal of optimization theory and applications 188 (2021): 744-769.

[4] Mishchenko, Konstantin, et al. "Asynchronous SGD beats minibatch SGD under arbitrary delays." Advances in Neural Information Processing Systems 35 (2022): 420-433.

---

> ### Author Response · Authors · 2024-12-20
> **Rebuttal - part 1**
>
> 1. **The selection of methods:** LBFGS is a popular second-order method proposed in [1] with 10,000+ citations. Unlike traditional BFGS, LBFGS only stores a limited number of past gradients and updates, making it much more memory-efficient, especially for large-scale problems. EASGD (Elastic Averaging SGD) is a popular decentralized optimization method proposed in [2], with over 700 citations. It efficiently balances local and global optimization in distributed training environments by allowing nodes to update locally and periodically synchronize with the global model. EASGD has significantly influenced the development of other decentralized methods
>
>     As stated earlier, our goal is to provide a theoretical handbook for optimization methods in deep learning. We include the papers on Leader Stochastic Gradient Descent (Teng et al., 2019) and Adjacent Leader Decentralized SGD (He et al., 2024) because they offer detailed theoretical analyses that can be broadly applied to other centralized and decentralized optimization methods. For example, the convergence proof for Leader SGD (Teng et al., 2019) can be adapted to prove the convergence of EASGD with slight adjustments to the assumptions, while maintaining the same proof structure. Moreover, we could not escape introducing extra hyperparementers when constructiong algorithms. The proof for Adjacent Leader Decentralized SGD (AL-DSGD) is valuable for understanding how to provide convergence guarantees for algorithms with many hyperparameters, since AL-DSGD has up to 4 hyperparameters: $\lambda_N, \lambda_\tau, w_N, w_\tau$​. This introduces additional complexity in bounding each hyperparameter to establish the overall convergence guarantee.
>
>     [1]On the limited memory BFGS method for large scale optimization
>
>     [2]Deep learning with elastic averaging SGD
> 2. **Generalization errors:**  Generalization guarantees are essential for ensuring that a machine learning model performs well on unseen data, not just the training set. They help prevent overfitting and provide theoretical confidence in the model's predictive performance. The uniform stability framework is widely used to derive such guarantees, as it connects the stability of a model to its ability to generalize. Essentially, if a model's predictions are not sensitive to small changes in the training data, it is less likely to overfit.
>
>     The Lipschitz assumption in non-convex settings is commonly used in optimization theory, particularly in deep learning, to derive generalization bounds and stability results. While these assumptions may not always hold precisely in practice, they provide a useful theoretical foundation for understanding how optimization algorithms can influence generalization.
> 3. **Clarification for Assumptions:** The assumptiosns used in the survey paper keep consistancy with the literature and are widely used in the proof of optimization in deep learning.
>
>     1)convexity: Deep learning typically involves non-convex loss landscapes, and all methods we discuss, except for BFGS, come with theoretical guarantees under non-convex assumptions. For those methods where we initially provide guarantees under strongly-convex assumptions, we also present corresponding proofs for the non-convex case, ensuring broader applicability to real-world deep learning scenarios.
>
>     2)Bounded gradient norm assumption, also now and Lipschitz continuous assumption, is commonly used and widely accepted in optimization fordeep learning. It is not a particularly strong one in the context of optimization for deep learning. It simply requires that the model's output does not change too drastically in response to small changes in the input, which is a relatively mild condition. Many commonly used loss functions and neural network architectures satisfy this condition, making it a useful and practical assumption for deriving theoretical guarantees. While it might not fully capture every aspect of real-world models, it provides a solid foundation for analyzing optimization behavior and generalization properties in deep learning.

---

> ### Author Response · Authors · 2024-12-20
> **Rebuttal - part 2**
>
> 4. **Clarification for citations:** While we acknowledge that recent advancements (e.g., [3] on Adam and [4] on quasi-Newton methods) offer valuable improvements, they do not invalidate the core contributions of earlier studies. For example, Theorem 12 proposed in 2016 [5] provides a valuable general proof scheme could be applied on multiple momentum-based SGD methods. Compared with [3] proposed in 2024 provides convergence proof for Adam, [5] we cited in this paper proposed in 2020 not only provides convergence proof for Adam, but also provides proof with Adagrad.
>
>     [3] Li, Haochuan, Alexander Rakhlin, and Ali Jadbabaie. "Convergence of Adam under relaxed assumptions." Advances in Neural Information Processing Systems 36 (2024).
>
>     [4] Rodomanov, Anton, and Yurii Nesterov. "New results on superlinear convergence of classical quasi-Newton methods." Journal of optimization theory and applications 188 (2021): 744-769.
>
>     [5] A Simple Convergence Proof of Adam and Adagrad
> 5. **GRAWA:** We soften the conclusion on convergence rate of GRAWA in the main body of the paper. The strongly-convex does not contradicts the PL condition. In the proof of deep learning, it is normal to assume that the model parameter $x$ lies in a bounded set and the loss function $F(x)$ is bounded. When $F(x)$ is bounded by some constant $M$, the bound for the PL assumption $\mu(F(x)-F(x^*))\leq2 \mu M$. When $v<2\mu M$, the function $F$ is a PL and $v-cone$ function.
> 6. **Feasibility of Assumption 32:** The assumption holds for quadratic function. If $F(x)=\frac{1/2}||Ax||^2$, The Hessain matrix of function $F(x)$ is $G(x)=\nabla^2 F(x)=A$. Denote the smallest and largest eigenvalue of matrix A is $\lambda_{min}$ and $\lambda_{max}$. if $M_1\leq \lambda_{min}$ and $M_2\geq \lambda_{max}$, the assumption holds.
> 7. **Appendix:** This is because the Appendix is defer to the supplementary meterials and the uploaded PDF file only includes the main body of the paper and the reference. We will include the Appendix in the PDF as well.

---

### Comment · Action_Editor_qqY5 · 2024-09-24
**First review and other comments**

Dear authors,

You've received the first review. Since the reviewer got back to you so soon, I think it'd be a good idea to properly address their concerns and update the paper without waiting for the other reviews. I checked the reviewer's comments and they make a lot of sense to me, in particular this comment: "The point is that it is expected that a survey can provide not only the factual information, but also authors' understanding on the differences between different methods." After quickly going through the paper, I feel that this feedback is very on point, and at times, the paper appears like a laundry list of optimization methods. It would be very valuable if the authors could add meaningful comparisons between the listed methods, providing the reader with the necessary guidance to choose what to use in practice.

Some minor comments from me:
(4.23) is identity, not inequality.
some equations are missing punctuation, e.g., first equation on page 10, (2.12), (2.14), (2.18), (2.48), (4.26), etc.
I'm confused by the use of subscripts in the norms, for instance, Definition 3 doesn't use subscripts but Theorem 7 and its proof do. Please make it consistent.
Punctuation in Algorithm 3.4 is inconsistent too.
Some equations use $\log$, for instance (2.46), whereas the rest of the paper seems to use $\ln$.
Some equations denote transposition with $^\top$, for instance, the first equation on page 40, the rest uses $^T$.
I'm a bit confused about the use of $m$ and $M$ on page 57, from looking at equation (4.38), it appears that they refer to the same thing, but for some reason, $M$ is used in most parts of the proof.
Citation style is inconsistent, some first names are replaced with the intial like "N. S. Keska", some are spelled fully like "Diederik P Kingma"; second names sometimes miss punctuation, for instance "Charles G Broyden"; the citation to Nesterov's paper is misspelled, "o(1/k2)"; some references only mention year and title without giving arxiv or journal information, for instance "On the generalization of stochastic gradient descent with momentum". Please make the references clean and consistent.
"LBFGS" and "L-BFGS" are used inconsistently.
On page 30, you use the notation $\Vert \cdot \Vert_M$, which I assume is the matrix-induced norm. Is it introduced anywhere? I couldn't find the relevant place. Same question about the Frobenius norm.

---

> ### Author Response · Authors · 2024-09-28
>
> Dear Editor,
>
> We read the review and is now writing for the detailed response. We would like to clarify several points regarding our manuscript. Our submission aims to serve as a comprehensive "theoretical" handbook on optimization methods for deep learning (DL), designed to provide valuable insights and understanding to both novice and seasoned researchers in the field. As explicitly stated in our abstract, our primary focus is on the theoretical underpinnings of various optimization techniques, rather than their practical performance. This approach is motivated by the belief that providing solid theoretical guarantees is equally important as the application perspective in convincing the research community of the efficacy of proposed methods.  While many survey papers emphasize methodologies and practical applications, they often overlook the theoretical foundations that are crucial for a deeper understanding of optimization techniques. Our manuscript seeks to fill this gap by offering a detailed exploration of the theoretical aspects of optimization methods, thereby supporting researchers in developing a robust theoretical framework.
>
> Regarding the comparison of different methods, we begin by providing a general overview of the evolution of optimization methods in the introduction. Each subsequent section then delves into the detailed properties of the methods discussed, with an introductory paragraph that outlines the key features and theoretical underpinnings of each method.
>
> However, since our submission is primarily a theoretical handbook, a significant portion of the content is dedicated to detailed proofs and derivations. As a result, the comparative analysis between different methods has been somewhat limited. We fully agree with the reviewer that providing such comparisons is crucial for a deeper understanding of the theoretical perspectives of optimization methods. Therefore, we are currently enriching the content to include more comprehensive comparisons between the various methods.
>
> Best

---

> > ### Comment · Action_Editor_qqY5 · 2024-10-01
> >
> > Thanks for your response. I understand your submission is primarily a theoretical handbook and not about the practical performance of the method. I appreciate the fact that you're working on a comparison between different methods, even if it is theoretical and based on what can be proved about them, it should benefit the reader. You can discuss this further with the reviewer, I only wanted to highlight their point as I thought addressing it would make the manuscript better.

---

### Decision · Action_Editor_qqY5 · 2025-02-03

**Recommendation:** Reject

**Comment:**

I encourage the authors to take their time, especially since it is a long paper and it covers so many topics, to properly revisit their work. If the authors add more comparisons to the paper and polish the presentation, it would be great to see it resubmitted to TMLR. If the authors do decide to go that way, I will do my best to assign the same reviewers and get it reviewed faster next time. I do believe it can be made into a nice survey and that many people will find it useful once the discussed issues are resolved, and I hope the authors will take the negative feedback constructively as the reviewers gave clear directions for improvement.

Some really minor points from me:
It would be nice if the authors made the citations consistent, in particular, either abbreviating or not abbreviating first names in all citations. The same also applies to conference names.
There are small issues with punctuation, e.g., it's missing in equations (2.48), (4.93)-(4.95), (D.4), (E.19), and placed wrongly after the equation line in the second equation on page 116.
The authors sometimes use the notation $=O(\cdot)$, e.g., eq. (4.66), and sometimes $\le O(\cdot)$, e.g., eq. (4.27). At least once in the last equation on page 48, the authors use $O(\cdot)$ instead of $\mathcal{O}(\cdot)$.
Eq. (4.122) has a double plus sign "++".
Figure 5's caption should have "Downpour" instead of "Dowpour".

These minor typos didn't affect my decision for the paper, but small issues like that very likely impacted the impression left on the reviewers that the paper wasn't ready. Reviewer 1VbJ also noted that the variance bound typo was only fixed in some places but not everywhere, so please consider double checking it. I want to emphasize that the authors are not expected to make the paper perfect, but reducing the number of small issues and typos to a reasonable level would nevertheless make the survey more appealing to the reader.

**Audience:**

This is where the reviewers do not have strong concerns, so I want to emphasize that the paper could be a very good fit for TMLR if it was written better and the previously mentioned issues were addressed.

**Claims And Evidence:**

There are multiple concerns raised by the reviewers regarding claims and evidence. All reviewers responded "No" to the "Claims and Evidence" part of the paper feedback. This is the main reason I have to suggest rejection of this paper.

First of all, Reviewer Vr6h and Reviewer HTdu both made the point, which Reviewer Vr6h also reinforced in their final revision, that as a survey paper, the submission is supposed to provide new comparative insights into the discussed topic.
The authors responded to Reviewer Vr6h by explaining that they have new convergence results that were not available in the original papers. The reviewer, however, clarified in their final decision that the problem is rather in the lack of comparative insights, while presenting new convergence guarantees isn't even necessary for a survey paper.
The authors also added more discussion related to decentralized optimization to address the concerns of Reviewer HTdu. However, the reviewer still found the section on distributed optimization lacking and suggested looking into the works of Angelia Nedic. The authors emphasized that their focus is not on federated learning and some techniques for handling data heterogeneity are not relevant to their survey. I find this to be a valid point, but in light of other concerns, I suggest the authors try to make the distinction more clear and make the section on distributed optimization more thorough. I found it a bit surprising that there is no discussion on gradient compression and very limited discussion on asynchronous methods.

Next, there is a lack of clarity stemming from how the paper is written. This was a particular cause of concern for Reviewer SP72 and Reviewer 1VbJ. It appears the work would benefit from a much more careful approach to how things are explained, fixing the typos and making sure that the extra insights provided in the survey are explained well. As noted by Reviewer 1VbJ, even critical steps are sometimes not explained properly,

**Resubmission Of Major Revision:**

The authors may consider submitting a major revision at a later time.